# Quaternary rodents of South Africa: A companion guide for cranio-dental identification

**Pierre Linchamps**[1,2]*, **D. Margaret Avery**[3], **Raphaël Cornette**[1], **Christiane Denys**[1], **Thalassa Matthews**[3], **Emmanuelle Stoetzel**[2]

**1** Institut de Systématique, Evolution, Biodiversité (ISYEB) UMR 7205, CNRS, Muséum National d'Histoire Naturelle, UPMC, EPHE, Sorbonne Universités, Paris, France, **2** Histoire Naturelle de l'Homme Préhistorique (HNHP) UMR 7194, CNRS, Muséum National d'Histoire Naturelle, UPVD, Sorbonne Universités, Paris, France, **3** Iziko Museums of South Africa, Cape Town, South Africa

\* pierre.linchamps@gmail.com

## Abstract

Rodentia is the most species-rich order among mammals. The Republic of South Africa harbours a high rodent diversity whose taxonomy and phylogeny have been extensively studied using genetic tools. Such advances have led to the establishment of new faunal lists for the country. Because rodents are frequently recovered from archaeological cave site material and owl pellets, and constitute prime material for studying both past and present environmental conditions, it is necessary to characterize their osteological remains. The skull and teeth are the most useful diagnostic skeletal elements preserved in modern and fossil accumulations. This key provides updated craniodental criteria for identifying rodent genera found in Quaternary deposits, and modern material from the Republic of South Africa, thus facilitating research on past and present rodent diversity.

**Data Availability Statement:** All relevant data are within the paper and its Supporting information files.

## 1. Introduction

The Republic of South Africa (RSA) has a rich storehouse of Quaternary archaeological sites, spanning the last 2.6 MYA from the Early Pleistocene to the late Holocene. It possesses one of the world's richest fossil hominin inventories documenting the origin of modern humans, as well as an infinite amount of animal and plant fossils which have enabled a detailed palaeo-reconstruction of the environments in which we evolved. Small mammals, especially rodents, have been widely used for reconstructing past environments from the Pleistocene to the Holocene [e.g., 1–10]. They have an advantage in that they are frequently abundant in the fossil record, and, because they do not migrate and have small home ranges, generally provide a clearer and more detailed picture of local conditions than the larger mammals.

Rodents are a very diverse group of mammals, with an exceptional taxonomic and phenotypic diversity, and constitute the most species-rich order among mammals, making up over 40% of all living mammal species [11–13]. This variability in body shapes and sizes, together with morphological and physiological adaptations, resulted in the successful colonization of

**Funding:** Funds were given for accessing specimens from Czech Republic and South Africa by the Partenariat Hubert Curien (PHC) Barrande and the international mobility program Transhumance of the doctoral school 227 "Sciences de la nature et de l'Homme" from the Muséum national d'Histoire naturelle-Sorbonne Université. The funders had no role in study design, data collection and analysis, decision to publish, or preparation of the manuscript.

**Competing interests:** The authors have declared that no competing interests exist.

most terrestrial environments [12–14]. Rodents are found in almost every habitat around the world; from open, dry deserts, to thick, wet rainforests, and display a wide array of ecomorphological adaptations including fossorial (burrowing), saltatorial (jumping), arboreal, subaquatic, and gliding forms [15].

Rodents are the preferred prey of many predators, including owls and small and meso-carnivores, which are often the agents responsible for the accumulation of micromammal remains in archaeological and palaeontological cave sites [16–19]. For instance, diurnal and nocturnal raptors regurgitate pellets that contain the undigested parts (mostly hair, bones, and teeth) of their prey at roost and nest sites; those pellets may accumulate over a long period of time, and bones and teeth become incorporated into the sediment as they break down. Analysis of modern owl pellets thus provides useful comparative information for identifying cranial fragments and teeth of fossil Pleistocene microfauna.

Whether the remains are fossil or (sub)contemporaneous, the material used for identification are the mandibles and maxillae (these are seldom complete, and rarely retain all the teeth) and also isolated teeth. The most effective method for identification of specimens at species level is a morpho-anatomical comparison of the fossil material with a modern reference collection. However, these are not always easy to access, and modern, comparative collections in museums, for example, may contain specimens which are misidentified, poorly documented, or which have unresolved taxonomies. A solution to these problems is the use of an identification key. There are several available keys for the identification of southern African rodents [20–22] but they generally rely on external characteristics such as length of the body, proportions of the tail or hindfeet, color and pattern of the pelage or number of nipples, and are therefore of little use for identifying cranio-dental remains. The only exhaustive existing keys based on cranial characters are those of Coetzee [23] and De Graaff [24], which are not up to date in terms of taxonomic research, and some other publications cover only specific families or subfamilies [25–27]. More recently, the systematics of many taxa has been partially resolved, with several new species being described and published in taxonomic reference publications [12, 13, 22], as well as in many systematic studies based upon genetic analyses and field surveys [e.g., 28–31]. Many recently described taxa are missing from earlier keys and this, together with the problems related to obtaining reliable comparative collections, has prevented the correct identification, and effective palaeoenvironmental reconstructions of quaternary rodent assemblages.

Here we propose a new identification key of cranio-dental morphological features which are of generic diagnostic importance, and apply them to material extracted from owl pellets and Quaternary fossil assemblages. The key, which follows the systematics of Wilson et al. [12, 13], covers all genera that occurred in South Africa during the Quaternary, including extinct fossil taxa, as well as more recently introduced taxa. Since cryptic diversity has been showed to occur within several genera, and because many uncertainties remain regarding the validity and/or taxonomic status of some extinct species, we chose not to provide identification guidance beyond the genus level, pending further systematic, taxonomic, and morphometric investigations. To assist in the identification of specimens in the field, and to get an idea of the relative size of each genus, some synthetic plates with full-scale photos of modern specimens are provided as supplementary material (S1 Fig for upper jaws, S2 Fig for lower jaws, S1 Table for references of photographed specimens; the plates S1 and S2 Figs should be printed at actual size to maintain full-scale).

As this key is mostly dedicated to researchers working in palaeontology, archaeology, and taphonomy, diagnostic information is based on features and materials that are most often recovered from fossil or taphonomic sites, i.e., upper and lower toothrows, and mandibles and maxillae with, and without, teeth. Some other cranial features which are often useful for

identifying rodents, such as the auditory bullae and width of the nasals, are not dealt with in detail here because they are not generally preserved on the material studied.

## 2. Past and present rodent diversity in South Africa

There are currently 35 genera of rodents living in RSA (see Table 1), with one additional valid fossil genus listed for the Quaternary. These 36 genera belong to nine families:

- The Muridae is the largest family of rodents, including many diverse species such of mice, rats and gerbils.

- Mole-rats of the family Bathyergidae are burrowing rodent with cylindrical bodies and short limbs that have distinct morphological adaptations to subterranean life.

- Dormice of the family Gliridae are small nocturnal rodents, largely arboreal and well adapted to climbing; in sub-Saharan Africa, they are only represented by the genus *Graphiurus*.

- Porcupines of the family Hystricidae are large, nocturnal rodents that have quills (spines) for defense against predators.

- The family Petromuridae is monospecific, it contains the dassie rat (*Petromus typicus*)—a medium size rodent which inhabits rocky outcrops.

- Springhares also belong to a single genus which constitutes the family Pedetidae; they are nocturnal rodents whose method of locomotion is hopping with their strong hind legs.

- The family Nesomyidae is diverse, and contains small to medium size morphologically varied rodents endemic to continental Africa and Madagascar.

- South African squirrels of the family Sciuridae are terrestrial (*Geosciurus*) or semi-arboreal (*Paraxerus*) diurnal rodents with an elongated body and bushy tail.

- Cane rats of the family Thryonomyidae are large, heavily built rodents that live in marshy areas and along riverbanks.

In the genus accounts of this work, we provide a short synthesis of the most recent research on the phylogeny and geographical distribution of each genus and species, as well as an overview of the fossil record in South Africa during the Quaternary period.

## 3. General anatomy and glossary for the lower and upper jaws

Despite the significant number of species, and a great diversity of morphological and ecological adaptations, rodents are remarkably uniform regarding the general morphology of the skull and dentition. The upper and lower jaws (Fig 1) each support a single pair of large, evergrowing scalpriform incisors. The enamel is mainly limited to the outer surface of the incisors. The presence of iron in the mineral phase of incisor enamel can give the surface of some rodent incisors a yellowish to orange appearance which can be diagnostic for some genera; however, the incisor colour criterion is not always usable in an owl pellet and fossil context due to taphonomic alteration (digestion, diagenesis, staining with sediment, etc.). A large diastema is present between the incisors and the cheekteeth, allowing the lips to fold inwards in order to prevent debris interfering with chewing activity when gnawing. The maximum number of cheekteeth is four for the lower jaw, and five for the upper jaw, with murids having only three molars in each jaw and an incisor in each mandible and premaxilla. The pattern on the

**Table 1. List of Pleistocene and modern rodent species from Republic of South Africa (RSA) based on Wilson et al. [2016, 2017].** Taxa are listed in alphabetical order. Published extinct taxa are identified by the symbol † in the table.

| Family | Subfamily | Genus | Species | Common generic name |
|---|---|---|---|---|
| Bathyergidae | Bathyerginae | *Bathyergus* | *janetta, suillus* | dune mole-rat |
| | | *Cryptomys* | *hottentotus, †robertsi* | mole-rat |
| | | *Fukomys* | *damarensis* | mole-rat |
| | | *Georychus* | *capensis* | Cape mole-rat |
| Gliridae | Graphiurinae | *Graphiurus* | *microtis, murinus, ocularis, platyops, rupicola* | dormouse |
| Hystricidae | | *Hystrix* | *africaeaustralis, †makapanensis* | crested porcupine |
| Muridae | Deomyinae | *Acomys* | *selousi, subspinosus* | spiny mouse |
| | Gerbillinae | *Desmodillus* | *auricularis* | Cape sort-tailed gerbil |
| | | *Gerbilliscus* | *afra, brantsii, leucogaster, paeba, vallinus* | gerbil & hairy-footed gerbil |
| | Murinae | *Aethomys* | *chrysophilus, ineptus* | veld rat |
| | | *Dasymys* | *capensis, incomtus, robertsii* | shaggy rat |
| | | *Grammomys* | *cometes, dolichurus* | thicket rat |
| | | *Lemniscomys* | *rosalia* | grass mouse |
| | | *Mastomys* | *coucha, natalensis* | multimammate mouse |
| | | *Micaelamys* | *granti, namaquensis* | lesser veld rat |
| | | *Mus* | *indutus, minutoides, musculus, neavei* | old world & pygmy mouse |
| | | *Myomyscus* | *verreauxii* | meadow mouse |
| | | *Otomys* | *angoniensis, auratus, †gracilis, irroratus, karoensis, laminatus, sloggetti, unisulcatus* | vlei rat or laminate-toothed rat |
| | | *Parotomys* | *brantsii, littledalei* | whistling rat |
| | | *Rattus* | *rattus, norvegicus, tanezumi* | rat |
| | | *Rhabdomys* | *bechuanae, dilectus, intermedius, pumilio* | four-striped grass mouse |
| | | *Thallomys* | *†debruyni, nigricauda, paedulcus, shortridgei* | acacia rat or tree rat |
| | | *Zelotomys* | *woosnami* | broad-headed mouse |
| Nesomyidae | Cricetomyinae | *Cricetomys* | *ansorgei* | giant pouched rat |
| | | *Saccostomus* | *campestris* | pouched mouse |
| | Dendromurinae | *Dendromus* | *melanotis, mesomelas, mystacalis, nyikae* | African climbing mouse |
| | | *Malacothrix* | *typica* | Long-eared Mouse |
| | | *Steatomys* | *krebsii, pratensis* | fat mouse |
| | Mystromyinae | *Mystromys* | *albicaudatus, †hausleitneri* | African white-tailed rat |
| | | *†Proodontomys* | *†cookei* | |
| | Petromyscinae | *Petromyscus* | *barbouri, collinus, monticularis* | pygmy rock mouse |
| Pedetidae | | *Pedetes* | *capensis, †hagenstadti* | springhare |
| Petromuridae | | *Petromus* | *typicus* | noki or dassie rat |
| Sciuridae | | *Geosciurus* | *inauris, princeps* | ground squirrel |
| | | *Paraxerus* | *cepapi, palliatus* | bush squirrel |
| Thryonomyidae | | *Thryonomys* | *swinderianus* | cane rat |

occlusal surface of molars is remarkably varied and diverse among rodents, and thus is of great diagnostic taxonomic value.

Below is a glossary of the main terms that will be used in the following anatomical descriptions:

**angular process**. Also called *processus angularis*; process at the posterior lower corner of the mandible that serves for muscular attachment.

**brachyodonty**. Cheekteeth with short crowns.

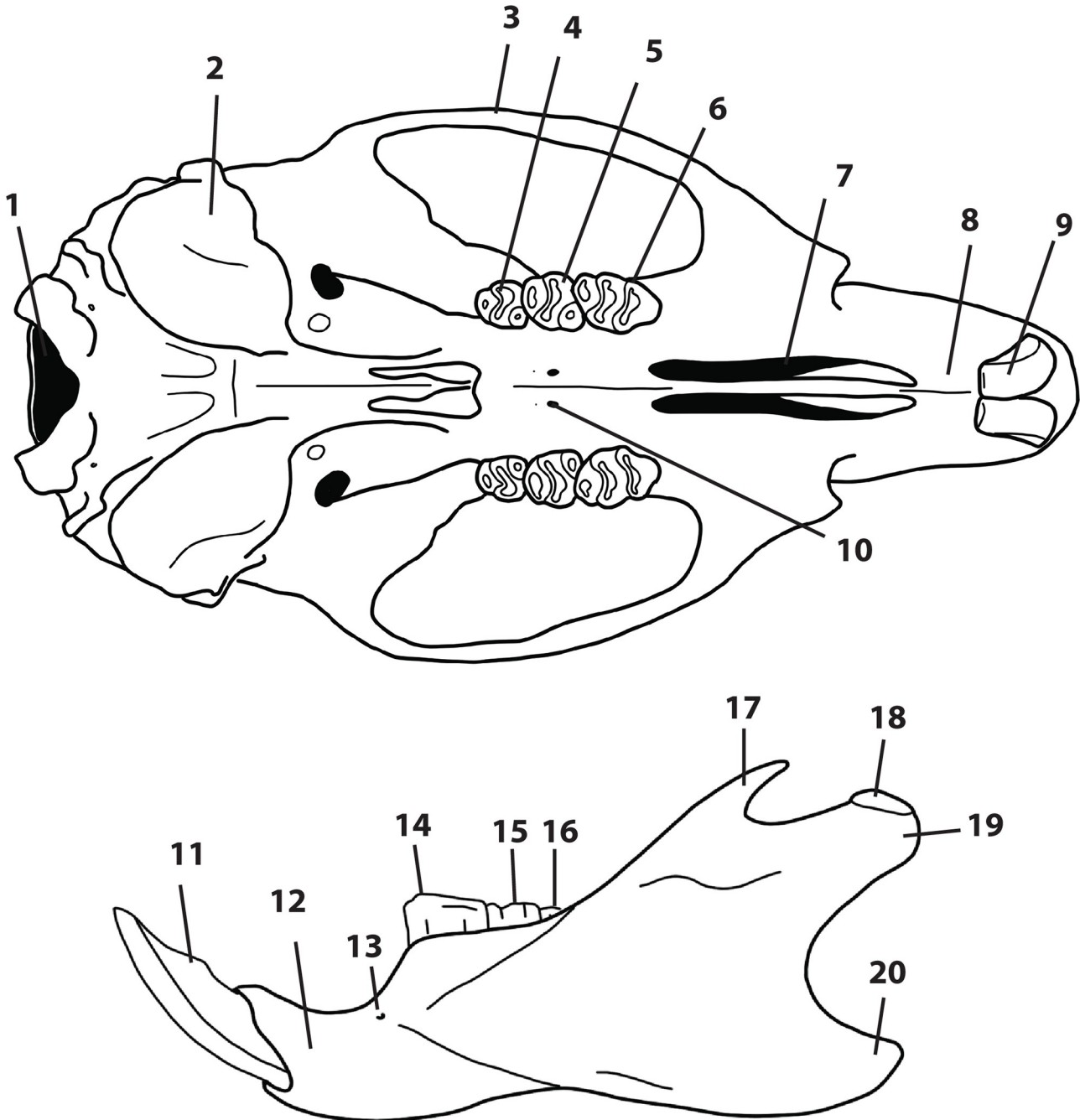

**Fig 1. Schematics and nomenclature of rodent cranium (1–10) and mandible (11–20).** 1: foramen magnum; 2: tympanic bulla; 3: zygomatic arch; 4: third upper molar ($M^3$); 5: second upper molar ($M^2$); 6: first upper molar ($M^1$); 7: first or anterior palatal foramen; 8: diastema; 9: incisor; 10: second or posterior palatal foramen; 11: incisor; 12: body of mandible; 13: mental foramen; 14: first lower molar ($M_1$); 15: second lower molar ($M_2$); 16: third lower molar ($M_3$); 17: coronoid process; 18: articular surface; 19: condylar process; 20: angular process.

> **bunodonty**. Cheekteeth in which the cusps are high and rounded on the occlusal surface of the crown.
>
> **buno-lophodonty**. Cheekteeth in which the cusps tend to connect transversally but remain partially individualized at low wear.

**condylar process**. Process at the posterior upper corner of the mandible that articulates with the glenoid fossa of the skull, forming the lower hinge of the jaw articulations.

**coronoid process**. Process at the anterior upper corner of the mandible ramus situated anteriorly to the condylar process. It serves as an attachment point for the temporalis muscle and does not participate in the jaw articulation.

**cusp**. Occlusal eminence on the surface of a tooth.

**cusplet**. Small cusp, often located on the edge of a tooth.

**diastema**. Gap between the incisor and the cheekteeth.

**foramen**. Orifice in a bone through which nerves and blood vessels pass.

**hypselodonty**. Ever-growing teeth with very long crown and short roots/rootless.

**hypsodonty**. Teeth with high crowns.

**hystricognathous**. Condition of the mandible with the angular process deflected lateral to the plane that includes the alveolus of the incisors (see Fig 2).

**hystricomorphy**. Type of rodent skull with enlarged infraorbital foramen (see Fig 3).

**incisor groove**. Longitudinal groove running lengthways along the anterior enamel surface of lower and/or upper incisor of several rodent genera. The number of grooves (maximum 3) is often specific to a genus, sometimes to species (e.g., in *Otomys* and *Parotomys*).

**incisor notch**. Notching on the incisal worn surface of the upper incisors (see Fig 4).

**incisor procumbency**. Orientation of the upper incisors, defined by the position of the cutting edge in relation to the vertical plane of the incisor (see Fig 5).

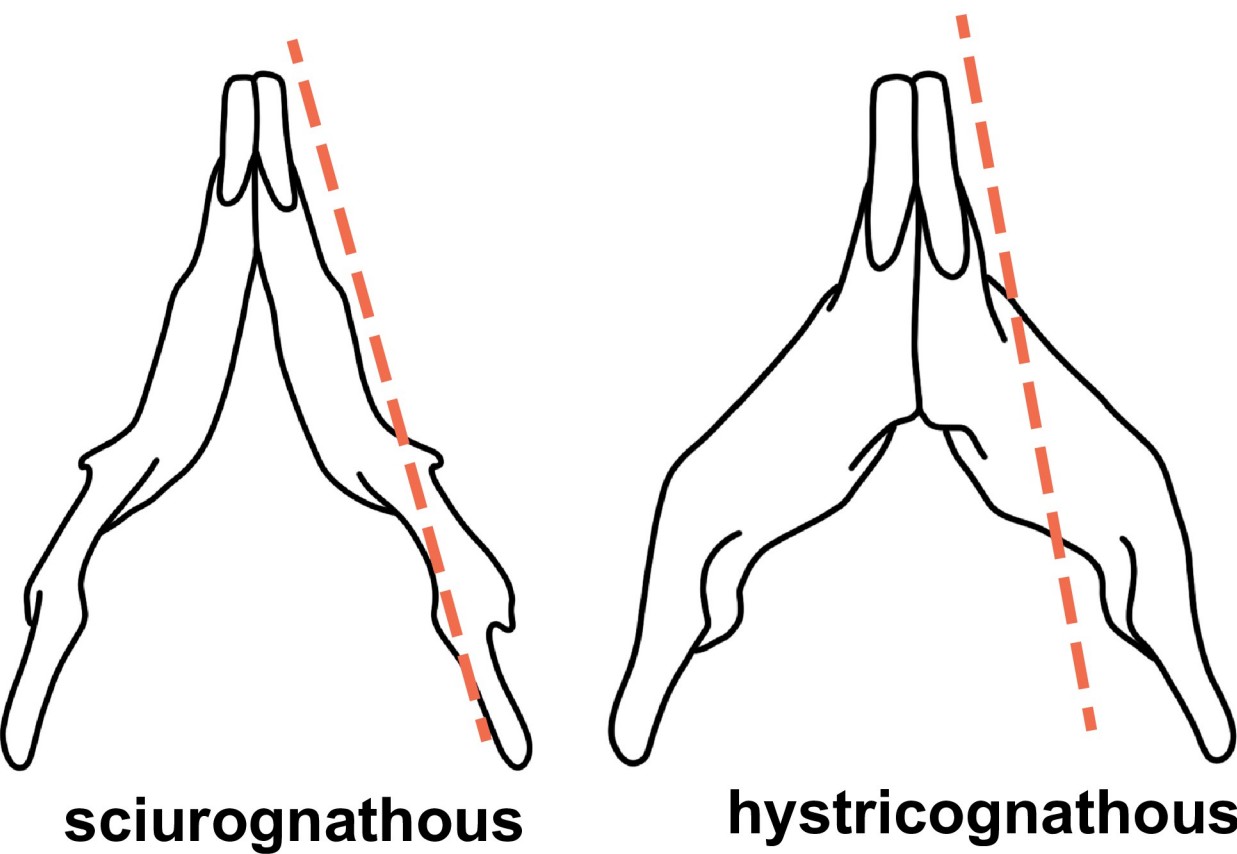

**Fig 2. Two mandibles in ventral view showing a sciurognathous condition (left) and hystricognathous condition (right).**

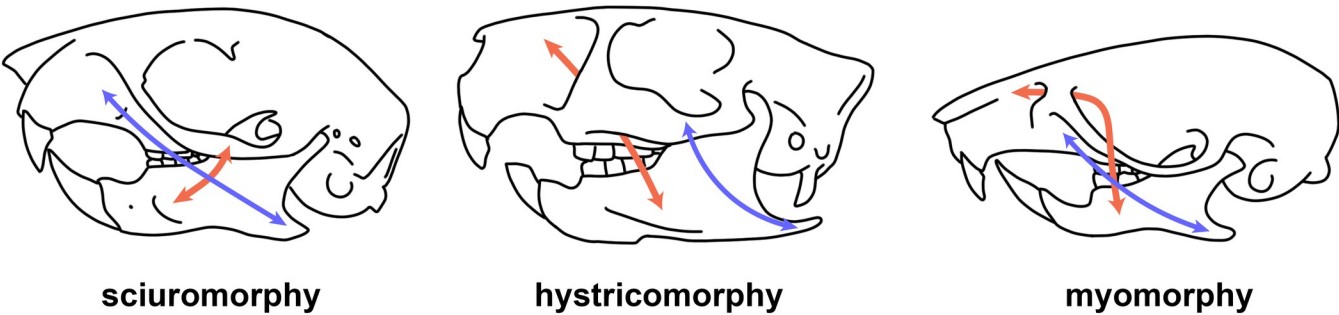

**Fig 3. Three types of zygomasseteric architecture of rodent skulls.** Arrows show the origin and the insertion of the masseter muscle (medial in red, lateral in blue).

**lophodonty**. Cheekteeth in which the cusps are fused to form transverse ridges (lophs).

**macrodonty**. Large teeth in proportion to the skull.

**masseter knob**. Small bony process located close to and below the anterior root of the zygoma (see Fig 6).

**microdonty**. Small teeth in proportion to the skull.

**myomorphy**. Type of rodent skull that combines a large zygomatic plate and a well-developed infraorbital foramen.

**opisthodonty**. Of the incisors, when the cutting extends posterior to the vertical plane (incisors are directed posteriorly; see Fig 5).

**orthodonty**. Of the incisors, when the cutting edge is perpendicular to the plane (incisors are directed more or less vertical plane; see Fig 5).

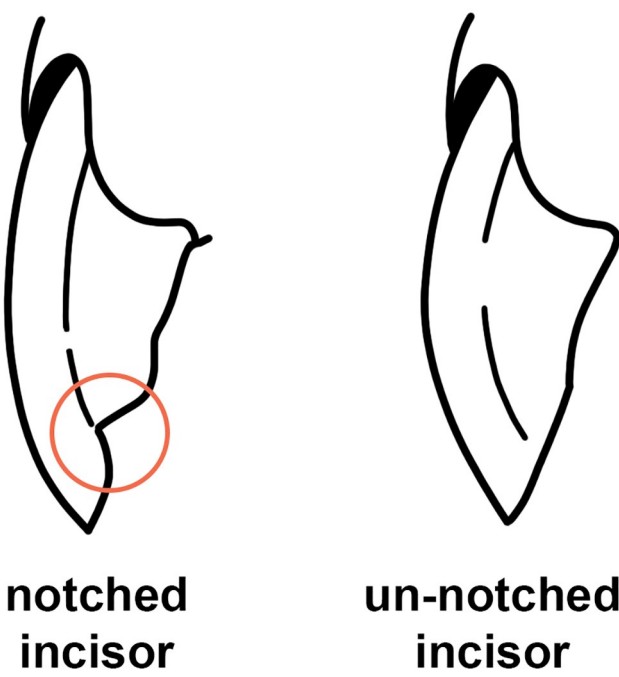

**Fig 4. Notched and un-notched left upper incisors in lateral view.**

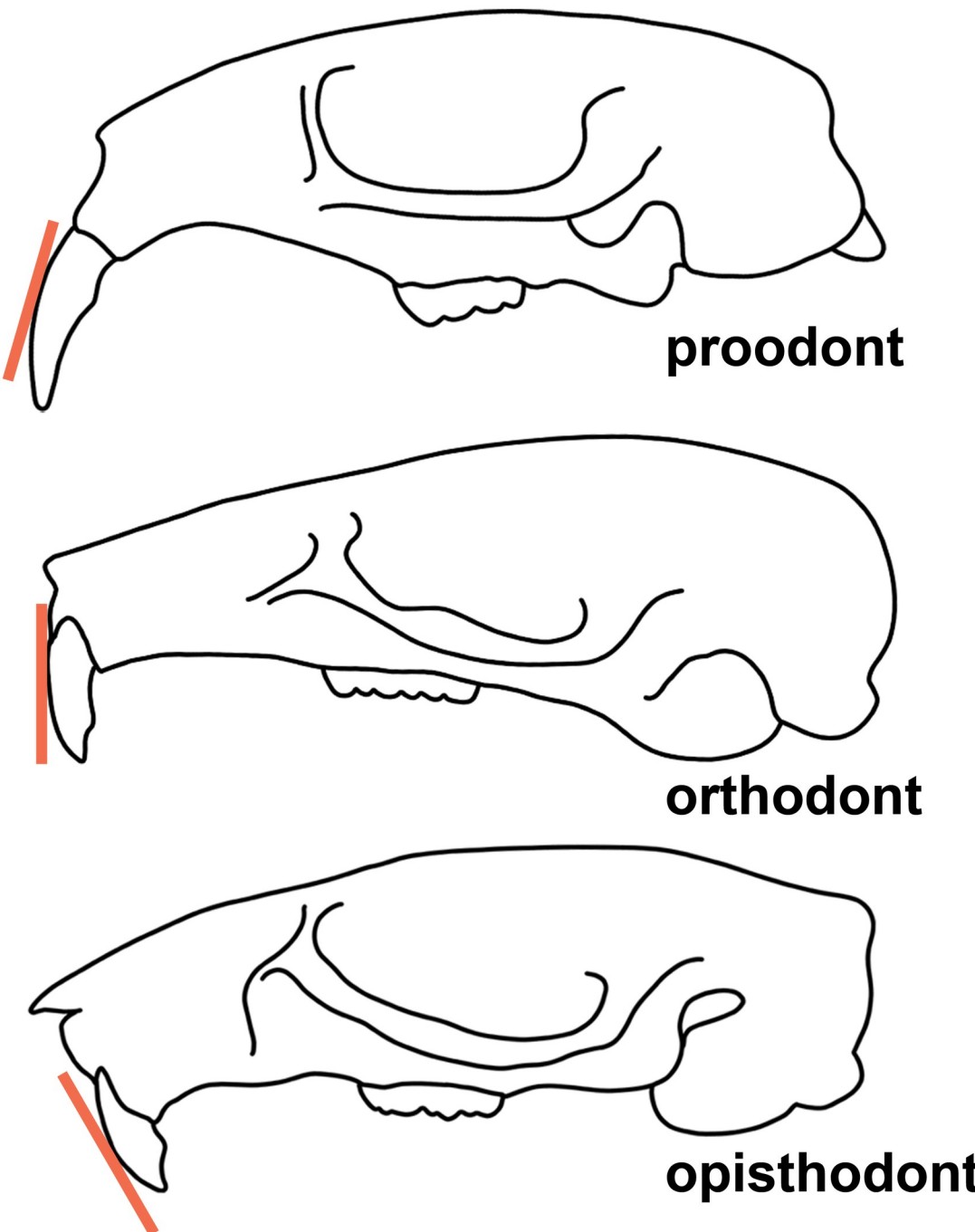

**Fig 5. Three different types of orientation of the upper incisor: Proodont (top), orthodont (middle), opisthodont (bottom).**

**palatal foramina**. Orifices in the bony palate for the transmission of palatine vessels and nerves. Anterior palatal foramina (also called incisive foramina) lie between the incisors and the cheekteeth. Posterior palatal foramina are situated between the two rows of cheekteeth.

**posterior cingulum**. Small, rounded structure of enamel on the posterior edge of the molar occlusal surface.

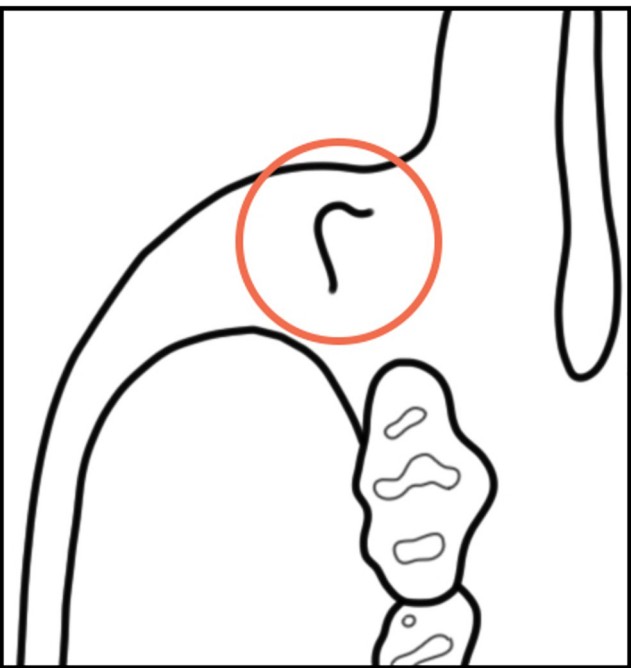

**Fig 6. Occlusal view of the upper right toothrow to show the masseter knob, circled in red.**

**proodonty**. Of the incisors, when the cutting edge extends anterior to the vertical plane (incisors are directed anteriorly; see Fig 5).

**scalpriform**. Shaped like a chisel.

**sciurognathous**. Condition of the mandible with the angular process in the same plane that the alveolus of incisors (see Fig 2).

**sciuromorphy**. Type of rodent skull with reduced infraorbital foramen and anterior part of the zygomatic arch developed into a large plate (see Fig 3).

**stephanodonty**. Cheekteeth in which ridges are connecting the various cusps in longitudinal rows.

## 4. Material and methods for constructing the key

### 4.1. Material

This key provides a simple step-by-step process for identifying rodents from fossil and owl pellet material in South Africa. For this work we have examined representative specimens from the collections of the Ditsong National Museum of Natural History, Republic of South Africa (DNMNH), Evolutionary Studies Institute of the University of the Witwatersrand, Republic of South Africa (ESI), Muséum national d'Histoire naturelle, France (MNHN), Musée royal de l'Afrique centrale, Belgium (RMCA), and the Institute of Vertebrate Biology, Czech Republic (IVB). List of examined museum specimens with associate catalogue numbers can be found in S1 Checklist. No permits were required for the described study, which complied with all relevant regulations.

This contribution has also relied heavily on information published in other works. For information related to biology, ecology and anatomy of the rodents, the comprehensive works of De Graaff [24], Happold [21], Monadjem et al. [22], and Wilson et al. [12, 13] have been of

great value. In addition to these works, a multitude of publications are recommended for supplementing nomenclature and anatomical description of each rodent family; they are explicitly stated in each section through literature citations.

## 4.2. Illustrations

Teeth and skull photographs were taken using Nikon digital camera D 5500 coupled with AF-S Micro NIKKOR 60 mm and macro extension tubes. Photos were stacked using Helicon Focus 8.1.4 and edited in Adobe Photoshop CC 21.1.3. We have tried to show the intrageneric variability in dental anatomy by including photographs of several species in each genus description. Schematics drawings of rodent craniodental morphology, including illustrations of some distinctive features for identification, were realized with Adobe Photoshop CC 21.1.3 and Adobe Illustrator CC 21.0.0. Distribution maps of each species have also been included. These maps are based on distributional data published in Monadjem et al. [22], Wilson et al. [12, 13], data from IUCN (*International Union for Conservation of Nature*) red list database and range maps from various expert sources available on the Map of Life's website at https://mol.org/ datasets/?dt = range&sg = Mammals [32]. They were designed using R software version 4.1.0.

## 4.3. How to identify families and genera of rodents

The first step of the key selects the name of the family to which a rodent belongs (Tables 2 and 3). For each family, further keys provide readers with a series of statements and two or three choices which will eventually lead to the correct identification of the organism. Some families contain only one genus in South Africa (Gliridae, Hystricidae, Pedetidae, Petromuridae and Thryonomyidae), facilitating easy identification. The keys to the genera (Tables 4 and 5, [32, 33, 43, 44]) are preceded by notes on the habits, preferred habitats, and potential predators, and are followed by a description of each genus and salient morphological characters. It contains dental and alveolar formulas (described below), images of the right upper and right lower molar rows, as well as distribution maps of each species in RSA. The reader may sometimes

**Table 2. Key to the rodent families: Upper jaw.**

| 1 | 3 cheekteeth in adults | 2 |
|---|---|---|
| | 4 or 5 cheekteeth in adults | 3 |
| 2 | first lobe of $M^1$ with 3 cusps | Muridae* |
| | first lobe of $M^1$ with 1 or 2 cusps | Nesomyidae** |
| 3 | occlusal pattern flat and simplified | 4 |
| | occlusal surfaces with infolds and/or islands of enamel | 5 |
| | occlusal surfaces with cusps and/or transverse ridges | 6 |
| 4 | cheekteeth rooted; incisors markedly proodont; occlusal surfaces simple, either ring or 8-shaped | Bathyergidae |
| | cheekteeth rootless; incisors opisthodont; teeth bilobed, occlusal surfaces with a single re-entrant fold on the lingual (lower toothrow) or labial (upper toothrow) side of each cheektooth | Pedetidae |
| 5 | massive, rounded molars; wavy enamel pattern with multiple crests and islands | Hystricidae |
| | upper molars have two labial folds and one lingual fold | Thryonomidae |
| | deep infolds that seem to divide each molar into two separate parts | Petromuridae |
| 6 | 4 cheekteeth; width of palate about equal to LUTR; toothrow $\leq$ 4 mm | Gliridae |
| | 4 or 5 cheekteeth; width of palate smaller than LUTR; toothrow $>$ 6 mm | Sciuridae |

* Except for the Gerbillinae and Otomyini that have a lophodont dentition

** Except for worn specimens of *Proodontomys* which have a semi-lophodont dentition with flat occlusal surface

**Table 3. Key to the rodent families: Lower jaw.**

| | | |
|---|---|---|
| 1 | 3 cheekteeth in adults; mandible sciurognath | 2 |
| | 4 cheekteeth in adults; mandible sciurognath or hystricognath | 3 |
| 2 | first lobe of $M_1$ with 2 cusps (with possible presence of an additional tma) | Muridae* |
| | first lobe of $M_1$ with 1 cusp (or two poorly differentiated in some *Saccostomus*) | Nesomyidae** |
| 3 | occlusal pattern flat and simplified | 4 |
| | occlusal surfaces with infolds and/or islands of enamel | 5 |
| | occlusal surfaces with cusps and/or transverse ridges | 6 |
| 4 | cheekteeth rooted; occlusal surfaces simple, either ring or 8-shaped; mandible hystricognath | Bathyergidae |
| | cheekteeth rootless; teeth bilobed, occlusal surfaces with a single re-entrant fold on the lingual (lower toothrow) or labial (upper toothrow) side of each cheektooth; mandible sciurognath | Pedetidae |
| 5 | massive, rounded molars; wavy enamel pattern with multiple crests and islands | Hystricidae |
| | lower molars have two lingual folds and one labial fold | Thryonomidae |
| | deep infolds that seem to divide each molar into two separate parts | Petromuridae |
| 6 | smaller: toothrow < 4 mm | Gliridae |
| | larger: toothrow ≥ 7 mm | Sciuridae |

* Except for the Gerbillinae and Otomyini that have a lophodont dentition

** Some Saccostomus display two poorly differentiated on first lobe of $M_1$

rely on geographical distributions in distinguishing genera; for example, in the case of *Mastomys* and *Myomyscus*, which are morphologically very close and have a limited overlapping distribution. However, the precision of distribution maps is limited for various reasons, such as species being falsely identified in the field or in collections (also morphologically cryptic species may not be distinguished), lack of knowledge of a taxon's range, errors introduced in the georeferencing procedure, etc. Furthermore, the range of species and genera have been noted to sometimes change significantly over time [19], so these maps should be used with extreme caution when identifying fossil specimens.

We tried to use mainly discriminating criteria, which correspond to diagnostic characters of a genus and are present in all the individuals. As discriminating criteria are not always available in cranio-dental morphology for some taxa, or for broken specimens within the fossil or pellet material, we also propose secondary criteria, *i.e.* character states that are not absolute in terms of identifying to genus. Secondary criteria are sometimes absent in taxa that exhibit great intraspecific variation in size and shape, or may be subject to subjective interpretation. The presence of several secondary criteria can lead to the confident identification of a taxon, but identification must be based on as many characters as possible. We specify in the key when a listed character is likely to display variability.

A modified version of the key can be accessed online with Xper[3], a free collaborative platform designed for computer-aided taxonomic description and identification [33, 34], at: https://rodentsouthafrica.identificationkey.org. The online key will be updated over time and integrate additional photographs for modern and fossil specimens.

We draw attention to the fact that the criteria used in this key are for South African taxa only and may not be applicable extralimitally. Furthermore, in an archaeological and palaeontological context, many specimens have been fragmented or damaged by taphonomic processes. Methods of genetic identification are not always feasible, and taxonomy based on geometric morphometrics is time-consuming and requires a certain amount of expertise. In such situations, the wisest attitude is to limit identification at a higher taxonomic

**Table 4. Key to the murid and nesomyid genera: Upper jaw.**

| | | |
|---|---|---|
| 1 | tooth are laminate; $M^3$ is the largest molar | 2 (*Otomys* or *Parotomys*) |
| | tooth are semi-laminate (lamelliform/buno-lophodont cusps); $M^1$ is the largest molar | 3 |
| | tooth are not laminate and have well-defined cusps; $M^1$ is the largest molar | 6 |
| 2 | bullae enlarged; $M^3$ has two or three complete laminae; upper incisors grooved or ungrooved | *Parotomys* |
| | bullae less inflated; $M^3$ has four or more laminae; upper incisors have one or more grooves | *Otomys* |
| 3 | lamelliforms cusps in $M^1$ and $M^2$ lack longitudinal connections | 4 |
| | lamelliforms cusps in $M^1$ and $M^2$ are connected longitudinally in their central region; not found in modern material (last occurrence around 1 MYA) | †*Proodontomys* |
| 4 | larger: LUTR $\geq$ 5.5 mm and $WM^1 \geq$ 2 mm | *Gerbilliscus* (*Gerbilliscus*) |
| | smaller: LUTR $\leq$ 5 mm and $WM^1 <$ 2mm | 5 |
| 5 | $M^1$ has three roots; $M^3$ has one lobe (or two poorly separated when unworn); bullae are very large proportionate to the skull | *Desmodillus* |
| | $M^1$ has four roots; $M^3$ has one or two lobes; bullae are smaller | *Gerbilliscus* (*Gerbillurus*) |
| 6 | first lobe of $M^1$ has one or two cusps | 7 |
| | first lobe of $M^1$ has three cusps | 13 |
| 7 | skull is large and LUTR > 8 mm | *Cricetomys* |
| | skull is smaller and LUTR < 8 mm | 8 |
| 8 | cusps of $M^1$ are arranged in a zigzag enamel pattern connected by a median longitudinal crest running the length of the tooth | 9 |
| | no median longitudinal crest in $M^1$ | 10 |
| 9 | LUTR < 5 mm | *Petromyscus* |
| | LUTR > 6 mm | *Mystromys* |
| 10 | $M^3$ has two lobes; cusps have a bulbous aspect; masseter knob absent | *Saccostomus* |
| | $M^3$ is reduced and has one lobe; molars show typical Dendromurinae pattern; masseter knob present | 11 |
| 11 | palate extends far beyond $M^3$; $M^1$ is very long, occupying half of the length of the toothrow; masseter knob ridge-shaped | *Malacothrix* |
| | palate ends close after $M^3$; $M^1$ is shorter proportionate to the toothrow; masseter knob not ridge-shaped | 12 |
| 12 | molars slightly larger; $M^1$ 3-rooted | *Steatomys* |
| | molars slightly smaller; $M^1$ 4-rooted | *Dendromus* |
| 13 | small size: LUTR < 4.2 mm | 14 |
| | medium or large size: LUTR > 4.2 mm | 15 |
| 14 | $M^3$ has a t3; masseter knob absent or reduced; palate extends far backwards | *Acomys* |
| | $M^3$ has no t3; conspicuous masseter knob; palate ends closer to the toothrow | *Mus* |
| 15 | pronounced stephanodonty; $M^3$ belongs to Group 6 in Fig 12 | 16 |
| | no or incomplete stephanodonty; $M^3$ belongs to Group 3, 4 or 5 in Fig 12 | 17 |
| 16 | smaller: mean LUTR $\leq$ 5 mm; t3 reduced or absent in $M^2$, which has four alveoli; large bullae (diameter usually > 6mm) | *Grammomys* |
| | larger: mean LUTR $\geq$ 5 mm; t3 present in $M^2$, which has five alveoli; smaller bullae (diameter usually < 6mm) | *Thallomys* |
| 17 | macrodonty; rows of cusps arranged in transverse rows with tendency to isolate and become laminate with wear | *Dasymys* |
| | molars of small or average size; rows of cusps showing slight or pronounced distortion | 18 |
| 18 | well-developed t9 projecting outwards, giving the impression that the $M^1$ is leaning obliquely | 19 |
| | t9 of small or average size, with $M^1$ positioned straight in the anteroposterior axis | 22 |

*(Continued)*

**Table 4.** (Continued)

| | | |
|---|---|---|
| 19 | $M^1$ 3-rooted; shorter toothrow; palatal foramina penetrating between the molars | 20 |
| | $M^1$ 5-rooted; longer toothrow; palate foramina stop at or just short of the root of $M^1$ | *Rattus* |
| 20 | larger: mean LUTR = 5.3 mm; $M^3$ belongs to Group 5 in Fig 12; strong incisors | *Zelotomys* |
| | smaller: mean LUTR = 4.8 mm; $M^3$ belongs to Group 3 in Fig 12; incisors less thick | 21 |
| 21 | posterior palatal foramina generally set between posterior part of $M^2$; wide distribution | *Mastomys* |
| | posterior palatal foramina generally set between anterior part of $M^2$; restricted distribution | *Myomyscus* |
| 22 | smaller: LUTR usually < 5mm; in the $M^1$ the two distal rows of cusps appear to be linked on the labial and lingual sides; in both $M^1$ and $M^2$ the t9 is reduced to a small ridge | *Rhabdomys* |
| | larger: LUTR usually > 5mm; in the $M^1$ the two distal rows of cusps may be linked or not on the labial and lingual sides; t9 not reduced | 23 |
| 23 | LUTR $\approx$ length of the upper palatal foramina; $M^1$ has four alveoli; distal lobe on $M^3$ has one or two poorly differentiated cusps | 24 |
| | LUTR < length of the upper palatal foramina; $M^1$ has five alveoli; distal lobe on $M^3$ has two differentiated cusps | *Lemniscomys* |
| 24 | molars slightly smaller; t1 is rather aligned with t2 and t3 | *Micaelamys* |
| | molars slightly larger; t1 is usually behind t2 and t3 | *Aethomys* |

determination level such as the family; it is inevitable that some specimens will be excluded from the fossil record.

**4.3.1. Morphological characters.** In rodent jaws, the most important morphological character for genus identification is the tooth cusp pattern. Other diagnostic characters for the upper jaw include: number of cheekteeth, incisor morphology, location of the primary and secondary palatal foramina, origination of the zygomatic, presence of a masseter knob, etc., and for the lower jaw: location of the mental foramen, muscle attachment, shape of the mandible, projection of the mental, coronoid and condylar processes, etc. The number of tooth types is written as a dental formula, with the upper and lower teeth shown consecutively. Incisors (I) are indicated first, canines (C) second, premolars (P) third, and molars (M) fourth, giving the formula: I-C-P-M:I-C-P-M.

As loss of teeth is frequent in predator-derived assemblages, alveolar pattern also provides identification information for several taxa. Avery [6] published a useful key (which relies mostly on the size and number of alveoli) to distinguish mandibles of Wonderwerk micromammals in the absence of diagnostic teeth. We use here a similar alveolar-molar root formula to indicate the number of alveoli of the various Muridae and Nesomyidae genera, but here the small, round alveoli resulting from rootlets are counted as independent alveoli. The alveolar formula provides the number of alveoli of each cheektooth and is written in a similar way to dental formula, with upper and lower alveolar patterns consecutively. Thus, the formula 4-3-3:3-2/3-2 means that the upper molars $M^1$, $M^2$ and $M^3$ have four, three, and three alveoli respectively, and the lower molars $M_1$, $M_2$ and $M_3$ have three, two or three, and two alveoli respectively. There may be variability in the number of roots and rootlets of some taxa, so indication of the alveoli should be used as a guiding, but not absolute, criterion.

**4.3.2. Size and measurements.** There is a great variation in size in South African rodents: the smallest (*Mus indutus*) weighs only 3 to 5 g and the largest (*Hystrix africaeaustralis*) weighs about 20 kg. The same is true for the size of the cheekteeth, and measurements of length and width can assist in identification at genus level. Fig 7 below presents measurements of the length of the upper toothrow (LUTR) and the length of the lower toothrow (LLTR) of each

**Table 5. Key to the murid and nesomyid genera: Lower jaw.**

| | | |
|---|---|---|
| 1 | tooth are laminate | 2 |
| | tooth are semi-laminate (lamelliform cusps) | 3 |
| | tooth have well defined cusps | 6 |
| 2 | lower incisors grooved (one or more grooves) | *Otomys* |
| | lower incisors ungrooved | *Parotomys* |
| 3 | lamelliforms cusps in $M_1$ and $M_2$ lack longitudinal connections | 4 |
| | lamelliforms cusps in $M_1$ and $M_2$ are connected longitudinally in their central region; not found in modern material (last occurrence around 1 MYA) | *†Proodontomys* |
| 4 | cusps in second row of $M_1$ fused in a transverse lamina | *Gerbilliscus (Gerbilliscus)* |
| | cusps in second row of $M_1$ unfused | 5 |
| 5 | $M_1$ has two alveoli | *Desmodillus* |
| | $M_1$ has four alveoli | *Gerbilliscus (Gerbillurus)* |
| 6 | cusps in $M_1$ are fused by a median longitudinal crest running the length of the tooth and arranged in a zigzag enamel pattern | 7 |
| | cusps show no median longitudinal crest | 8 |
| 7 | small: LLTR < 4 mm; lower incisors smooth | *Petromyscus* |
| | larger: LLTR > 5 mm; characteristic enamel band on lower incisors | *Mystromys* |
| 8 | mandible is very large (length between incisor alveolus and condylar process > 4 cm), with a wide, elongated alveolar region; restricted to the Limpopo province; LLTR > 9,5mm | *Cricetomys* |
| | mandible is smaller; LLTR < 9,5mm | 9 |
| 9 | first lobe of the $M_1$ has a single median anterior cusp; muscle attachment on the mandible is not right next to the mental foramen | 10 |
| | first lobe of the $M_1$ has a two or three cusps; position of the muscle attachment in relation to the mental foramen variable | 13 |
| 10 | cups in $M_1$ not alternated; lower incisors display small raised band of enamel | *Saccostomus* |
| | typical Dendromurinae pattern with alternated cusps in $M_1$; lower incisors smooth | 11 |
| 11 | $M_1$ is very long, occupying more than half of the length of the toothrow | *Malacothrix* |
| | $M_1$ is shorter proportionate to the toothrow | 12 |
| 12 | LLTR $\leq$ 3 mm; hd and ed not fused in $M_1$ | *Dendromus* |
| | LLTR $\geq$ 3 mm; hd and ed fused in $M_1$ | *Steatomys* |
| 13 | broad molars (macrodonty) with $WM_1 \geq 2$ mm; first lobe of $M_1$ has three cusps; $M_1$ has no clear pc | *Dasymys* |
| | $WM_1 < 2$ m; first lobe of $M_1$ has two to three cusps; presence of a pc in $M_1$ variable | 14 |
| 14 | LLTR < 3.8 mm | 15 |
| | LLTR > 3.8 mm | 16 |
| 15 | $M_1$ has a typically enlarged alg; $M_3$ has one or two lobes; well-developed coronoid process | *Mus* |
| | alg and alb roughly the same size; $M_3$ has two lobes; poorly developed coronoid process | *Acomys* |
| 16 | $M_1$ has a well-developed anteromedian cusp | 17 |
| | $M_1$ has a poorly developed or no anteromedian cusp | 18 |
| 17 | $M_1$ has a pc and a stephanodont crest | *Grammomys* |
| | $M_1$ has no pc and no stephanodont crest. | *Micaelamys* |
| 18 | conspicuous stephanodont crest in both $M_1$ & $M_2$ | *Thallomys* |
| | no well-marked stephanodont crest in both $M_1$ & $M_2$ | 19 |
| 19 | $M_1$, $M_2$ and $M_3$ with two main roots | 20 |
| | $M_1$, $M_2$ and $M_3$ with three or more roots | 22 |
| 20 | strong incisor; LLTR ± 5.1 mm | *Zelotomys* |
| | incisor of average size; LLTR ± 4.5 mm | 21 |

*(Continued)*

**Table 5.** (Continued)

| 21 | wide distribution; mandible and teeth length slightly larger on average | *Mastomys* |
|---|---|---|
| | restricted distribution; mandible and teeth length slightly smaller on average | *Myomyscus* |
| 22 | much smaller: LLTR < 5mm | *Rhabdomys* |
| | larger: LLTR > 5 mm | 23 |
| 23 | $M_1$ and $M_2$ have a well-developed pc; molars proportionally small in relation to the mandible | *Rattus* |
| | $M_1$ often has no pc; molars of average size in relation to the mandible | 24 |
| 24 | the two most posterior roots of $M_1$ are fused; $M_2$ has three alveoli; additional cusplets often occur on labial side of $M_1$ and $M_2$ | *Aethomys* |
| | the two most posterior roots of $M_1$ are unfused; $M_2$ has six alveoli (presence of two rootlets); lateral cusplets less marked | *Lemniscomys* |

genus from modern museum collections (except for the extinct *Proodontomys* whose measurements were taken on fossils):

We use four types of measurements in this key: LUTR = length of the upper toothrow (including molars and premolar(s) if present); $WM^1$ = width of the first upper molar; LLTR = length of the lower toothrow; $WM_1$ = width of the first lower molar. All measurements were taken with a digital calliper to the nearest 0.01 mm. The length and width of the teeth correspond to the maximum values along the mesiodistal and labiolingual axes of the teeth on the basis of the crown, as illustrated in Fig 8:

Although these measurements may be useful for preliminary identification, they must not be considered absolute as some taxa may present biogeographic or temporal variations in size [9].

**4.3.3. Comments on the age of the specimens.** This key is mostly intended for adult and sub-adult specimens, as juvenile or old specimens may not always key out correctly. In juvenile specimens, the size and eruption of the teeth may differ. This is especially true for larger species, which have a slower growth rate than smaller species and therefore reach maturity after a longer period. For instance, in *Rhabdomys pumilio* (mean weight = 45 g) the $M^1$ and $M^2$ starts erupting at the age of two weeks, while the $M^3$ begins to erupt by four to five weeks of age [35]. In *Hystrix africaeaustralis* (weight up to 20 kg), deciduous premolars begin erupting at about 14 days of age, the $M^1$ between 2 to 3 months, the $M^2$ between five and six months, and the $M^3$ at about 11 months; the permanent premolars are fully erupted at the age of two years [21]. The age at which species reach sexual maturity is also of concern, since most rodents are nidifugous (i.e., they leave the nest shortly after hatching or birth) and will leave the family nest and disperse once they are sexually mature. For instance, the Cairo spiny mice, *Acomys cahirinus*, become sexually mature and leaves the nest at the age of 2–3 months [36]. In *Cryptomys hottentotus*, the pups remain in the maternal burrow system for about 60 days until they are expelled from the burrow by the mother [36]. In *Otomys sloggetti*, males reach sexual maturity in 11 weeks, and females in 16 weeks, before dispersing [37] while subadult males of *Geosciurus inauris* do not disperse until eight months of age [38]. In older specimens, wear on the occlusal surface, which results in the removal of enamel and dentine, may obscure or obliterate the cusp pattern, and previously separate cusps may become fused. At a most advanced stage of wear, molars can be reduced to flattened, ovoid lobes with exposed dentine (Fig 9). At this stage, it is often more prudent to limit identification to the level of the family or subfamily.

**4.3.4. Effects of digestion on tooth morphology.** The passage through the digestive tract of predators can cause corrosive damage to the teeth, which in turn may sometimes cause difficulties in taxonomic identification. Numerous works have been dedicated to categorizing

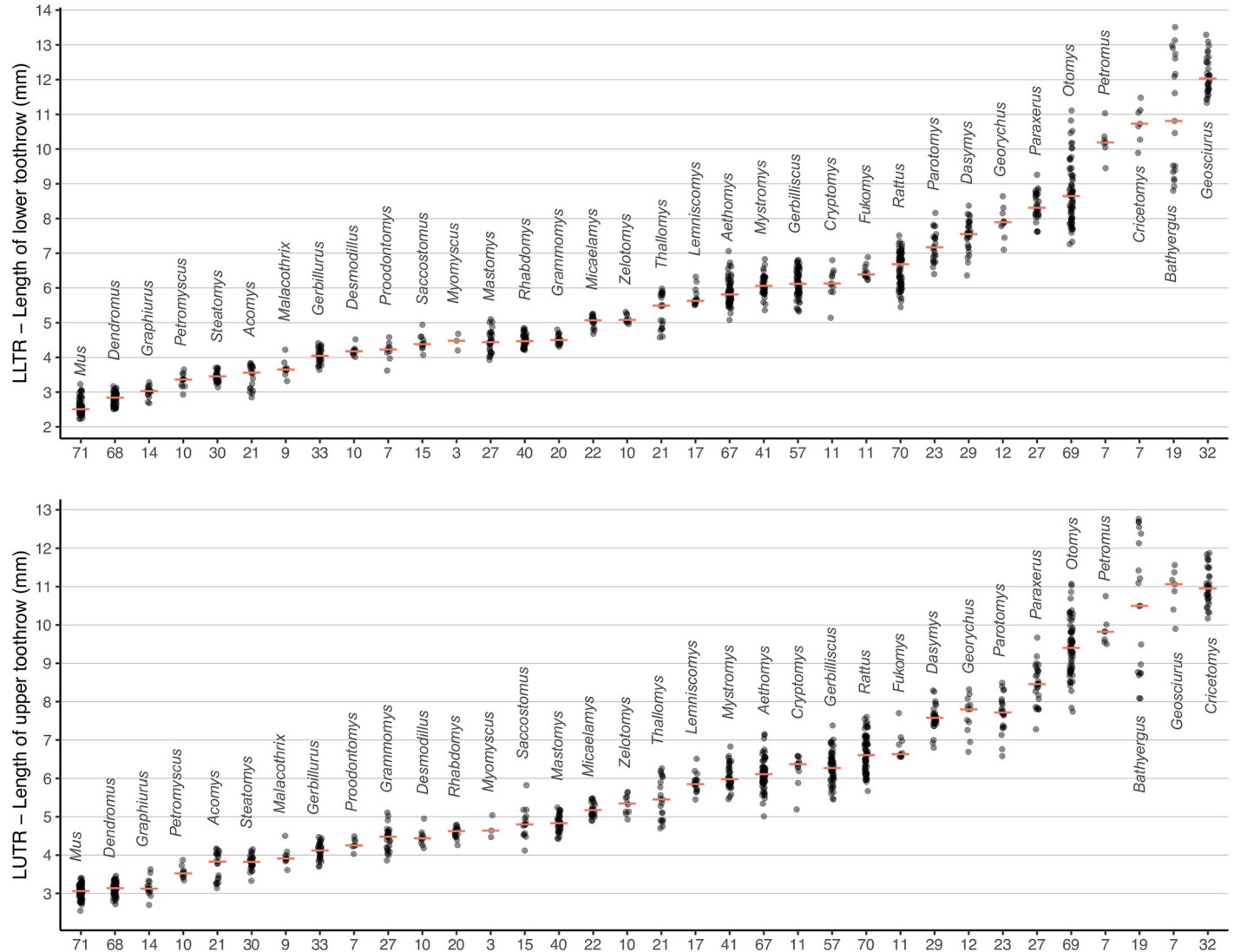

**Fig 7. Length of lower toothrow (LLTR) and upper toothrow (LUTR) for rodent genera from RSA.** Red dashes represent medians. Numbers under the x-axis represent the number of specimens used for the measurements. The following genera were not included because both LUTR and LLTR are much greater: *Pedetes*, *Thryonomys*, and *Hystrix* (see measurement details in genus accounts).

predators into distinct categories based on their digestion patterns [16, 18, 39, 40]. Predators that cause heavy or extreme digestion modifications, such as diurnal raptors or small carnivores, can alter the enamel outline of molars, resulting in some cusps or other features of diagnostic importance almost eaten away or completely missing (Fig 10).

## 5. Key to the families and genera of rodents in South Africa

The keys presented below for upper and lower jaws provides the main identification criteria to identify specimens to family level.

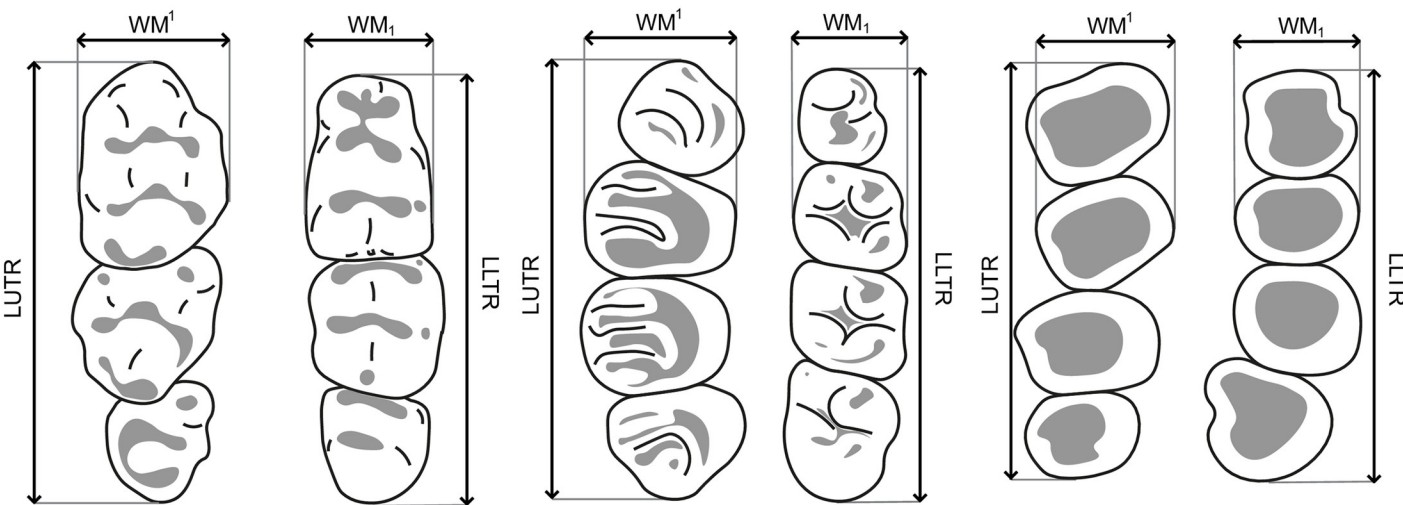

**Fig 8. Definition of the measurements used in this key for upper toothrow and lower toothrow of typical Murinae (left), Sciuridae (center) and Bathyergidae (right).** LUTR: length of the upper toothrow; WM1: width of the first upper molar; LLTR: length of the lower toothrow; WM1: width of the first lower molar.

## Muridae & Nesomyidae

Following Wilson et al. [12, 13], the family Muridae includes three subfamilies in RSA (Deomyinae, Gerbillinae, Murinae) and the family Nesomyidae includes four subfamilies (Cricetomyinae, Dendromurinae, Mystromyinae, Petromyscinae). Each subfamily of the Nesomyidae

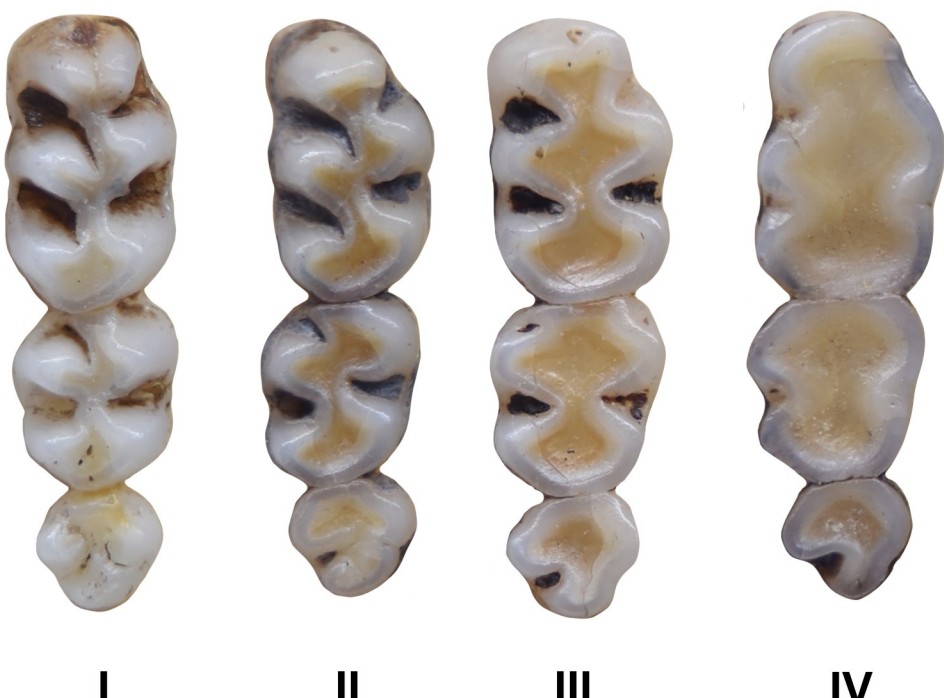

**Fig 9. Various dental wear stages of right upper toothrows in the same species *Mystromys albicaudatus*, ranging from no wear (stage 1, juvenile specimen) to advanced wear with dentine exposed (stage 4, old individual).**

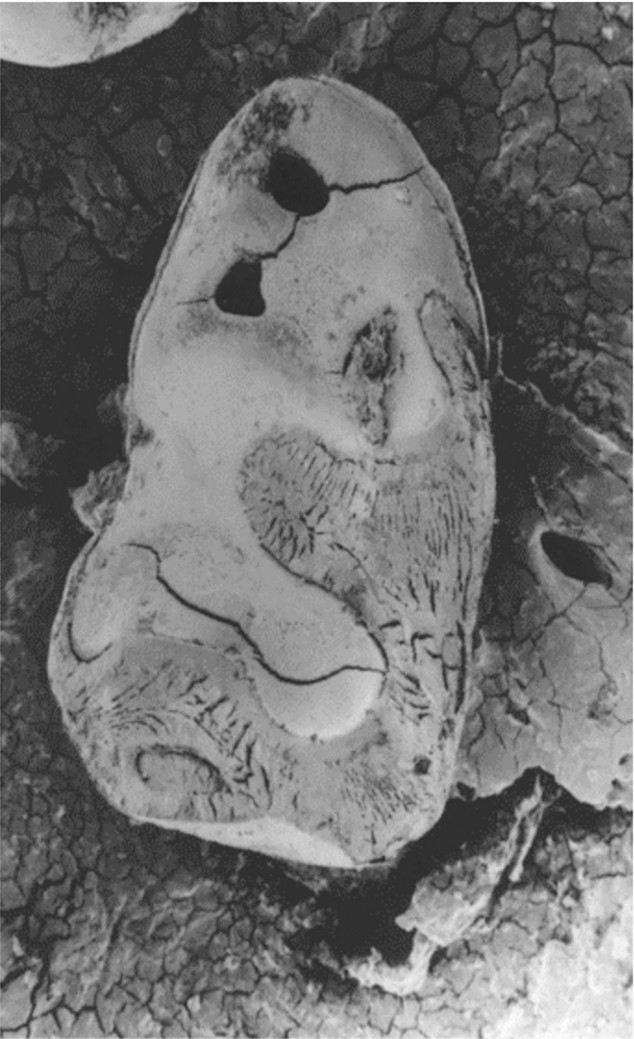

**Fig 10. Scanning electron micrograph of a murid first lower molar showing heavy digestion.** Enamel is locally removed, dentine is also affected.

is morphologically well characterized, but there are no known morphological dental features that distinguish the family itself from the Muridae. We have therefore grouped the two families together. We use cusp nomenclature from Misonne [26] and Denys et al. [27] for describing the cusps of the lower and upper molars of Murinae and Dendromurinae:

In Muridae and Nesomyidae, the upper molars are usually the most diagnostic teeth. Denys & Michaux [41] grouped some African genera by the structure of the third upper molar; these structures are presented in Fig 11 for genera from South Africa. Unfortunately, skulls are more prone to fragmentation than mandibles, which explains why they are generally proportionately less well-represented in archaeological and paleontological assemblages. In the absence of teeth, the number and structure of alveoli, as well as the length of the anterior palatal foramina, provide useful criteria for identifying Muridae and Nesomyidae taxa (Fig 12).

Family **MURIDAE** Illiger, 1811

Subfamily **DEOMYINAE** Thomas, 1888

Genus *Acomys* I. Geoffroy, 1838 (Spiny Mice)

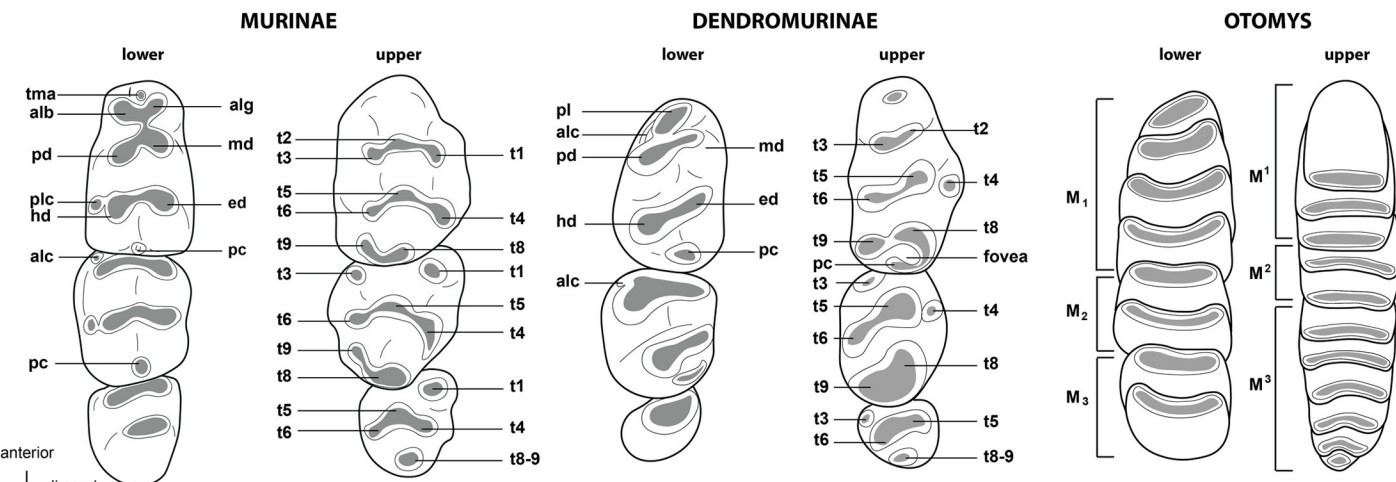

**Fig 11. Typical Murinae, Dendromurinae, and Otomys left lower toothrows and right upper toothrows, with nomenclature of the cusps.** Lower toothrow: alb: labial anteroconid; alc: anterolabial cingulum; alg: lingual anteroconid; hd: hypoconid; md: metaconid; pd: protoconid; pc: posterior cingulum; pl: prelobe; plc: posterolabial cusplet; tma: anteromedian cusp. Upper toothrow: t1: anterostyle; t2: lingual anterocone; t3: labial anterocone; t4: anterostyle; t5: protocone; t6: paracone; t8: pseudohypocone; t9: metacone (modified from Denys et al., 1992).

Figs 13–15; Table 6

Dental formula is 1-0-0-3:1-0-0-3. Alveolar formula is 3-3-3/2-2-2 (Figs 16 and 17).

**Upper jaw.** Upper incisors are opisthodont and ungrooved. The palatal foramina taper to the t4 of the $M^1$. Molars are small and show a superficially *Mus*-like cusp configuration, with the t1 displaced backwards (sometimes almost in line with t5 and t6) and the t4 subsequently low. $M^3$ is rather small and has a t3 but no t1 (group 2 in Fig 12), in contrast to Dendromurinae and most Murinae, including *Mus*.

**Lower jaw.** The $M_1$ is the longest in the lower toothrow. The lingual anteroconid is approximately the same size as or slightly larger than the labial anteroconid. A posterolabial cusplet is sometimes present on the tooth. In South African species, the posterior cingulum is absent or very small in both the $M_1$ and the $M_2$. The molars $M_2$ and $M_3$ have two rows of cusps. The $M_2$ displays a short anterolophid on its anterolabial zone, while the $M_3$ displays a small anterolabial cusplet (more developed in *A. selousi* than in *A. subspinosus*) which tends to obliterate with wear (this cusplet is absent in *Mus*). The sciurognath mandible has a poorly

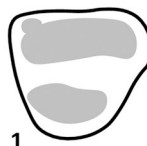 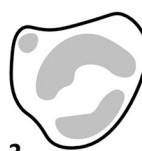 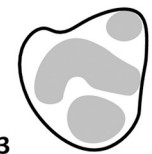 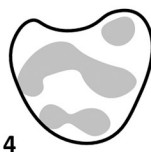 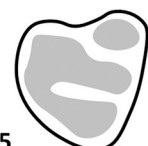 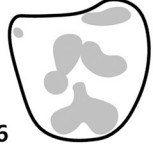 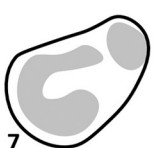

**Fig 12. Shape and configuration of the right upper $M^3$ of the Muridae and Nesomyidae (after Denys & Michaux [41], reproduced in Monadjem et al. [22]).** 1) t3 present, t1 absent, t3 connected to the first row of cusps: *Cricetomys, Dendromus, Malacothrix, Petromyscus, Saccostomus, Steatomys = Nesomyidae*; 2) t3 present, t1 absent, t3 isolated: *Acomys = Deomyinae*; 3) t1 present, a distal cusp, t3 absent: some *Aethomys, Mastomys*, some *Micaelamys, Myomyscus, Rattus, Rhabdomys*; 4) t1 present, t3 absent, second lobe with 2 fused or distinct distal cusps: some *Aethomys, Dasymys, Lemniscomys*, some *Micaelamys*; 5) rather large t1, labial link between first lobe and second lobe: *Zelotomys*; 6) presence of t1 and tiny t3, trace of median longitudinal link: *Grammomys, Thallomys*; 7) small molars with t1 and link between first and second lobe, cusps poorly differentiated: *Mus*.

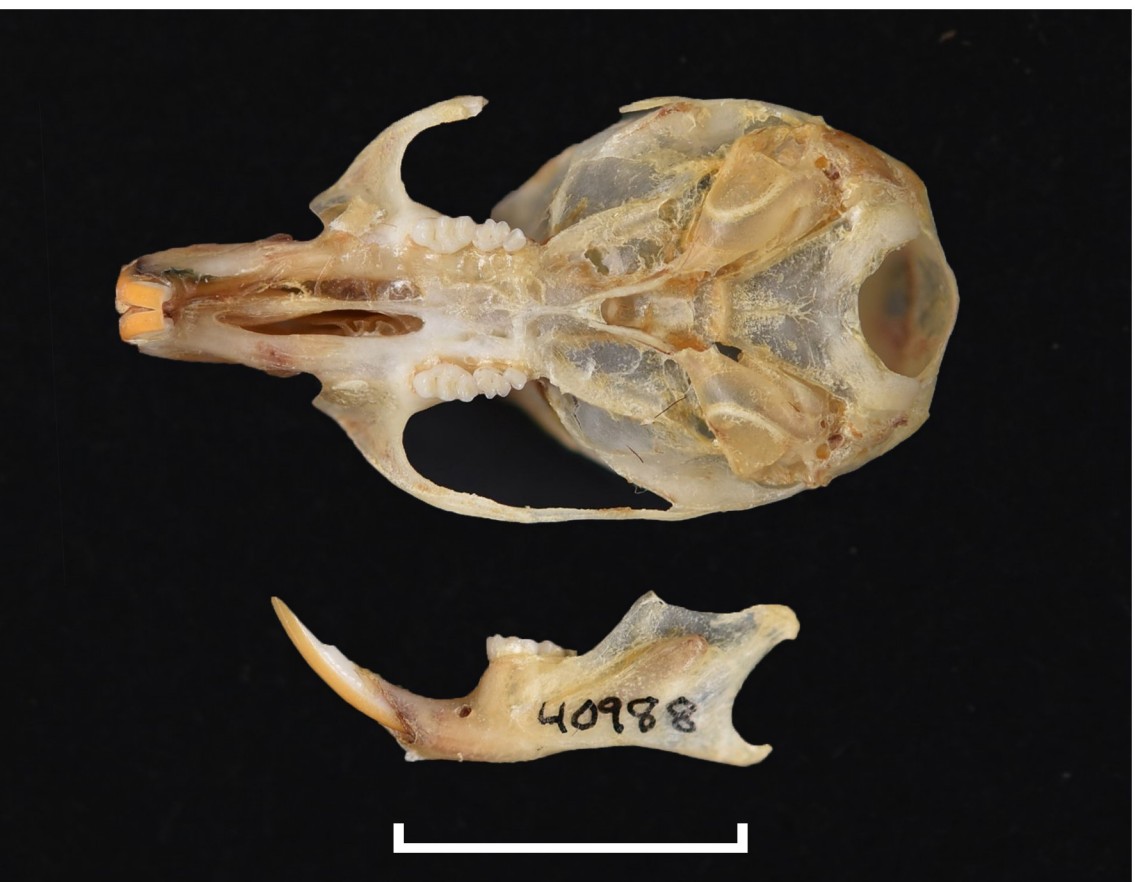

**Fig 13. Cranium of *Acomys subspinosus* (DNMN-40988), with a scale bar of 1 cm.**

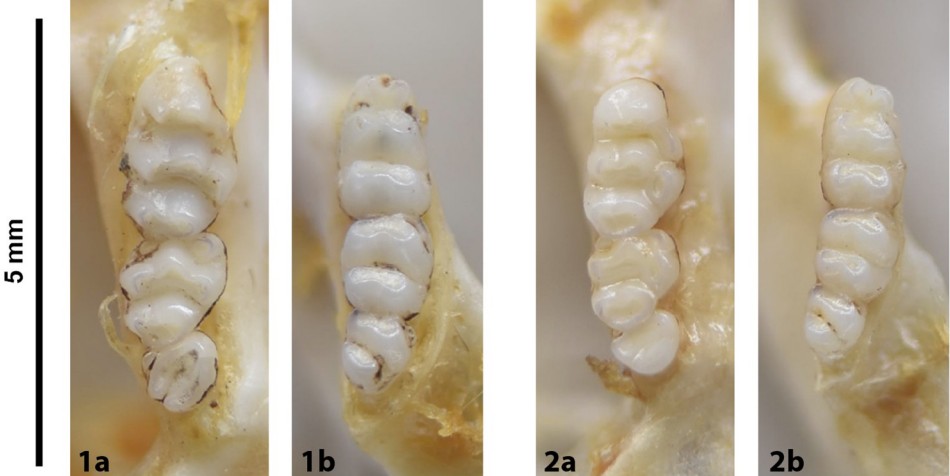

**Fig 14. Cheekteeth of *Acomys*. 1)** Upper (a) and lower (b) right toothrow of *A. selousi* (DNMNH-2833); **2)** Upper (a) and lower (b) right toothrow of *A. subspinosus* (DNMNH-40989).

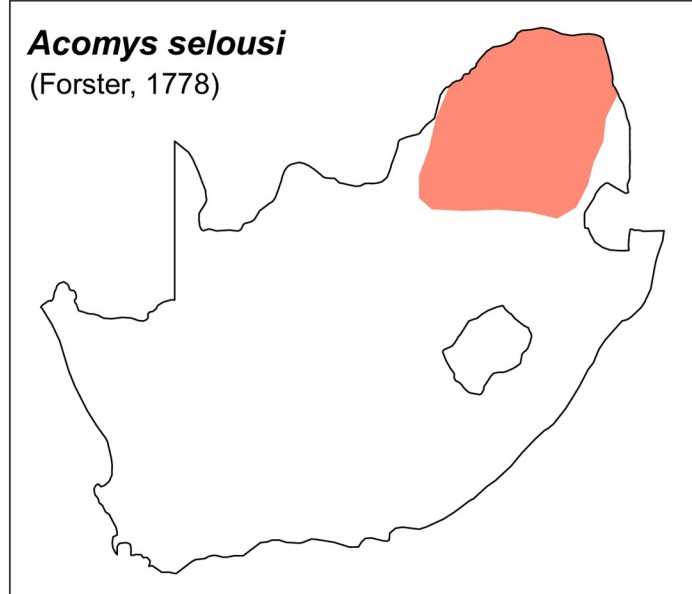
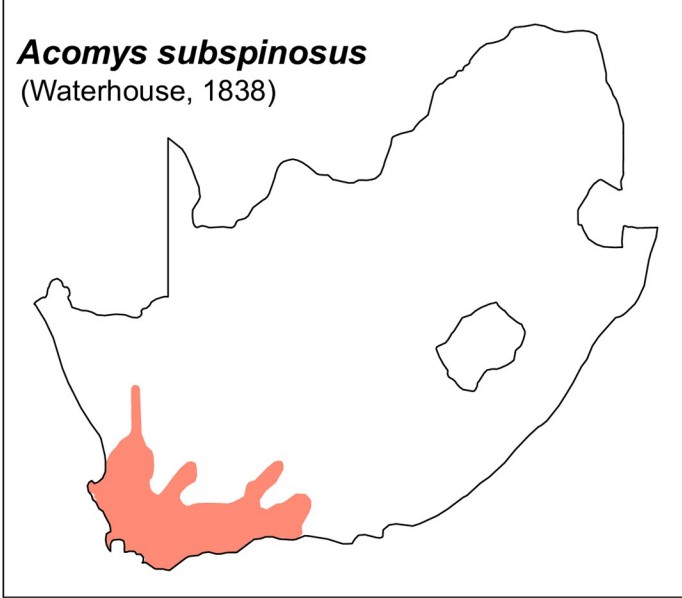

**Fig 15. Distribution maps.**

developed coronoid process. The mandible is small, being of same average size than in *Steatomys*.

**Systematic notes and South African fossil record.** Two species are currently recognized in South Africa:

- *A. selousi* (FORSTER, 1778)

- *A. subspinosus* (WATERHOUSE, 1838)

The species *A. selousi* has been described as a split from as *A. spinosissimus* PETERS, 1852 [22, 42], the latter occurring further north in South-Central Africa. Further taxonomic and biogeographic investigation of these species is required [43]. Two additional fossil species have been described:

- †*Acomys mabele* DENYS, 1990 known only from the Pliocene site of Langebaanweg, and that constitutes the first occurrence of the genus in the South African fossil record

- *Acomys spinosissimus* Peters, 1852 described in various Pleistocene sites

Fossils of this genus have been recorded from many Quaternary fossil deposits. Based on the recent split between *A. selousi* and *A. spinosissimus*, which are morphologically undistinguishable, South African fossil specimens previously attributed to *A. spinosissimus* may be more parsimoniously assigned to *A. selousi*.

**Table 6. Dental measurements (in mm) for *Acomys* from South Africa, sexes and species combined.**

|  | Mean | Min | Max | n |
|---|---|---|---|---|
| LLTR | 3.5 | 2.9 | 3.8 | 20 |
| $WM_1$ | 1.0 | 0.8 | 1.1 | 20 |
| LUTR | 3.8 | 3.1 | 4.2 | 21 |
| $WM^1$ | 1.2 | 1.0 | 1.4 | 21 |

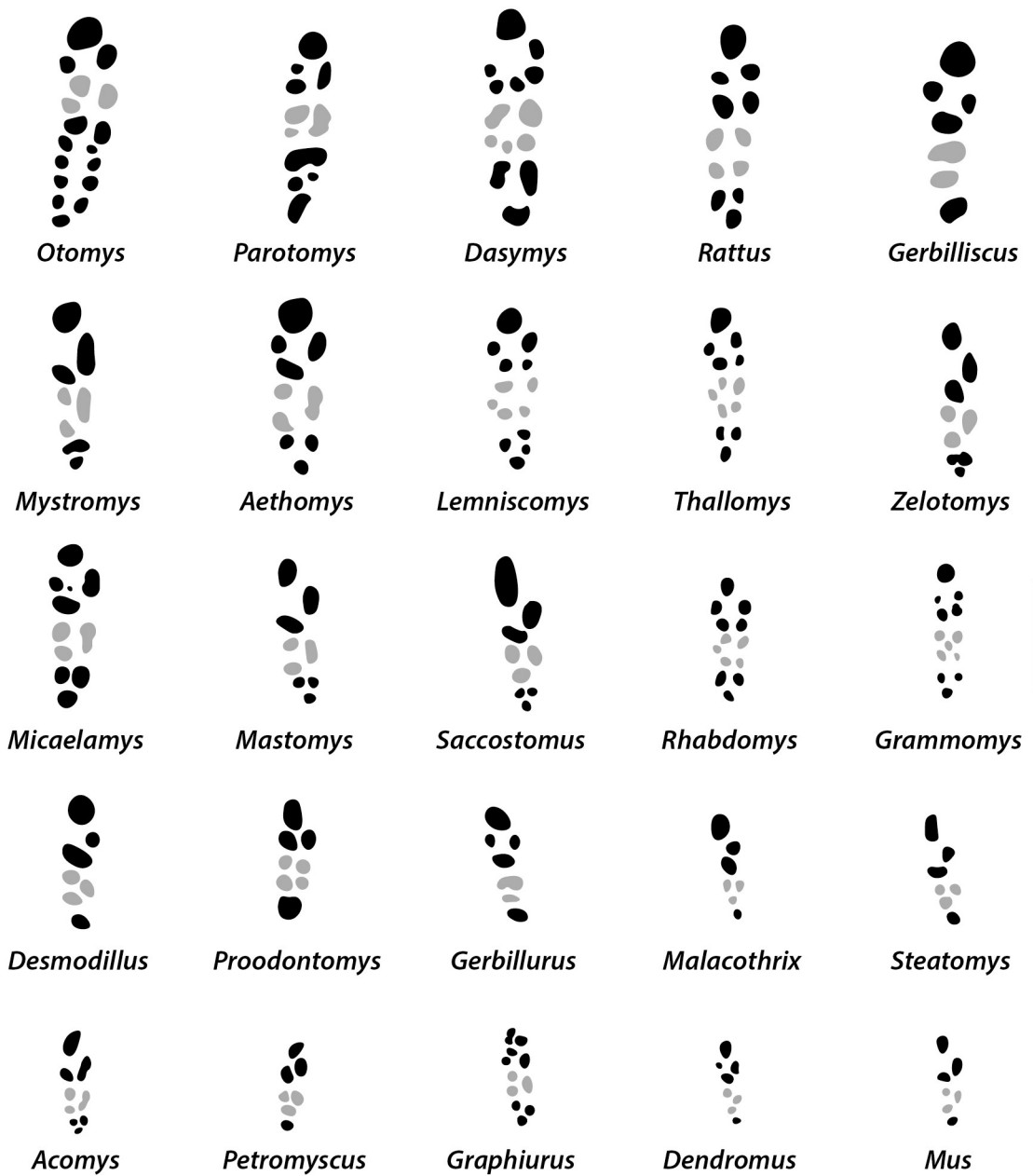

**Fig 16. Right maxilla alveolar patterns of modern Muridae and Nesomyidae (with the exception of the larger *Cricetomys*) and *Graphiurus* from South Africa, with a scale bar of 4 mm.** Alveoli of the molars M² are indicated in grey. Adapted and modified from [6].

Subfamily **GERBILLINAE** Gray, 1825
Genus ***Desmodillus*** Thomas & Schwann, 1904 (Cape Short-tailed Gerbils)
Figs 18–20; Table 7
Dental formula is 1-0-0-3:1-0-0-3. Alveolar formula is 3-3-1:2-2-1 (Figs 16 and 17).
**Upper jaw.** Upper incisors are long with a shallow groove. The anterior palatal foramina end short of the $M^1$, the posterior palatal foramina extend between the second lamina of $M^1$ to the beginning of $M^3$. Molars show a semi-lophodont (or buno-lophodont) condition. $M^1$ has

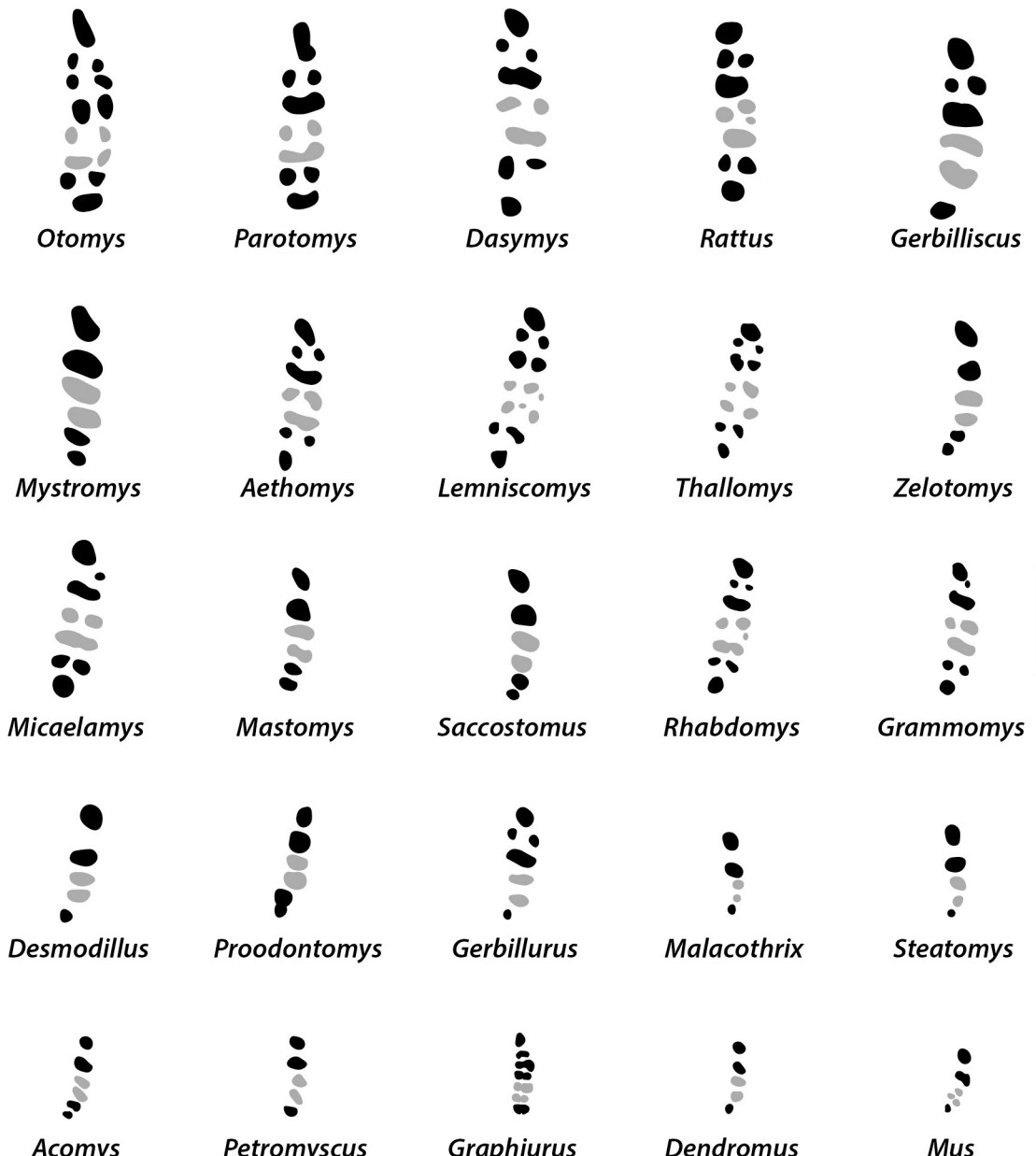

**Fig 17. Right mandible alveolar patterns of modern Muridae and Nesomyidae (with the exception of the larger *Cricetomys*) and *Graphiurus* from South Africa, with a scale bar of 4 mm.** Alveoli of the molars $M_2$ are indicated in grey. Adapted and modified from [6].

three lobes, $M^2$ has two, and $M^3$ has one lobe only (species of *Gerbilliscus (Gerbillurus)* have one or two lobes). The central lobe in $M^1$ has two circular cusps. Tympanic bullae are greatly inflated; they can even be seen when the skull is viewed from above. Fossil material that preserves only the toothrow is not always easily distinguished from *Gerbillurus*, although the length of the toothrow is greater in *Desmodillus*.

**Lower jaw.** Lower incisors are plain. Young specimens have two cusps in the first lobe of the $M_1$ but they fuse with the age to form the typical horseshoe shape of Taterillini; the two

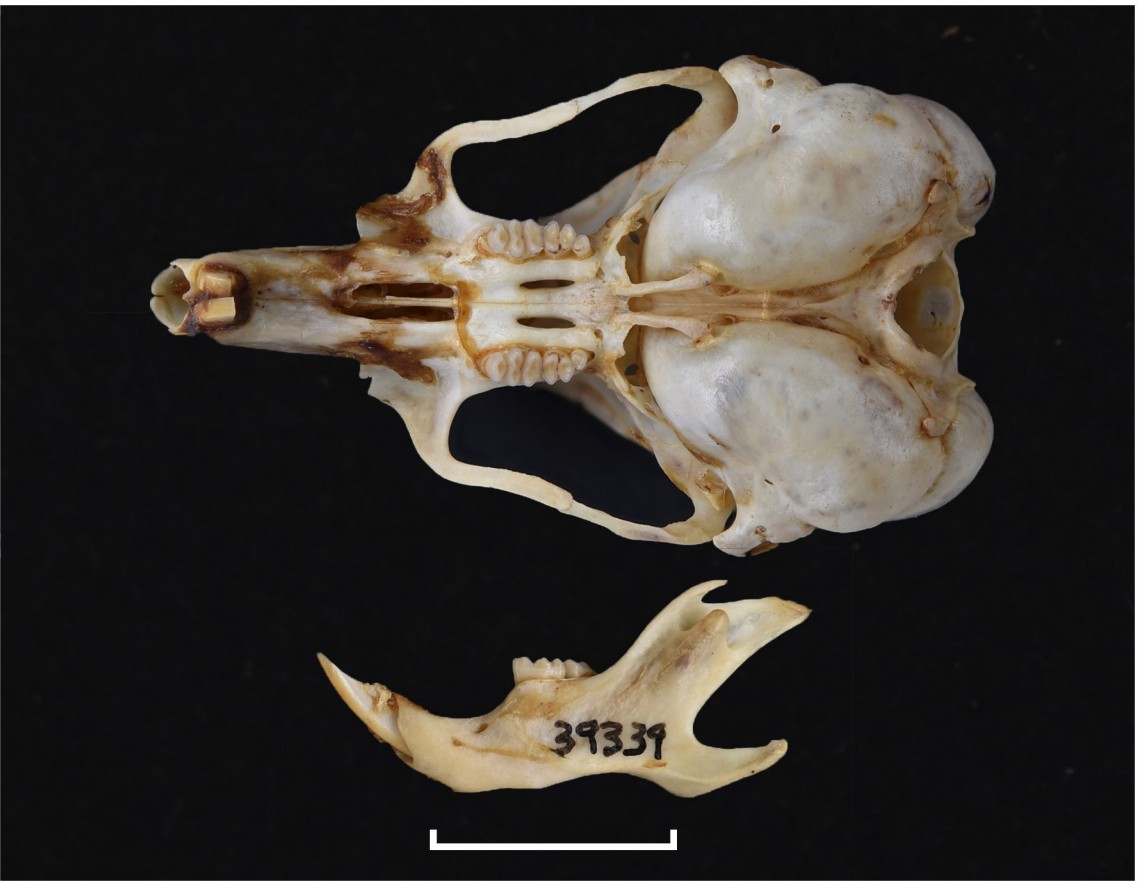

**Fig 18. Cranium of *Desmodillus auricularis* (DNMN-39339), with a scale bar of 1 cm.**

cusps of the second lobe of $M_1$ remain unfused, while they are fused in a lamina in the third lobe. $M_2$ has two lobes. $M_3$ is very small and consists of one tiny cusp. The angular process of the mandible is sharp and elongated, and its ventral edge makes an angle with the ventral margin of the mandibular body. The coronoid process is higher than in *Gerbillurus*.

**Systematic notes and South African fossil record.** The genus is monotypic:

- *Desmodillus auricularis* (Smith, 1834)

An additional fossil species has been described:

- *†Desmodillus magnus* Denys and Matthews, 2017 from Langebaanweg

Since the Early Pliocene, until the present, fossils of *Desmodillus* have been recorded from numerous fossil deposits.

Genus **Gerbilliscus** Thomas, 1897 (Gerbils & Hairy-footed Gerbils)
Subgenus **Gerbilliscus (Gerbilliscus)**
Figs 21–23, Table 8
Dental formula is 1-0-0-3:1-0-0-3. Alveolar formula is 4-2-1:4-2-1 (Figs 16 and 17).

**Upper jaw.** Upper incisors are slightly opisthodont, yellow to orange in colour, and have a single groove. The anterior palatal foramina barely reach the alveolus of $M^1$; the posterior palatal foramina are not as developed as in *Desmodillus* and *Gerbilliscus*. Molars show a lophodont

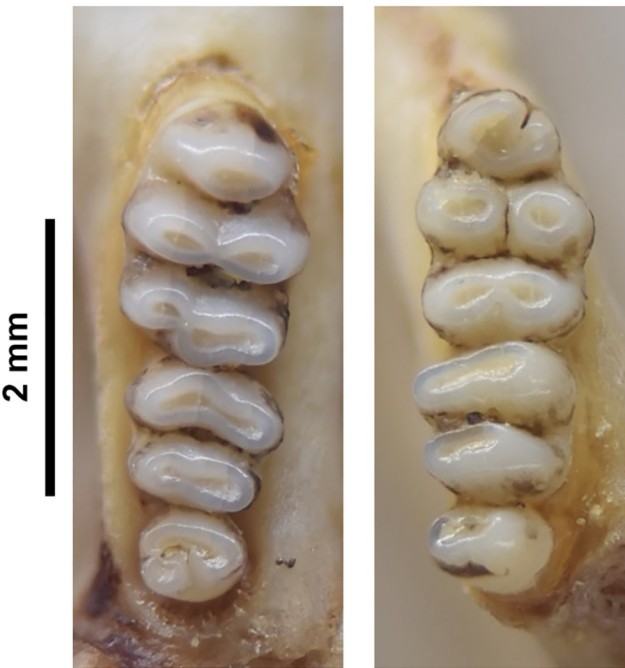

**Fig 19. Cheekteeth of *Desmodillus*.** Upper (a) and lower (b) right toothrow of *Desmodillus auricularis* (DNMNH-39365).

condition, with cusps fused in transverse laminae lacking longitudinal connections. The molar $M^1$ has three laminae, the first consisting of a single cusp; molars $M^2$ and $M^3$ both have two laminae when unworn.

**Lower jaw.**    Molars show lophodont condition, with cusps fused in transverse laminae that lack longitudinal connections. The anteroconid of $M_1$ has a horseshoe shape whose appearance is highly variable. The $M_2$ has two laminae, and the $M_3$ has one. The mandible is elongated and seems vertically compressed, with coronoid and condylar processes projecting backwards.

**Systematic notes and South African fossil record.**    Species of the genus *Gerbilliscus* were previously included in *Gerbillus*, then in *Tatera*, but are now grouped in their own distinct genus [44, 45]. The genera *Gerbilliscus* and *Gerbillurus* have long been treated as two distinct genera, but recent molecular and chromosomal analysis suggest that they should be combined into one genus, *Gerbilliscus* [46]. In South Africa, three species of *Gerbilliscus* (excluding *Gerbillurus*) are currently recognised:

- *Gerbilliscus afra* (GRAY, 1830),

- *Gerbilliscus brantsii* (SMITH, 1836)

- *Gerbilliscus leucogaster* (PETERS, 1852).

The first occurrences of this genus are from Makapansgat [47, 48] and Taung [1, 47, 49] in the Pliocene. Remains of *Gerbilliscus* are found in many deposits throughout the Quaternary [50].

Subgenus ***Gerbilliscus*** (***Gerbillurus***)
Figs 24–26; Table 9

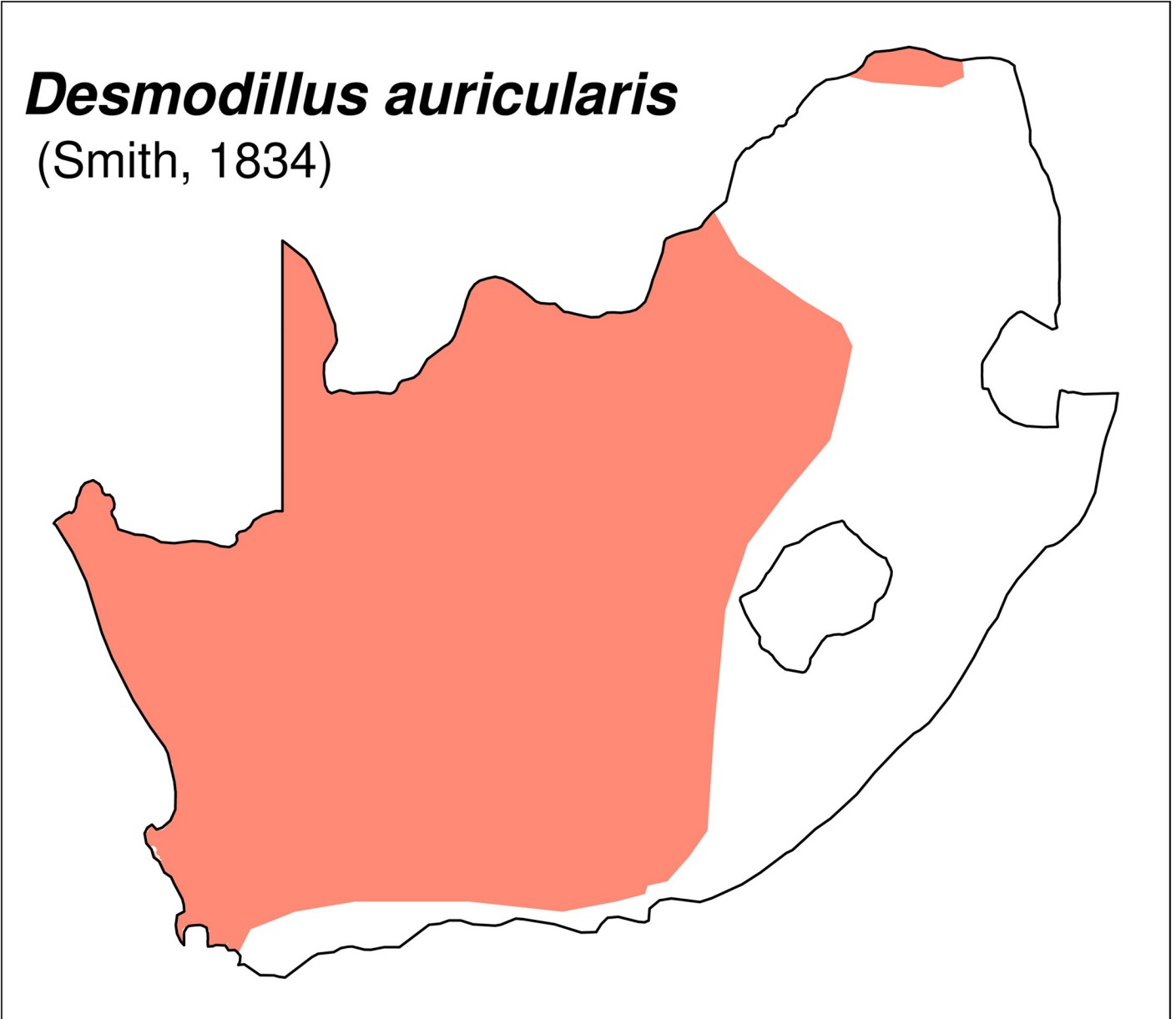

**Fig 20. Distribution map.**

Dental formula is 1-0-0-3:1-0-0-3. Alveolar formula is 4-2-1:4-2-1 (Figs 16 and 17).

**Upper jaw.** Upper incisors are strongly opisthodont, yellow to orange, and have a single groove. The anterior palatal foramina end short of the $M^1$, the posterior palatal foramina extend between the second lamina of $M^1$ to the beginning of $M^3$. Molars show semi-lophodont condition, with cusps fused in transverse laminae that lack longitudinal connections. $M^1$ has three lobes, $M^2$ has two lobes, and $M^3$ has two lobes, poorly, or completely, fused (in *Desmodillus* the second lobe is always much reduced). Fossil material that preserves only the teeth is not always easily distinguishable from *Desmodillus*, although the length of the toothrow is smaller in *Gerbilliscus*.

**Table 7. Dental measurements (in mm) for *Desmodillus auricularis*, sexes combined.**

|        | Mean | Min | Max | n  |
|--------|------|-----|-----|----|
| LLTR   | 4.2  | 4.0 | 4.5 | 10 |
| $WM_1$ | 1.4  | 1.4 | 1.5 | 10 |
| LUTR   | 4.5  | 4.2 | 5.0 | 10 |
| $WM^1$ | 1.6  | 1.4 | 2.0 | 10 |

**Lower jaw.** As opposed to *Gerbilliscus (Gerbilliscus)*, cusps are high and round and do not fuse together in transverse laminae as in other *Gerbilliscus* spp. Lower molars are very similar to those of *Desmodillus*, but the $M_1$ has four alveoli and the incisor is thinner. Mandible is small, vertically compressed and not as high as in *Desmodillus*.

**Systematic notes and South African fossil record.** Until recently, specimens of *Gerbillurus* were placed in their own genus based on the marked morphological difference with other *Gerbilliscus*, but recent chromosomal and molecular analyses indicate they should be assigned to *Gerbilliscus*. In South Africa, two species of *Gerbilliscus (Gerbillurus)* are currently recognized:

- *Gerbilliscus paeba* (A. Smith, 1836)

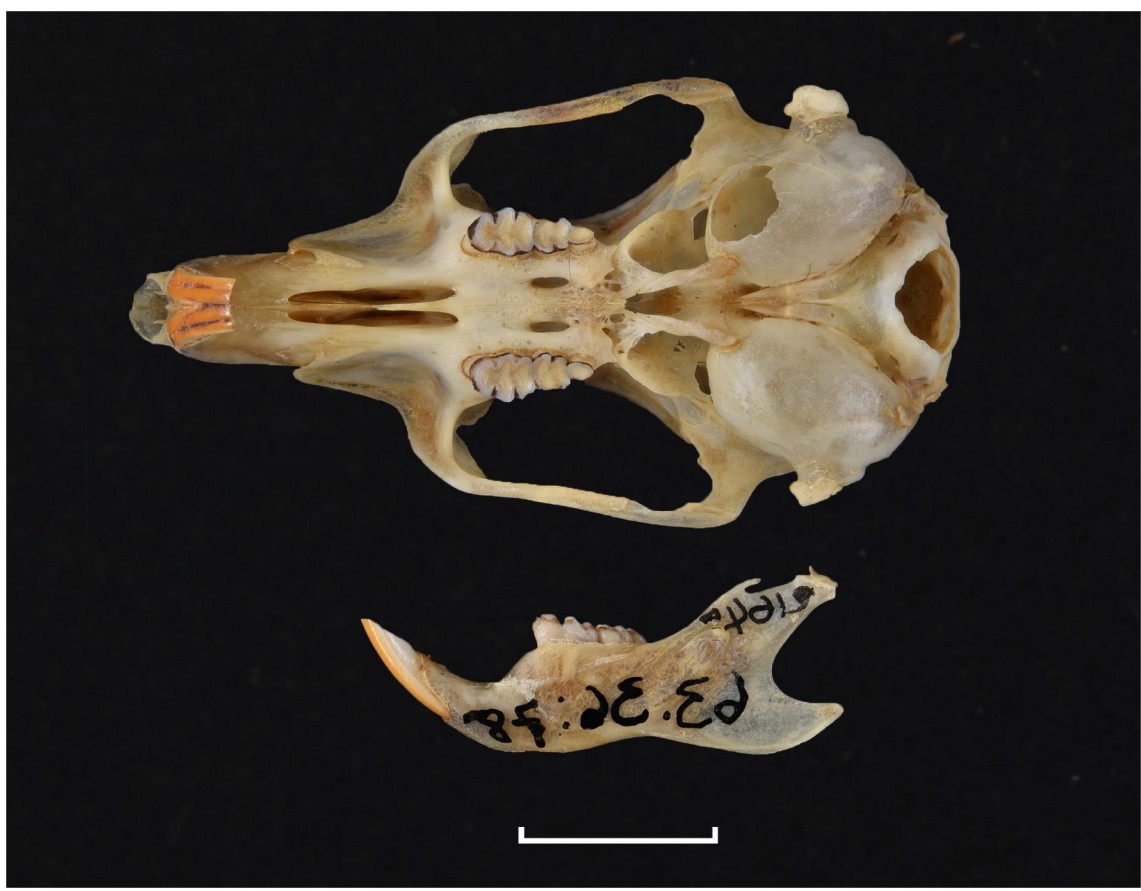

**Fig 21. Cranium of *Gerbilliscus afra* (DNMN-21640), with a scale bar of 1 cm.**

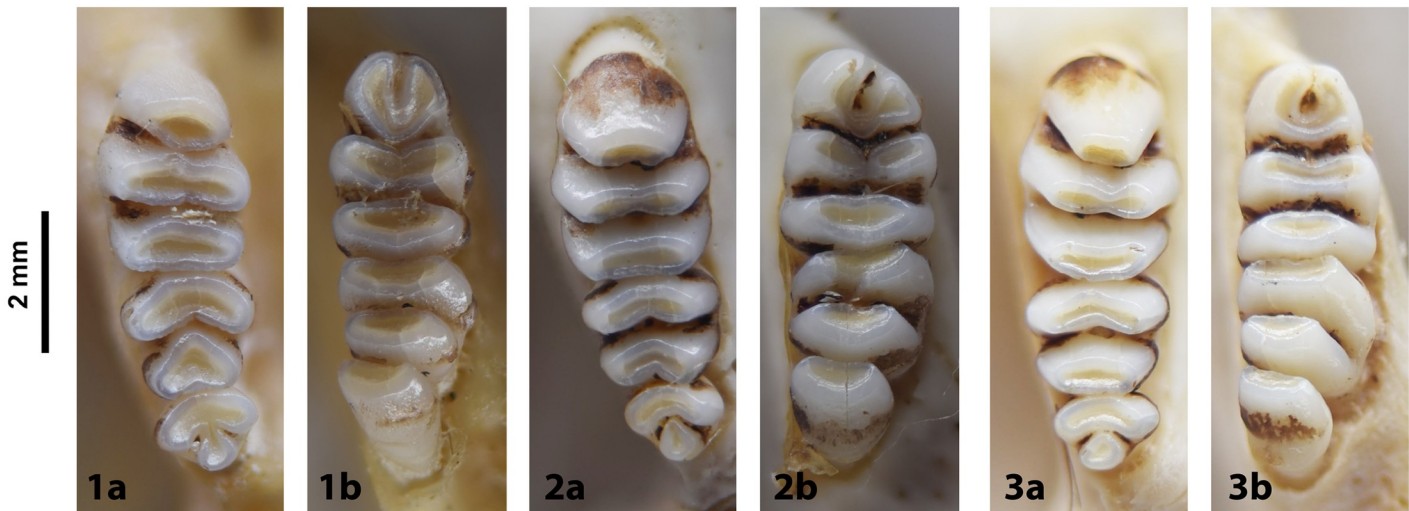

**Fig 22. Cheekteeth of *Gerbilliscus (Gerbilliscus)*. 1)** Upper (a) and lower (b) right toothrow of *G. afra* (DNMNH-21634); **2)** Upper (a) and lower (b) right toothrow of *G. brantsii* (DNMNH-27755); **3)** Upper (a) and lower (b) right toothrow of *G. leucogaster* (DNMNH-44280).

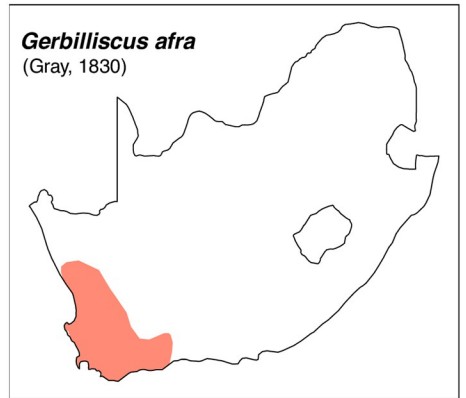
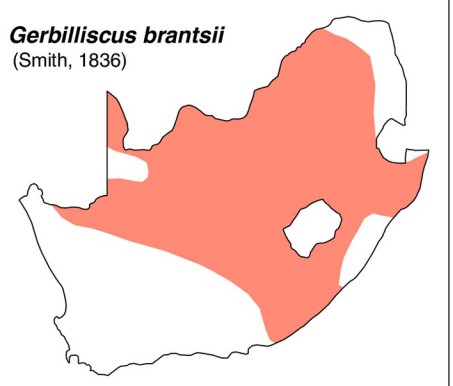
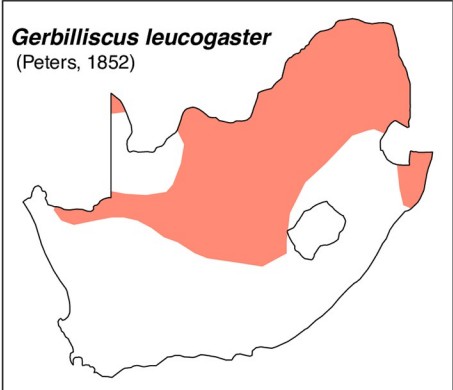

**Fig 23. Distribution maps.**

**Table 8. Dental measurements (in mm) for *Gerbilliscus* (*Gerbilliscus*) from South Africa, sexes and species combined.**

|  | Mean | Min | Max | n |
|---|---|---|---|---|
| LLTR | 6.1 | 5.3 | 6.8 | 59 |
| $WM_1$ | 2.2 | 1.8 | 2.5 | 59 |
| LUTR | 6.2 | 5.5 | 7.4 | 57 |
| $WM^1$ | 2.3 | 2.0 | 2.6 | 57 |

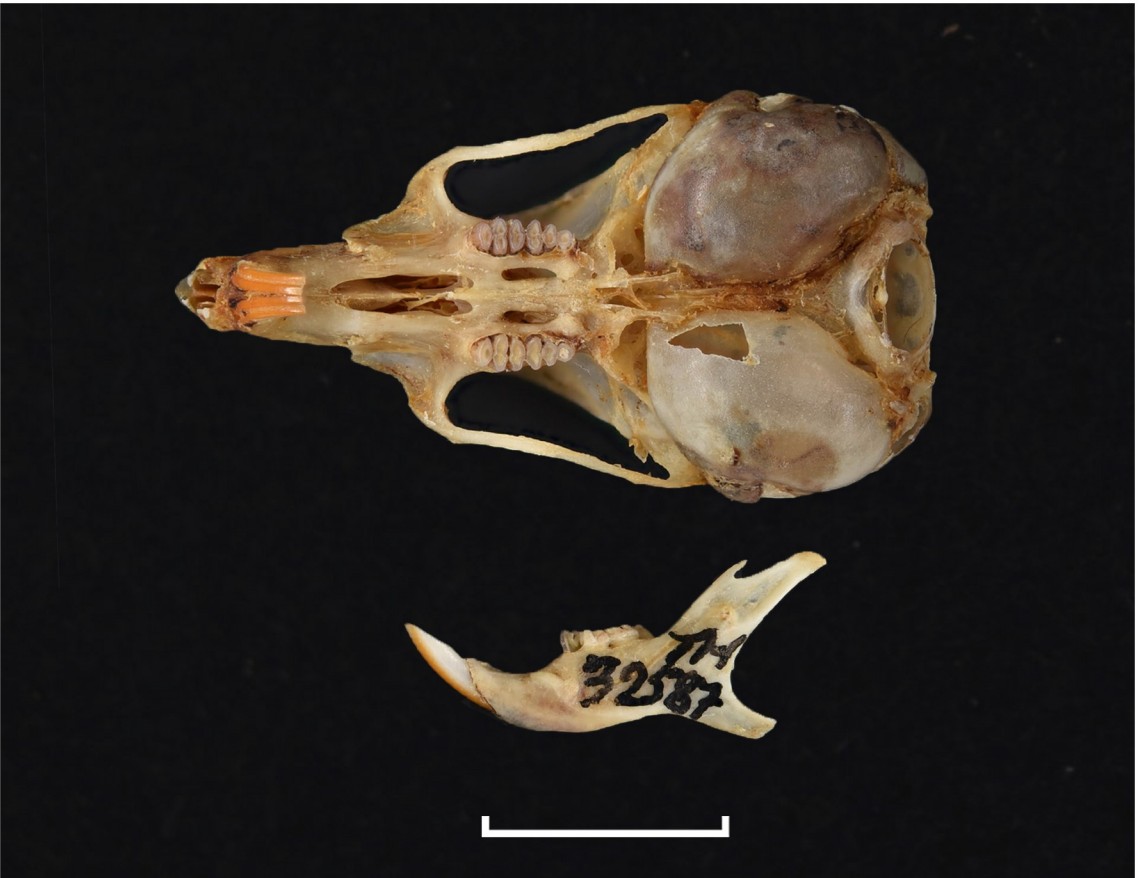

**Fig 24. Cranium of *Gerbilliscus (Gerbillurus)* vallinus (DNMN-21640), with a scale bar of 1 cm.**

- *Gerbilliscus vallinus* (Thomas, 1918)

The oldest remains of *Gerbillurus* in the South African fossil record are found in the Early Pleistocene site of Wonderwerk [6, 51].

Subfamily **MURINAE** Illiger, 1811

Genus ***Aethomys*** Thomas, 1915 (Veld Rats)

Figs 27–29; Table 10

Dental formula is 1-0-0-3:1-0-0-3. Alveolar formula is 4-3-3:4-3-3 (Figs 16 and 17).

**Upper jaw.** Incisors are ungrooved and opisthodont. The anterior palatal foramina are long and extend beyond the level of the $M^1$. Molars are relatively broad. In $M^1$, the t1 is situated slightly or well behind t2 and t3; there is often a small stephanodont crest uniting t6 and t9 on $M^1$ and on $M^2$; t7 is absent. The configuration of $M^3$ corresponds to group 3 or 4 in Fig 12, with the distal lobe having often one elongated cusp (or two poorly differentiated, as opposed to two well differentiated in *Lemniscomys*) although this criterion shows variability in few specimens.

**Lower jaw.** The $M_1$ sometimes display lateral cusplets or small ridges that can form a stephanodont crest; it has no conspicuous posterior cingulum (a condition similar in *Micaelamys* and *Lemniscomys*), although this feature is not constant. In the $M_1$ there is no tma (it is present in *Micaelamys*) but some specimens may display a tiny anteromedian cusplet. The $M_2$ typically has three alveoli (two isolated and two fused ones).

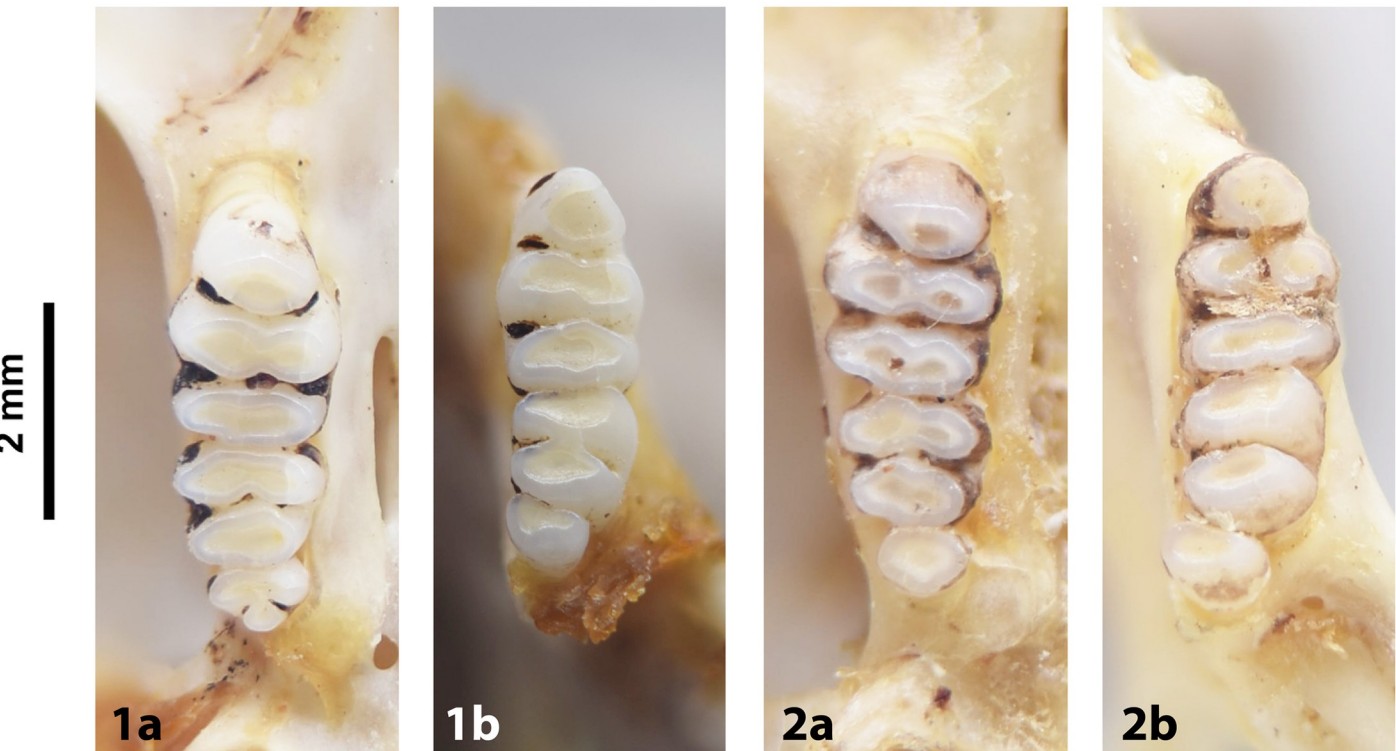

**Fig 25. Cheekteeth of *Gerbilliscus* (*Gerbillurus*). 1)** Upper (a) and lower (b) right toothrow of *G. paeba* (DNMNH-32636); **2)** Upper (a) and lower (b) right toothrow of *G. vallinus* (DNMNH-32580).

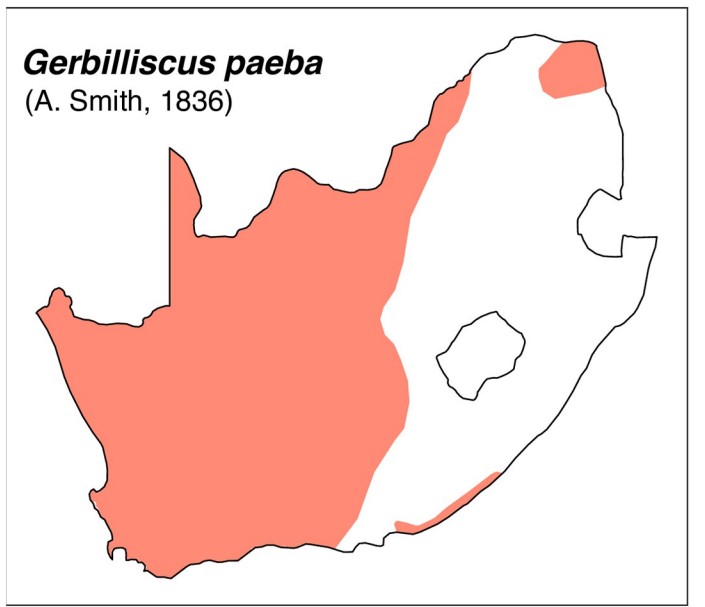
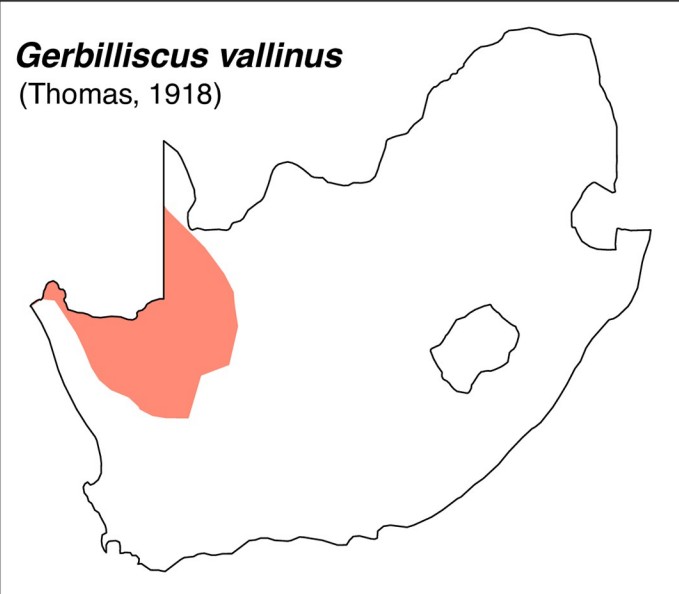

**Fig 26. Distribution maps.**

**Table 9. Dental measurements (in mm) for *Gerbilliscus* (*Gerbillurus*) from South Africa, sexes and species combined.**

|  | Mean | Min | Max | n |
|---|---|---|---|---|
| LLTR | 4.1 | 3.6 | 4.4 | 32 |
| $WM_1$ | 1.3 | 1.2 | 1.5 | 32 |
| LUTR | 4.1 | 3.7 | 4.5 | 33 |
| $WM^1$ | 1.5 | 1.3 | 1.6 | 33 |

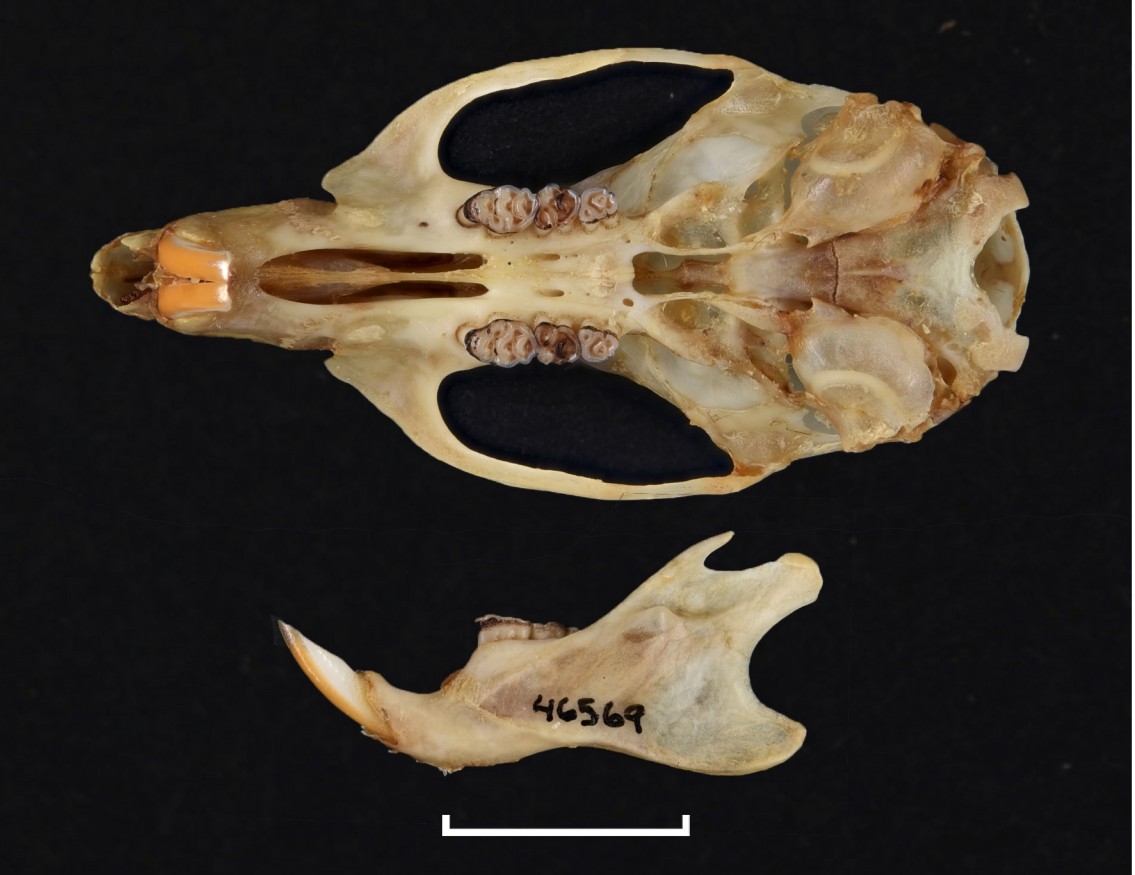

**Fig 27. Cranium of *Aethomys ineptus* (DNMNH-4659), with scale bar of 1 cm.**

**Systematic notes and South African fossil record.** Species of the genus *Micaelamys* were previously included in *Aethomys*, but they were later placed in their own genus based on molecular and morphological data [52]. Two species of *Aethomys* are currently recognized in South Africa:

- *Aethomys chrysophilus* (DE WINTON, 1897)

- *Aethomys ineptus* (THOMAS AND WROUGHTON, 1908)

Additional fossil species have been identified:

- †*Aethomys adamanticola* DENYS, 1990 from the Early Pliocene locality of Langebaanweg

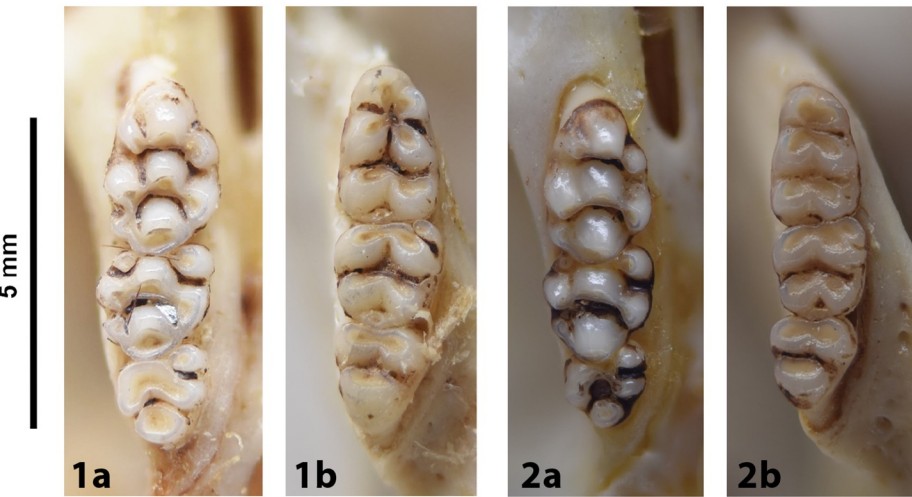

**Fig 28. Cheekteeth of *Aethomys*. 1)** Upper (a) and lower (b) right toothrow of *A. chrysophilus* (DNMNH-4659); **2)** Upper (a) and lower (b) right toothrow of *A. ineptus* (DNMNH-46903).

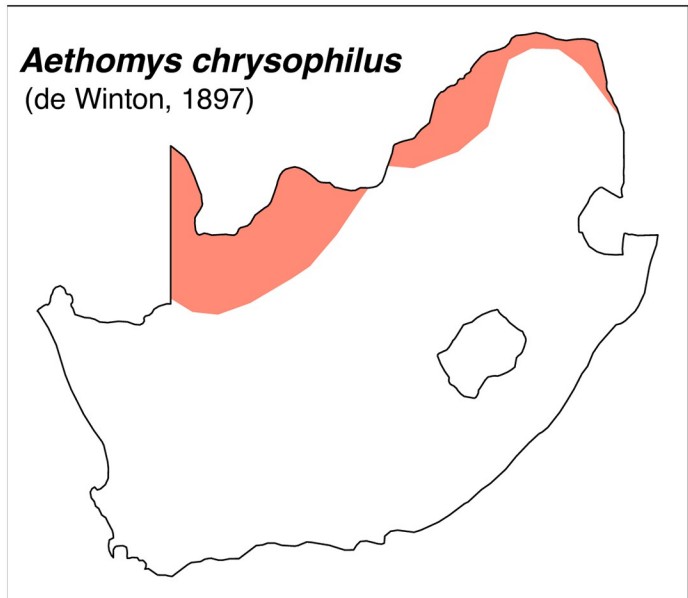

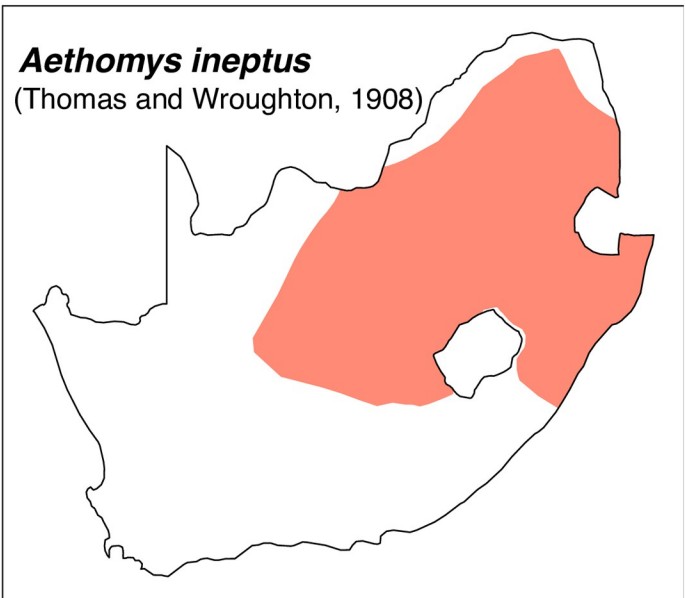

**Fig 29. Distribution maps.**

**Table 10. Dental measurements (in mm) for *Aethomys* from South Africa, sexes and species combined.**

|          | Mean | Min | Max | n  |
|----------|------|-----|-----|----|
| LLTR     | 5.1  | 5.1 | 7.1 | 65 |
| WM$_1$   | 1.7  | 1.5 | 2.3 | 65 |
| LUTR     | 6.0  | 5.0 | 7.2 | 67 |
| WM$^1$   | 2.0  | 1.7 | 2.4 | 67 |

- †*Aethomys modernis* Denys, 1990 from the Early Pliocene locality of Langebaanweg

Fossils of *Aethomys* are known from many Pleistocene and Holocene sites from South Africa [50]. Many of these specimens were identified as *Aethomys chrysophilus* at a time when *A. ineptus* was not recognized yet as a cryptic species distinct from *A. chrysophilus*. As these two species appear to be morphologically indistinguishable using the cranio-dental anatomy, caution is recommended when making an osteological identification

Genus ***Dasymys*** Peters, 1875 (Shaggy Rats)

Figs 30–32; Table 11

Dental formula is 1-0-0-3:1-0-0-3. Alveolar formula is 6-5-3:4/5-3/4-3 (Figs 16 and 17).

**Upper jaw.** Upper incisors are broad and ungrooved. The palatal foramina end just before the alveolus of the first root of the $M^1$. Molars are large and heavily cusped, exhibiting macrodonty. Rows of cusps are arranged in transverse rows, with t1 in line with t2 and t3 in $M^1$. With age, rows of tubercles are obliterated, and the spaces between the original rows of cusps may isolate as characteristic enamel islands. The configuration of $M^3$ corresponds to Group 4 in Fig 12.

**Lower jaw.** Lower incisors are ungrooved. Molars are large and heavily cusped, showing macrodonty. $M_1$ has three cusps on the prelobe, and no posterior cingulum (or a tiny one). Most of the cusps are fused in laminae that tend to isolate as enamel islands with the age. $M_2$

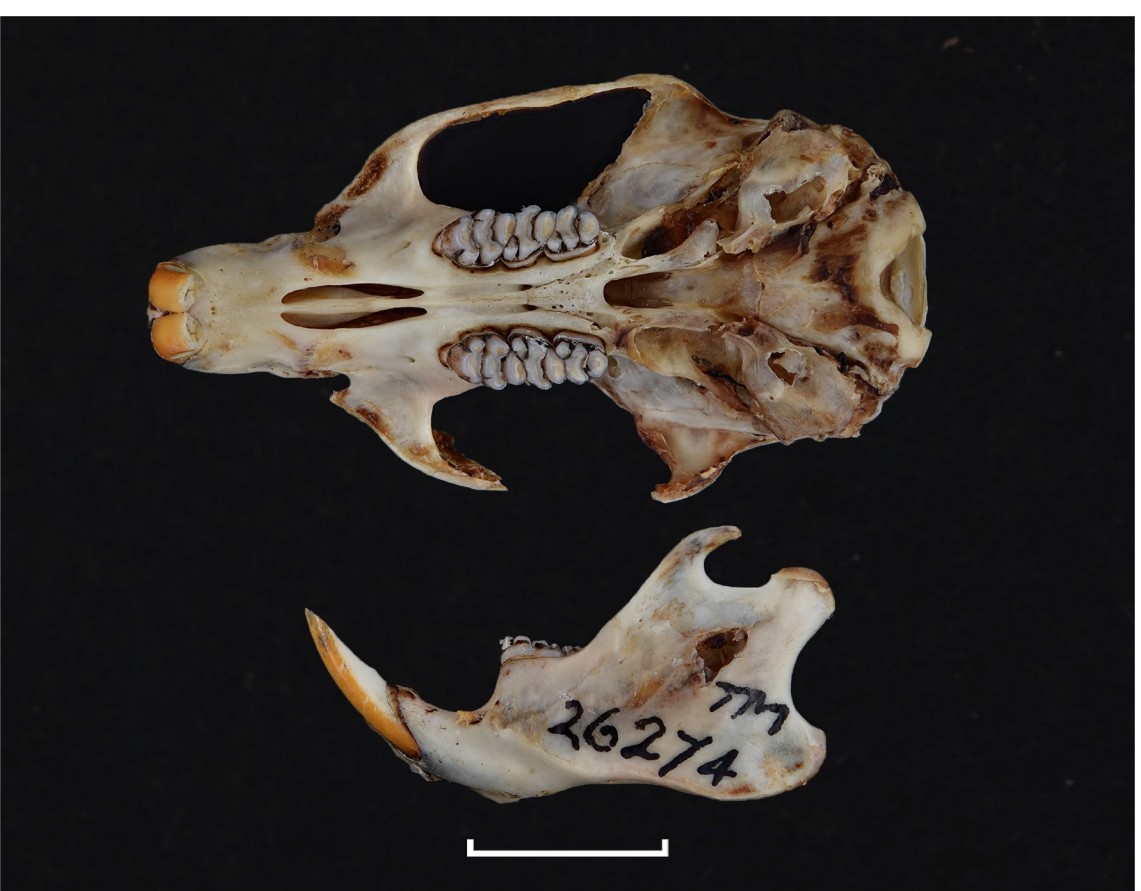

**Fig 30. Cranium of *D. capensis* (DNMNH-26274), with scale bar of 1 cm.**

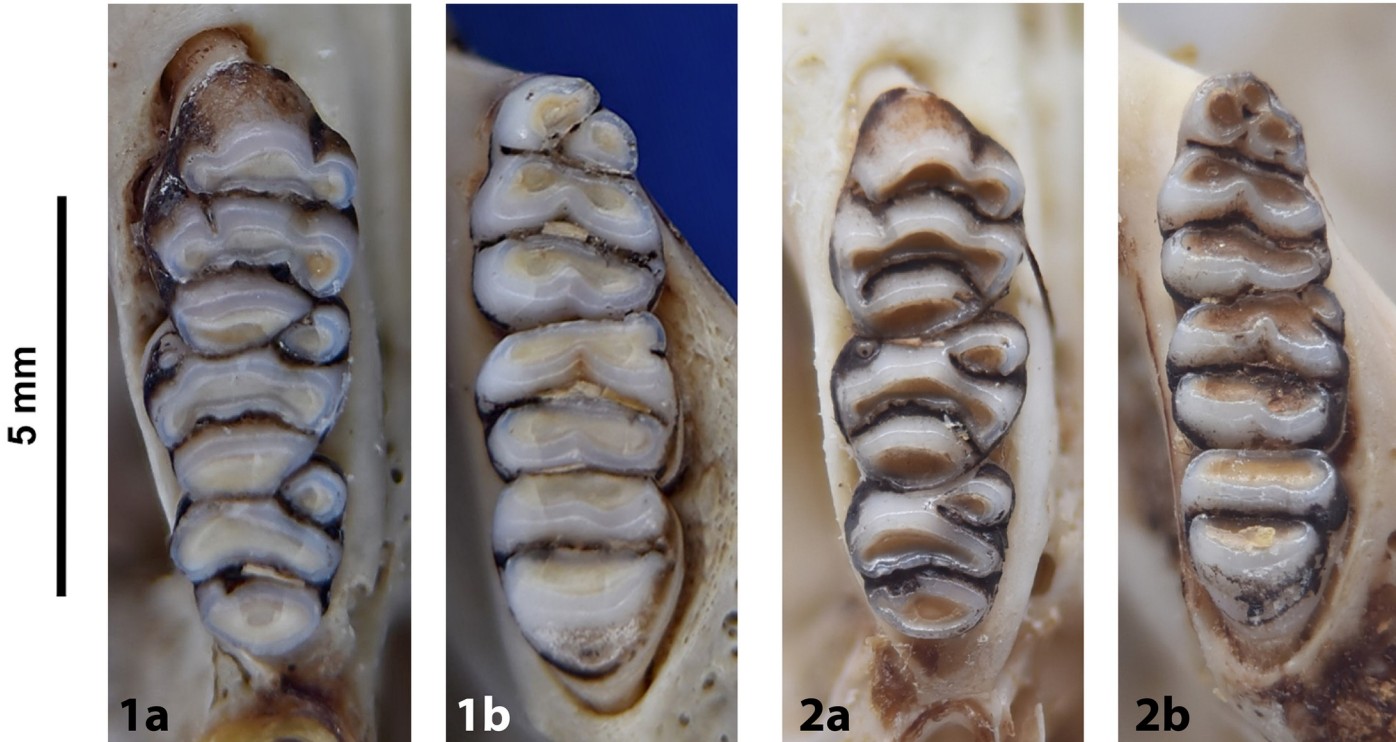

**Fig 31. Cheekteeth of *Dasymys*. 1)** Upper (a) and lower (b) right toothrow of *D. capensis* (DNMNH-26274); **2)** Upper (a) and lower (b) right toothrow of *D. robertsii* (DNMNH-30639).

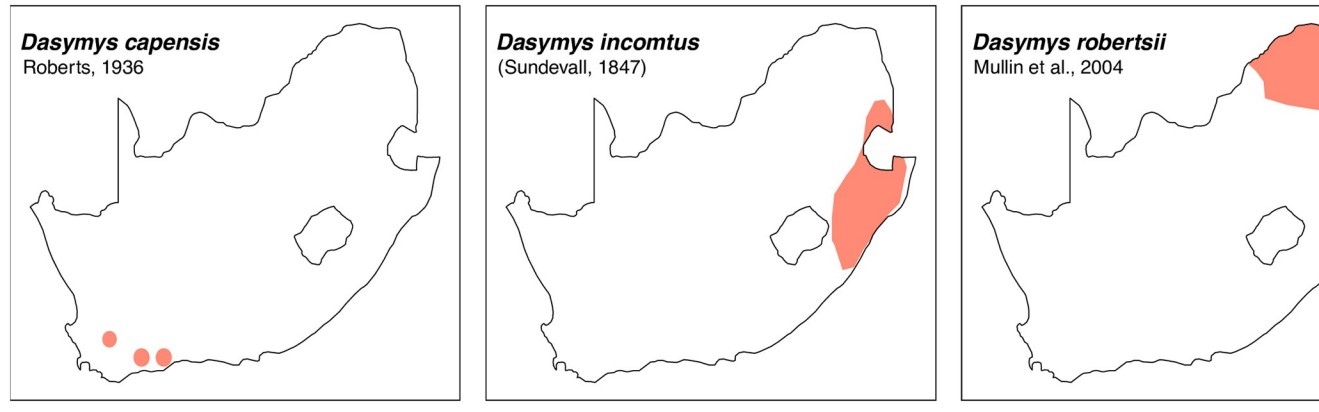

**Fig 32. Distribution maps.**

and $M_3$ have two lobes each and are large. The posterior root of the $M_3$ is large. The mandible is rather massive and high.

**Systematic notes and South African fossil record.** The number of species recognised within this genus has changed several times over the past decades, and species limits are still not fully resolved [22]. Currently, three species are described in South Africa:

- *Dasymys capensis* ROBERTS, 1936

**Table 11. Dental measurements (in mm) for *Dasymys* from South Africa, sexes and species combined.**

|  | Mean | Min | Max | n |
|---|---|---|---|---|
| LLTR | 7.5 | 6.4 | 8.4 | 31 |
| WM$_1$ | 2.3 | 1.8 | 2.6 | 32 |
| LUTR | 7.6 | 6.8 | 8.3 | 29 |
| WM$^1$ | 2.6 | 2.1 | 2.8 | 29 |

- *Dasymys incomtus* (SUNDEVALL, 1847)

- *Dasymys robertsii* MULLIN ET AL., 2004

Both *D. capensis* and *D. robertsii* were previously identified as *D. incomtus*. Although the three species should be distinguishable on a craniometric basis (if the preservation of the fossils allows it), most of the fossils of *Dasymys* were identified at a time when only *D. incomtus* was recognized. This material therefore should be re-evaluated. Additional fossil species from South African deposits have been described:

- *†Dasymys bolti* BROOM [Unpublished]

- *†Dasymys broomi* BROOM [Unpublished]

- *†Dasymys lavocati* BROOM [Unpublished]

*D. bolti* was described by Denys [47], but the other two fossil species *D. broomi* and *D. lavocati* have not been published and are considered as invalid species [4].

Genus **Grammomys** Thomas, 1915 (Thicket Rats)

Figs 33–35; Table 12

Dental formula is 1-0-0-3:1-0-0-3. Alveolar formula is 4/5-4/5-3:2/4-3/4-3 (Figs 16 and 17).

**Upper jaw.** Upper incisors are ungrooved and orthodont. The anterior palatal foramina are long and penetrate between the first upper molars. The palate is broad, and the molars show microdonty. The cusps are well separated from each other and linked by stephanodont crests. In the M$^1$, t1 is very slightly behind t2 and t3, and t4 is slightly behind t5 and t6. The M$^1$ has also a small crestiform t7, and an accessory anterior median cusp is occasionally present. The cusp t3 is reduced or absent in M$^2$, and the t9 is well visible and situated close to the t6. On the M$^3$, presence of a t1 and a tiny t3, and trace of median longitudinal link (group 6 in Fig 12).

**Lower jaw.** Lower incisors are ungrooved. Molar cusps are high and well separated. There are three cusps on the first lobe of M$_1$, a stephanodont crest uniting the first lobe with second and third lobes, and a conspicuous posterior cingulum. The M$_2$ has two lobes with two cusps each, a small anterolabial cusp and lateral cusplets or small ridges that can form stephanodont crest, as well as a posterior cingulum. The M$_3$ displays a tiny antero-external cusp.

**Systematic notes and South African fossil record.** *Grammomys* was previously considered as subgenus of *Thamnomys* but this genus is now elevated to genus rank based on morphological and phylogenetic analysis [14, 21, 53]. According to Monadjem et al. [22], this genus critically needs revision. Two species are currently recognized in South Africa:

- *Grammomys cometes* THOMAS & WROUGHTON, 1908

- *Grammomys dolichurus* (SMUTS, 1832)

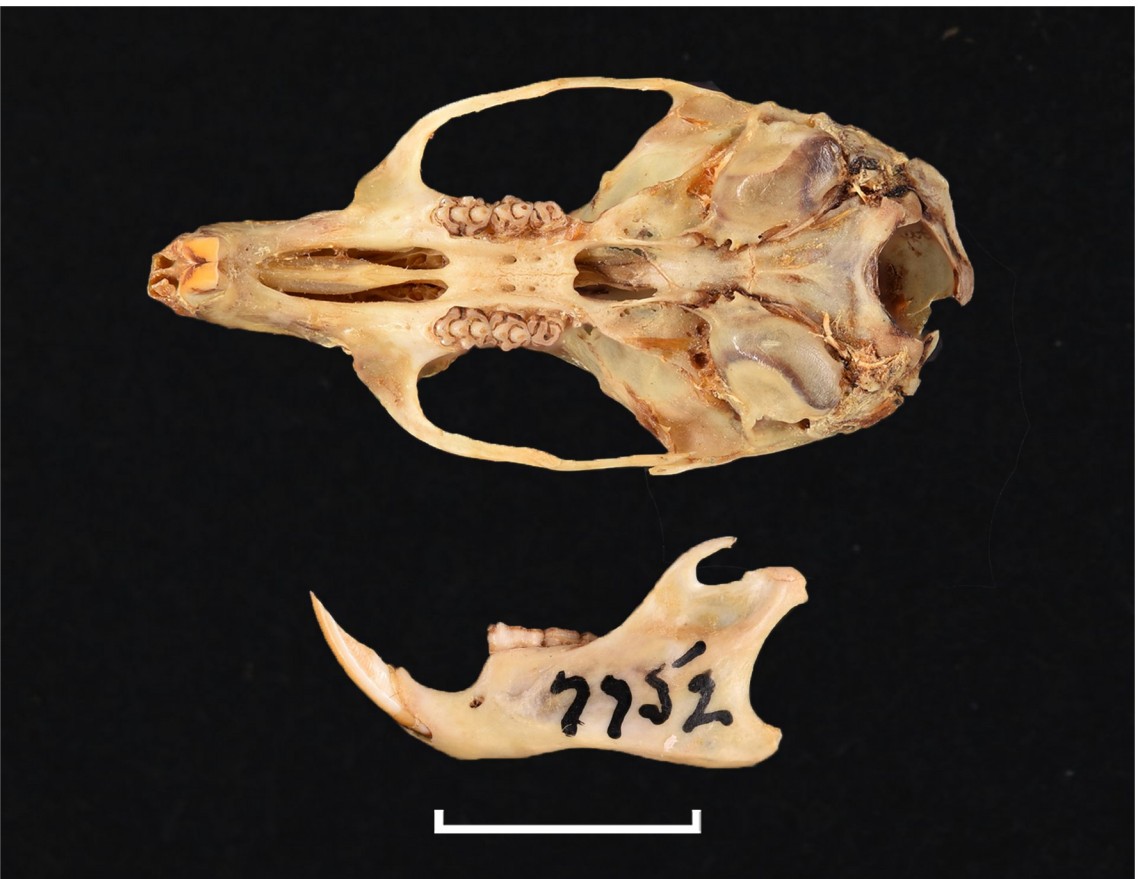

**Fig 33. Cranium of *Grammomys cometes* (DNMNH-7752), with scale bar of 1 cm.**

Fossils are known from Pliocene to Holocene deposits, with the oldest occurrence in Makapansgat Limeworks [1, 49].

Genus ***Lemniscomys*** Trouessart, 1881 (Grass Mice)

Figs 36–38; Table 13

Dental formula is 1-0-0-3:1-0-0-3. Alveolar formula is 5-5-4:5-5/6-3 (Figs 16 and 17).

**Upper jaw.**  Incisors are opisthodont. The anterior palatal foramina generally reach the level of the t2 in $M^1$. Molars are large, with broad central cusps (t2, t5, t8); this feature is mostly patent in the $M^2$. In molar $M^1$, the t4 is not connected to the t8 by a small crest as in *Aethomys* and is usually placed higher, sometimes in line with the cusp t5. There can be a small stephanodont crest uniting t6 and t9 on $M^1$ and on $M^2$. The $M^3$ is relatively large, its distal lobe has two well differentiated cusps (as opposed to one or two fused cusps in *Aethomys* or *Micaelamys*); it belongs to Group 4 in Fig 12.

**Lower jaw.**  The distinction between *Aethomys* and *Lemniscomys* based on the lower toothrow is sometimes not straightforward. In *Lemniscomys* the lateral cusplets are rarely well marked, while *Aethomys* often displays lateral cusplets or small ridges that can form some stephanodont crest. Molar size is overlapping between the two genera. As with *Micaelamys* and *Aethomys*, the $M_1$ has no clear cingulum posterior. The $M_2$ has two rootlets, resulting in six alveoli.

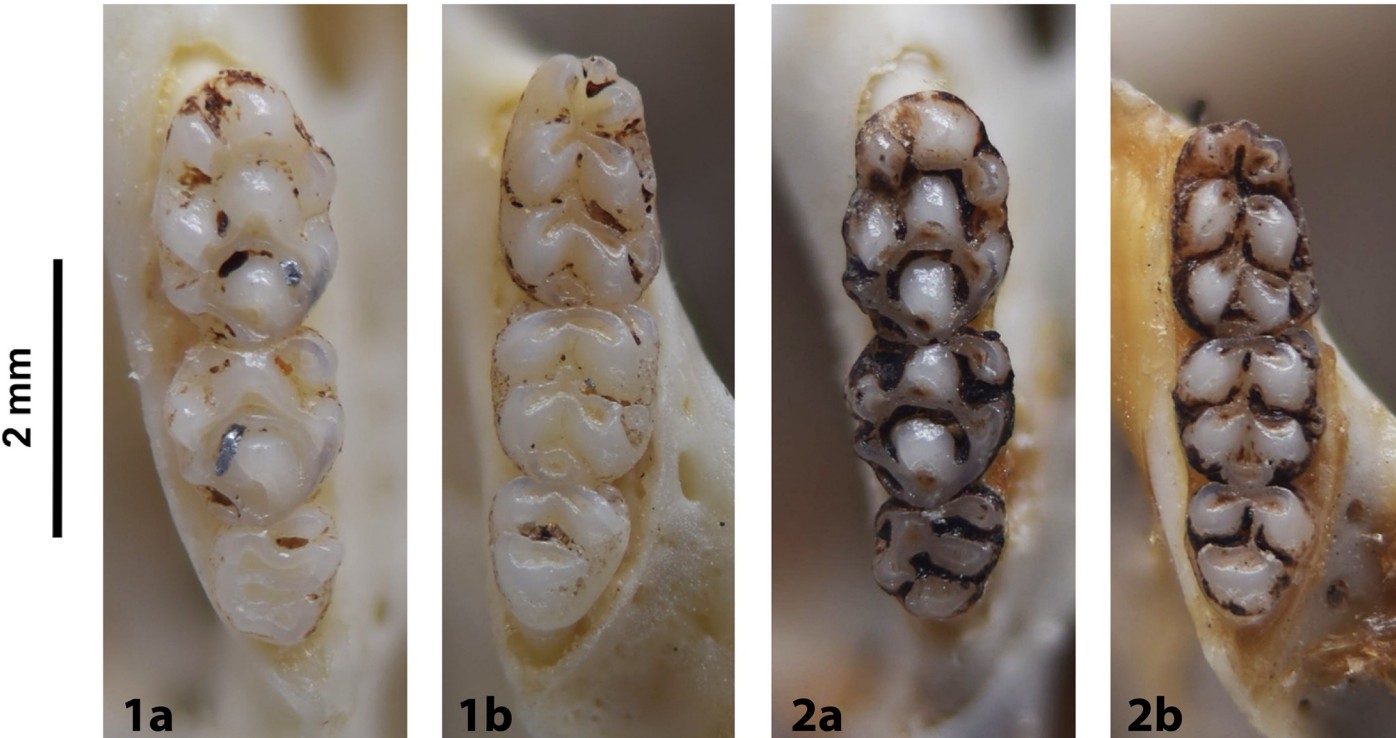

**Fig 34. Cheekteeth of *Grammomys*. 1)** Upper (a) and lower (b) right toothrow of *G. cometes* (DNMNH-40467); **2)** Upper (a) and lower (b) right toothrow of *G. dolichurus* (DNMNH-10403).

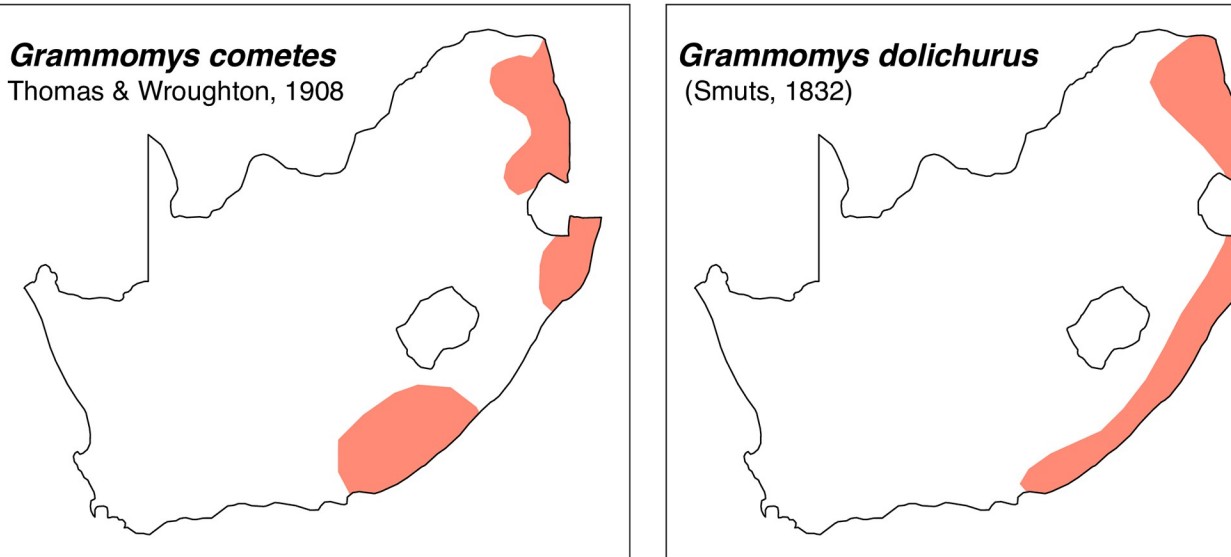

**Fig 35. Distribution maps.**

**Table 12. Dental measurements (in mm) for _Grammomys_ from South Africa, sexes and species combined.** Note that there is a significant size difference between the two species.

|  | Mean | Min | Max | n |
|---|---|---|---|---|
| LLTR | 4.5 | 3.9 | 5.1 | 28 |
| WM$_1$ | 1.3 | 1.0 | 1.6 | 28 |
| LUTR | 4.4 | 3.9 | 5.1 | 27 |
| WM$^1$ | 1.4 | 1.3 | 1.8 | 27 |

**Systematic notes and South African fossil record.** There is only one species recognized in South Africa:

- _Lemniscomys rosalia_ (Thomas, 1904)

Fossils are known from the Early Pleistocene in various karst deposits from the Sterkfontein Valley [50].

Genus **_Mastomys_** Thomas, 1915 (Multimammate Mice)

Figs 39–41; Table 14

Dental formula is 1-0-0-3:1-0-03. Alveolar formula is 3-3-2:2-2-2 (Figs 16 and 17).

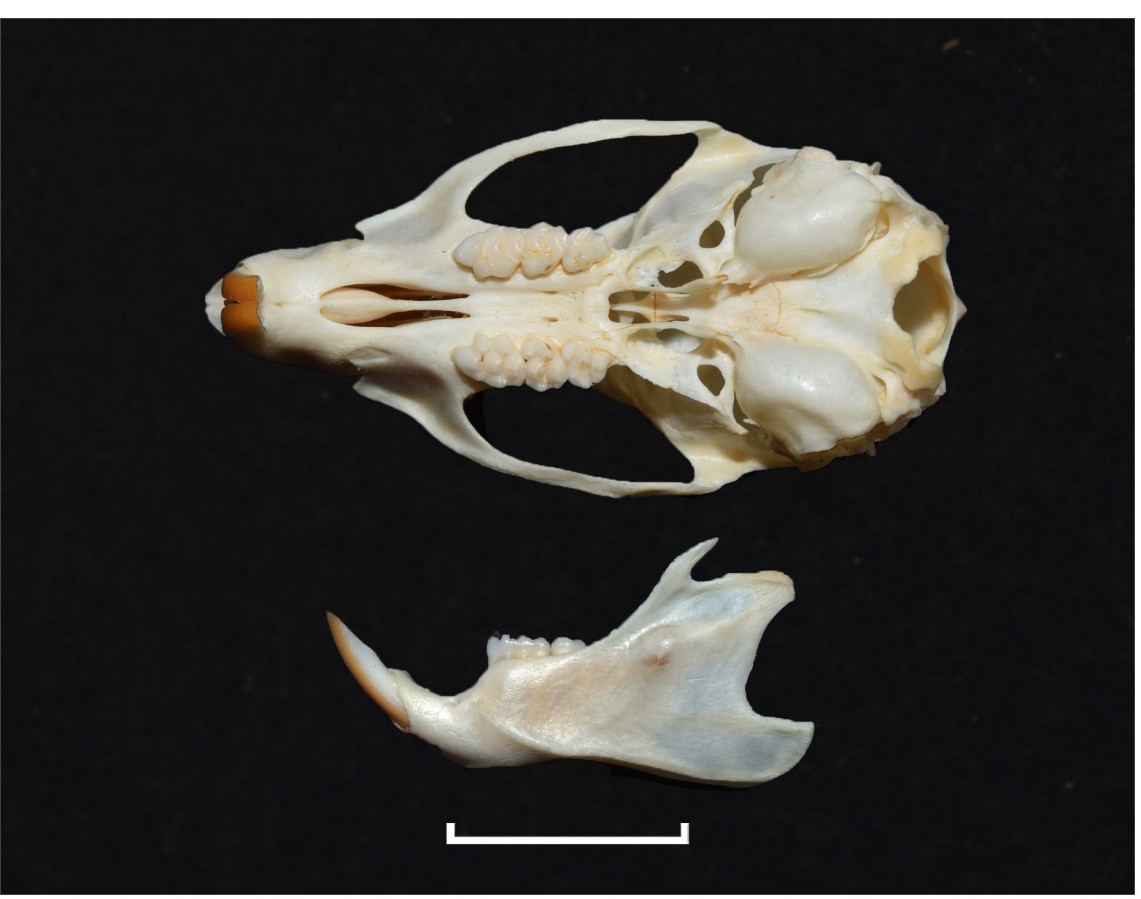

**Fig 36. Cranium of _Lemniscomys rosalia_ (IVB-RS3796), with scale bar of 1 cm.**

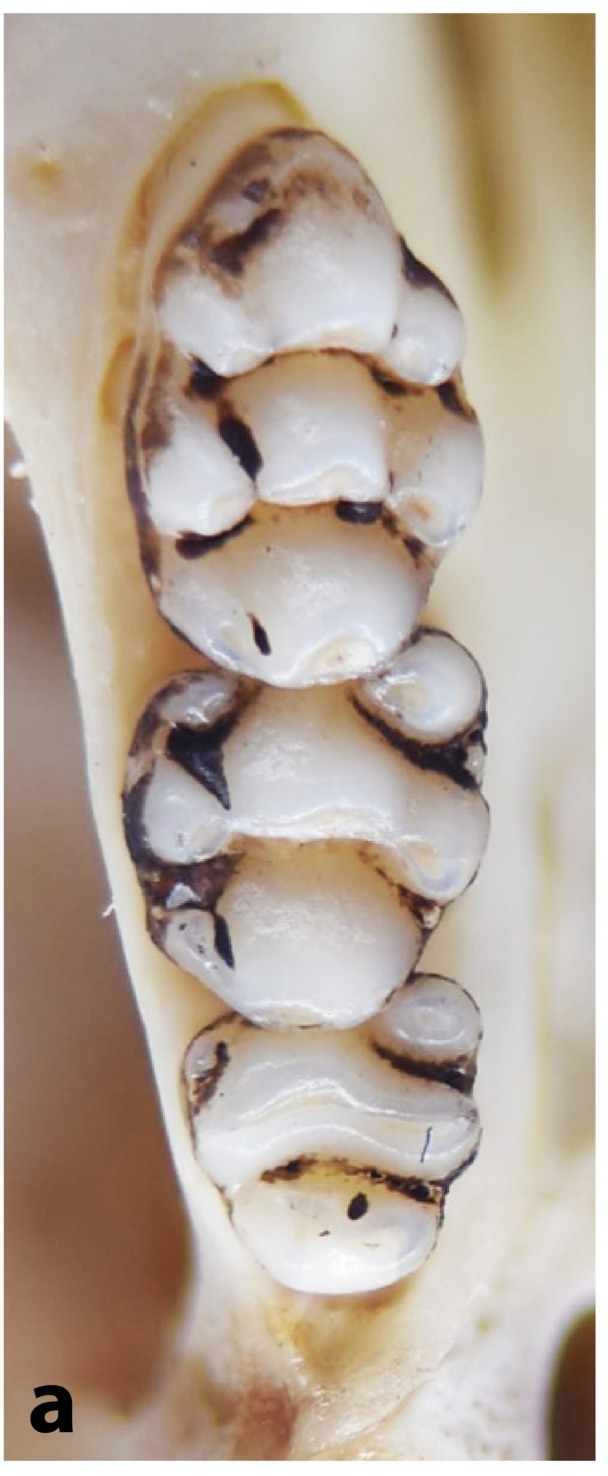
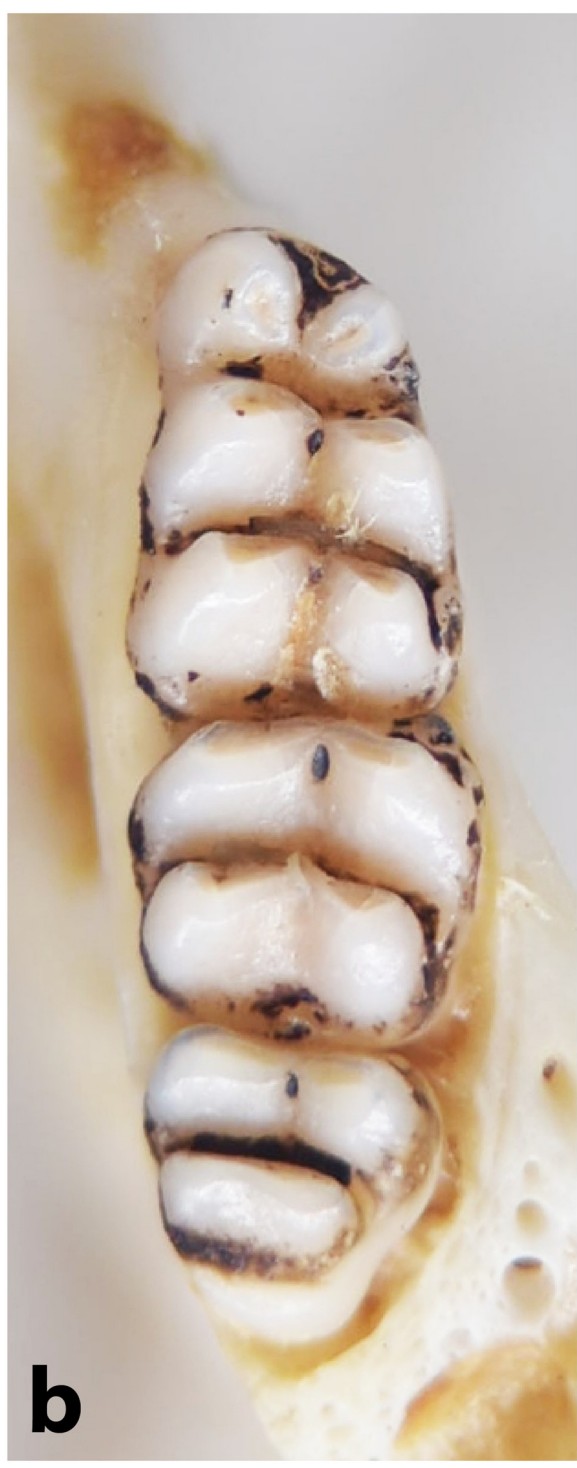

**Fig 37. Upper (a) and lower (b) right toothrow of *L. rosalia* (DNMNH-29961).**

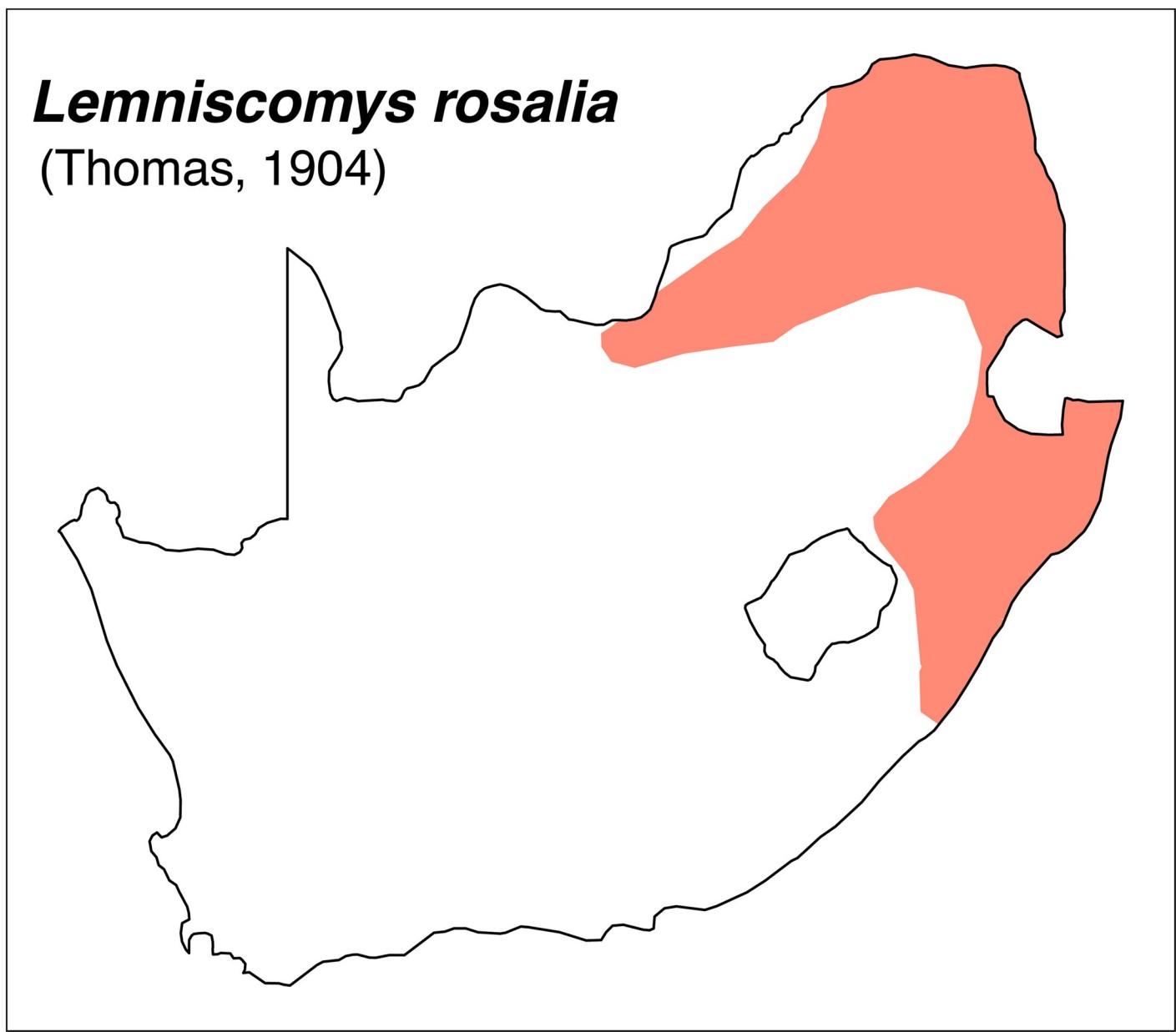

**Fig 38. Distribution map.**

**Table 13. Dental measurements (in mm) for *Lemniscomys rosalia*, sexes and species combined.**

|  | Mean | Min | Max | n |
|---|---|---|---|---|
| LLTR | 5.7 | 5.5 | 6.3 | 17 |
| $WM_1$ | 1.7 | 1.5 | 1.9 | 17 |
| LUTR | 5.9 | 5.5 | 6.5 | 17 |
| $WM^1$ | 1.9 | 1.8 | 2.2 | 17 |

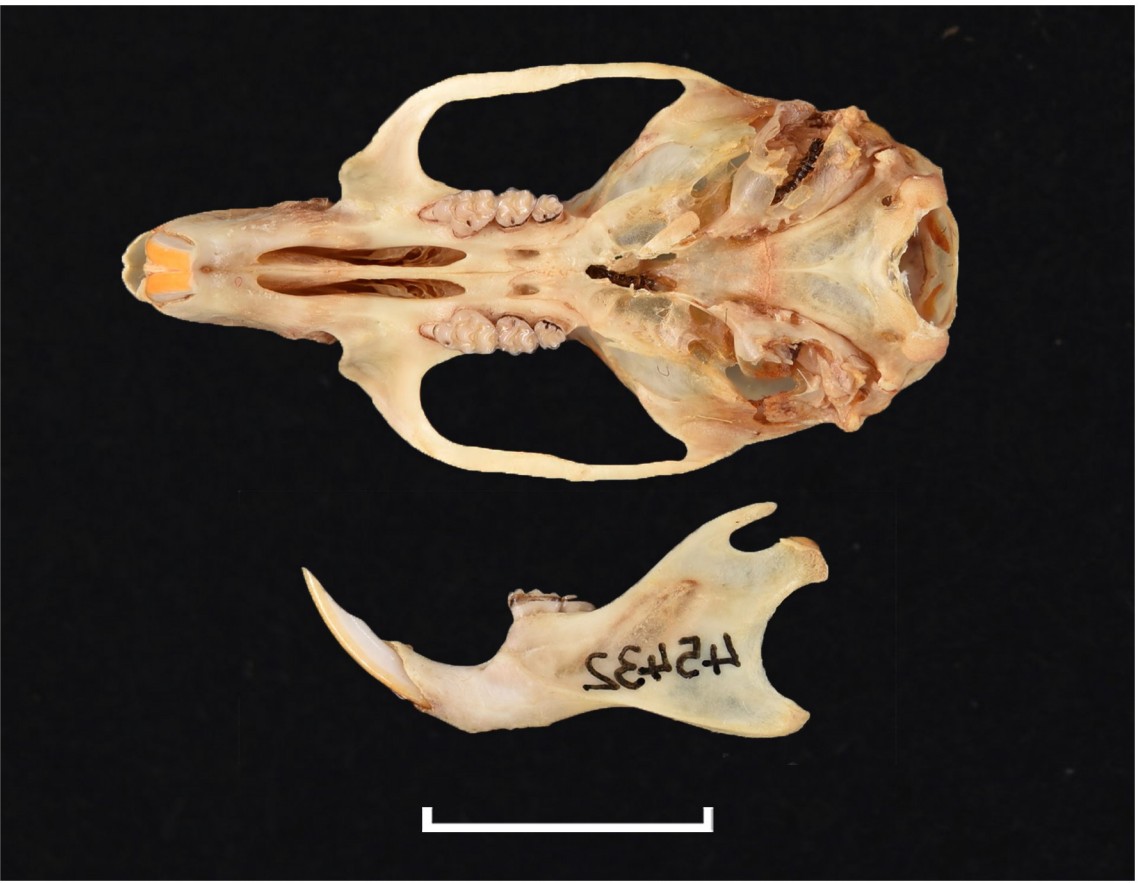

**Fig 39. Cranium of *Mastomys coucha* (DNMNH-45432), with scale bar of 1 cm.**

**Upper jaw.** *Mastomys* is morphologically very similar to *Myomyscus* but has a much wider distribution. Upper incisors are smooth and orthodont. The anterior palatal foramina are long and reach the t1 of the $M^1$. Rows of cusps are rather distorted in the $M^1$, with the t1 located behind t2 and t3, and the t4 also located behind t5 and t6. As in *Myomyscus* and *Zelotomys* (two other members of the Praomyini tribe), the t9 is well separated from the t6, and located far in the labial side. The $M^1$ has three roots. The $M^2$ has a t1 and a small t3, and the t9 is low and well distinct. The $M^3$ has two lobes and corresponds to Group 3 in Fig 12.

**Lower jaw.** Lower incisors are smooth. Molars show a typical Praomyini pattern (as *Myomyscus* and *Zelotomys*). Both $M_1$ and $M_2$ have a posterior cingulum. The $M_1$ displays a small posterior cingulum and a posterolabial cusplet (plc). The $M_2$ has both anterolabial and posterolabial cusplets. The mandible is approximately the same size as in *Rhabdomys*, but both genera can be distinguished based on the number of alveoli when teeth are absent (alveolar formula is 2-2-2 in *Mastomys*, 4-5-3 in *Rhabdomys*). As in *Myomyscus* and *Zelotomys*, the alveolar region of the mandible is well-developed in proportion to the rest of the mandible.

**Systematic notes and South African fossil record.** *Mastomys* was previously included in *Praomys*, but is now placed in its own genus [22]. Two species are currently recognized in South Africa:

- *Mastomys coucha* (Sмıтн, 1834)

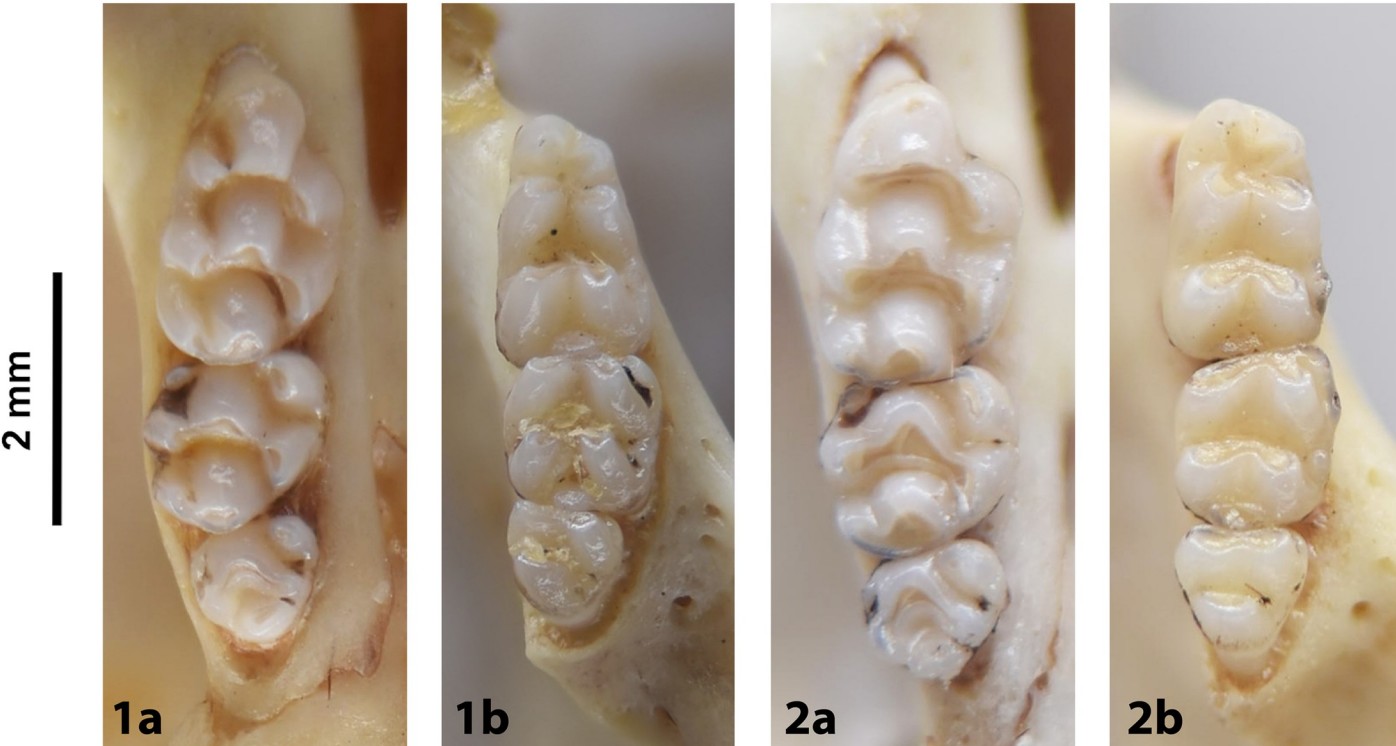

**Fig 40. Cheekteeth of *Mastomys*. 1)** Upper (a) and lower (b) right toothrow of *M. coucha* (DNMNH-45433); **2)** Upper (a) and lower (b) right toothrow of *M. natalensis* (DNMNH-37511).

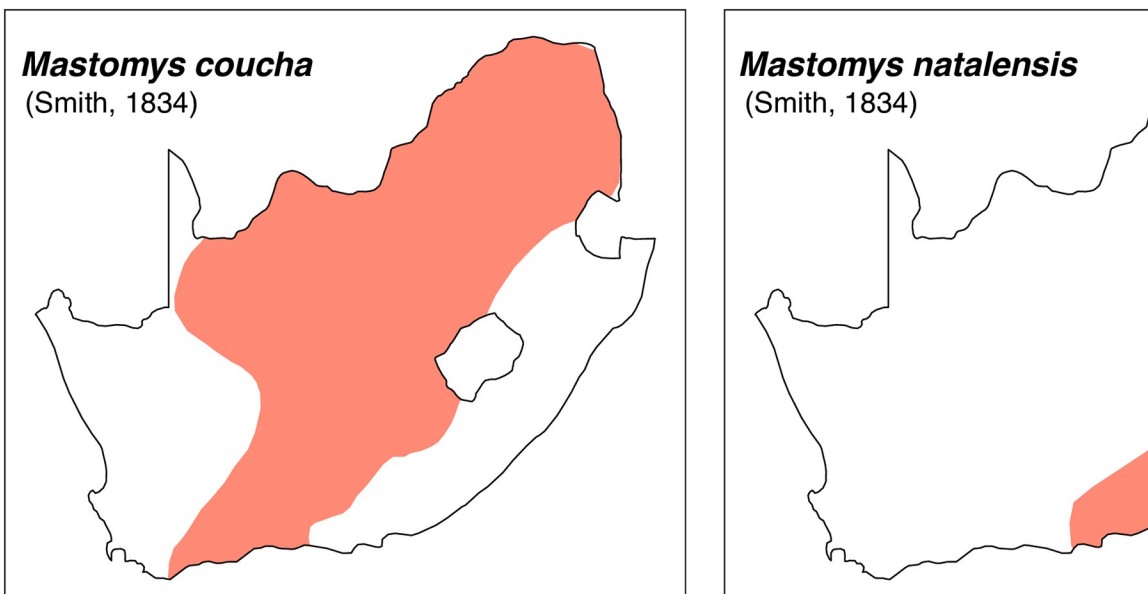

*Mastomys coucha*
(Smith, 1834)

*Mastomys natalensis*
(Smith, 1834)

**Fig 41. Distribution maps.**

**Table 14. Dental measurements (in mm) for *Mastomys* from South Africa, sexes and species combined.**

|          | Mean | Min | Max | n   |
|----------|------|-----|-----|-----|
| LLTR     | 4.5  | 4.2 | 4.8 | 40  |
| $WM_1$   | 1.3  | 1.2 | 1.5 | 40  |
| LUTR     | 4.8  | 4.4 | 5.2 | 40  |
| $WM^1$   | 1.6  | 1.4 | 1.8 | 40  |

- *Mastomys natalensis* (Smith, 1834)

Fossils of *Mastomys* have been recovered from many Pleistocene fossil deposits from South Africa. Most of this material is attributed to *M. natalensis*, but the remains were identified at a time when *M. coucha* was not recognized. These two species appear undistinguishable using cranio-dental features [54], so specimens from the Early Pleistocene should be conservatively attributed to *Mastomys* sp.

Genus ***Micaelamys*** Ellerman, 1941 (Lesser Veld Rats)

Figs 42–44; Table 15

Dental formula is 1-0-0-3:1-0-0-3. Alveolar formula is 4/5-3/4-3:3-3-3 (Figs 16 and 17).

**Upper jaw.** Incisors are opisthodont. In the $M^1$, cusps are few distorted with t1 slightly behind t2 and t3, and t4 behind t5 and t6. There are sometimes traces of stephanodonty. The

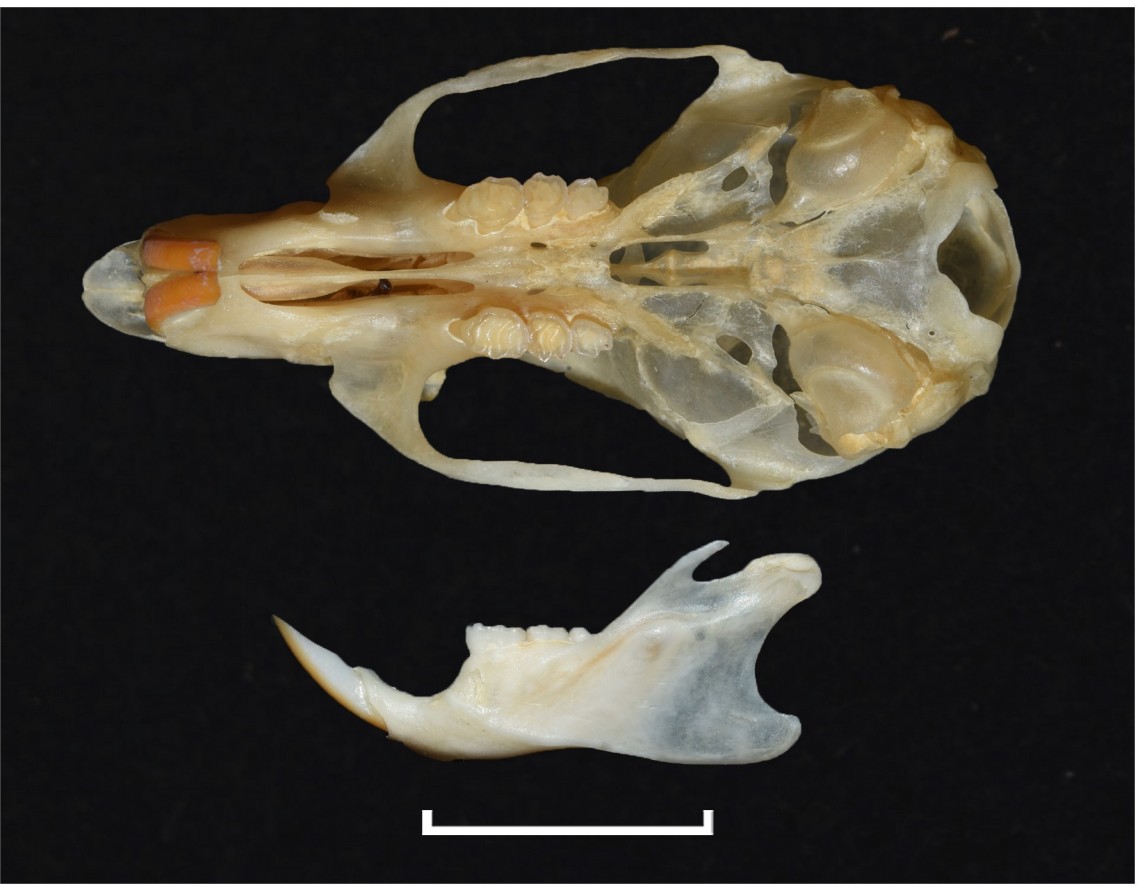

**Fig 42. Cranium of *Micaelamys namaquensis* (IVB-M4883), with scale bar of 1 cm.**

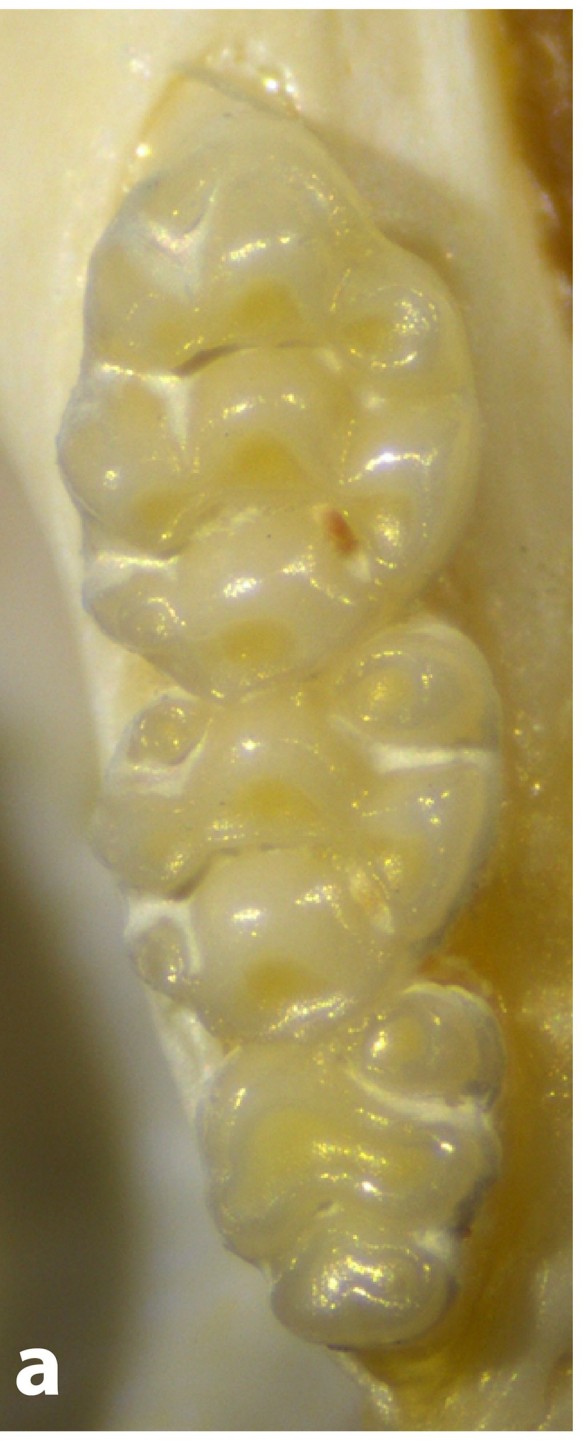 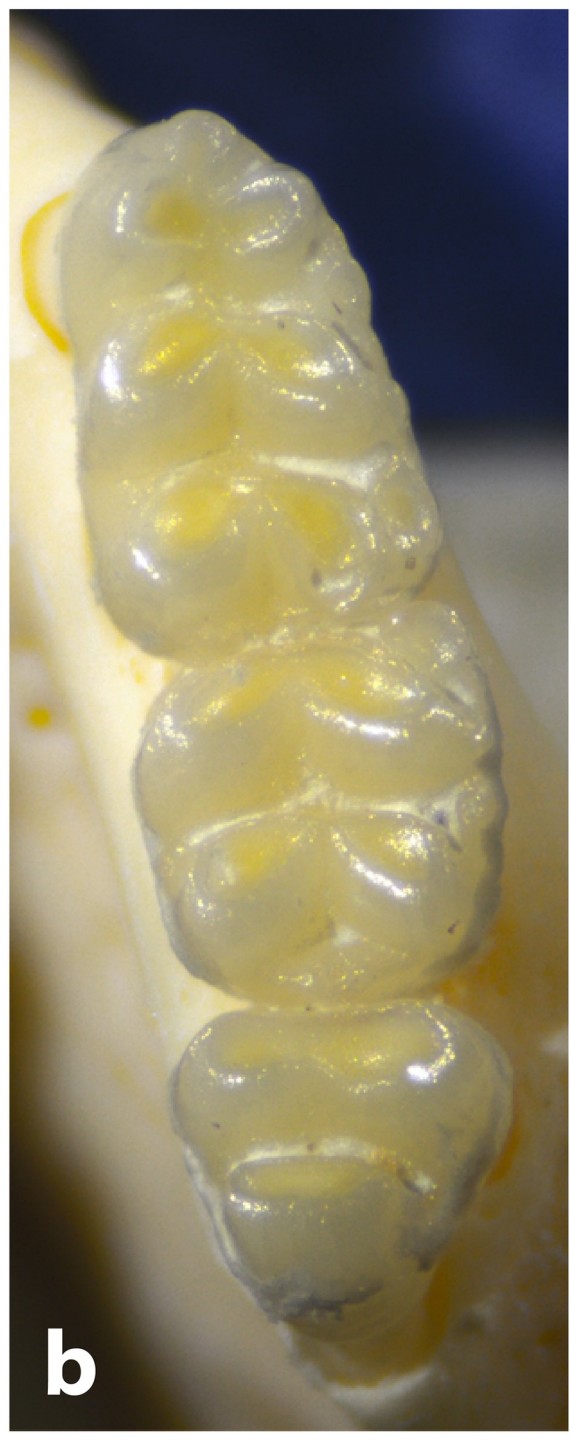

**Fig 43. Upper (a) and lower (b) right toothrow of *M. namaquensis* (MNHN-ZM-MO-1990-323).**

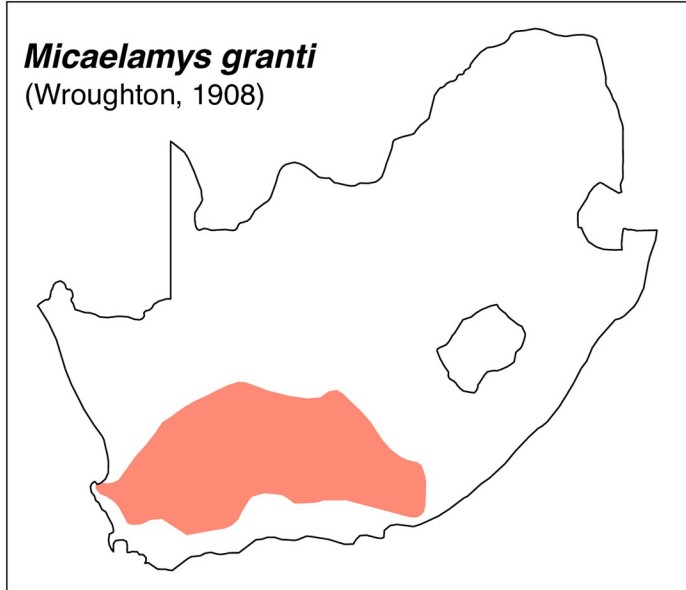
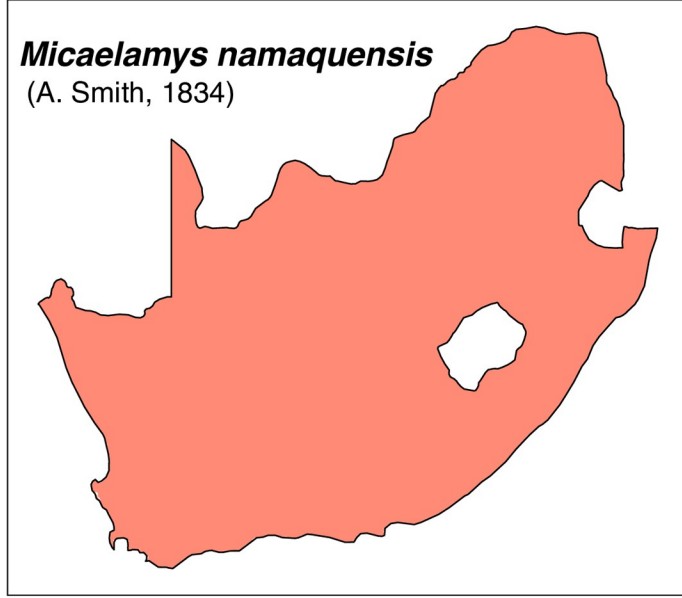

**Fig 44. Distribution maps.**

$M^3$ lacks the t3 its distal lobe has one elongated cusp (or 2 very few differentiated, as opposed to two well differentiated in *Lemniscomys*); it corresponds to Group 3 in Fig 12. Molars are smaller than in *Aethomys* and *Lemniscomys*.

**Lower jaw.** $M_1$ has always three cusps in the anterior lobe (presence of a well-developed tma), as opposed to two in most specimens of *Aethomys* (some *Aethomys* display a very small anteromedian cusplet). As *Aethomys* and *Lemniscomys*, the $M_1$ has no posterior cingulum or a very small one in few specimens. Similarly, the $M_2$ has a very small pc contrary to *Aethomys* where there is often a round median pc. A row of additional cusplets often occurs on the labial side of the $M_1$ and $M_2$.

**Systematic notes and South African fossil record.** Species of this genus were previously included within *Aethomys* but are now placed in their own genus based on morphological and molecular ground [28, 52, 55]. Two species are currently recognized in South Africa:

- *Micaelamys granti* (WROUGHTON, 1908)

- *Micaelamys namaquensis* (A. SMITH, 1834)

The genus *Micaelamys* has been identified in many Pleistocene deposits in South Africa, often referred to as *Aethomys* (for instance, *Aethomys namaquensis*). The oldest *Aethomys* cf. *namaquensis* has been described by Pocock [48] in Makapansgat.

**Table 15. Dental measurements (in mm) for *Micaelamys namaquensis*, sexes and species combined.**

|        | Mean | Min | Max | n  |
|--------|------|-----|-----|----|
| LLTR   | 5.0  | 4.7 | 5.3 | 22 |
| $WM_1$ | 1.5  | 1.3 | 1.8 | 22 |
| LUTR   | 5.2  | 4.9 | 5.5 | 22 |
| $WM^1$ | 1.7  | 1.6 | 1.9 | 22 |

Genus *Mus* Linnaeus, 1758 (Old World Mice and Pygmy Mice)

Figs 45–47; Table 16

Dental formula is 1-0-0-3:1-0-0-3. Alveolar formula is 3-3-1/2:2-2/3-1/2 (Figs 16 and 17).

**Upper jaw.** Upper incisors are ungrooved and opisthodont, with the worn posterior surface characteristically notched for the three South African species. The palatal foramina reach the first or the second row of cusps in $M^1$. The skull is small, and so are the molars. The cusps on $M^1$ are distorted, with t1 situated far posterior of t2 and t3. Molar $M^3$ is small with a reduced cusp configuration: it has a t1, but cusps are poorly differentiated and there is a link between first and second lobe (Group 7 in Fig 12). Presence of a masseteric knob.

**Lower jaw.** The $M_1$ has a typically enlarged lingual anteroconid that distorts the tooth, while the labial anteroconid is smaller and close to the protoconid. The $M_3$ is tiny and has one or two lobes with one or two corresponding alveoli (always two lobes and two alveoli in *Acomys*). The mandible is small, being of the same average size as *Dendromus*; the $M_2$ is often 3-rooted in *Mus* whereas it is 2-rooted in *Dendromus*.

**Systematic notes and South African fossil record.** Four species are currently recognized in South Africa:

- *Mus indutus* (Thomas, 1910)

- *Mus minutoides* Smith, 1834

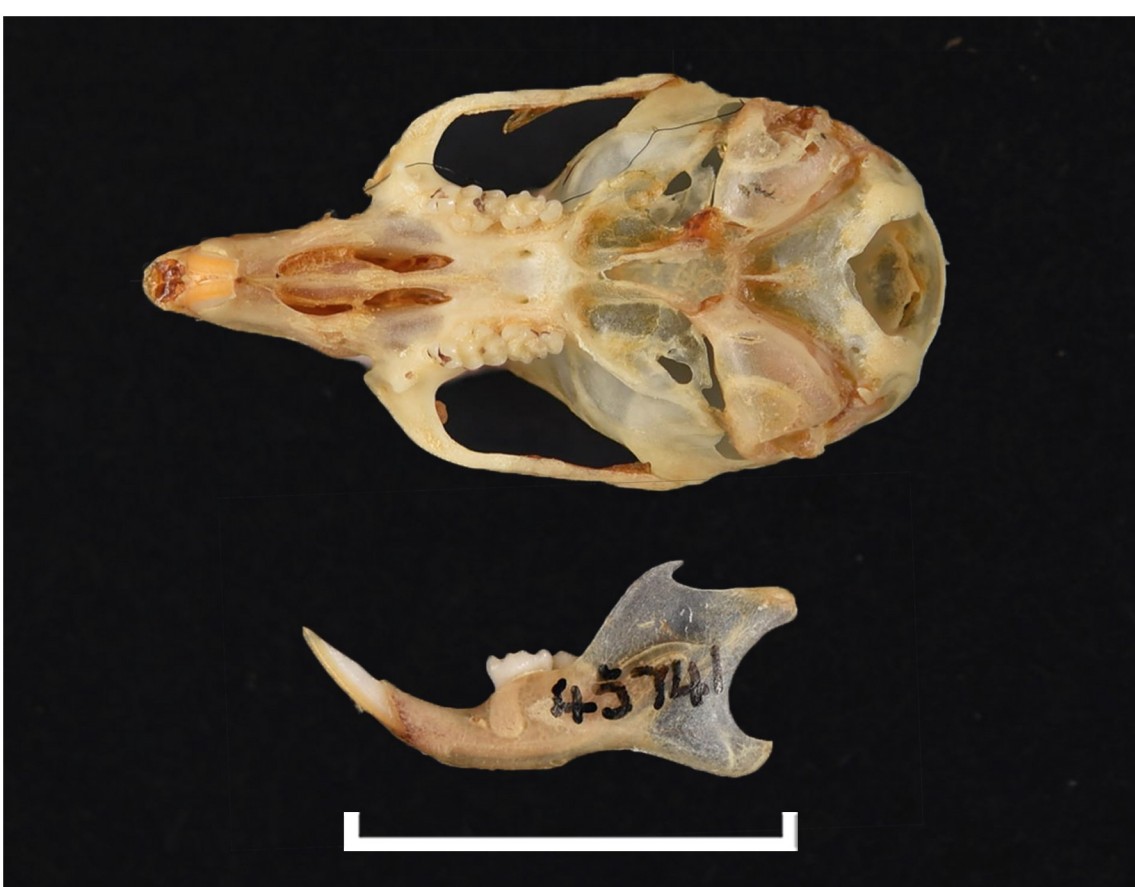

**Fig 45. Cranium of *Mus indutus* (DNMNH-45741), with scale bar of 1 cm.**

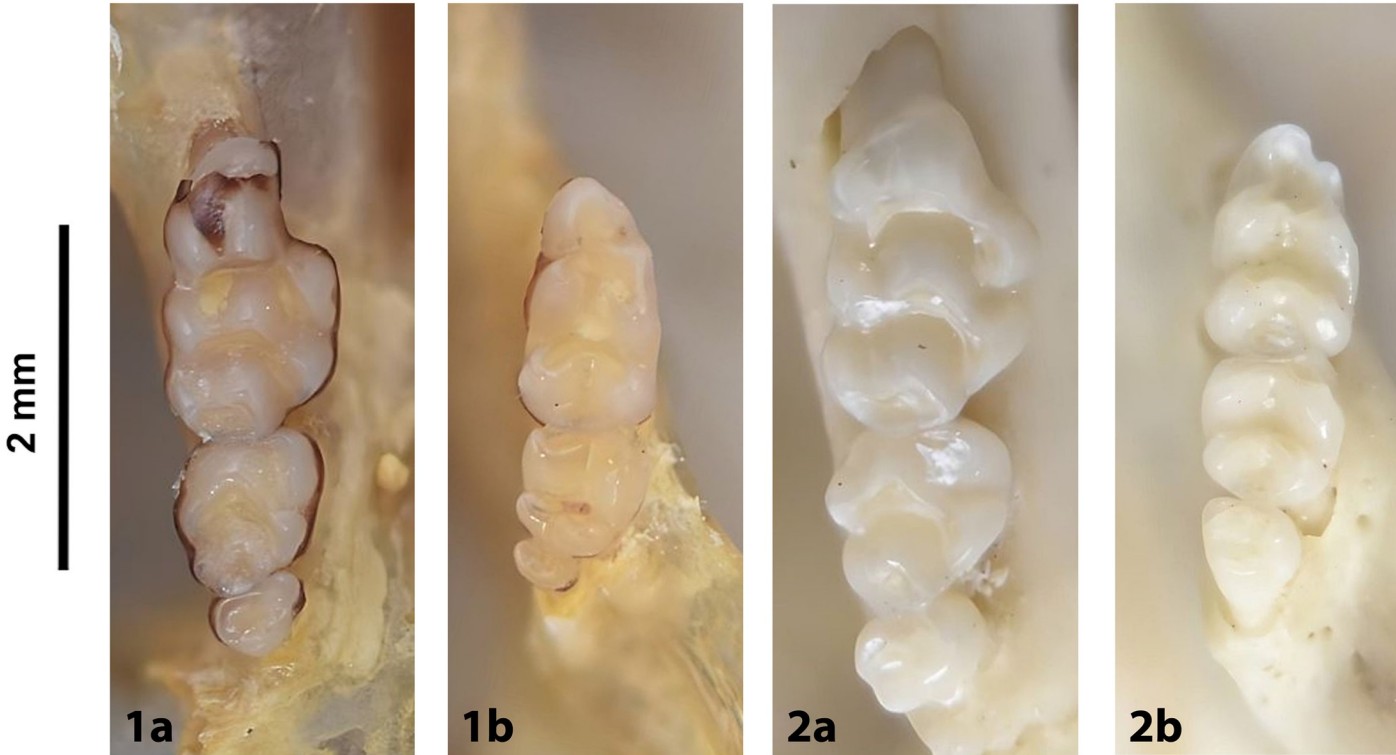

**Fig 46. Cheekteeth of *Mus*. 1)** Upper (a) and lower (b) right toothrow of *M. indutus* (DNMNH-45737); **2)** Upper (a) and lower (b) right toothrow of *M. musculus* (MNHN-ZM-MO-1994-2350).

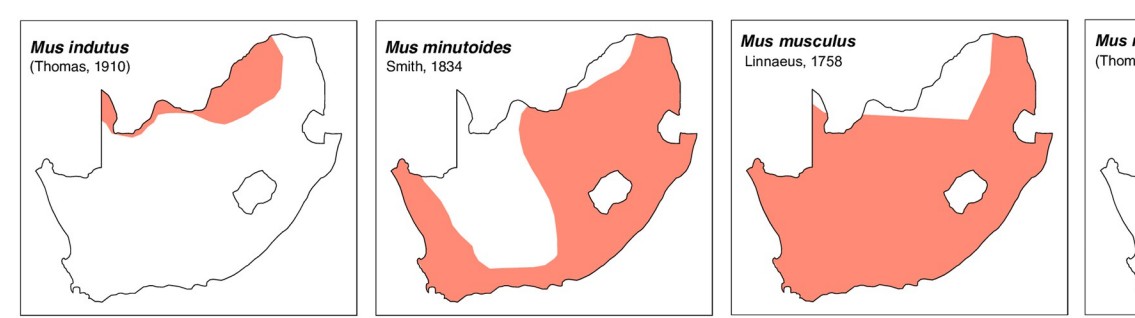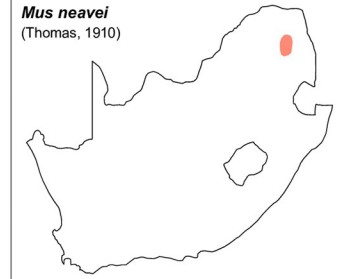

**Fig 47. Distribution maps.**

**Table 16. Dental measurements (in mm) for *Mus* from South Africa, sexes and species combined.**

|  | Mean | Min | Max | n |
|---|---|---|---|---|
| LLTR | 2.6 | 2.2 | 3.2 | 72 |
| $WM_1$ | 0.9 | 0.7 | 1.1 | 72 |
| LUTR | 3.1 | 2.6 | 3.4 | 71 |
| $WM^1$ | 1.0 | 0.9 | 1.4 | 71 |

- *Mus musculus* LINNAEUS, 1758

- *Mus neavei* (THOMAS, 1910)

*M. musculus* belongs to the subgenus *Mus*, while *M. indutus*, *M. minutoides* and *M. neavei* belong to the subgenus *Nannomys*, formerly named *Leggada* [e.g., 24, 56]. The genus *Mus* is known from the Pliocene of Makapansgat [48] and has been recovered from many Pleistocene sites. The species *M. musculus* was introduced and is thus not found in the Pleistocene deposits. The oldest and single record of *M. musculus* in Holocene deposits is from Hope Hill Shelter dated to 4,400±100 B.P. [57]

Genus **Myomyscus** Shortridge, 1942 (Meadow Mice)

Figs 48–50; Table 17

Dental formula is 1-0-0-3:1-0-0-3. Alveolar formula is 3-3-2:2-2-2 (Figs 16 and 17).

**Upper jaw.** *Myomyscus* is morphologically very similar to *Mastomys* and is mostly diagnosable by its restricted distribution to the Fynbos biome. The anterior palatal foramina reach the t1 of the $M^1$. The rows of cusps are slightly distorted in the $M^1$, with the t1 located behind t2 and t3, and the t4 also located behind t5 and t6. As with other members of the Praomyini tribe, the t9 is well separated from the t6, and located far in the labial side. The $M^1$ has three roots. The $M^2$ has a t1 and a small t3, and the t9 is low and well distinct. Like in *Mastomys*, the $M^3$ is large with a t1 and a t3 and two lobes and corresponds to Group 3 in Fig 11.

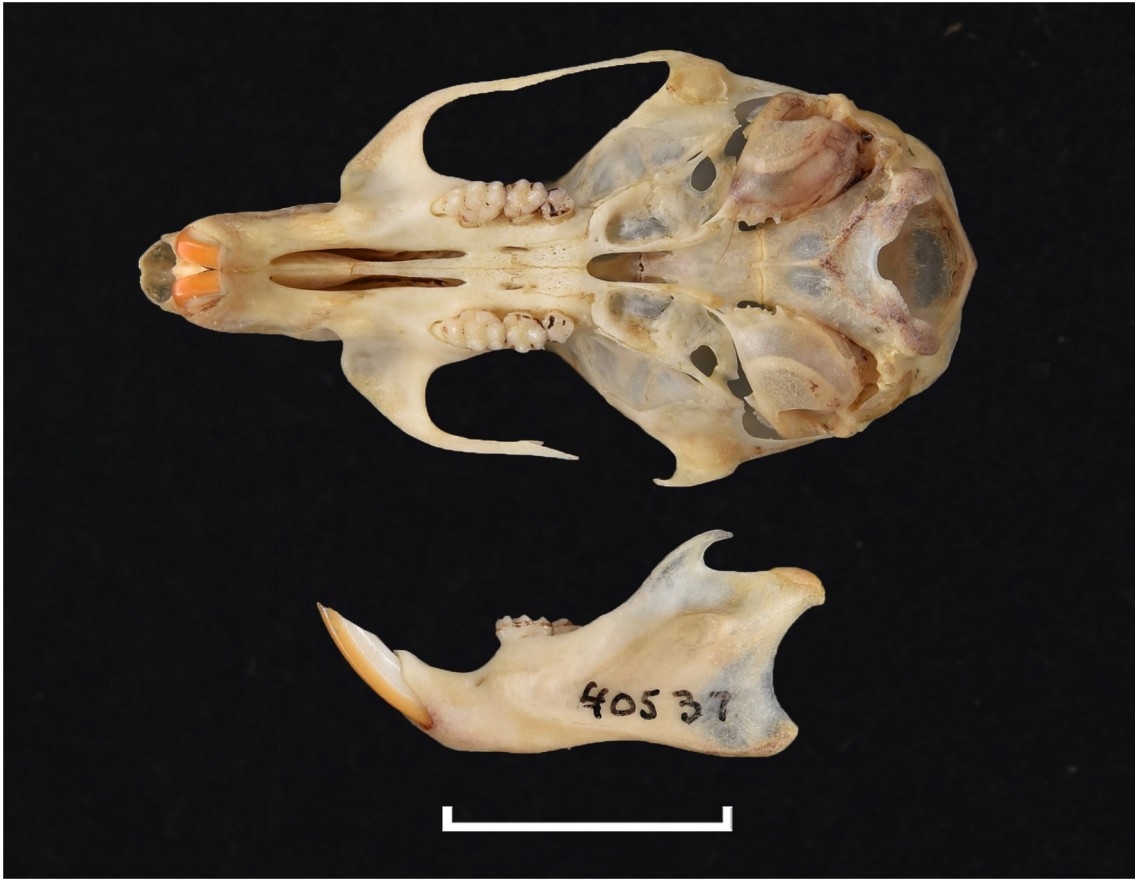

**Fig 48. Cranium of *Myomyscus verreauxii* (DNMNH-40537), with scale bar of 1 cm.**

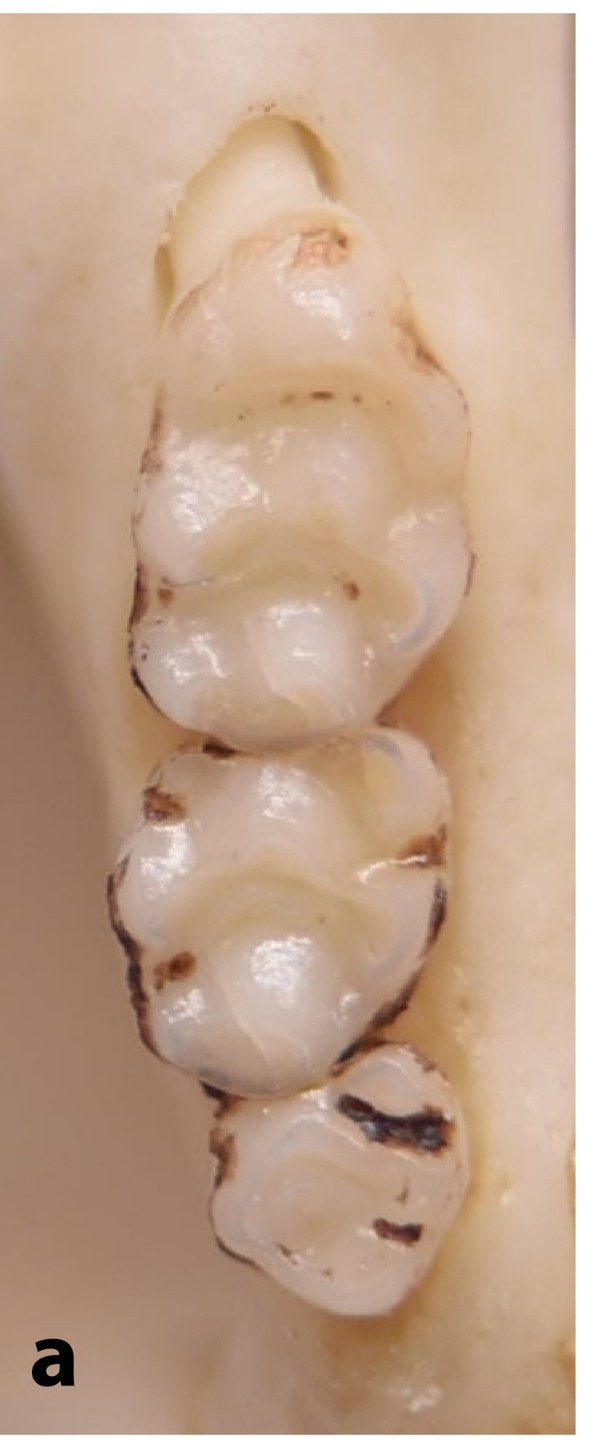
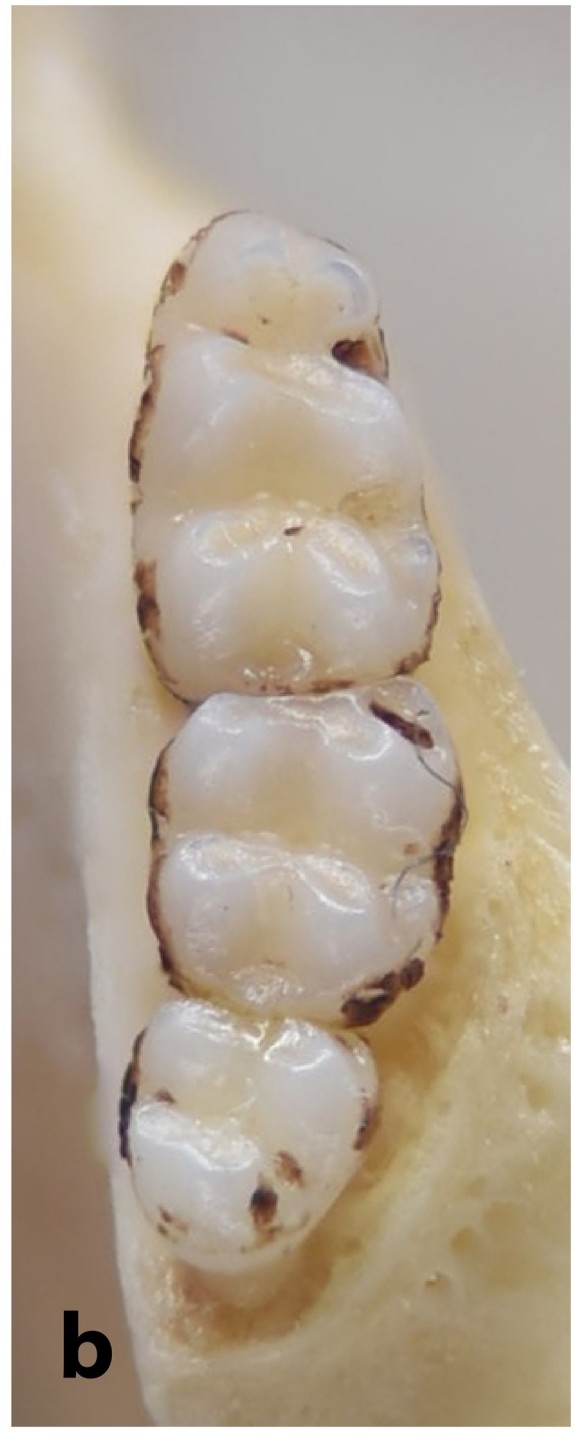

**Fig 49. Upper (a) and lower (b) right toothrow of *M. verreauxii* (DNMNH-40537).**

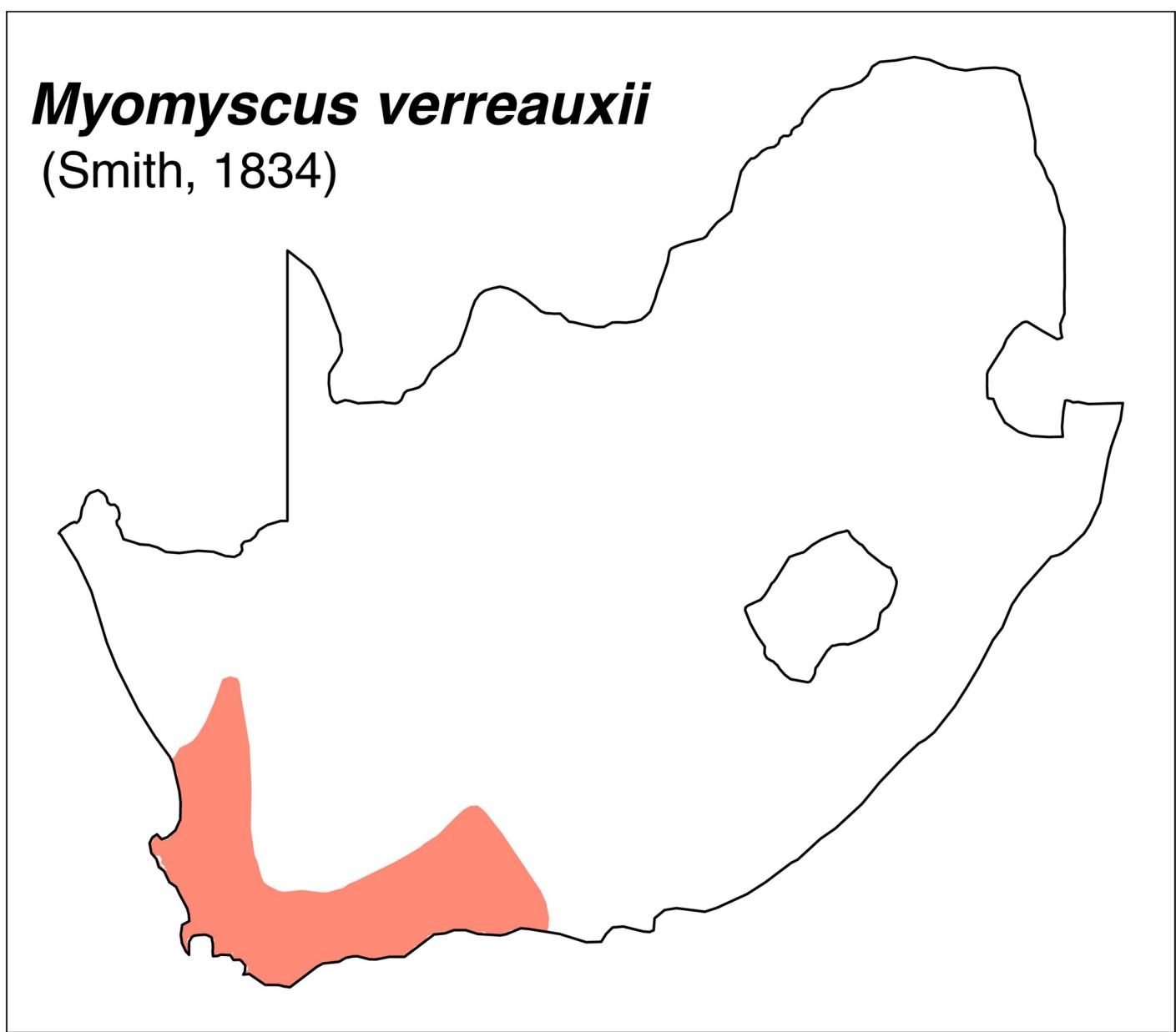

**Fig 50. Distribution map.**

**Table 17. Dental measurements (in mm) for *Myomyscus verreauxii*, sexes combined.**

|  | Mean | Min | Max | n |
|---|---|---|---|---|
| LLTR | 4.5 | 4.2 | 4.7 | 3 |
| WM$_1$ | 1.2 | 1.2 | 1.3 | 3 |
| LUTR | 4.7 | 4.5 | 5.0 | 3 |
| WM$^1$ | 2.4 | 1.4 | 1.5 | 3 |

**Lower jaw.** As in *Mastomys*, molars show a typical Praomyini pattern. Both $M_1$ and $M_2$ have a posterior cingulum. The $M_1$ displays a small posterior cingulum and a posterolabial cusplet (plc). The $M_2$ has both anterolabial and posterolabial cusplets. The mandible is of average size.

**Systematic notes and South African fossil record.** A single species is recognized in South Africa:

- *Myomyscus verreauxii* (Smith, 1834)

In the past, this species had been placed in the genera *Praomys* and *Myomys* [22]. In South Africa, remains of *Myomyscus* are known since the Late Pliocene [50, 58].

Genus ***Otomys*** Cuvier, 1824 (Vlei Rats or Laminate-toothed Rats)

Figs 51–53; Table 18

Dental formula is 1-0-0-3:1-0-0-3. Alveolar formula is variable, for instance 3-3-11:7-4-3 (Figs 16 and 17).

**Upper jaw.** Incisors are opisthodont and always grooved (the number of grooves varies by species). The anterior palatal foramina rarely reach the $M^1$. The palate is very narrow with broad region of contact between the two halves. The cusps of the cheekteeth are fused into several transverse laminae: $M^1$ has three laminae; $M^2$ has two; $M^3$ is the longest cheekteeth, having four or more laminae. The auditory bullae are hardly, or not, inflated.

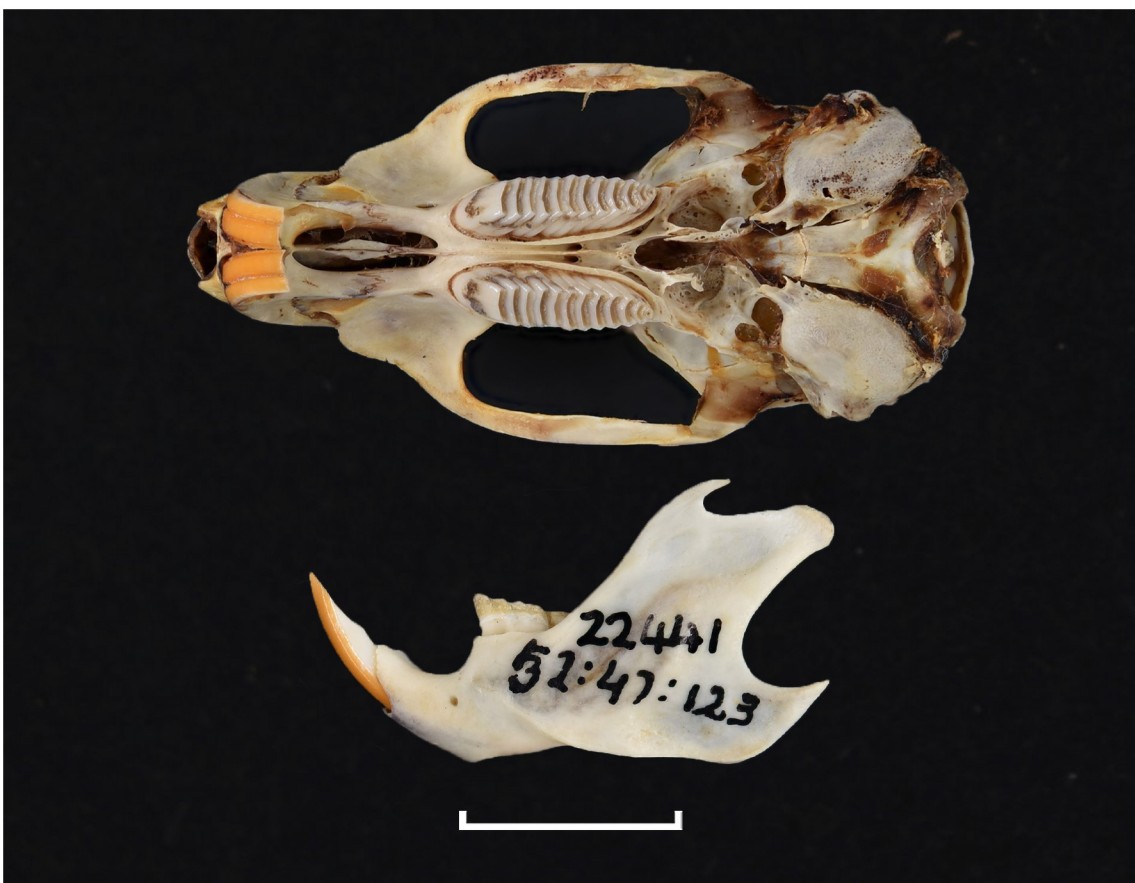

**Fig 51. Cranium of *Otomys angoniensis* (DNMNH-27526), with scale bar of 1 cm.**

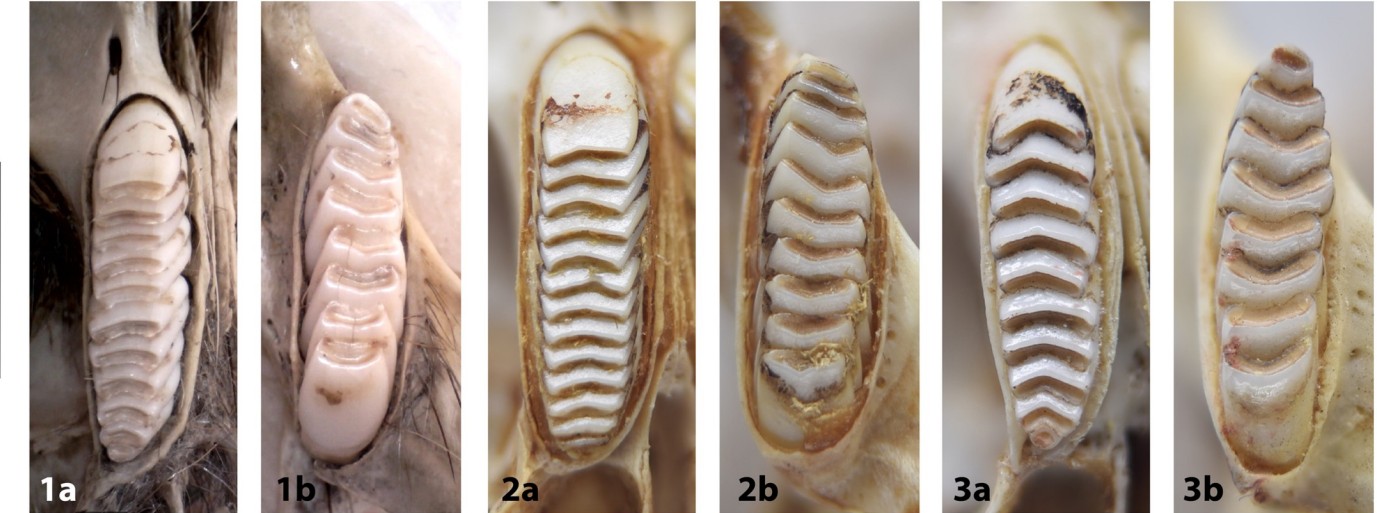

**Fig 52. Cheekteeth of *Otomys*. 1)** Upper (a) and lower (b) right toothrow of *O. angoniensis* (MNHN- ZM-2020-574 BO24); **2)** Upper (a) and lower (b) right toothrow of *O. laminatus* (DNMNH-4647); **3)** Upper (a) and lower (b) right toothrow of *O. sloggetti* (DNMNH-7781).

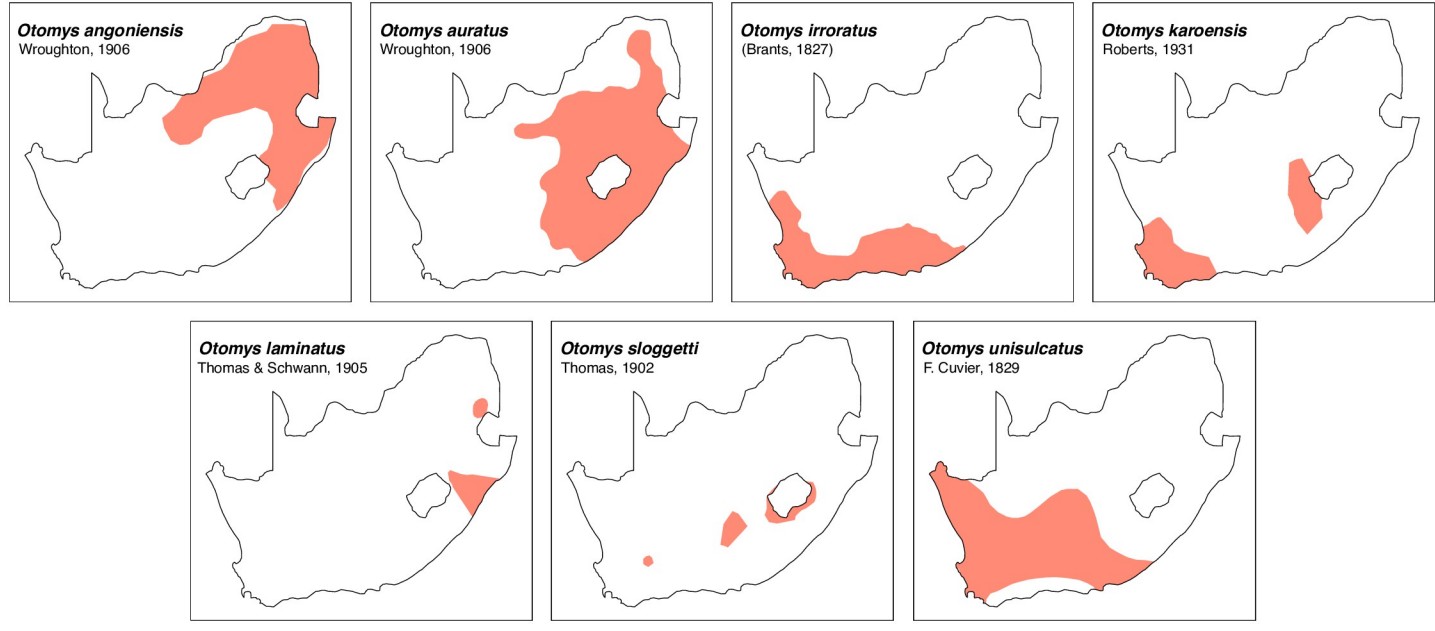

**Fig 53. Distribution maps.**

**Table 18. Dental measurements (in mm) for *Otomys* from South Africa, sexes and species combined.**

|            | Mean | Min | Max  | n  |
|------------|------|-----|------|----|
| LLTR       | 8.7  | 7.6 | 11.1 | 70 |
| $WM_1$     | 2.4  | 1.9 | 3.0  | 70 |
| LUTR       | 9.4  | 7.7 | 11.1 | 69 |
| $WM^1$     | 2.5  | 2.1 | 3.8  | 69 |

**Lower jaw.** Lower incisors have one deep or faint groove, or no groove at all. The $M_1$ is the largest molar of the toothrow, having between three (*O. unisulcatus*) and seven (*O. laminatus*) laminae. The $M_2$ and the $M_3$ have two laminae each. The mandible is stocky with a broad ascending ramus.

**Systematic notes and South African fossil record.** There has been debate about the position of *Otomys* within its own subfamily Otomyinae, but most recent phylogenetic analyses nest it within the Murinae [28, 55]. There has also been debate about the genus boundaries of *Otomys*, *Parotomys*, and *Myotomys*. The latter genus, which comprised *M. unisulcatus* and *M. sloggetti*, has been proved paraphyletic [59]. For this reason, only the two genera, *Otomys* and *Parotomys*, are currently recognized [22]. Recent systematic analyses have revealed multiple cryptic species, which resulted in a considerable increase in the total number of species currently recognized in South Africa:

- *Otomys angoniensis* Wroughton, 1906

- *Otomys auratus* Wroughton, 1906

- *Otomys irroratus* (Brants, 1827)

- *Otomys karoensis* Roberts, 1931

- *Otomys laminatus* Thomas & Schwann, 1905

- *Otomys sloggetti* Thomas, 1902

- *Otomys unisulcatus* F. Cuvier, 1829

Further taxonomic investigation is required for this genus. Additional fossil species have been described from the fossil deposits of South Africa:

- †*Myotomys campbelli* Broom and Schepers, 1946 from the Pliocene deposit of Taung

- †*Otomys gracilis* Broom, 1937 identified in many Pleistocene karstic deposits from the Sterkfontein Valley

The fossil species †*O. gracilis* is regarded by Avery [4] as a synonym of modern *O. saundersiae*. The Otomys fossils need to be revised in light of recent advancements in systematic research.

Genus ***Parotomys*** Thomas, 1918 (Whistling Rats)

Figs 54–56; Table 19

Dental formula is 1-0-0-3:1-0-0-3. Alveolar formula is 4/5-3/4-3/4:4/5-3/4-3/4 (Figs 16 and 17).

**Upper jaw.** Incisors are opisthodont, ungrooved in *P. littledalei* and grooved (a single groove) in *P. brantsii*. The anterior palatal foramina do not reach the $M^1$. Cusps of the cheekteeth are fused into a number of transverse laminae: $M^1$ has three laminae; $M^2$ has two; $M^3$ is the longest cheekteeth, having three (*P. brantsii*) or two *(P. littledalei)* differentiated laminae and two distal laminae fused by enamel ridge (while they remain separate in *Otomys*). Auditory bullae are distinctively inflated (hardly or not inflated in *Otomys*).

**Lower jaw.** Lower incisors are ungrooved. The $M_1$, which is the longest tooth of the toothrow, has four laminae, the first two most anterior of which are joined by an enamel ridge. The $M_2$ and the $M_3$ have two laminae each.

**Systematic notes and South African fossil record.** Currently, there are two species recognized in South Africa:

- *Parotomys brantsii* (A. Smith, 1834)

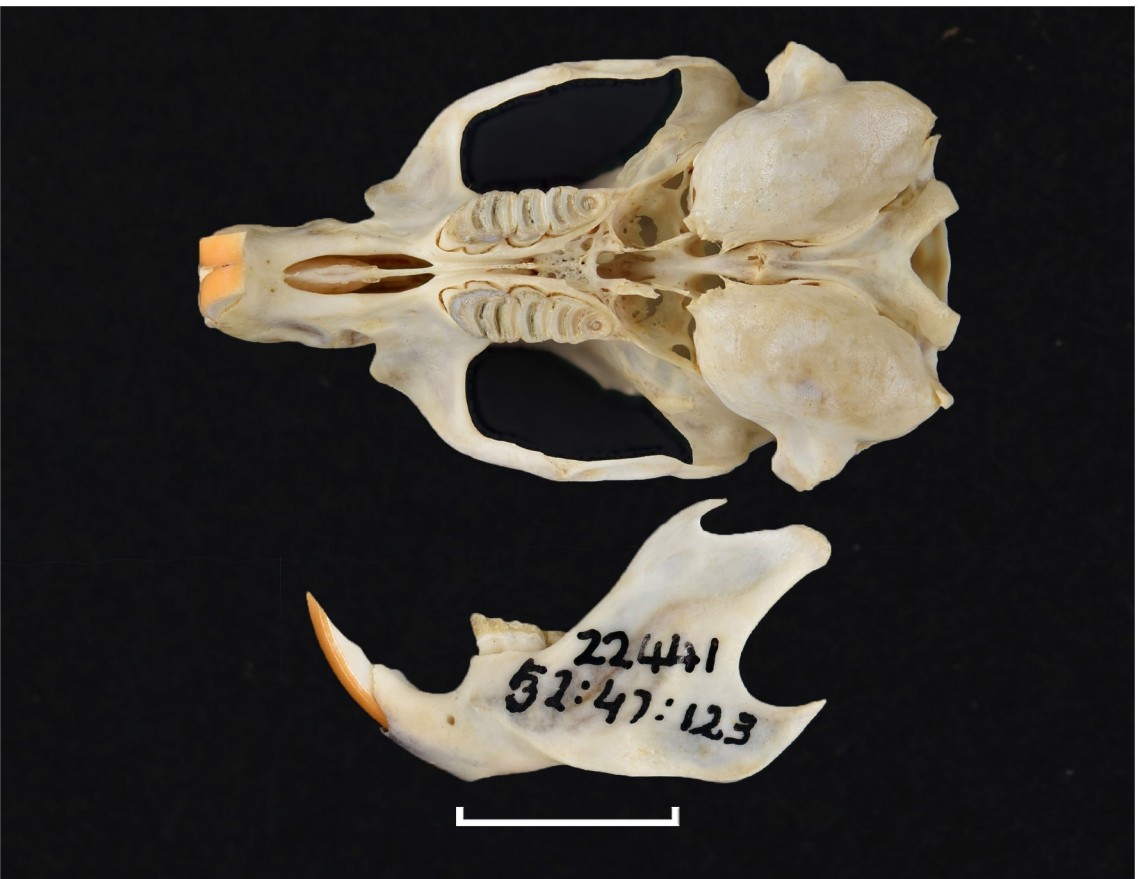

**Fig 54. Cranium of *Parotomys littledalei* (DNMNH-22441), with scale bar of 1 cm.**

- *Parotomys littledalei* (Thomas, 1918)

Fossils of *Parotomys* have been recorded in numerous South African deposits since the Middle Pleistocene [50, 58].

Genus ***Rattus*** Waldheim, 1803

Figs 57–59; Table 20

Dental formula is 1-0-0-3:1-0-0-3. Alveolar formula is 5-4-3/4:4-3/4-3 (Figs 16 and 17).

**Upper jaw.** Upper incisors are smooth and orthodont. The anterior palatal foramina are rather short, barely reaching short of the root of $M^1$. The palate is broad, extending well beyond the $M^3$. The dentition is proportionally small with relatively narrow teeth. The central cusps of the upper molars are pronounced. In the $M^1$, the t1 is displaced backwards to the t2 and connected to it by a ridge, and cusp t7 is absent. In both $M^1$ and $M^2$, t9 is reduced and very close to the t8 (forming a single lamina when worn). In $M^2$, the t3 is small or absent. The $M^3$ displays a prominent t1 but lacks t3, corresponding to Group 3 in Fig 12; it is not markedly smaller than the $M^2$.

**Lower jaw.** The alveolar region of the mandible is well-developed in relation to the whole mandible. Both the $M_1$ and the $M_2$ have a posterior cingulum (often oval-shaped); they lack anterior cusps but up to three small posterolabial cusps may be present.

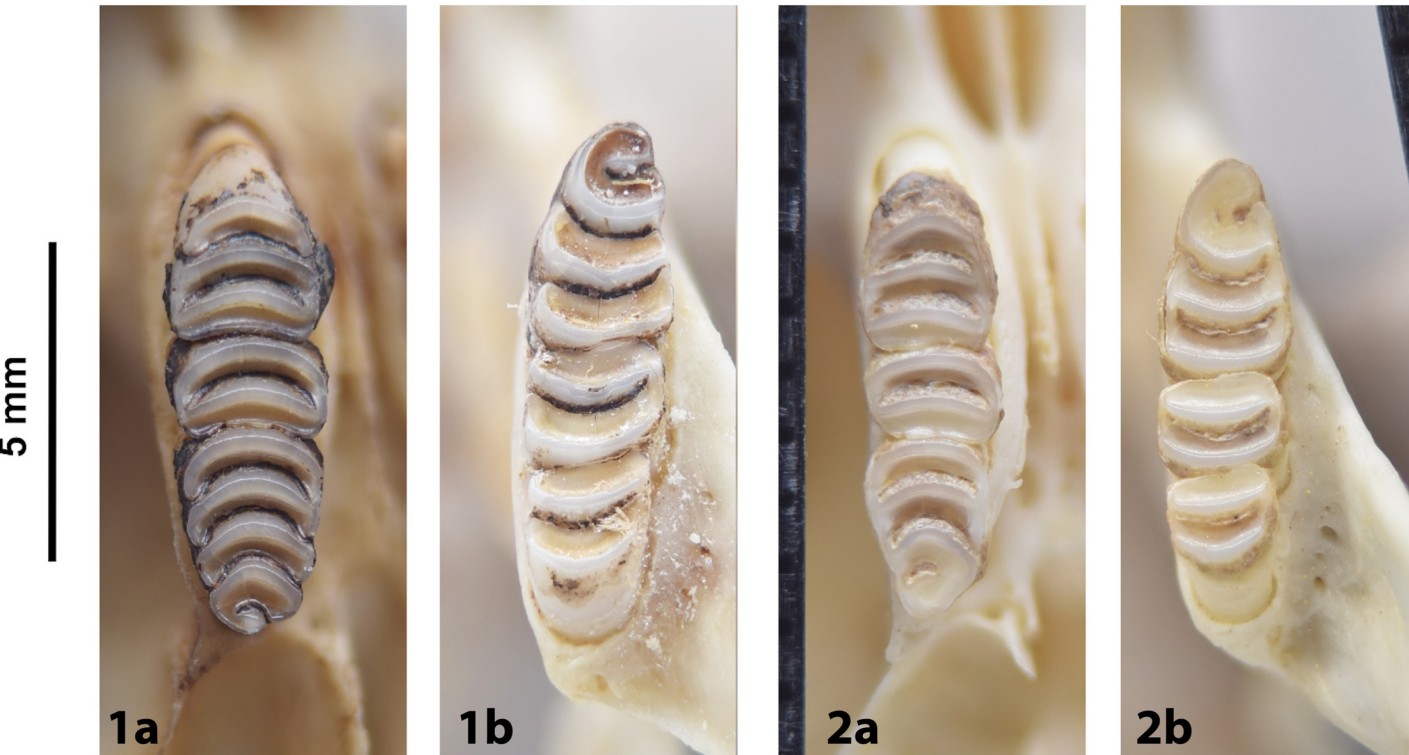

**Fig 55. Cheekteeth of *Parotomys*. 1)** Upper (a) and lower (b) right toothrow of *P. brantsii* (DNMNH-22612); **2)** Upper (a) and lower (b) right toothrow of *P. littledalei* (DNMNH-22446).

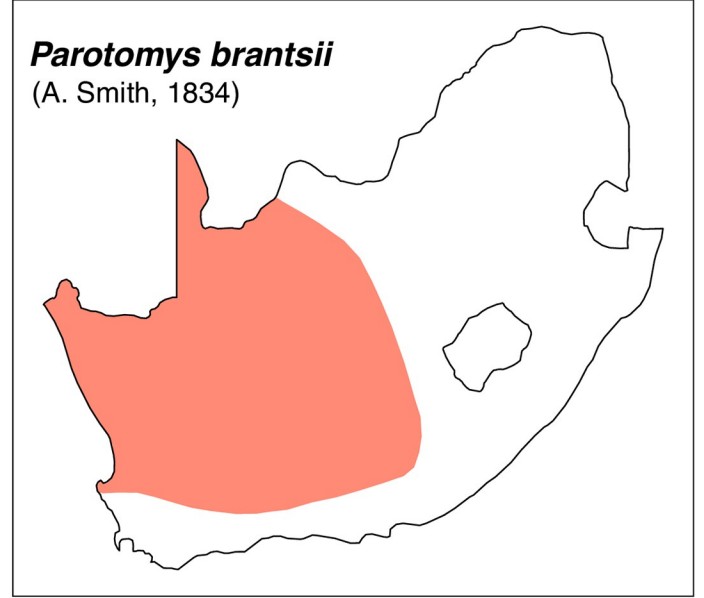
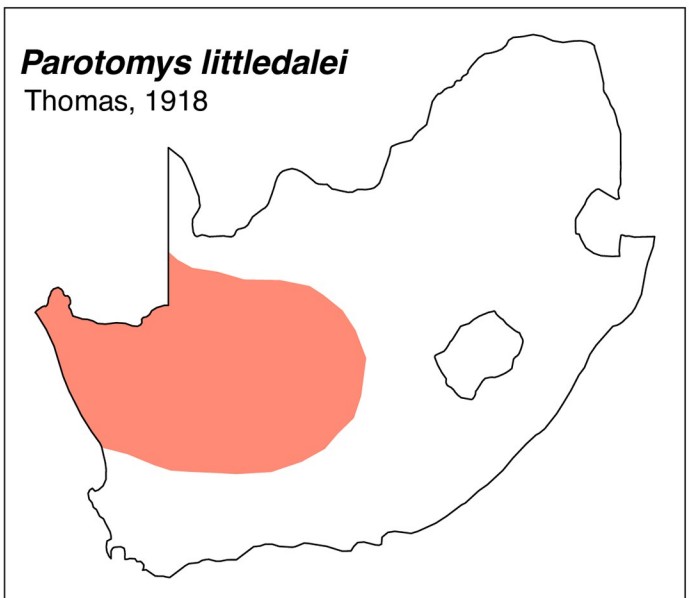

**Fig 56. Distribution maps.**

**Table 19. Dental measurements (in mm) for *Parotomys* from South Africa, sexes and species combined.**

|  | Mean | Min | Max | n |
|---|---|---|---|---|
| LLTR | 7.2 | 6.4 | 8.2 | 23 |
| $WM_1$ | 2.1 | 1.9 | 2.4 | 23 |
| LUTR | 7.7 | 6.6 | 8.5 | 23 |
| $WM^1$ | 2.2 | 2.0 | 2.4 | 23 |

**Systematic notes and South African fossil record.** This genus was unintentionally introduced to South Africa. Three species are currently recognized:

- *Rattus norvegicus* (BERKENHOUT, 1769)

- *Rattus rattus* (LINNAEUS, 1758)

- *Rattus tanezumi* TEMMINCK, 1844

*R. tanezumi* was first discovered in South Africa by Bastos et al. [60] from two sites in Limpopo Province, and has been identified since then from further localities. There is debate about the species status of this taxon, and some authors classify it rather as belonging to a

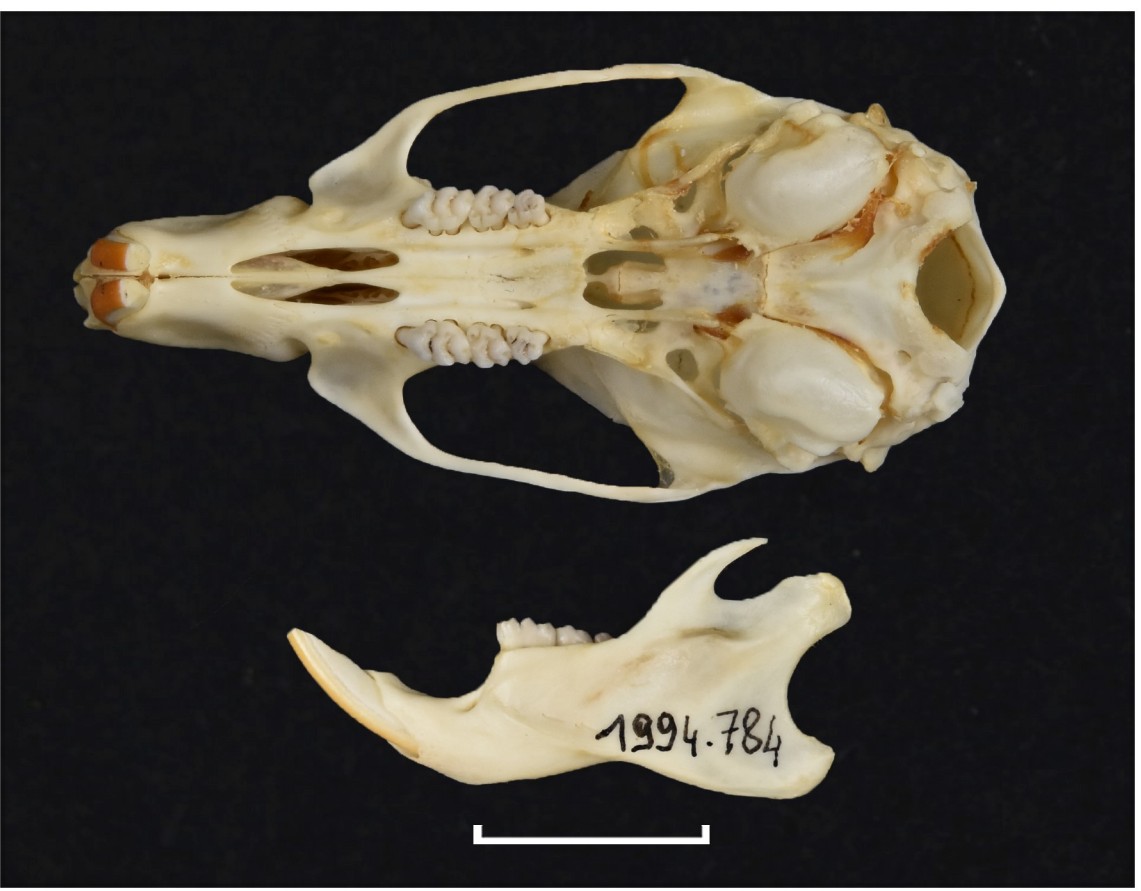

**Fig 57. Cranium of *Rattus rattus* (MNHZM-MO-1994-784), with scale bar of 1 cm.**

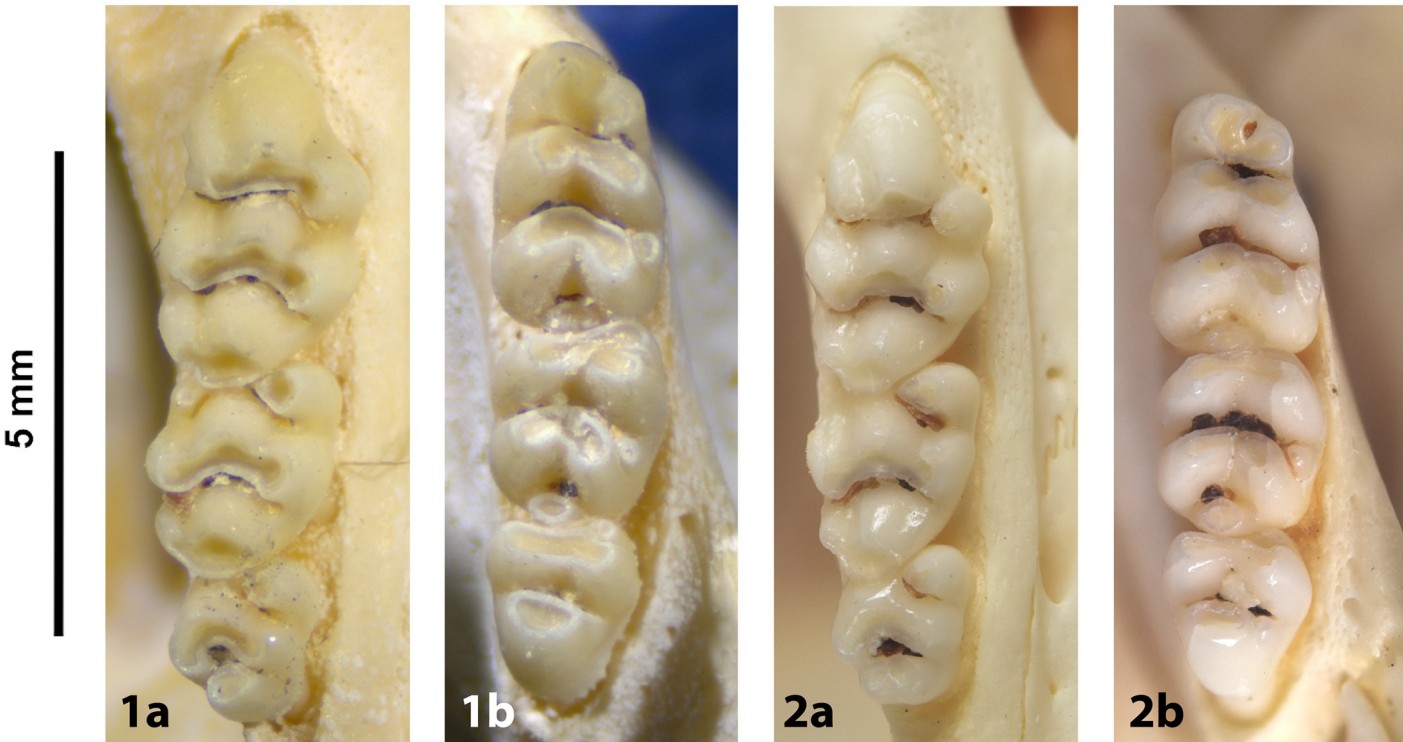

**Fig 58. Cheekteeth of *Rattus*. 1)** Upper (a) and lower (b) right toothrow of *R. norvegicus* (MNHN-ZM-MO-1888-382); **2)** Upper (a) and lower (b) right toothrow of *R. rattus* (MNHN-ZM-MO-1981-352).

lineage of *R. rattus* [61]. The three species *R. norvegicus*, *R. rattus* and *R. tanezumi* are commensal to humans. According to archaeological data, it seems that *R. rattus* followed human migrations into southern Africa during Iron Age [62] and that *R. norvegicus* may have arrived from European ships during the 19[th] century [63].

Genus ***Rhabdomys*** Thomas, 1916 (Four-striped Grass Mice)

Figs 60–62; Table 21

Dental formula is 1-0-0-3:1-0-0-3. Alveolar formula is 5-5-3:4-4/5-3 (Figs 16 and 17).

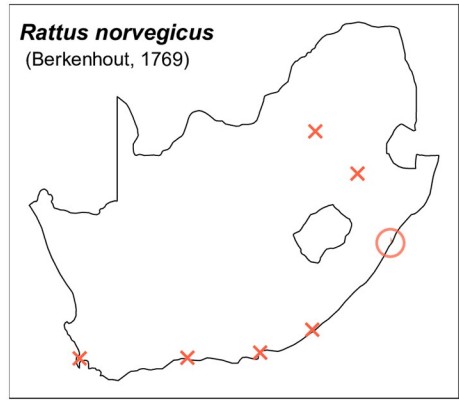
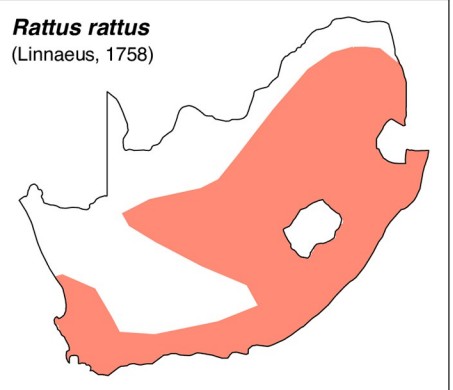
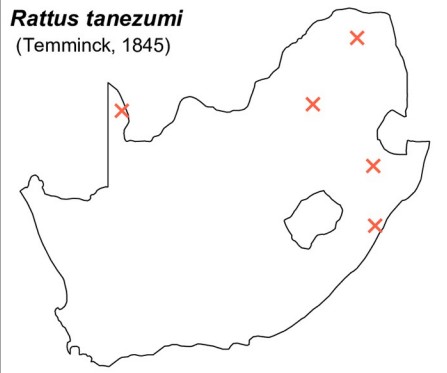

**Fig 59. Distribution maps.**

**Table 20. Dental measurements (in mm) for *Rattus* from South Africa, sexes and species combined.**

|        | Mean | Min | Max | n  |
|--------|------|-----|-----|----|
| LLTR   | 6.5  | 5.5 | 7.5 | 68 |
| WM$_1$ | 1.8  | 1.5 | 2.0 | 67 |
| LUTR   | 6.6  | 5.7 | 7.6 | 70 |
| WM$^1$ | 2.0  | 1.7 | 2.3 | 69 |

**Upper jaw.** Upper incisors are ungrooved and orthodont to opisthodont. The anterior palatal foramina reach just to the anterior root alveolus of the M$^1$. Upper cheekteeth display large central cusps. In the M$^1$, the t1 is located behind t2 and t3, and t4 behind t5; the second and third rows of cusps appear to be linked on the lingual and labial sides. In both the M$^1$ and M$^2$, the t9 is small and often reduced to a small ridge that connects with the t6. The M$^3$ lacks

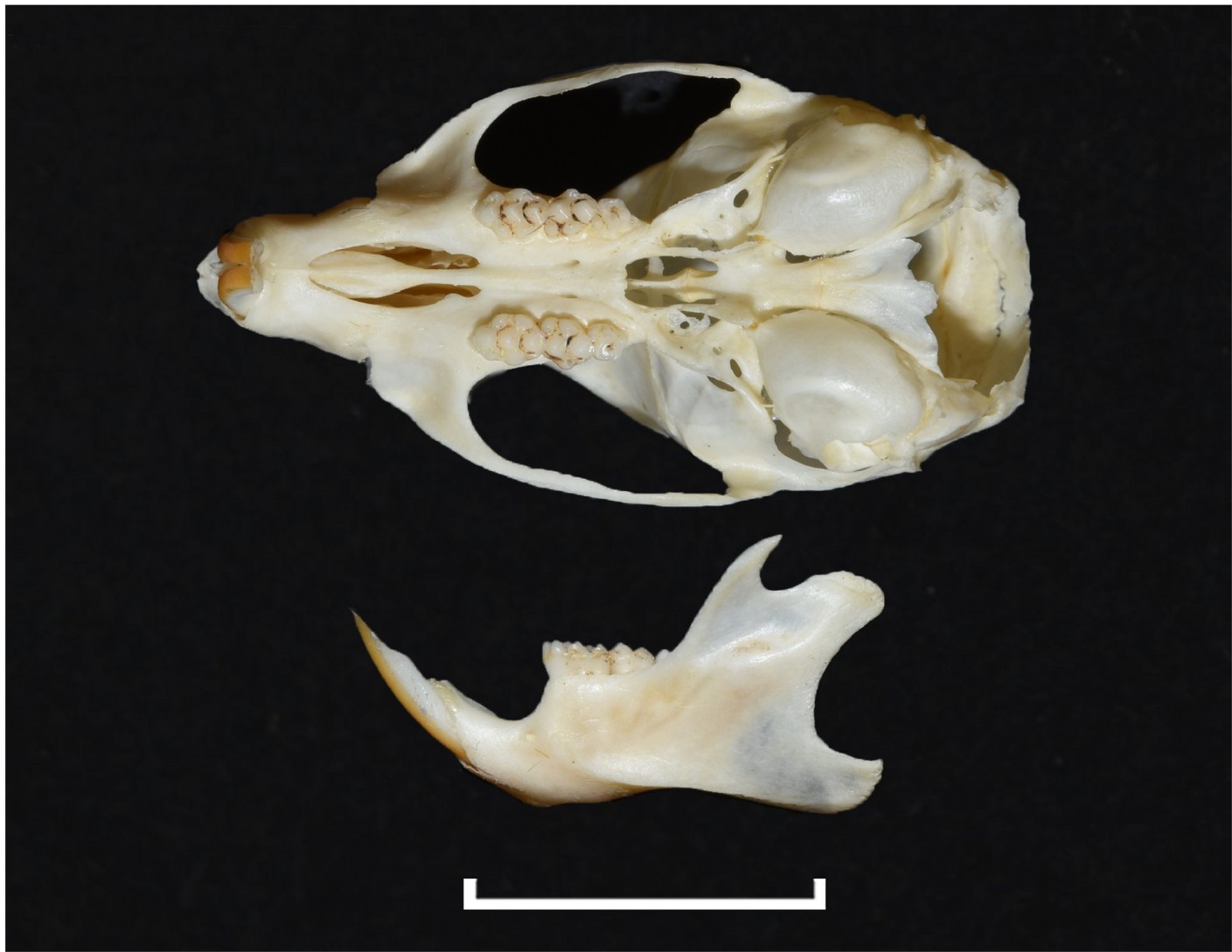

**Fig 60. Cranium of *Rhabdomys dilectus* (IVB-M-T8x353), with scale bar of 1 cm.**

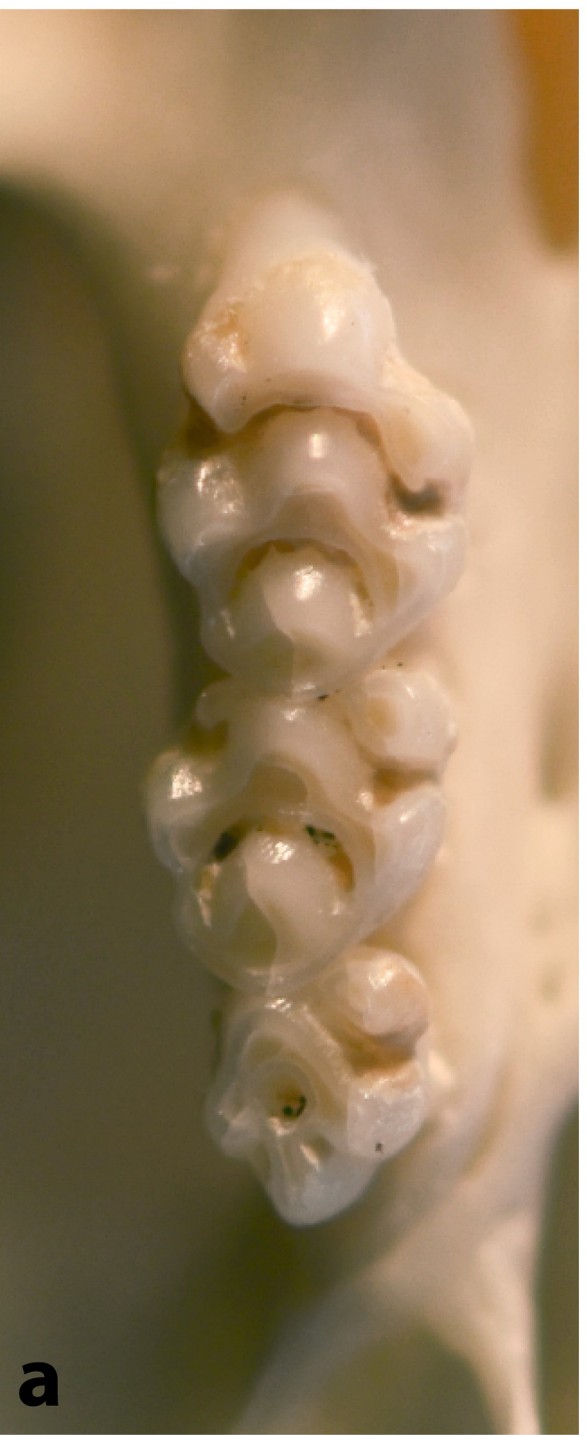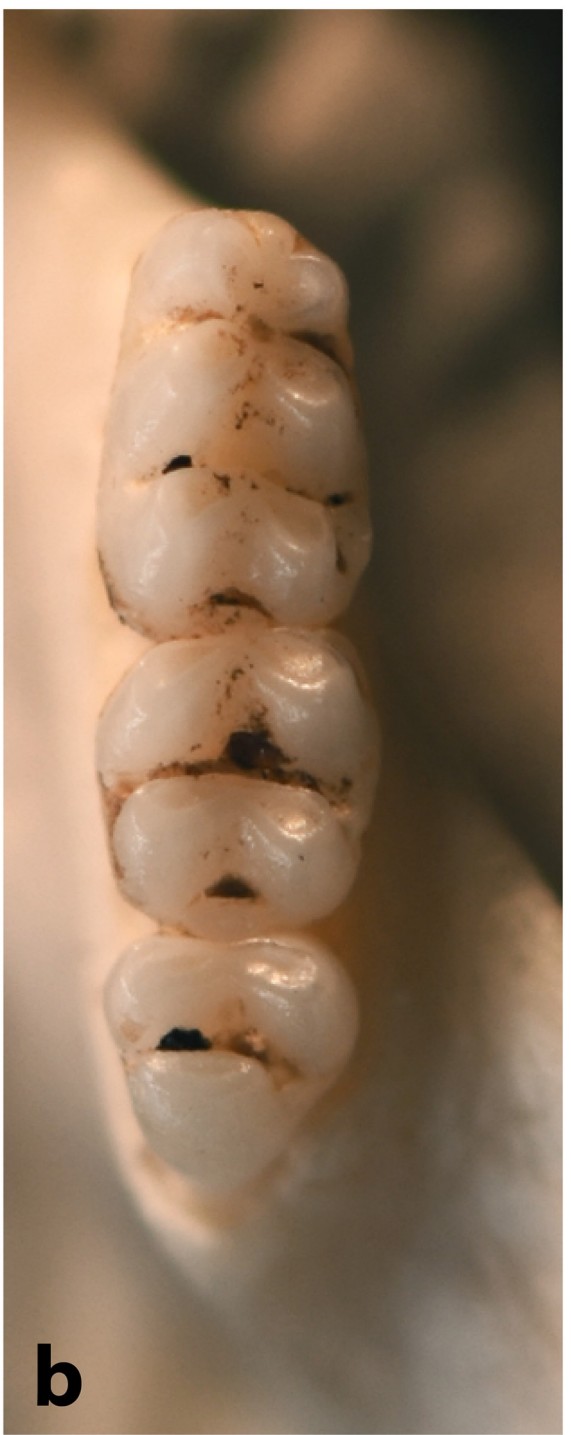

**Fig 61. Upper (a) and lower (b) right toothrow of *R. dilectus* (upper IVB-T8x336; lower IVB-T8x353).**

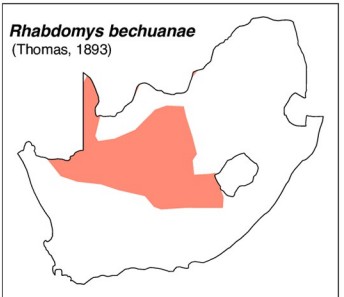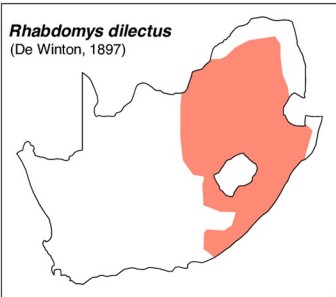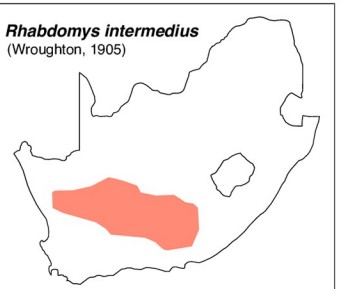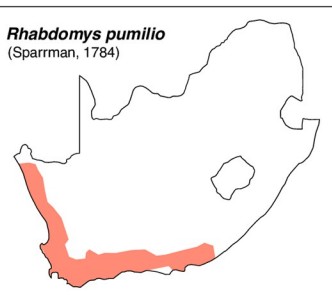

**Fig 62. Distribution maps.**

**Table 21. Dental measurements (in mm) for *Rhabdomys* from South Africa, sexes and species combined.**

|         | Mean | Min | Max | n  |
|---------|------|-----|-----|----|
| LLTR    | 4.5  | 4.3 | 4.8 | 20 |
| $WM_1$  | 1.2  | 1.1 | 1.3 | 20 |
| LUTR    | 4.6  | 4.3 | 4.8 | 20 |
| $WM^1$  | 1.5  | 1.3 | 1.7 | 20 |

the t3 and has one distal cusp. Length of upper cheekteeth is similar to *Mastomys* and *Myomyscus*, but the number of roots and alveoli is different ($M^1$ is 5-rooted in *Rhabdomys*, 3-rooted in *Mastomys* and *Myomyscus*).

**Lower jaw.** The $M_1$ has two anterior cusps and an additional small tma is often present. Both $M_1$ and $M_2$ have labial cusplets and reduced to absent posterior cingulum. The lower dentition is of similar size than *Mastomys* but both genera can be distinguished almost unambiguously on the basis of the number of roots and corresponding alveoli: the $M_1$ of *Rhabdomys* is 4-rooted, with the central pair of roots usually visible on the sides of the tooth *in situ*, while the $M_1$ of *Mastomys* is 2-rooted.

**Systematic notes and South African fossil record.** Until recently, the genus was considered as monotypic [21] with *R. pumilio* being the only known species, but new karyotypic and genotypic analyses revealed a more complex diversity [64, 65]. Today, four species are recognized in South Africa:

- *Rhabdomys bechuanae* (THOMAS, 1893)

- *Rhabdomys dilectus* (DE WINTON, 1897)

- *Rhabdomys intermedius* (WROUGHTON, 1905)

- *Rhabdomys pumilio* (SPARMAN, 1784)

The oldest known *Rhabdomys* have been discovered around 5 MYA in Langebaanweg [66] and 3.3 MYA in Makapansgat [48]. Fossils are known from a variety of Pleistocene localities. This material was identified as *R. pumilio* at a time when the specific diversity of *Rhabdomys* had not been recognized and is likely to include material attributable to other species.

Genus ***Thallomys*** Thomas, 1920 (Acacia Rats or Tree Rats)

Figs 63–65; Table 22

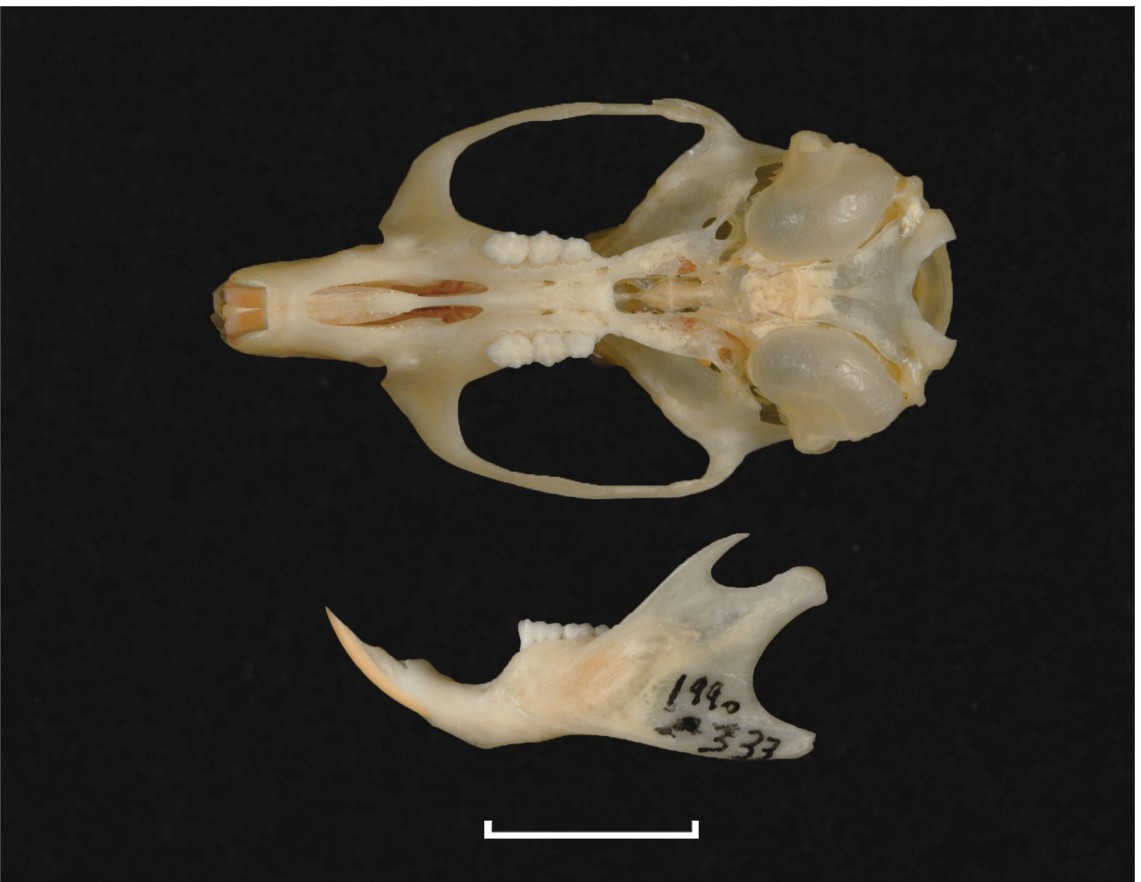

**Fig 63. Cranium of *Thallomys paedulcus* (MNHN-ZM-MO-1990-333), with scale bar of 1 cm.**

Dental formula is 1-0-0-3:1-0-0-3. Alveolar formula is 5-4/5-3/4:4/5-4-3 (Figs 16 and 17).

**Upper jaw.** Incisors are orthodont and ungrooved. The palatal foramina are large and penetrate between the second rows of cusps on $M^1$. Molars have an embossed, angular appearance, and show microdonty. In the $M^1$, the t7 is absent or much reduced, and there are small stephanodont crests on t1 and t3 and joining t6 with t9. The central cusps (t2, t5 and t8) are prominent. In the $M^2$ the cusps t2 and t7 are absent. The molar $M^3$ is the smallest of the toothrow, but not greatly reduced; it corresponds to Group 6 in Fig 12.

**Lower jaw.** Cusps of lower molars are prominent and sharply defined. The $M_1$ and $M_2$ have equal-sized posterior cingula, and a longitudinal crest that connects the lobes on the labial side. The $M_1$ often displays small labial extra cusplets or ridges, and sometimes a small antero-median cusplet. Both the $M_2$ and $M_3$ have antero-external cusplets. The toothrow is relatively small compared to the size of the mandible (microdonty).

**Systematic notes and South African fossil record.** According to Monadjem et al. [22], this genus requires urgent revision. Currently, three species are listed in South Africa:

- *Thallomys nigricauda* THOMAS, 1882

- *Thallomys paedulcus* (SUNDEVALL, 1846)

- *Thallomys shortridgei* THOMAS AND HINTON, 1923

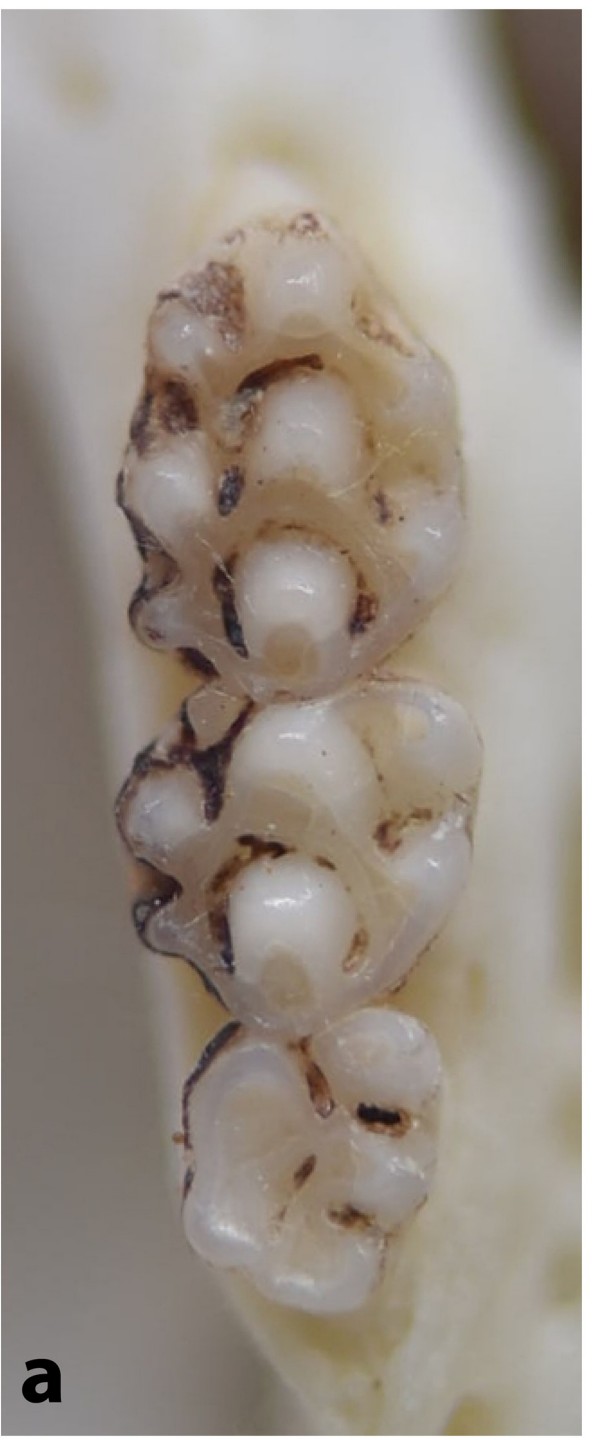
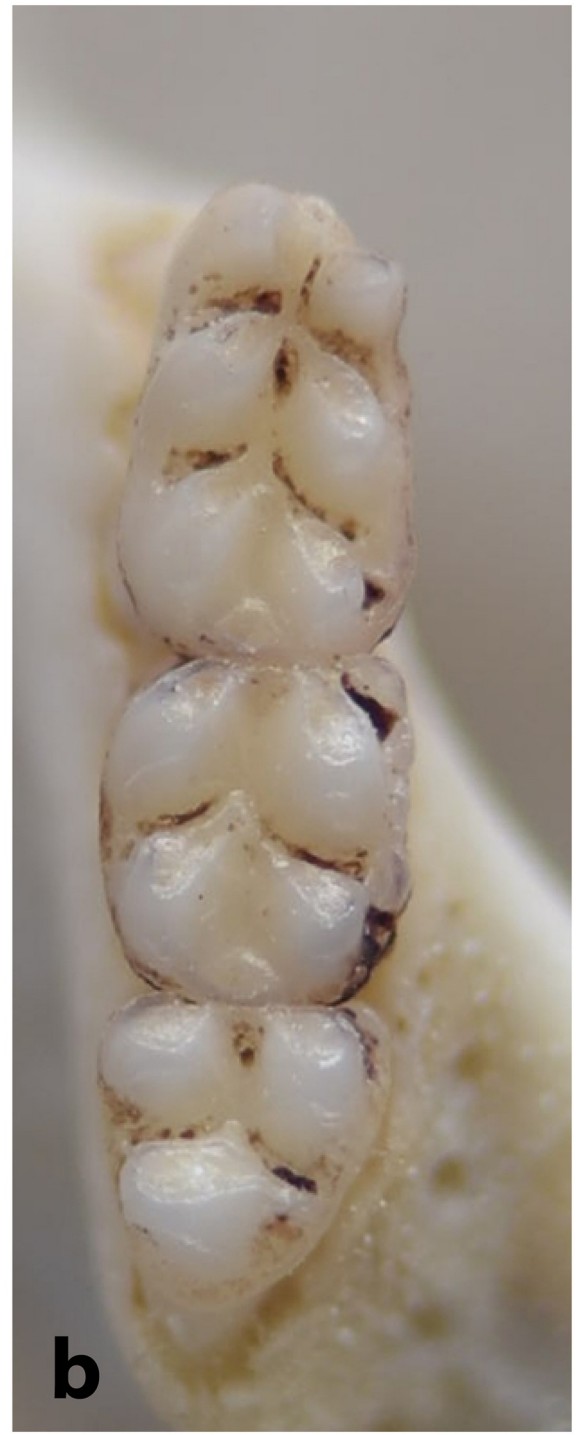

**Fig 64. Upper (a) and lower (b) right toothrow of *T. paedulcus* (DNMNH-30229).**

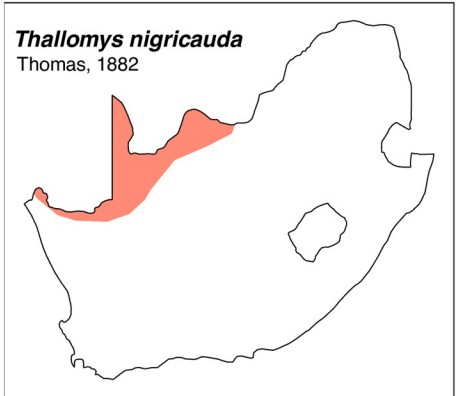 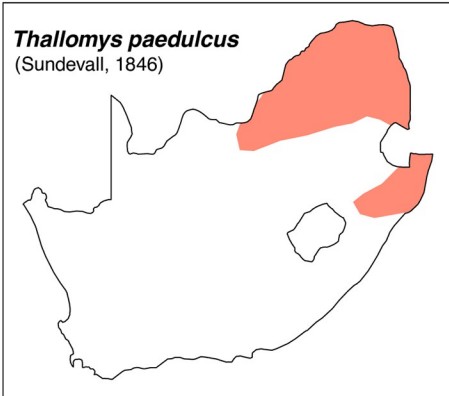 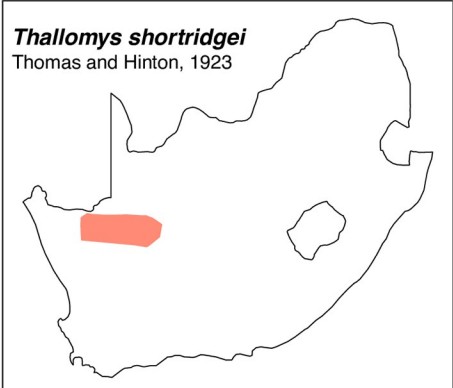

**Fig 65. Distribution maps.**

**Table 22. Dental measurements (in mm) for *Thallomys* from South Africa, sexes and species combined.**

|  | Mean | Min | Max | n |
|---|---|---|---|---|
| LLTR | 5.4 | 4.6 | 6.0 | 21 |
| $WM_1$ | 1.4 | 1.2 | 1.6 | 21 |
| LUTR | 5.5 | 4.7 | 6.3 | 21 |
| $WM^1$ | 1.6 | 1.4 | 1.8 | 21 |

An additional fossil species has been described in South Africa:

- *† Thallomys debruyni* Broom, 1948

The oldest known fossil of *Thallomys* was found in Langebaanweg around 5MYA [66]. Fossils of this genus are known from various Pleistocene localities in South Africa.

Genus ***Zelotomys*** Osgood, 1910 (Broad-headed Mice)

Figs 66–68; Table 23

Dental formula is 1-0-0-3:1-0-0-3. Alveolar formula is 3-3-2/3:2-2-2 (Figs 16 and 17).

**Upper jaw.** Upper incisors are ungrooved and opisthodont. The palatal foramina reach the second root of the three-rooted $M_1$. The labial row of cusps is well developed and tend to project outwards in a characteristic way. In the $M^1$, t1 and t2 tend to fuse with wear. As with other members of the Praomyiny tribe (*Mastomys* and *Myomyscus*), the t9 is well separated from the t6, and located far in the labial side. The $M^2$ is as broad as or broader than long, with a low t9 and a t3 very small, often reduced to a tiny process. The $M^3$ is markedly reduced, with traces of two laminae (group 5 in Fig 12) and a big t1.

**Lower jaw.** Lower incisors are plain and ungrooved; they are relatively strong, with the alveolar region of the mandible well-developed, and generally extend far beyond the alveolus. The prelobe of the $M_1$ has two nearly longitudinal, poorly differentiated cusps that are connected to those of the second lobe, together forming a characteristic compact "trefoil" like pattern, while the third lobe is located further below. There is a small cingulum posterior on $M_1$ and $M_2$. The $M_3$ is reduced.

**Systematic notes and South African fossil record.** Currently, a single species is recognised in South Africa:

- *Zelotomys woosnami* (Schwann, 1906)

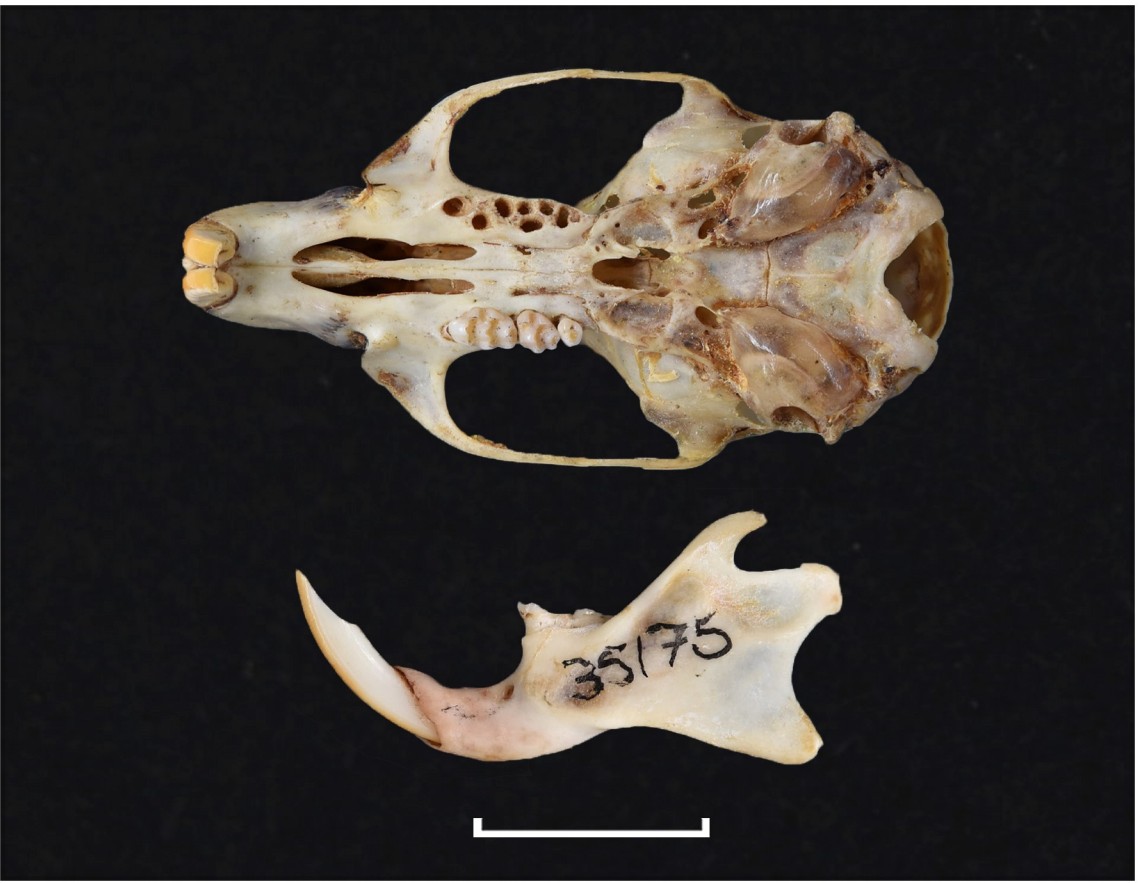

**Fig 66. Cranium** *of Zelotomys woosnami* **(DNMNH-30229), with scale bar of 1 cm.**

The oldest known fossil of *Zelotomys* was found in Langebaanweg around 5MYA [66]. Fossils of this genus are known from various Pleistocene localities in South Africa.

Family **NESOMYIDAE** Major, 1897

Subfamily **CRICETOMYINAE** Roberts, 1951

Genus *Cricetomys* Waterhouse, 1840 (Giant Pouched Rats)

Figs 69–71; Table 24

Dental formula is 1-0-0-3:1-0-0-3. Alveolar formula is 3-3-3:2-2/3-2/3 (Figs 16 and 17).

**Upper jaw.** Upper incisors are opisthodont and ungrooved but display characteristic striations on their anterior enamel surface, forming a slightly raised band (this feature is also present in *Saccostomus* and *Mystromys*). The anterior palatal foramina are short, located before the basis of the zygomatic arches. Molars are large, with high, well-separated cusps. The $M^1$ has three lobes, The $M^2$ and $M^3$ two lobes each. In the first lobe of $M^1$, the t2 and t3 are large and connected, being well separated from the t1 which is smaller and isolated below; similarly in the second lobe, the t5 and t6 are large and separated from the t4; the third lobe has no t7, but a large posterior cingulum. The $M^2$ has two lobes, with t4 separated from t5 and t6, and a conspicuous posterior cingulum. The $M^3$ is almost as large as the $M^2$ and corresponds to Group 1 in Fig 12.

**Lower jaw.** Lower incisors are ungrooved but display the same ridged band as the upper incisors. The molar $M_1$ has three lobes: the first lobe has one elongated cusp, the second and

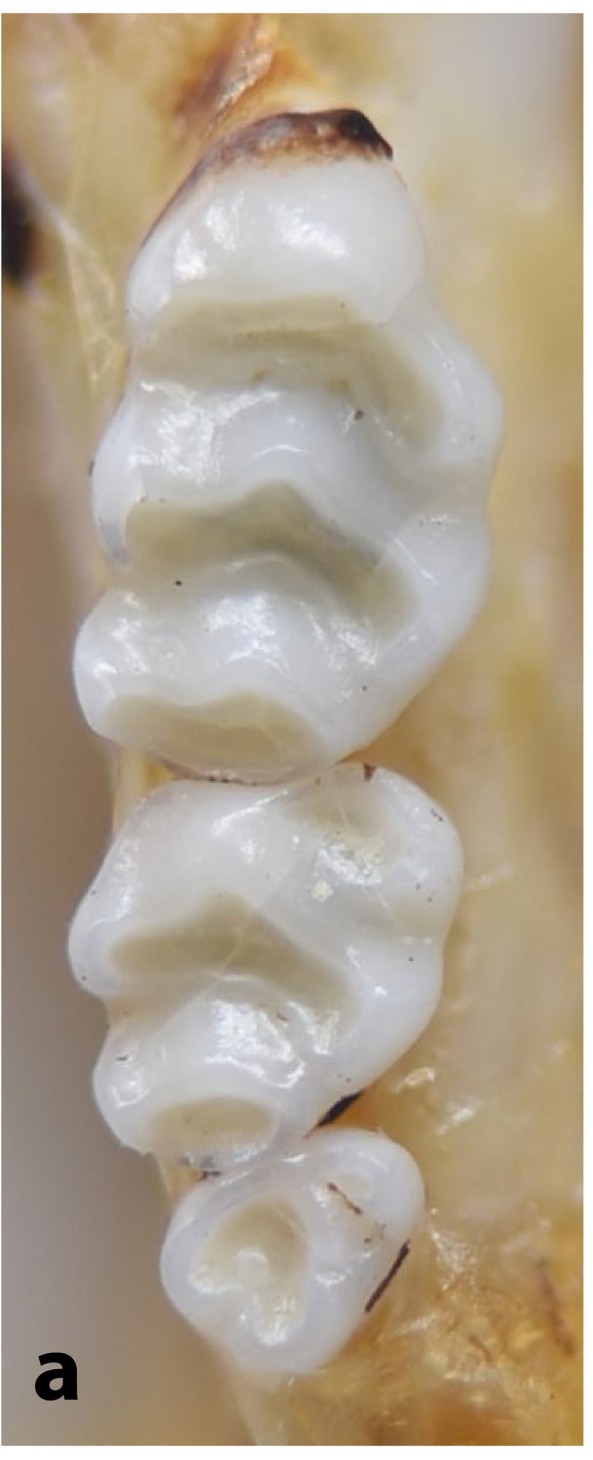 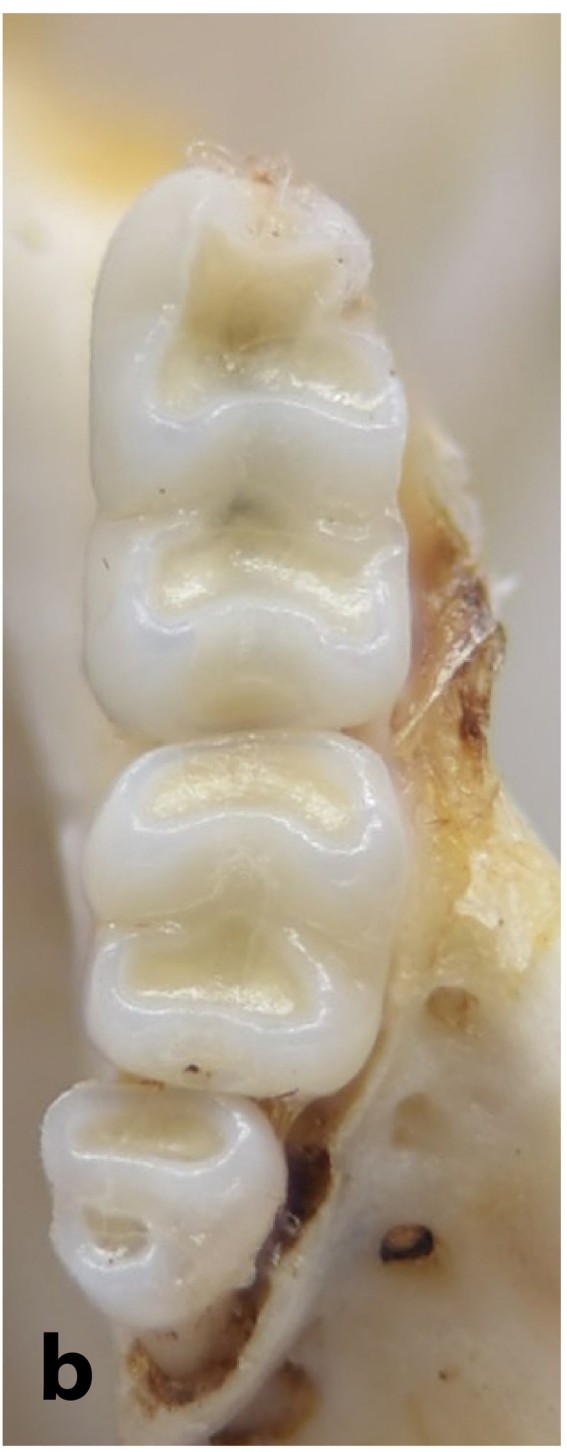

**Fig 67. Upper (a) and lower (b) right toothrow of *Z. woosnami* (upper DNMNH-6413; lower DNMNH-35175).**

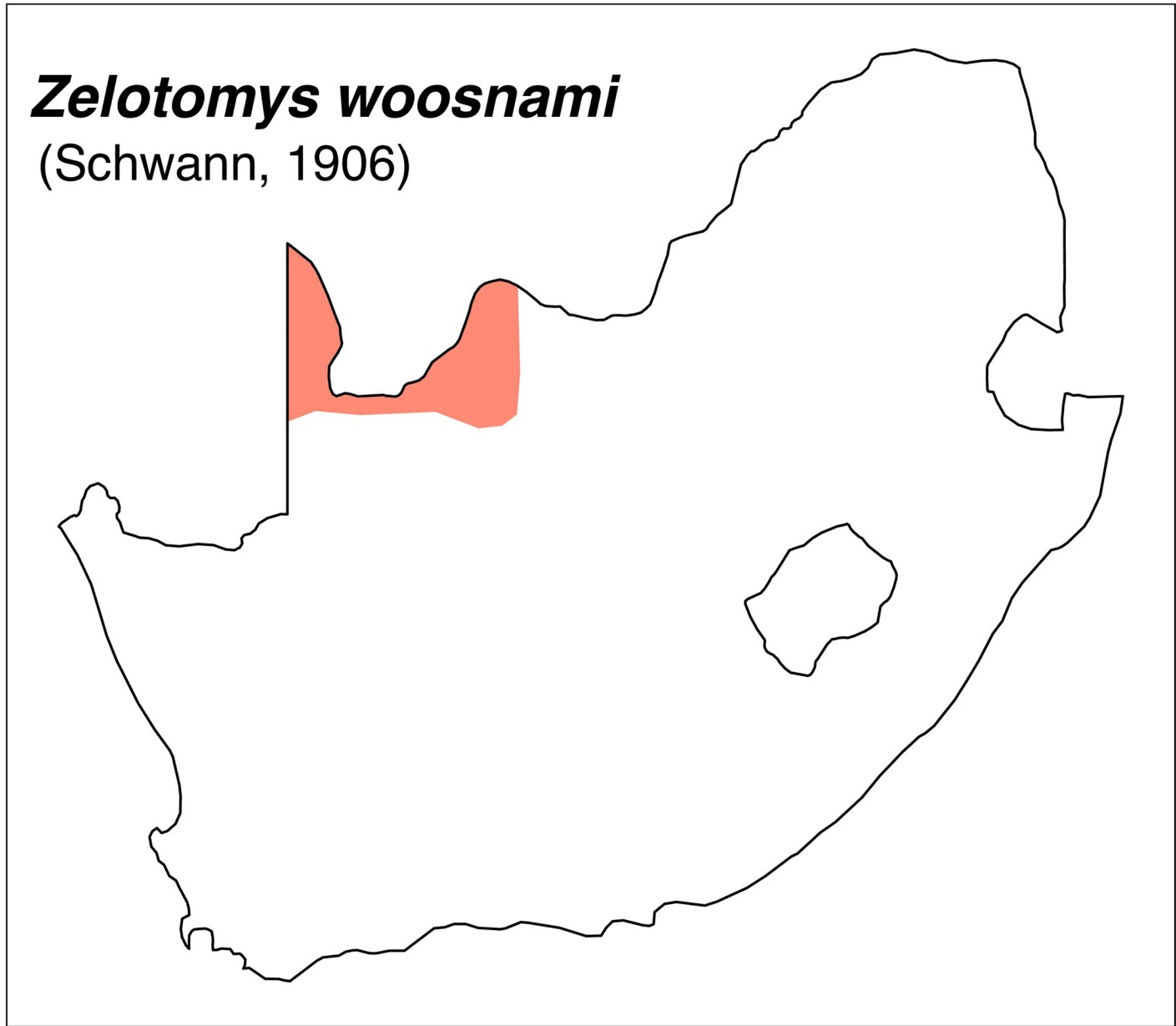

**Fig 68. Distribution map.**

**Table 23. Dental measurements (in mm) for *Zelotomys woosnami*, sexes combined.**

|  | Mean | Min | Max | n |
|---|---|---|---|---|
| LLTR | 5.1 | 5.0 | 5.3 | 10 |
| WM$_1$ | 1.4 | 1.3 | 1.5 | 10 |
| LUTR | 5.3 | 4.9 | 5.7 | 10 |
| WM$^1$ | 1.8 | 1.7 | 1.9 | 10 |

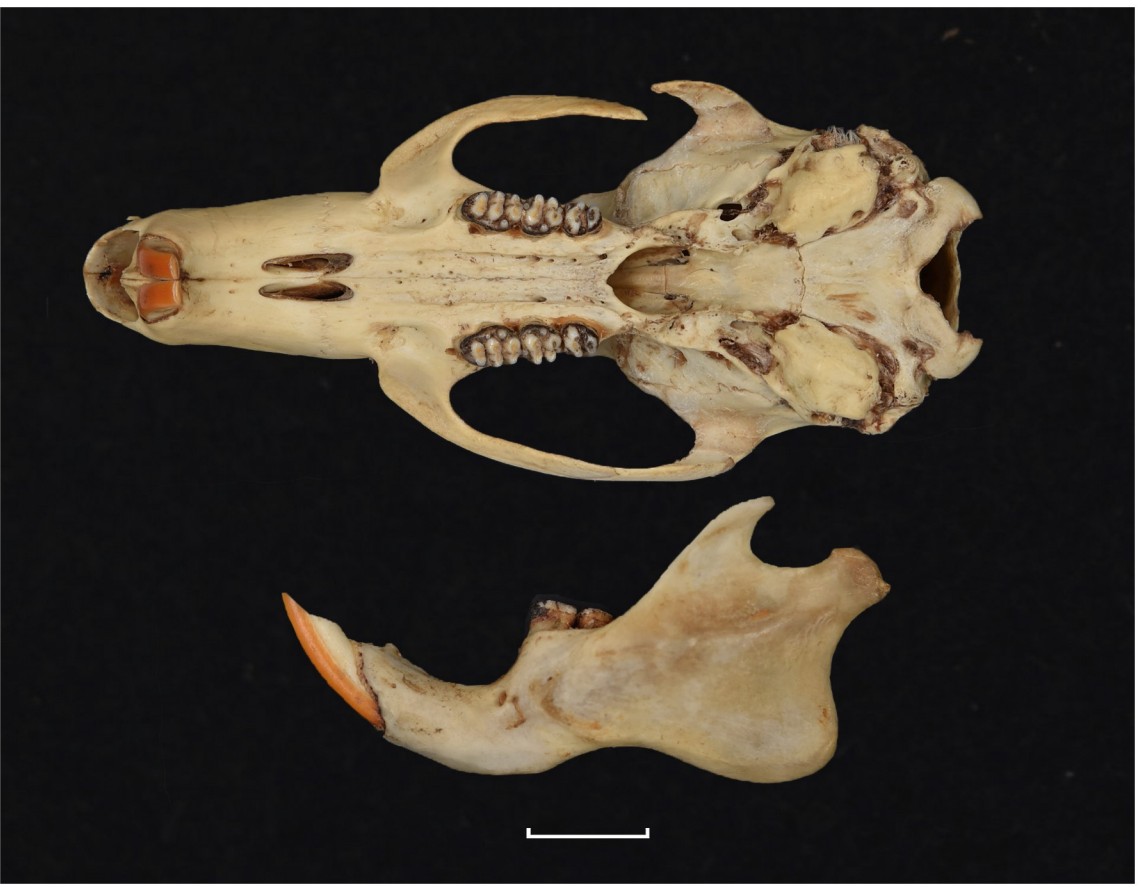

**Fig 69. Cranium of *Cricetomys ansorgei* (DNMNH-13954), with scale bar of 1 cm.**

third lobes have two cusps each that are aligned transversally. It has a well-developed posterior cingulum and a variable number (generally 2) of supplementary cusps on the labial side. As in *Saccostomus* (another member of the subfamily Cricetomyinae), the coronoid process is well-developed while the angular process is not projected backwards.

**Systematic notes and South African fossil record.** South African specimens of *Cricetomys* were previously attributed to *C. gambianus* WATERHOUSE, 1840, but recent morphological and molecular analyses revealed the existence of multiple species [67]. Today, the specimens recorded in Southern Africa are attributed to a single species:

• *Cricetomys ansorgei* THOMAS, 1904

Fossils of *Cricetomys* have been identified from very few Quaternary deposits in South Africa. Its oldest record is from the Late Pleistocene from Sibudu [68] and Rose Cottage Cave [69].

Genus *Saccostomus* Peters, 1846 (Pouched Mice)

Figs 72–74; Table 25

Dental formula is 1-0-0-3:1-0-0-3. Alveolar formula is 3-3-3:2-2-2 (Figs 16 and 17).

**Upper jaw.** Upper incisors are opisthodont and ungrooved, but they display characteristic striations on their anterior enamel surface, forming a slightly raised band (this feature is also present in *Cricetomys* and *Mystromys*). The anterior palatal foramina reach the anterior root of the first upper molar. Molars have bulbous cusps that are connected transversally. The M$^1$ has

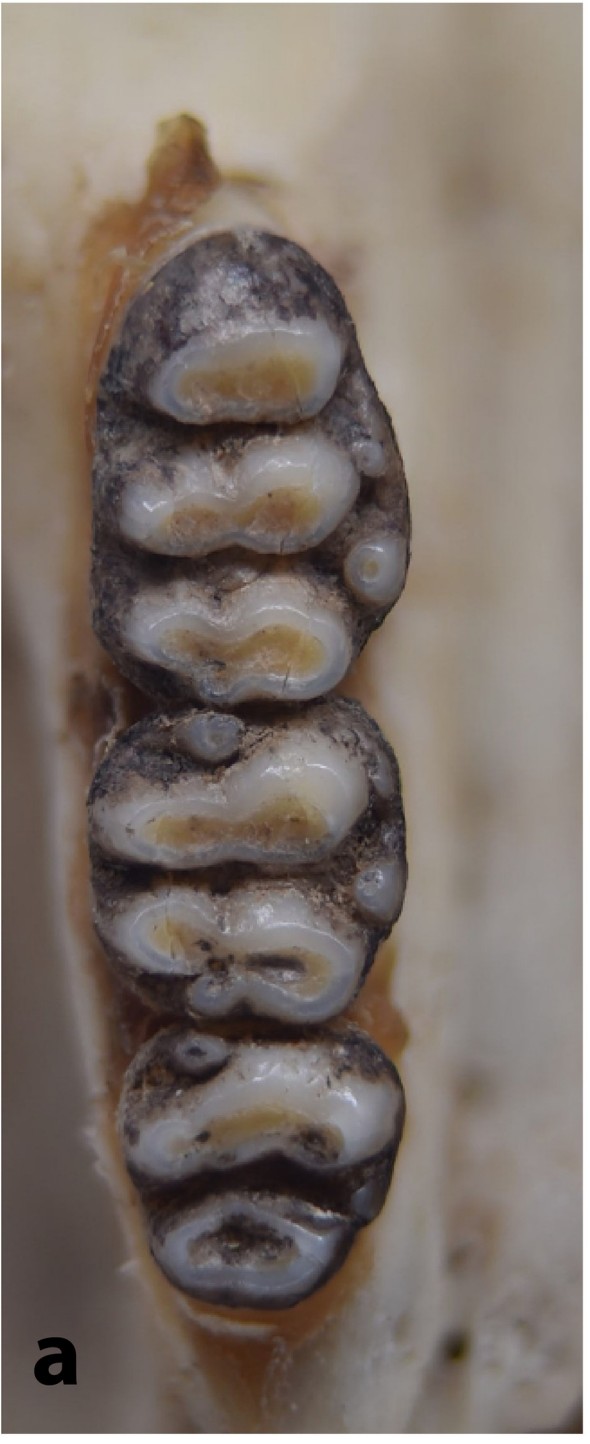
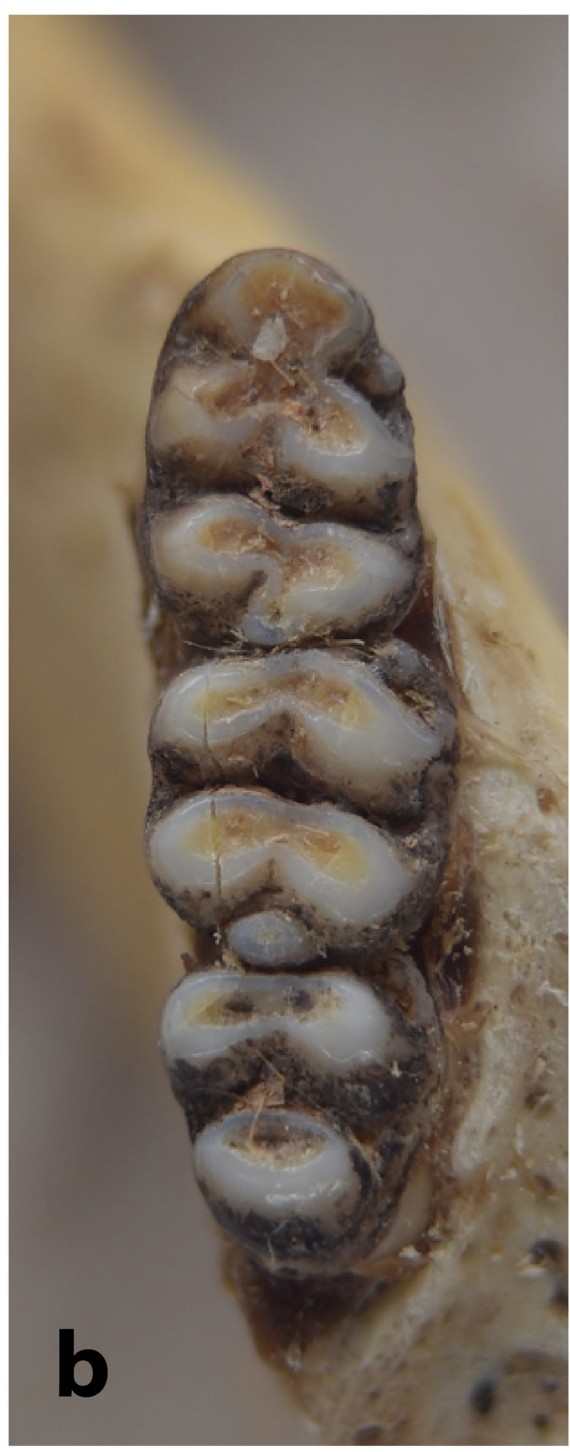

**Fig 70. Upper (a) and lower (b) right toothrow of *C. ansorgei* (DNMNH-30736).**

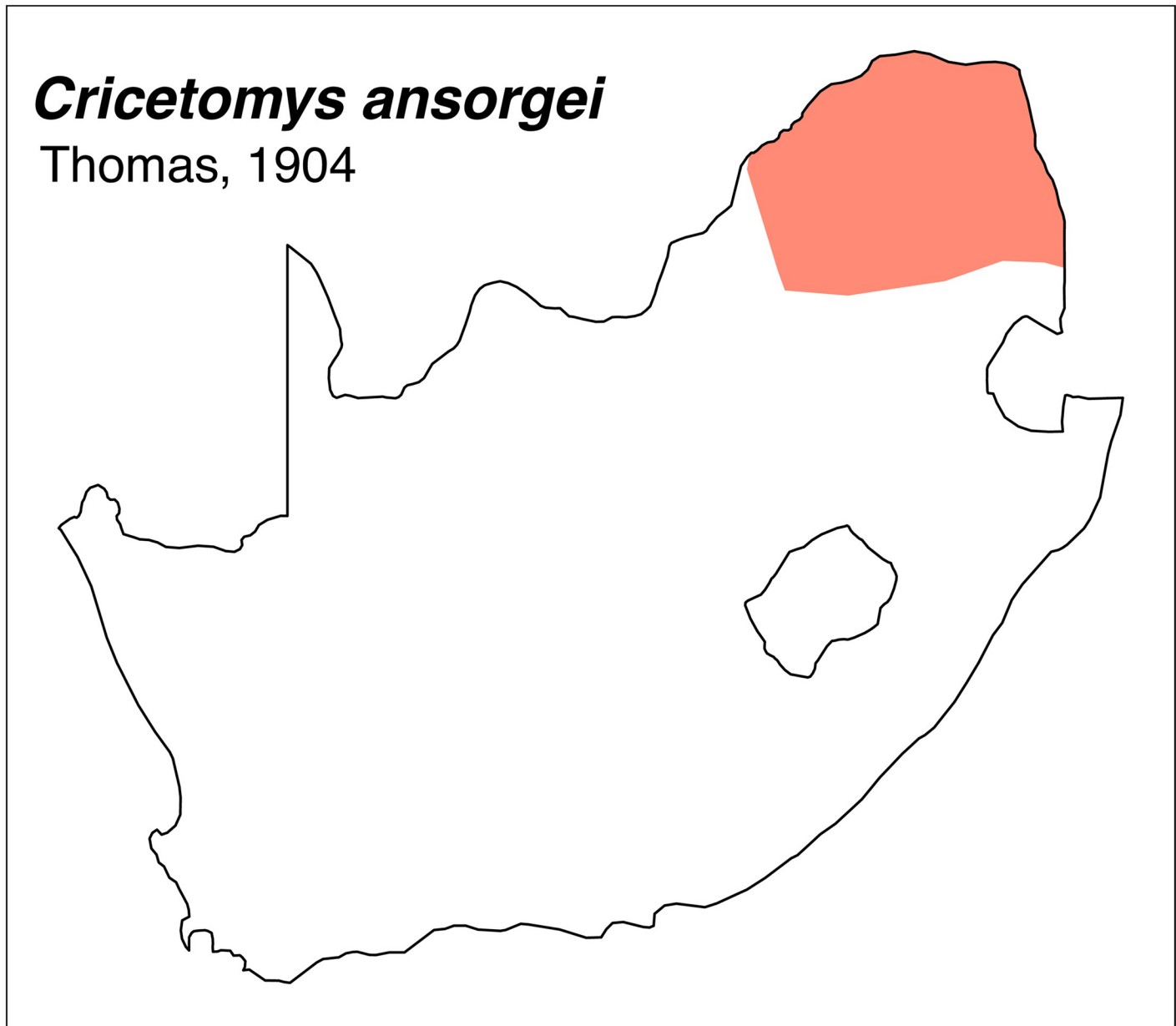

**Fig 71. Distribution map.**

Table 24. Dental measurements (in mm) for *Cricetomys ansorgei*, sexes combined.

| | Mean | Min | Max | n |
|---|---|---|---|---|
| LLTR | 10.7 | 10.3 | 11.1 | 3 |
| $WM_1$ | 3.0 | 2.9 | 3.1 | 3 |
| LUTR | 11.3 | 10.9 | 11.6 | 3 |
| $WM^1$ | 3.3 | 3.2 | 3.6 | 3 |

three lobes with only the outer two cusps in the first row (it lacks the t1), and three cusps on the second and third rows. Both the $M^1$ and the $M^2$ have a relatively well-developed posterior cingulum. The $M^2$ and $M^3$ have two lobes each. The $M^3$ is not greatly reduced, as in the Dendromurinae; it corresponds to Group 1 in Fig 12.

**Lower jaw.**   Lower incisors are ungrooved but display the same ridged band than in the upper incisors. The $M_1$ has three lobes and a small posterior cingulum. The first lobe consists of a single elongated cusp (sometimes two poorly differentiated cusps are visible), which may be connected to the second row of cusps. The second and third lobes have two fused cusps each. The $M_2$ has two lobes, with an antero-external cusps of variable size and a posterior cingulum. The $M_3$ has two lobes with a tiny antero-external cingulum.

**Systematic notes and South African fossil record.**   One species is currently recognized in South Africa:

- *Saccotomus campestris* PETERS, 1846

This genus seems absent from South Africa until the late Pleistocene, with the oldest remains discovered in Border Cave around 0.2 MYA [70], in Sterkfontein StP6 around 0.1 MYA [5], in younger levels of Makapansgat Cave of Hearths [1] and in Gladysvale Pink Breccia, S18.E6, and S19.6 of uncertain age [71].

Subfamily **DENDROMURINAE** G.M. Allen, 1939

Genus *Dendromus* Smith, 1829 (African Climbing Mice)

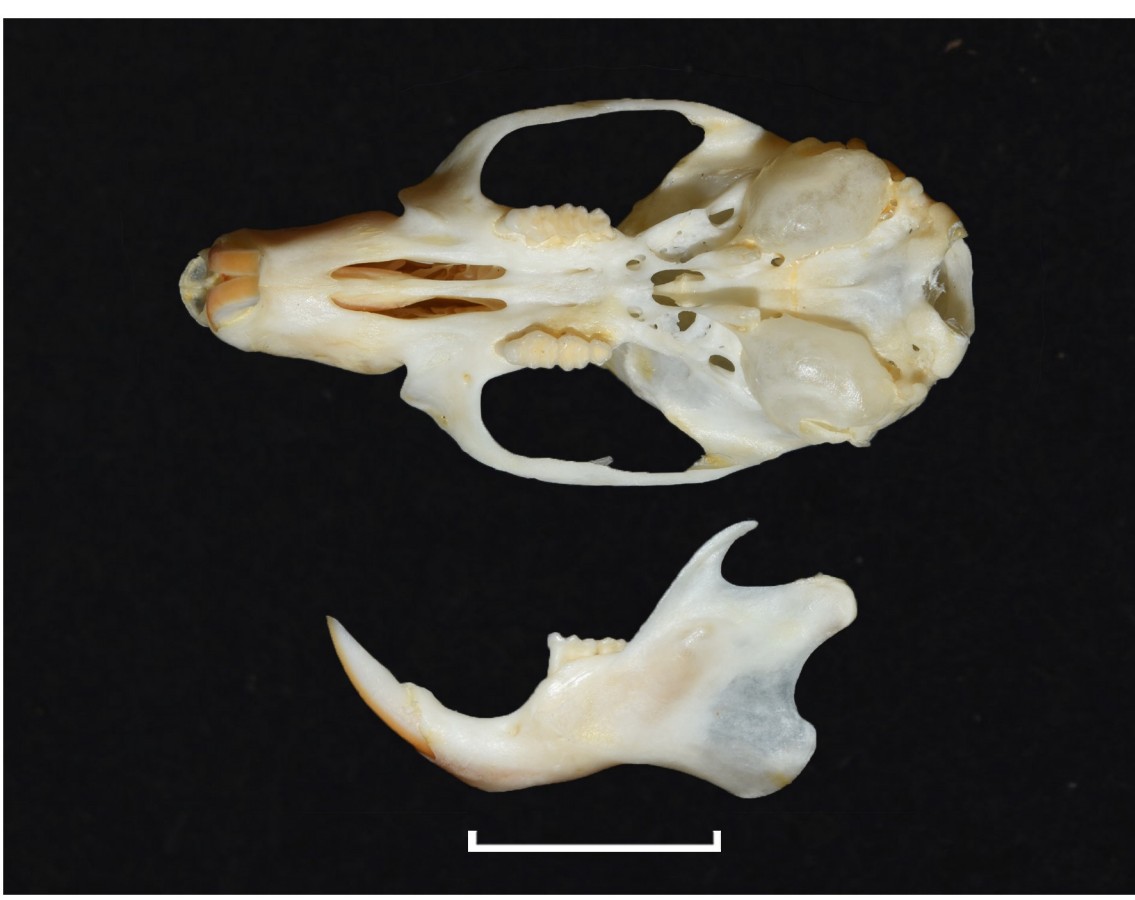

**Fig 72.  Cranium of *Saccostomus campestris* (DNMNH-4203), with scale bar of 1 cm.**

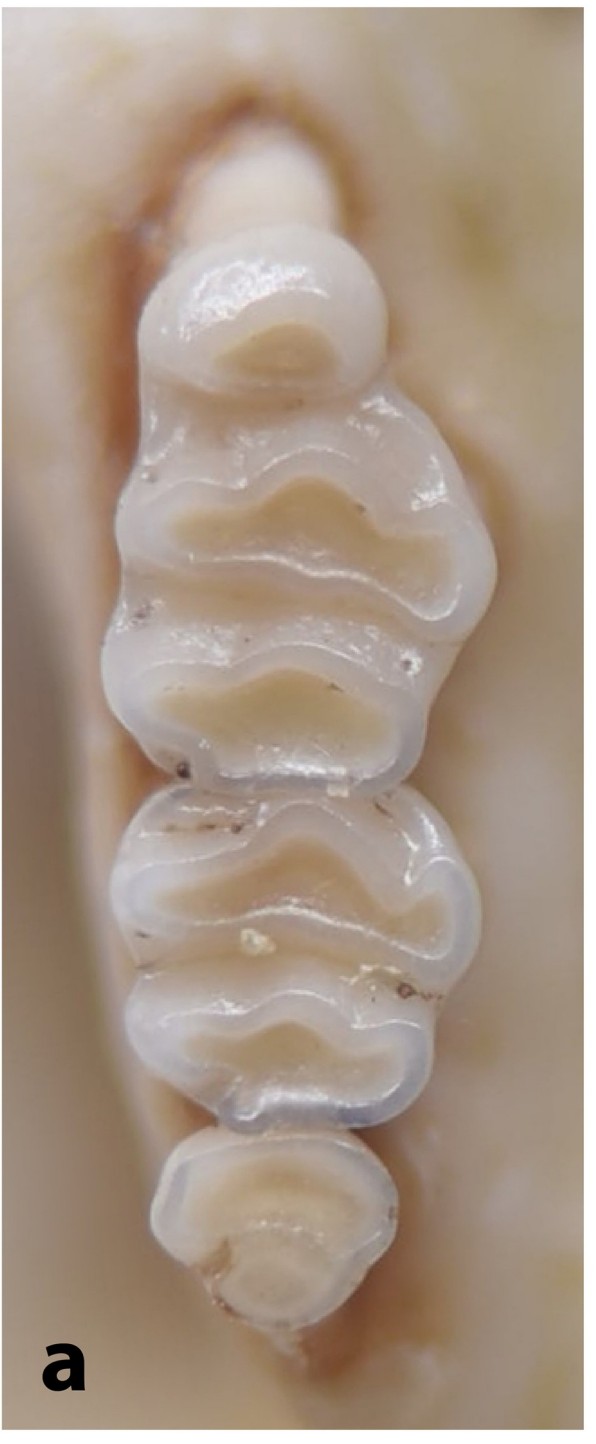
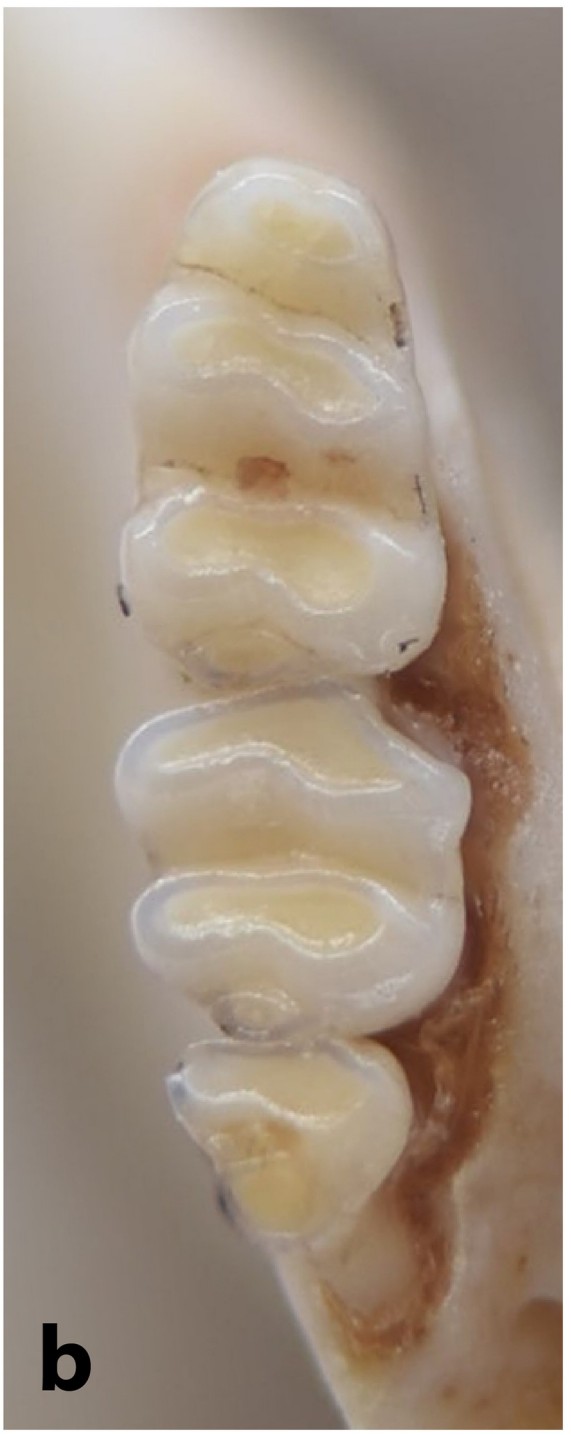

**Fig 73. Upper (a) and lower (b) right toothrow of *S. campestris* (DNMNH-4203).**

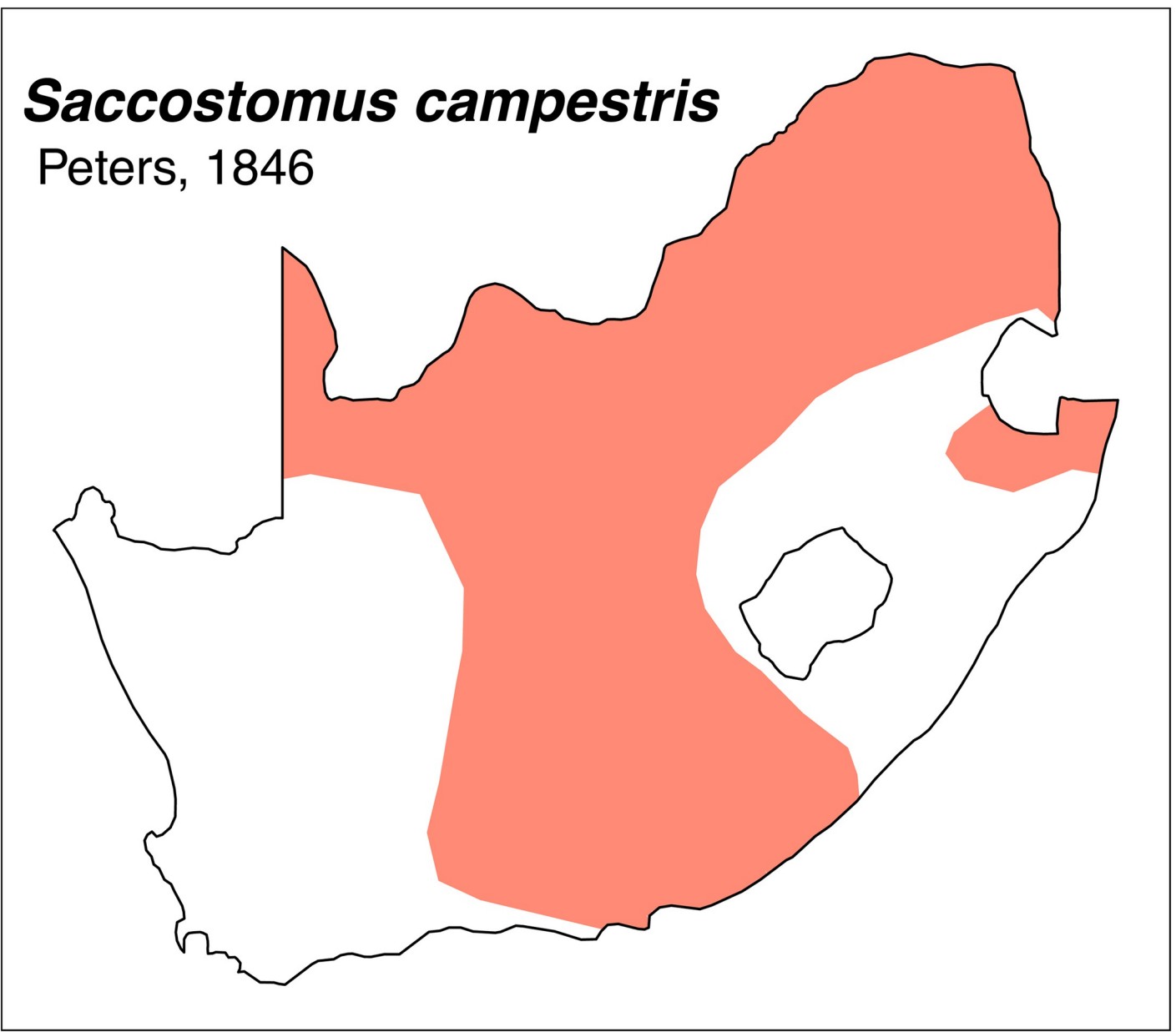

**Fig 74. Distribution map.**

**Table 25. Dental measurements (in mm) for *Saccostomus campestris*, sexes combined.**

| | Mean | Min | Max | n |
|---|---|---|---|---|
| LLTR | 4.4 | 4.1 | 4.9 | 11 |
| $WM_1$ | 1.3 | 1.2 | 1.4 | 11 |
| LUTR | 4.8 | 4.1 | 5.9 | 15 |
| $WM^1$ | 1.5 | 1.3 | 1.6 | 15 |

Figs 75–77; Table 26

Dental formula is 1-0-0-3:1-0-0-3. Alveolar formula is 4-3-1:2-2-1 (Figs 16 and 17).

**Upper jaw.** Upper incisors are opisthodont and have one pronounced groove, located closer to the outer margin than the inner (a character shared with other Dendromurinae). There is a masseteric knob at the lower anterior corner of the zygomatic plate. Anterior palatal foramina extend beyond the first row of cusps of $M^1$. Molars have a typical Dendromurinae dental pattern, but cusps are more individualized than in *Steatomys*. The $M^1$ is the longest of the toothrow, being twice as large as the $M^2$. It lacks the cusp t1 but has a rather large t4 connected to the t5. There is a groove separating the central (t2, t5 and t8) and labial (t3, t6, t9) cusps (not marked in *Steatomys*). The $M^2$ lacks the t1 and has a small to tiny t3; the t4 is well-developed and connected to the t5. Molar $M^3$ is very small and corresponds to Group 1 in Fig 12.

**Lower jaw.** Lower incisors are ungrooved (one groove in upper incisors). The prelobe of $M_1$ consists of a single median cusp. It is connected to the first row of cusps, which is oblique and constituted by a lingual metaconid connected to the labial protoconid located further back. The second row of cusps is equally oblique, with a well-developed hypoconid and a very large posterior cingulum located in the postero-lingual angle of the tooth. The $M_2$ has two oblique rows of cusps, with conspicuous anterolabial and posterior cingula. The $M_3$ is very small.

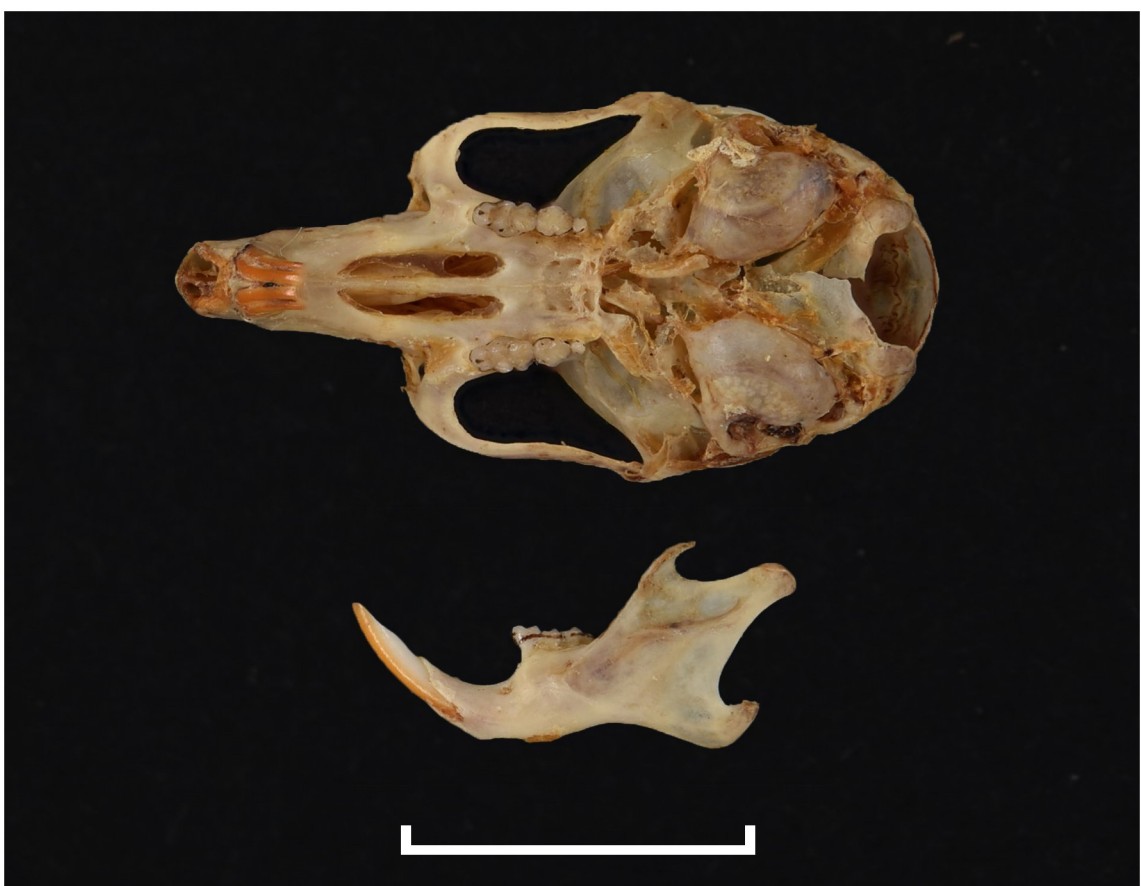

**Fig 75. Cranium of *Dendromus (Poemys) nyikae* (DNMNH-34625), with scale bar of 1 cm.**

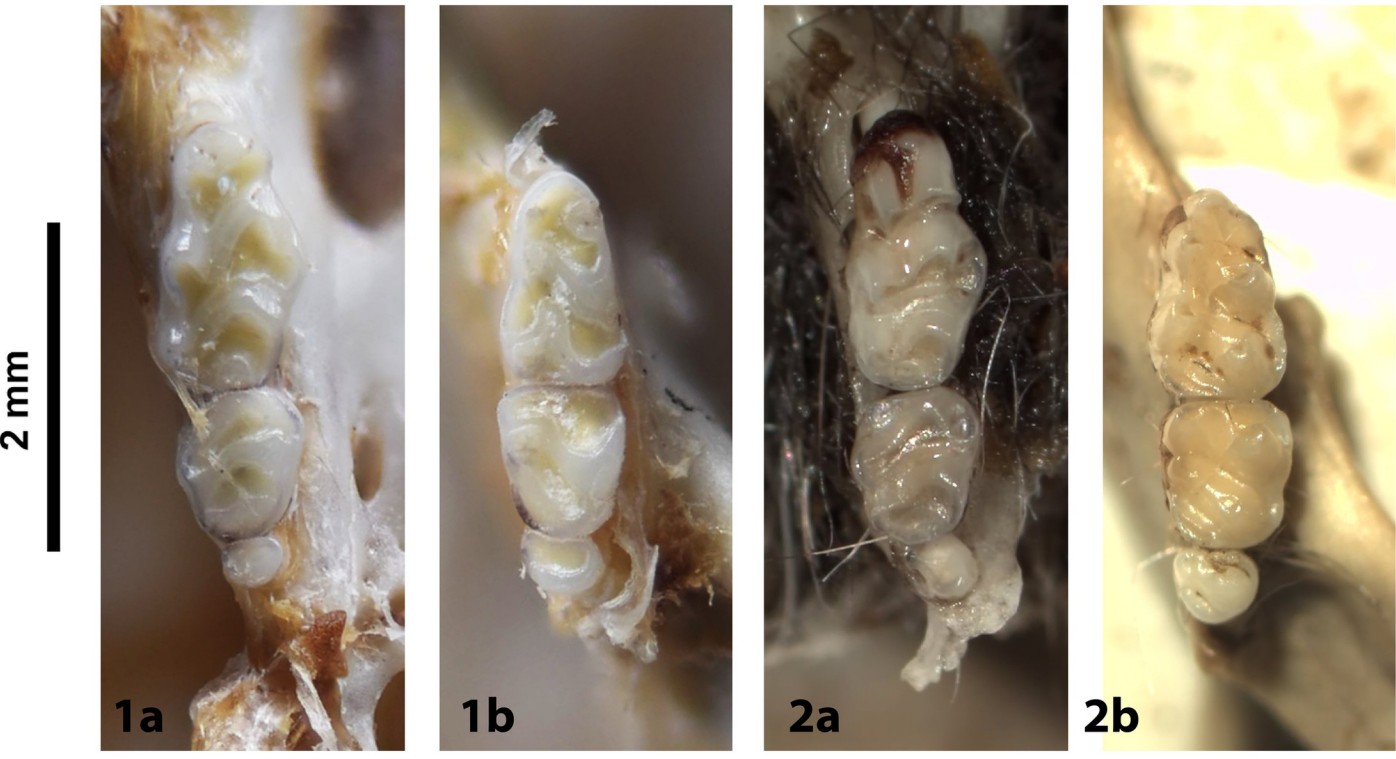

**Fig 76. Cheekteeth of *Dendromus*. 1)** Upper (a) and lower (b) right toothrow of *D. mesomelas* (DNMNH-2445); **2)** Upper (a) and lower (b) right toothrow of *Dendromus* sp. (ESI modern owl pellet collection).

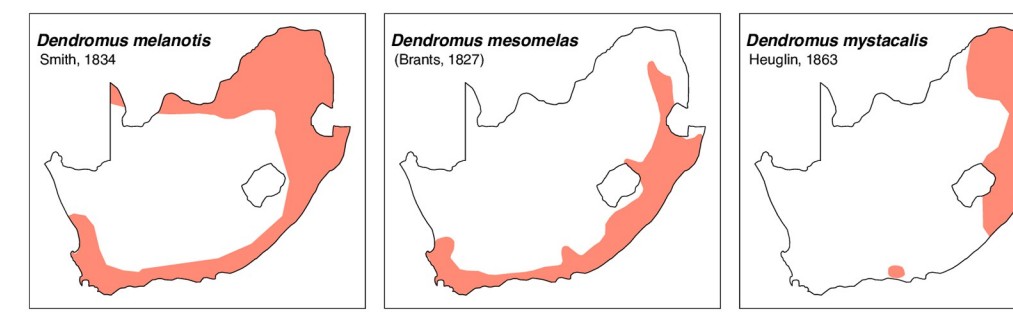
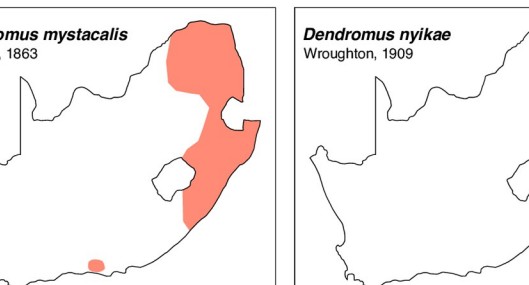

**Fig 77. Distribution map.**

**Table 26. Dental measurements (in mm) for *Dendromus* from South Africa, sexes and species combined.**

|         | Mean | Min | Max | n  |
|---------|------|-----|-----|----|
| LLTR    | 2.8  | 2.5 | 3.2 | 67 |
| $WM_1$  | 0.8  | 0.6 | 0.9 | 67 |
| LUTR    | 3.1  | 2.7 | 3.5 | 68 |
| $WM^1$  | 0.9  | 0.8 | 1.1 | 68 |

As in other Dendromurinae, muscle attachment on the mandible is situated rather far from the mental foramen, which can help to distinguish *Dendromus* from *Mus*.

**Systematic notes and South African fossil record.** The classification and species delimitation of the genus *Dendromus* has changed considerably over the last decades. Four species of *Dendromus* are listed in Monadjem et al. [22] for South Africa:

- *Dendromus melanotis* Smith, 1834

- *Dendromus mesomelas* (Brain, 1827)

- *Dendromus mystacalis* Heuglin, 1863

- *Dendromus nyikae* Wroughton, 1909

Recent molecular systematic assessment published by Voelker et al. [72] led to the resurrection of the genus *Poemys*. According to this, *D. nyikae* and *D. melanotis* would rather be identified as *P. nyikae and P. melanotis*, with the latter species probably comprising at least four species (*P. arenarius*, *P. basuticus*, *P. melanotis*, *P. vulturnus*). The two species *D. mesomelas* and *D. mystacalis* remain attributed to *Dendromus* and include many lineages and cryptic species not yet described. Consequently we prefer to keep the traditional taxonomy pending further revisions of South African *Dendromus* species. Additional fossil species have been described in South Africa:

- *†Dendromus antiquus* Broom, 1946 from the Late Pliocene of Taung, considered by Avery [4] as *nomen nudum*

- *†Dendromus averyi* Denys, 1994 from the Early Pliocene of Langebaanweg [73]

- *†Dendromus darti* Denys, 1994 from the Early Pliocene of Langebaanweg [73]

Material assigned to this genus has been recovered from many Quaternary fossil deposits in South Africa [50].

Genus *Malacothrix* Wagner, 1843 (Long-eared Mouse)

Figs 78–80; Table 27

Dental formula is 1-0-0-3:1-0-0-3. Alveolar formula is 3-3-1:2-2-1 (Figs 16 and 17).

**Upper jaw.** Upper incisors are opisthodont and have one groove (as other members of Dendromurinae). Anterior palatal foramina extend beyond the first row of cusps of $M^1$. Molars exhibit the typical Dendromurinae pattern, but the $M^1$ is very long, occupying half of the length of the toothrow. The $M^1$ has a well-marked antero-median cusp. The $M^3$ is very small and corresponds to Group 1 in Fig 12. The three alveoli of the $M^1$ are almost in line, with the lingual alveolus less displaced lingually than in *Dendromus* and *Steatomys*. There is a ridge-shaped masseter knob on the lower anterior of the zygomatic plate (sub-circular in *Dendromus* and *Steatomys*).

**Lower jaw.** Lower incisors are narrow and ungrooved. The molars display round, well separated bunodont cusps, with the typical Dendromurinae cusp pattern. The $M_1$ is narrow and very long, occupying more than half of the length of the molar toothrow; it displays seven highly alternated cusps separated by a longitudinal crest. The $M_2$ has an antero-external cusp. The $M_3$ is tiny and has a single cusp. The ventral edge of the mandible body between the ramus and the angular process is markedly curved. The posterior curvature between angular and coronoid process is well pronounced. As in other Dendromurinae, muscle attachment on the mandible is situated rather far from the mental foramen.

**Systematic notes and South African fossil record.** This genus is monotypic:

- *Malacothrix typica* (A. Smith, 1834)

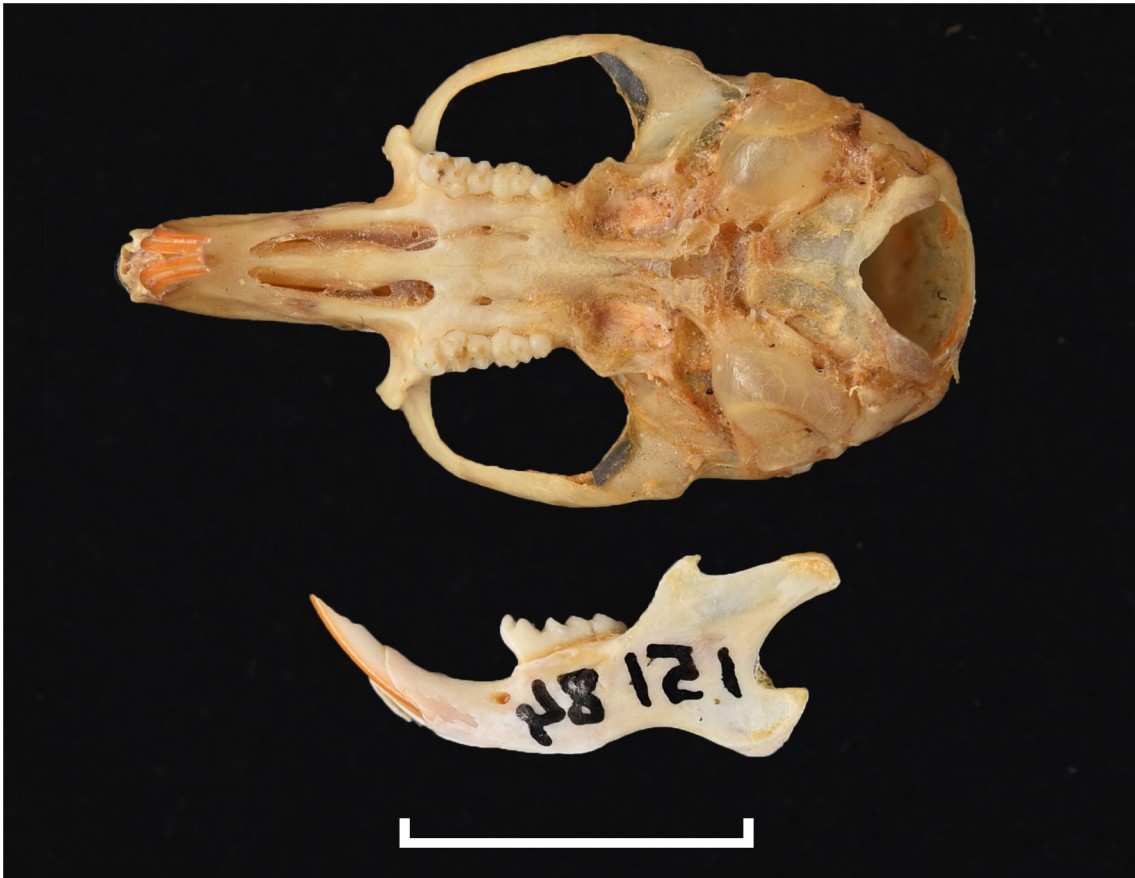

**Fig 78. Cranium of *Malacothrix typica* (skull-DNMNH 4965; mandible DNMNH-15184), with scale bar of 1 cm.**

An additional fossil species has been described in South Africa:

- *†Malacothrix makapani* DE GRAAFF, 1961

The oldest remains of *Malacothrix* are from Makapansgat around 3.3 MYA [48]. It has been recorded in several Quaternary fossil deposits in South Africa [50].

Genus ***Steatomys*** Peters 1846 (Fat Mice)

Figs 81–83; Table 28

Dental formula is 1-0-0-3:1-0-0-3. Alveolar formula is 3-3-1:2-2-1 (Figs 16 and 17).

**Upper jaw.** Upper incisors are opisthodont and have one pronounced groove, located closer to the outer margin than the inner. Anterior palatal foramina are long and extend beyond the first row of cusps of $M^1$. Molars have a typical Dendromurinae cusp pattern (with t1 and t7 missing in $M^1$ and $M^2$) but cusps are rounder than in *Dendromus* and *Malacothrix*. The $M^1$ is the longest of the toothrow and has three lobes: the first lobe has two cusps that are connected and project posteriorly; the second lobe has three cusps with a t4 closely connected to the t5 but located slightly behind; the third lobe has two cusps and a posterior cingulum, which is connected to the t8 and delimits a fovea. The $M^2$ has two lobes and a small t3. The $M^3$ is small (although generally not as reduced as in *Dendromus*) and corresponds to Group 1 in Fig 12.

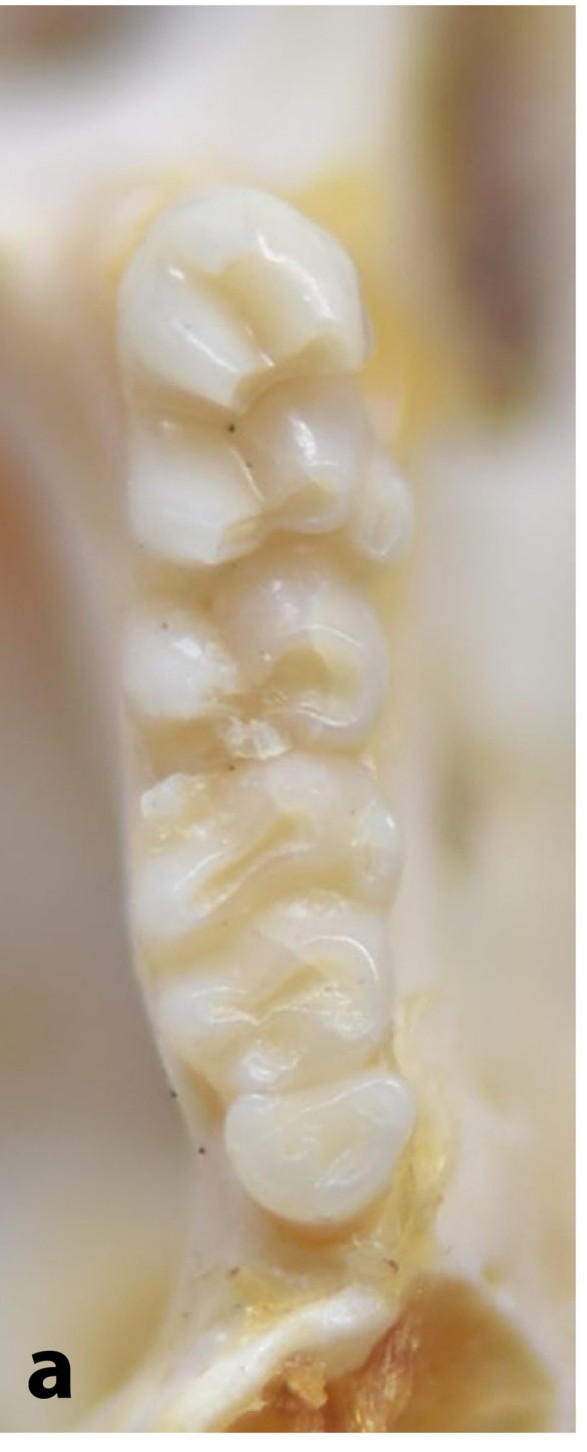
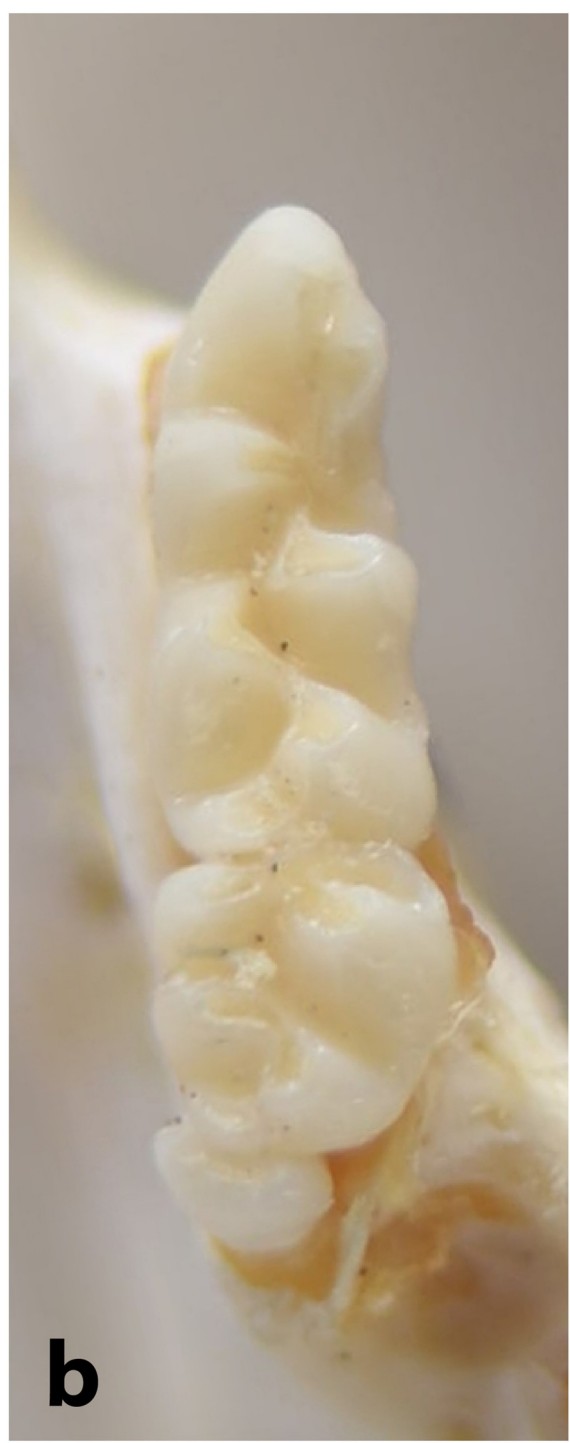

**Fig 79. Upper (a) and lower (b) right toothrow of *M. typica* (DNMNH-15184).**

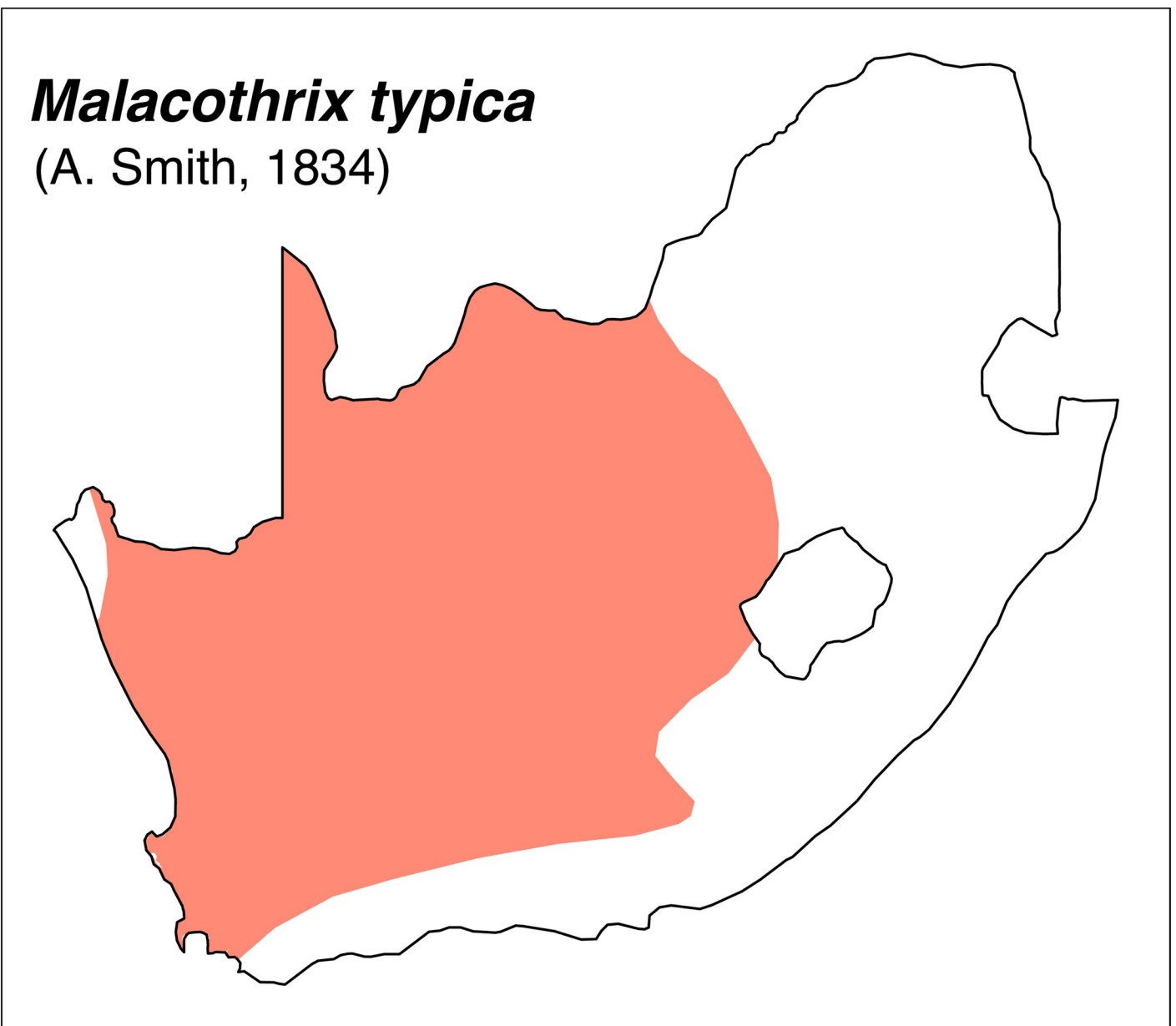

**Fig 80. Distribution map.**

**Table 27. Dental measurements (in mm) for *Malacothrix typica*, sexes combined.**

|  | Mean | Min | Max | n |
|---|---|---|---|---|
| LLTR | 3.7 | 3.3 | 4.2 | 9 |
| $WM_1$ | 1.0 | 0.9 | 1.1 | 9 |
| LUTR | 4.0 | 3.6 | 4.5 | 9 |
| $WM^1$ | 1.1 | 1.0 | 1.2 | 9 |

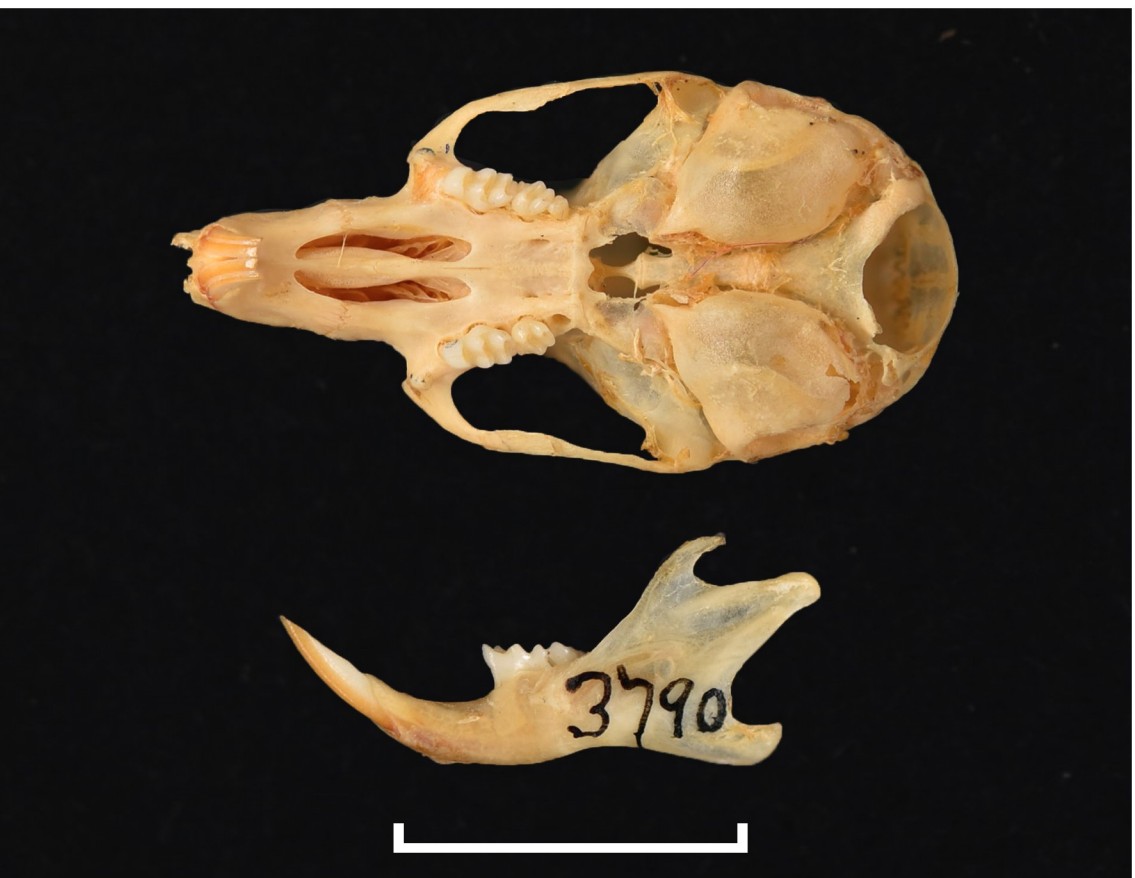

**Fig 81. Cranium of *Steatomys krebsii* (skull DNMNH-3792, mandible skull DNMNH-3790), with scale bar of 1 cm.**

**Lower jaw.**   Lower incisors are ungrooved. Like *Dendromus*, the $M_1$ has a median prelobe connected to the first oblique row of cusps and a median posterior cingulum below the second oblique row. The $M_2$ has two lobes with an anterolabial cingulum and a small to medium posterior cingulum. The $M_3$ is small and consists of a single cusp. As in other Dendromurinae, muscle attachment on the mandible is situated rather far from the mental foramen.

**Systematic notes and South African fossil record.**   Two species are recognised in South Africa:

- *Steatomys krebsii* Peters, 1852

- *Steatomys pratensis* Peters, 1846

In South Africa, the oldest known fossils are from Makapansgat around 3.3 MYA [48]. Remains of this genus have been recorded in several Quaternary fossil deposits [50].

Subfamily **MYSTROMYINAE** Vorontsov, 1966

Genus ***Mystromys*** Wagner, 1841 (African White-tailed Rat)

Figs 84–86; Table 29

Dental formula is 1-0-0-3:1-0-0-3. Alveolar formula is 3-3-2:2-2-2 (Figs 16 and 17).

**Upper jaw.**   Upper incisors are orthodont and display typical striations on the anterior enamel surface that form a slightly raised band. The palatal foramina are long and extend past

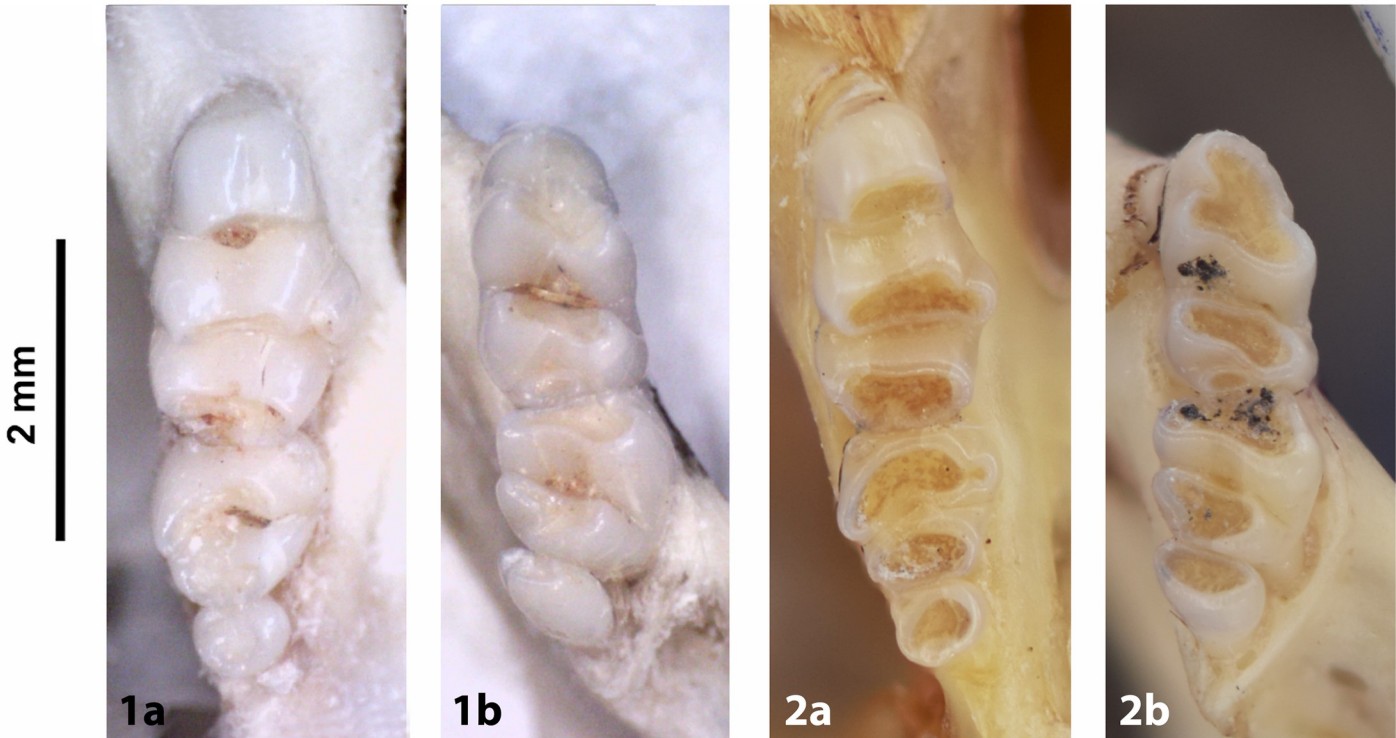

**Fig 82. Cheekteeth of *Steatomys*. 1)** Upper (a) and lower (b) right toothrow of *S. pratensis* (RMCA-96-037-M-5108); **2)** Upper (a) and lower (b) right toothrow of *Steatomys* sp. (DNMNH-3458).

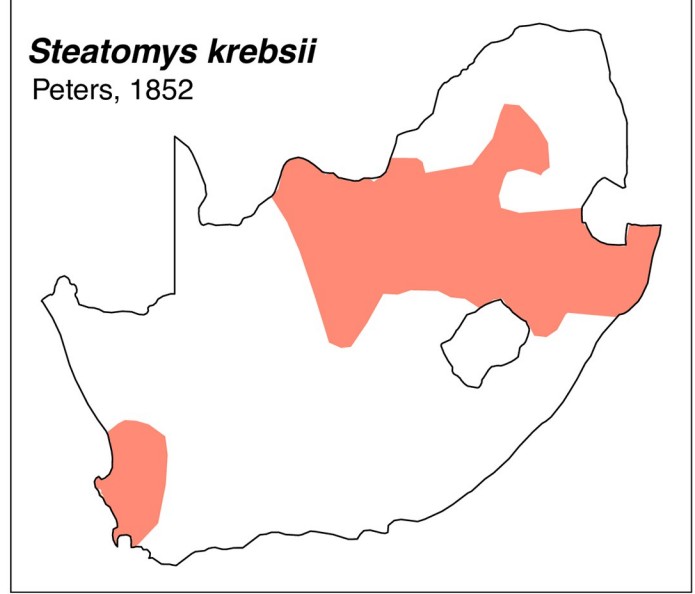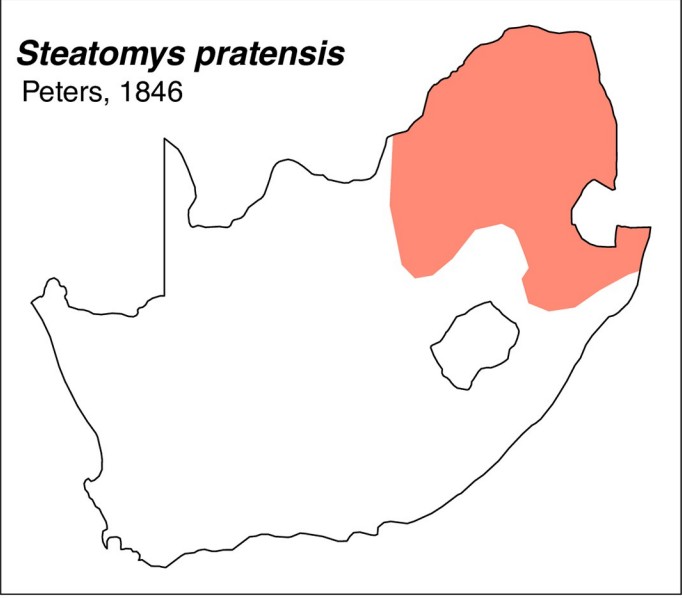

**Fig 83. Distribution maps.**

**Table 28. Dental measurements (in mm) for *Steatomys* from South Africa, sexes and species combined.**

|            | Mean | Min | Max | n  |
|------------|------|-----|-----|----|
| LLTR       | 3.4  | 3.1 | 3.7 | 30 |
| $WM_1$     | 1.1  | 0.9 | 1.2 | 30 |
| LUTR       | 3.8  | 3.3 | 4.2 | 30 |
| $WM^1$     | 1.3  | 1.1 | 4.1 | 30 |

the $M^1$. The cheekteeth have well separated opposed cusps that are connected by a longitudinal crest, resulting in a characteristic zigzag pattern. $M^1$ has three lobes with two lingual and two labial folds; the first lobe has one to two fused cusps, the second and third lobe have two alternated cusps. $M^2$ has two lobes each with two cusps. $M^3$ is small and has two lobes, the last one consisting of a single cusp.

**Lower jaw.**   Lower incisors display characteristic raised enamel bands. Cusps of lower molars have alternate cusps connected by a longitudinal crest, which results in a distinctive zigzag pattern. Molar $M_1$ has three lingual and two labial folds. Molar $M_2$ has two lingual and two labial folds. Molar $M_3$ has three cusps with a small antero-external cingulum.

**Systematic notes and South African fossil record.**   This genus is monotypic today:

- *Mystromys albicaudatus* (A. Smith, 1834)

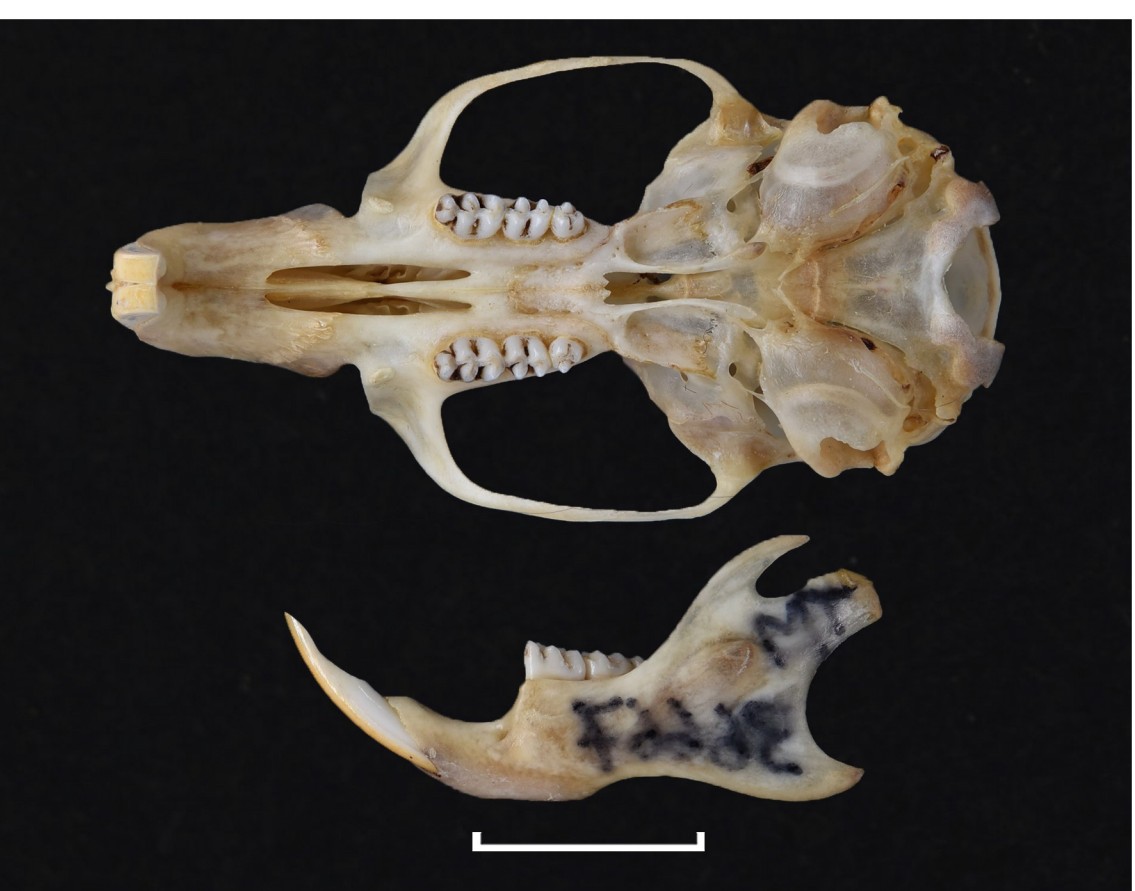

**Fig 84. Cranium of *Mystromys albicaudatus* (skull DNMNH-5986, mandible DNMNH-8813), with scale bar of 1 cm.**

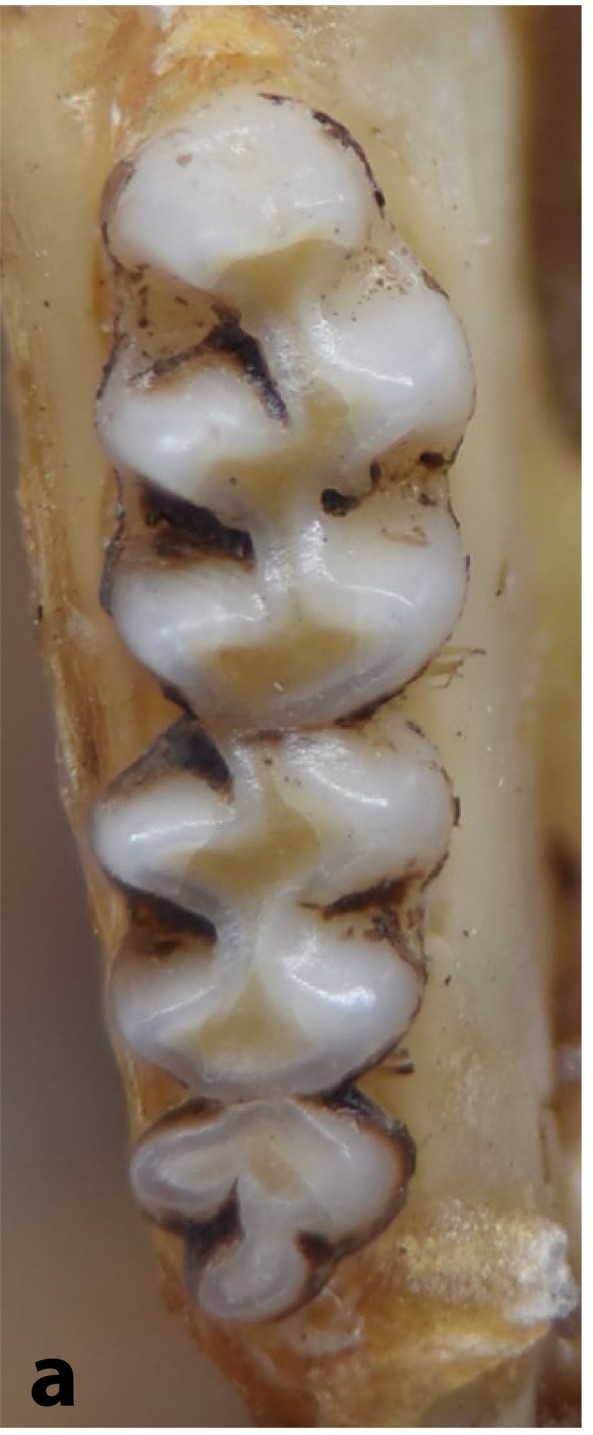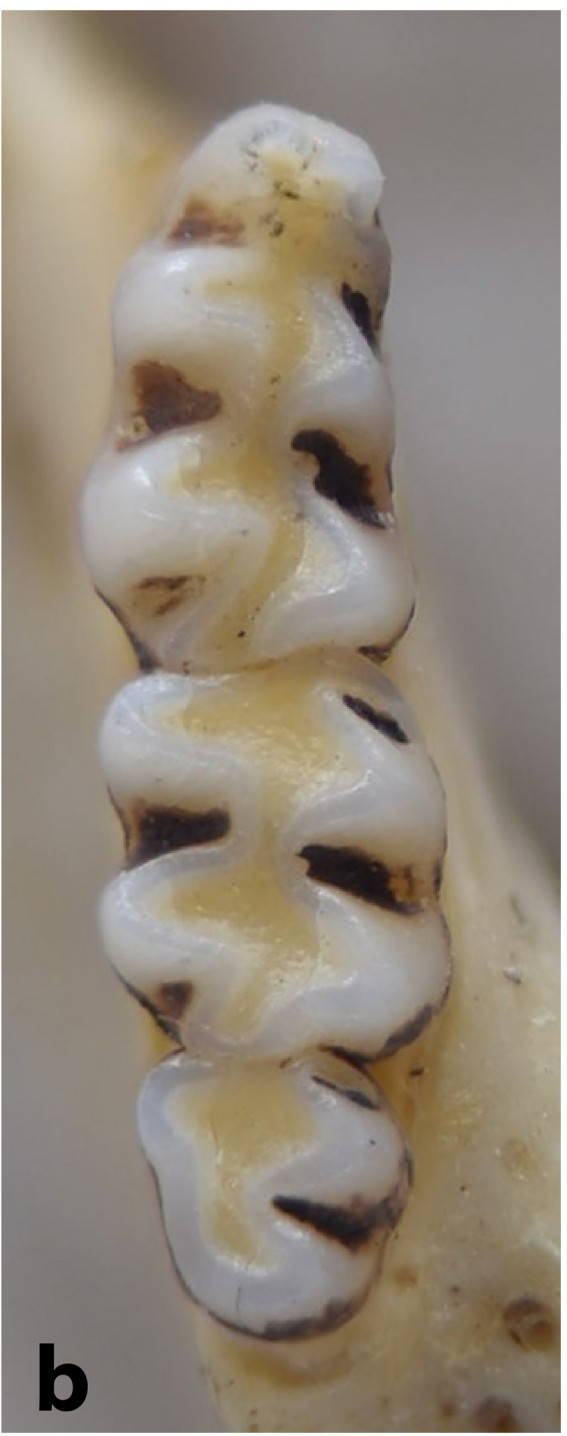

**Fig 85. Upper (a) and lower (b) right toothrow of *M. albicaudatus* (DNMNH-3666).**

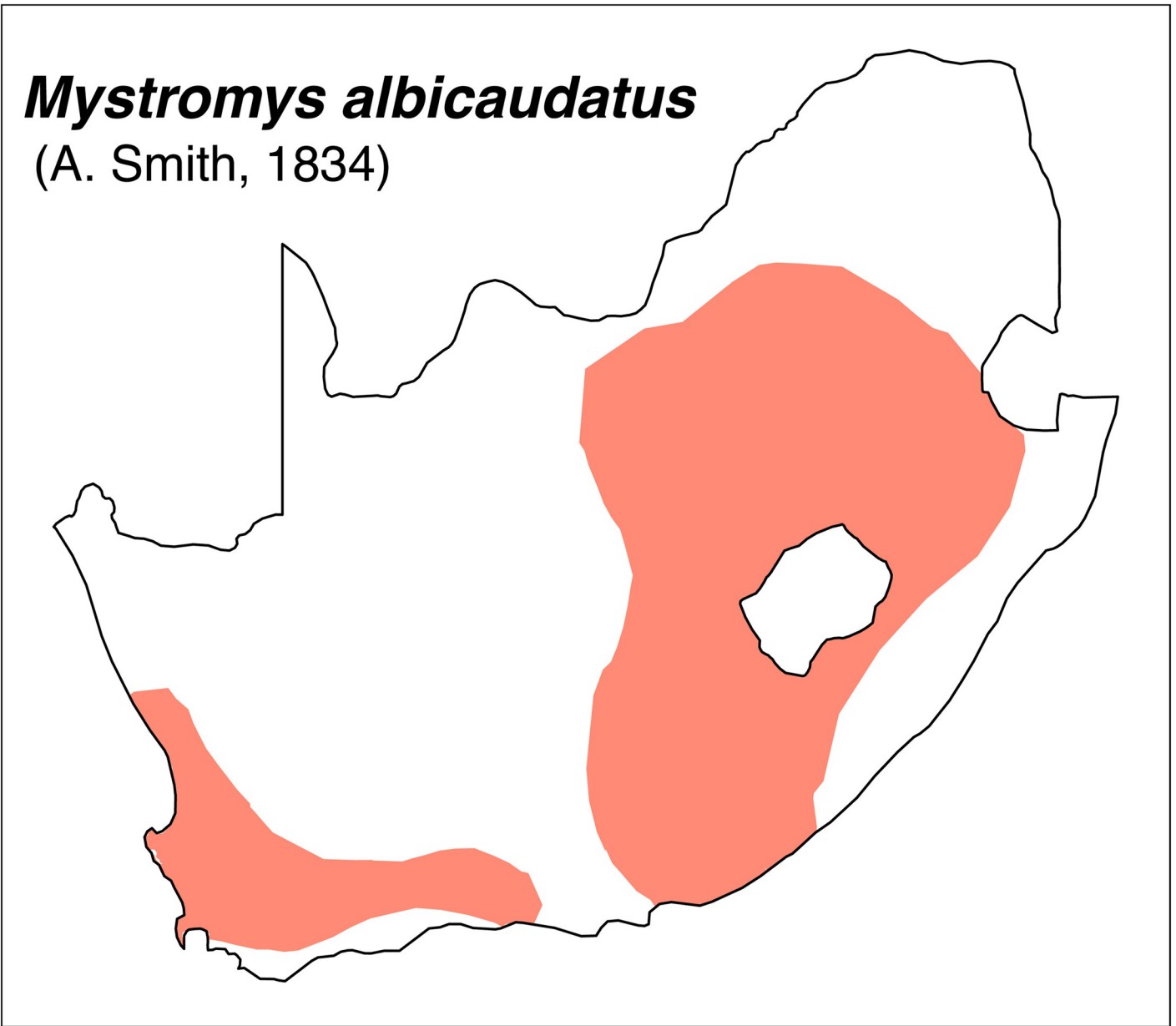

**Fig 86. Distribution map.**

**Table 29. Dental measurements (in mm) for *Mystromys albicaudatus*, sexes combined.**

|  | Mean | Min | Max | n |
|---|---|---|---|---|
| LLTR | 6.1 | 5.4 | 6.8 | 41 |
| WM$_1$ | 1.7 | 1.4 | 1.9 | 40 |
| LUTR | 6.0 | 5.5 | 6.8 | 41 |
| WM$^1$ | 1.8 | 1.6 | 2.0 | 41 |

Additional species have been described in South African fossil deposits:

- *†Mystromys hausleitneri* Broom, 1937 from various Pleistocene sites, which is often regarded as a chronospecies that closely resembles the present-day form, albeit slightly smaller [4, 47, 74].

- *†Mystromys pocockei* Denys, 1991 from the Early Pliocene site of Langebaanweg

Fossils of this genus are known from many Pleistocene sites in South Africa [50]. In the karst deposits of the Sterkfontein Valley from the Gauteng Province, it is often one the most abundant taxa from the micromammal assemblages.

† Genus *Proodontomys* Pocock, 1987

Figs 87 and 88; Table 30

Dental formula is 1-0-0-3:1-0-0-3. Alveolar formula is 3-3-1:2-2-1/2 (Figs 16 and 17).

**Upper jaw.**   Upper incisors are highly proodont and ungrooved; they do not display the typical ridge seen in *Mystromys*. Anterior palatal foramina are long and penetrate between the second rows of cusps in $M^1$. Molars show a tendency to hypsodonty, with weak cusps that quickly wear down to a plain occlusal surface. As in *Mystromys*, the cheekteeth lack lateral cusps, but they differ by having opposite, rather than alternate, molar cusps, resulting in a semi-lophodont dentition. The $M^1$ has three lobes, the $M^2$ has two lobes, and the $M^3$ has two poorly differentiated lobes that tend to fuse with wear to form a simple cylindrical tooth.

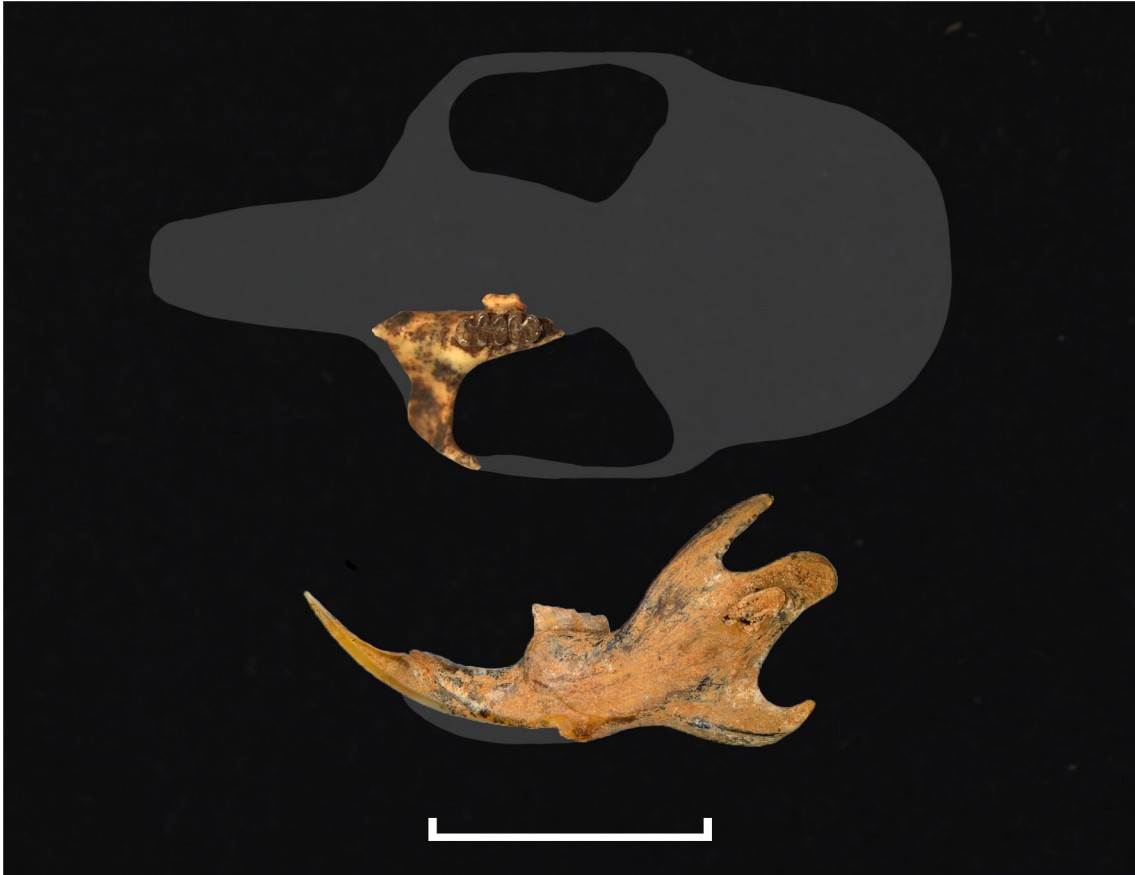

**Fig 87. Cranium of *Proodontomys cookei* (ESI T.N. Pocock microfaunal fossil collection), with scale bar of 1 cm.**

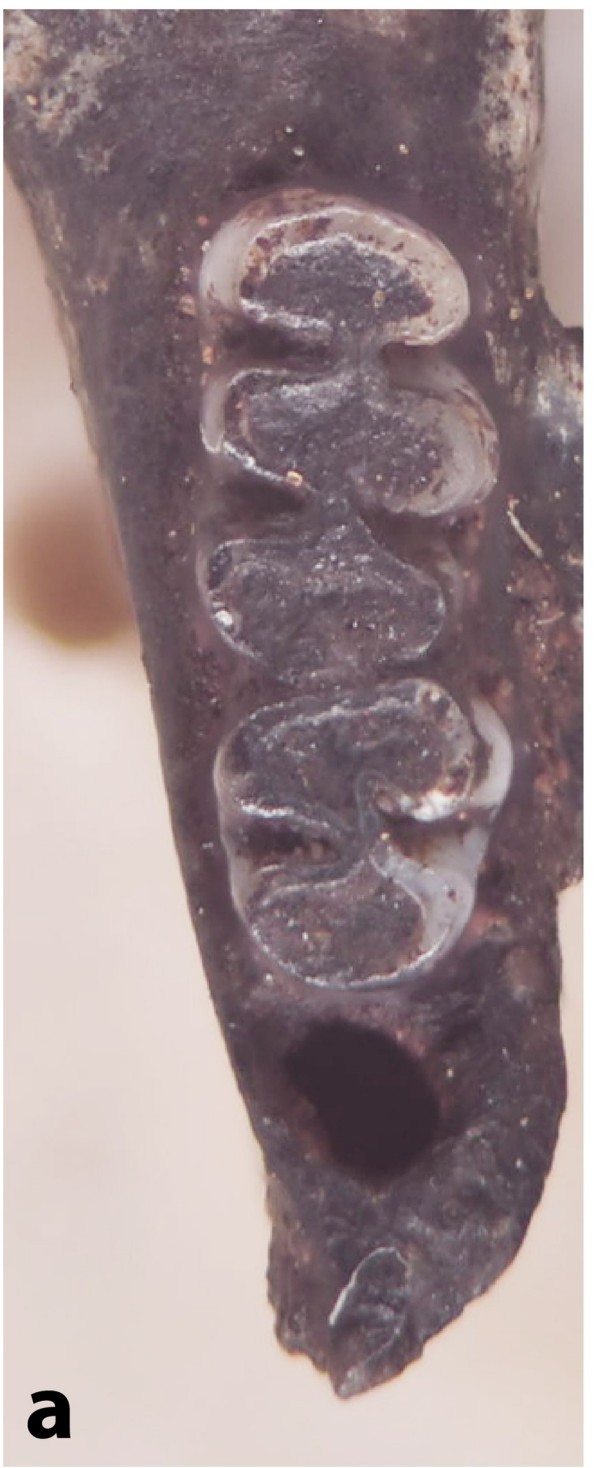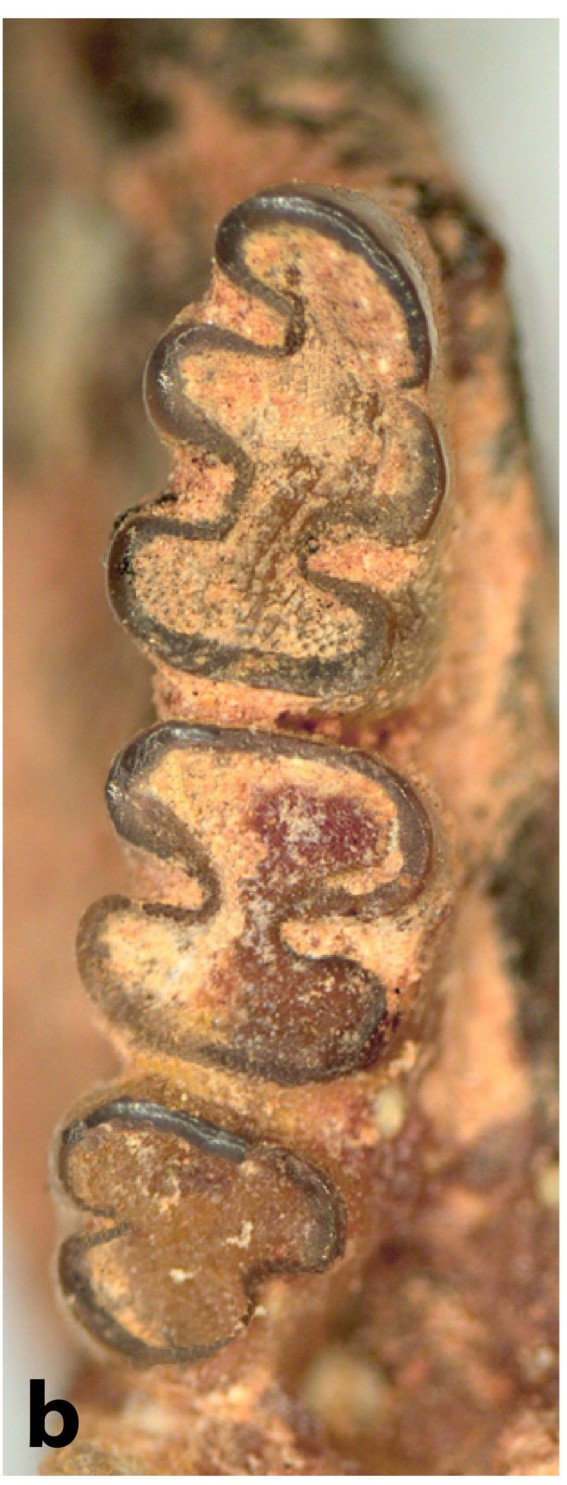

**Fig 88. Cheekteeth of *Proodontomys*.** Upper (a) and lower (b) right toothrow of *P. cookei* (upper ESI, Cooper's D fossil collection CD1980; lower ESI T.N. Pocock microfaunal fossil collection, EXQRM47). The molar $M^3$ is missing.

**Table 30. Dental measurements (in mm) for *Proodontomys cookei*, sexes combined.**

|  | Mean | Min | Max | n |
|---|---|---|---|---|
| LLTR | 4.2 | 3.6 | 4.6 | 9 |
| $WM_1$ | 1.1 | 1.0 | 1.2 | 35 |
| LUTR | 4.3 | 4.0 | 4.5 | 7 |
| $WM^1$ | 1.3 | 1.2 | 1.4 | 2 |

**Lower jaw.** Lower incisors are ungrooved and do not display the typical ridge present in *Mystromys*. As in upper molars, molars have week cusps that flatten in early wear into a plain, semi-lophodont occlusal surface. The molar $M_1$ shows a trilophodont pattern, and both the $M_2$ and $M_3$ are bilophodont. The coronoid process of the mandible is very large.

**Systematic notes and South African fossil record.** This genus got extinct around 1 MYA, and a single species has been described in South Africa fossil deposits:

- †*Proodontomys cookei* Pocock, 1987

The oldest remains of *Proodontomys* come from the locality of Limeworks Makapansgat around 3.3 MYA [48].

Subfamily **PETROMYSCINAE** Roberts, 1951

Genus *Petromyscus* Thomas, 1926 (Pygmy Rock Mice)

Figs 89–91; Table 31

Dental formula is 1-0-0-3-:1-0-0-3. Alveolar formula is: 3-3-1:2-2-1 (Figs 16 and 17).

**Upper jaw.** Upper incisors are opisthodont and ungrooved. Cheekteeth are relatively small. Anterior palatal foramina reach the margin or the first lobe of the $M^1$. Molars display round cusps connected by a median longitudinal crest, resulting in a zigzag pattern reminiscent of *Mystromys*. The $M^1$ has three lobes: the first has one or two fused cusps, the second has three cusps including a t4 attached to the t5 (which distinguishes this genus from the members of *Mystromyinae*), the third has two cusps. The $M^2$ has two lobes as well as an antero-external cusp (t3) but no t4. The $M^3$ is small, having a tiny t3 (t1 is absent) and two rows of cusps connected lingually. It corresponds to Group 1 in Fig 12.

**Lower jaw.** Lower incisors are ungrooved. The molar $M_1$ has three lobes, which are connected in their median part by a longitudinal crest. Molar $M_3$ is small and display a tiny antero-external cingulum.

**Systematic notes and South African fossil record.** Three species are presently recognized in South Africa:

- *Petromyscus barbouri* SHORTRIDGE & CARTER, 1938

- *Petromyscus collinus* (THOMAS & HINTON, 1925)

- *Petromyscus monticularis* (THOMAS & HINTON, 1925)

Remains of *Petromyscus* are scanty in the Quaternary fossil record, and the oldest fossils in South Africa are known from Holocene sites [50].

## Bathyergidae

Mole-rats of the family Bathyergidae (Tables 32 and 33) are fossorial (adapted to digging or burrowing) rodents and exhibit numerous morphological adaptations for an underground life. Some four genera occur in South Africa. They live in tunnel systems and are almost never seen above the ground's surface [21]. They mostly become vulnerable during the mound formation,

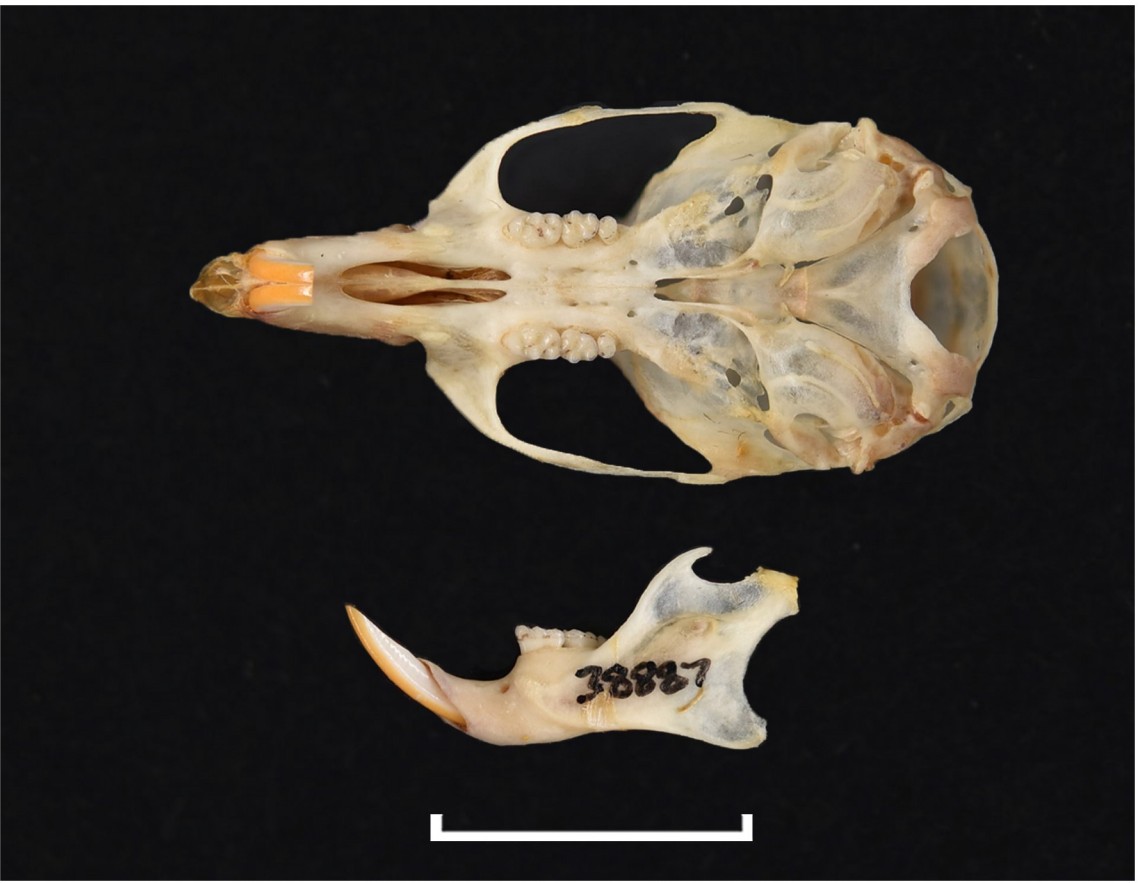

**Fig 89. Cranium of *Petromyscus collinus* (DNMNH-38887), with scale bar of 1 cm.**

when predators are able to locate them with accuracy. Only juveniles of *Bathyergus* and *Georychus* are small enough to be captured by owls such as *Tyto alba* (Barn owl), and *Bubo africanus* (Spotted eagle-owl), which are the main accumulators of Quaternary fossil micromammal deposits in South Africa. As a result, it is common to find mainly young individuals among fossil assemblages accumulated by these predators, with specimens showing only two or three completely erupted cheekteeth *versus* the four cheekteeth found in adult specimens. Members of Bathyergidae display distinctly proodont upper incisors. The lower jaw is hystricomorph in character, showing a characteristic flared and inflected angular process on the mandible. In South African mole-rats, the dental formula is always 1-0-1-3:1-0-1-3 and the anterior palatal foramina are small, ending well before the $M^1$.

Family **BATHYERGIDAE** Waterhouse, 1841

Genus ***Bathyergus*** Illiger, 1811 (Dune Mole-rats)

Figs 92–94; Table 34

Dental formula is 1-0-1-3:1-0-1-3. Alveolar formula varies with age of specimen.

**Upper jaw.**   Upper Incisors are proodont, robust and heavily grooved; they are less protruding than in other genera and are rooted above the anterior cheekteeth. Anterior palatal foramina are short, ending before molar $M^1$. Cheeekteeth are hypsodont and simple (rounded and uncusped), but young animals display re-entrant folds.

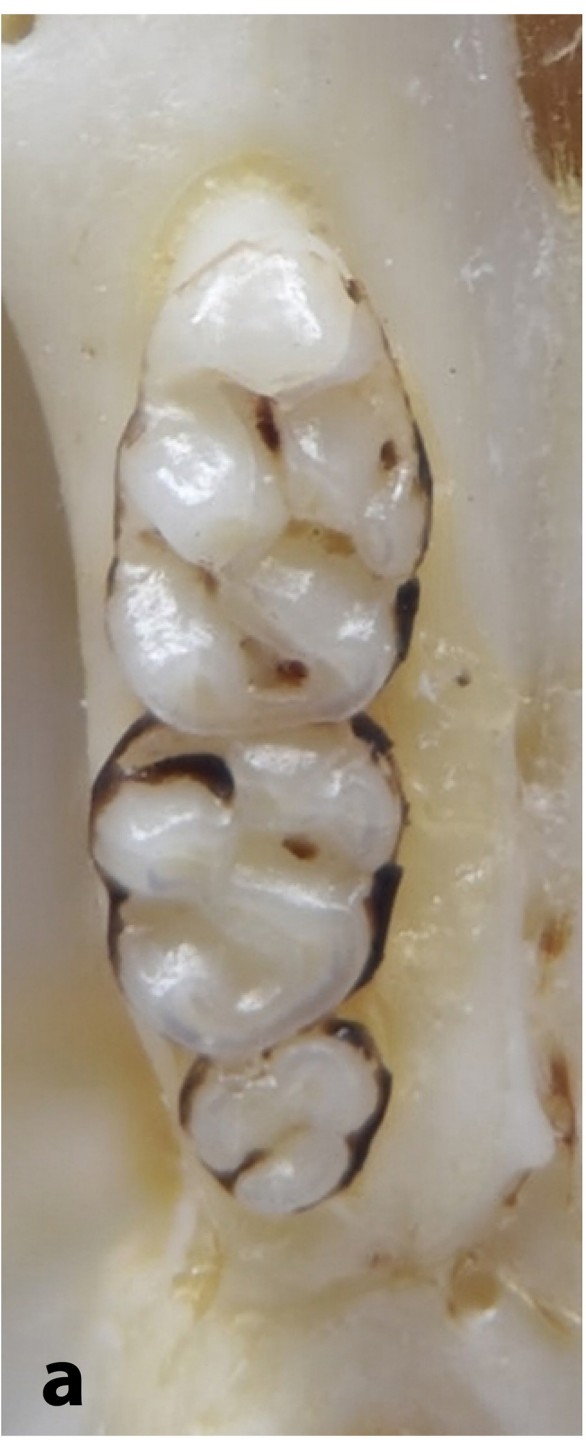
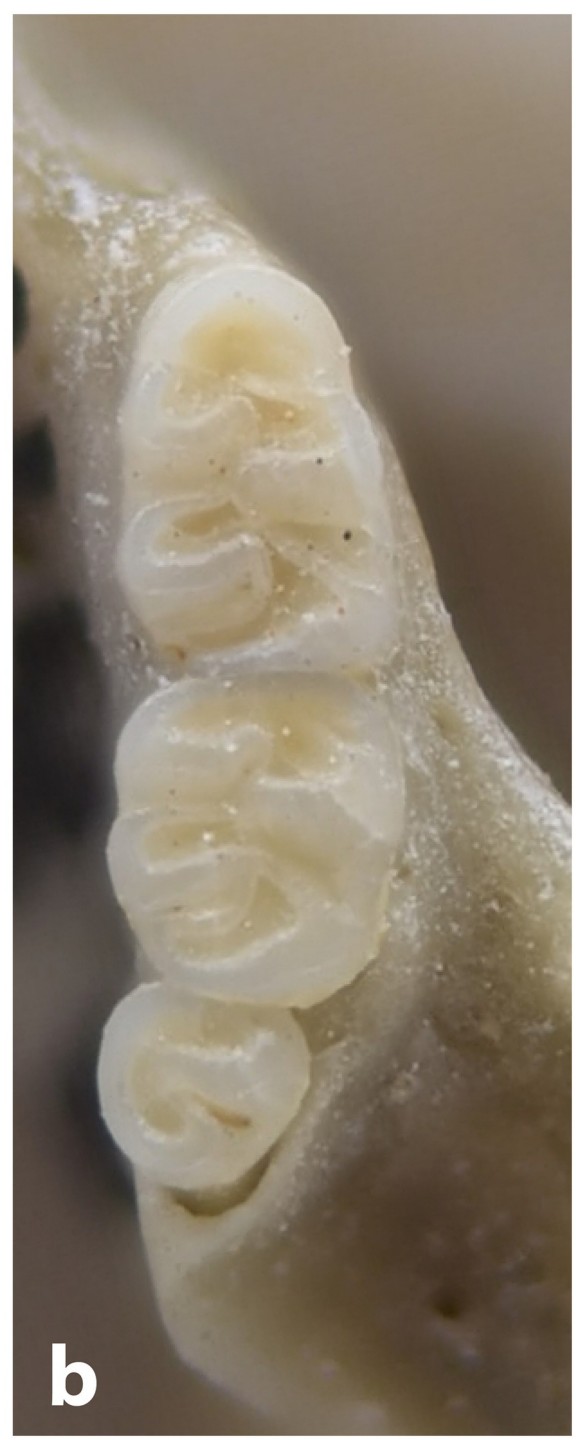

**Fig 90. Upper (a) and lower (b) right toothrow of *P. shortridgei* (upper DNMNH-23111; lower DNMNH-29253).**

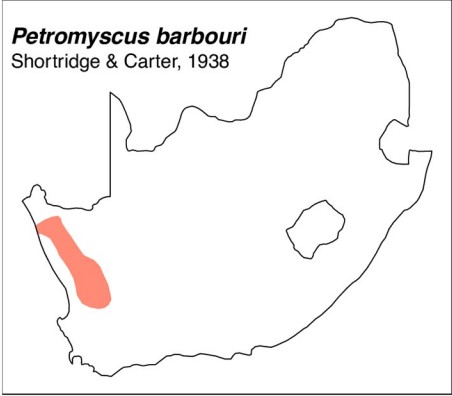 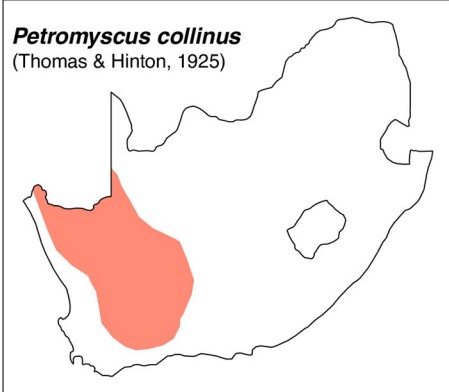 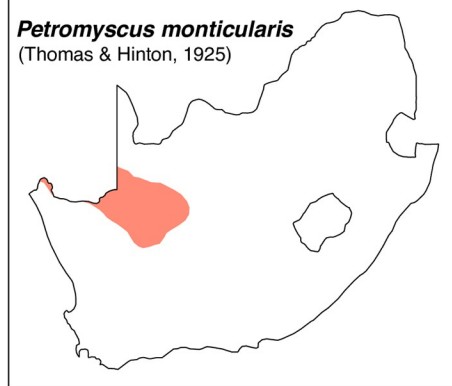

**Fig 91. Distribution map.**

**Table 31. Dental measurements (in mm) for *Petromyscus* from South Africa, sexes and species combined.**

|  | Mean | Min | Max | n |
|---|---|---|---|---|
| LLTR | 3.3 | 2.9 | 3.7 | 13 |
| $WM_1$ | 1.0 | 0.8 | 1.0 | 13 |
| LUTR | 3.5 | 3.3 | 3.9 | 10 |
| $WM^1$ | 1.1 | 1.0 | 1.2 | 12 |

**Table 32. Key to the bathyergid genera: Upper jaw.**

| 1 | molars with folds of enamel | *Georychus* |
|---|---|---|
|  | molars without folds of enamel | 2 |
| 2 | all molars are approximately the same size; incisors grooved | *Bathyergus* |
|  | $M^3$ is smaller; incisors are ungrooved | *Cryptomys/Fukomys* |

**Table 33. Key to the bathyergid genera: Lower jaw.**

| 1 | molars with folds of enamel | *Georychus* |
|---|---|---|
|  | molars without folds of enamel | 2 |
| 2 | LLTR > 8 mmm | *Bathyergus* |
|  | LLTR < 7 mm | *Cryptomys/Fukomys* |

**Lower jaw.**   Lower incisors are plain. Lower cheekteeth are similar to upper cheekteeth. The mandible is hystricognathous and stickily built, with the angular portion extending well posteriorly. The mental foramen is positioned posteriorly to the diastema, aligned with the posterior border of the $M_1$.

**Systematic notes and South African fossil record.**   Two species of *Bathyergus* occur in South Africa:

- *Bathyergus janetta* Thomas & Schwann, 1904

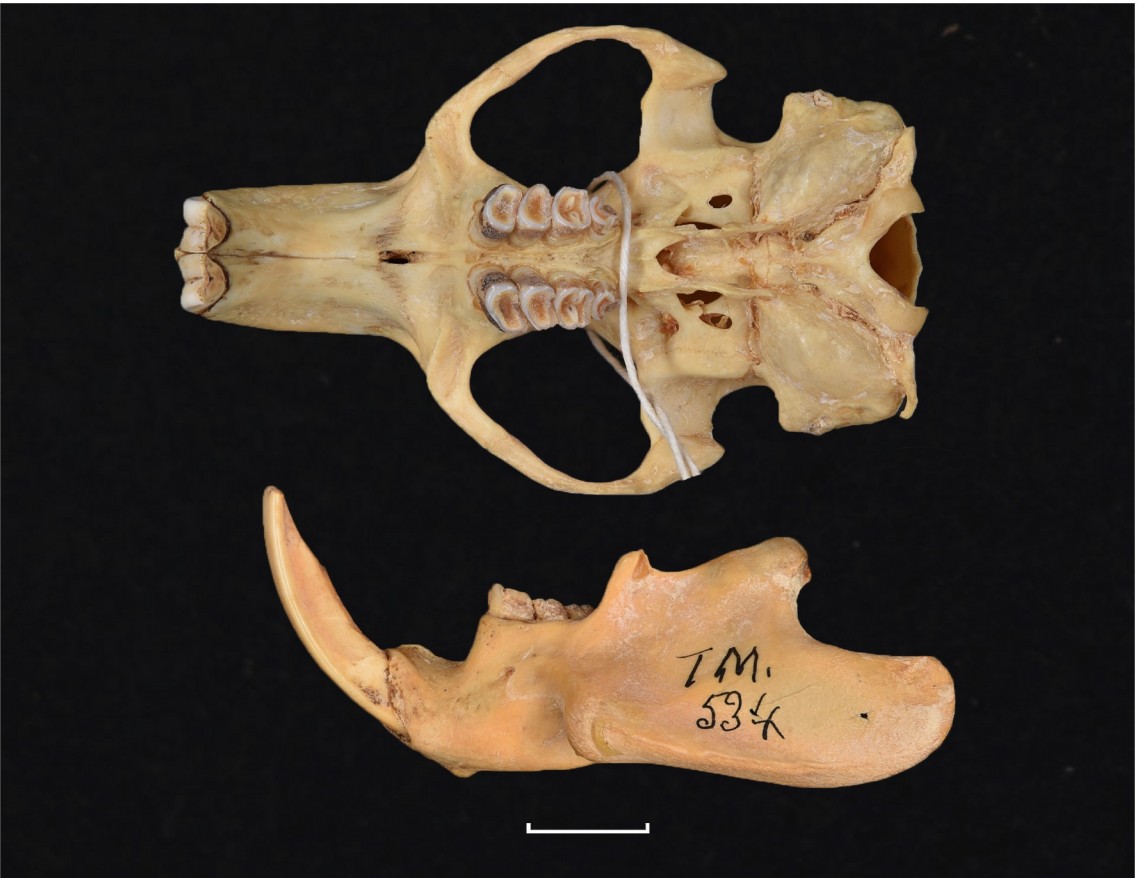

**Fig 92. Cranium of *Bathyergus suillus* (skull DNMNH 523; mandible DNMNH-534), with scale bar of 1 cm.**

- *Bathyergus suillus* (Schreber, 1782)

  An additional fossil species has been described:

- †*Bathyergus hendeyi* Denys, 1998 from the Early Pliocene of Langebaanweg

  Genus ***Cryptomys*** Gray, 1864 (Mole-rats)
  Figs 95–97; Table 35
  Dental formula is 1-0-1-3:1-0-1-3. Alveolar formula varies with age of specimen.
  **Upper jaw.**   *Cryptomys* and *Fukomys* appear to be indistinguishable on morphological grounds. Upper incisors are robust, ungrooved and proodont. Anterior palatal foramina are very short, ending before molar $M^1$. Cheekteeth are simple, rounded and uncusped, but young animal display re-entrant folds. The skull is the smallest found in the Bathyergidae.
  **Lower jaw.**   Lower incisors are smooth on their anterior surfaces. Their width is less than 2 mm, even for older specimens (in *Bathyergus*, the width of the incisor is superior to 2 mm before the $M_3$ is erupted). Lower cheekteeth are similar to upper cheekteeth: simple, rounded and uncusped, although young animals display re-entrant folds. The coronoid process of the hystricognathous mandible is high and projects backwards. The mental foramen is positioned posteriorly to the diastema, aligned with the posterior border of the $M_1$.
  **Systematic notes and South African fossil record.**   The species delimitation of *Cryptomys* has changed multiple times over last decades [75]. This genus formerly encompassed all the

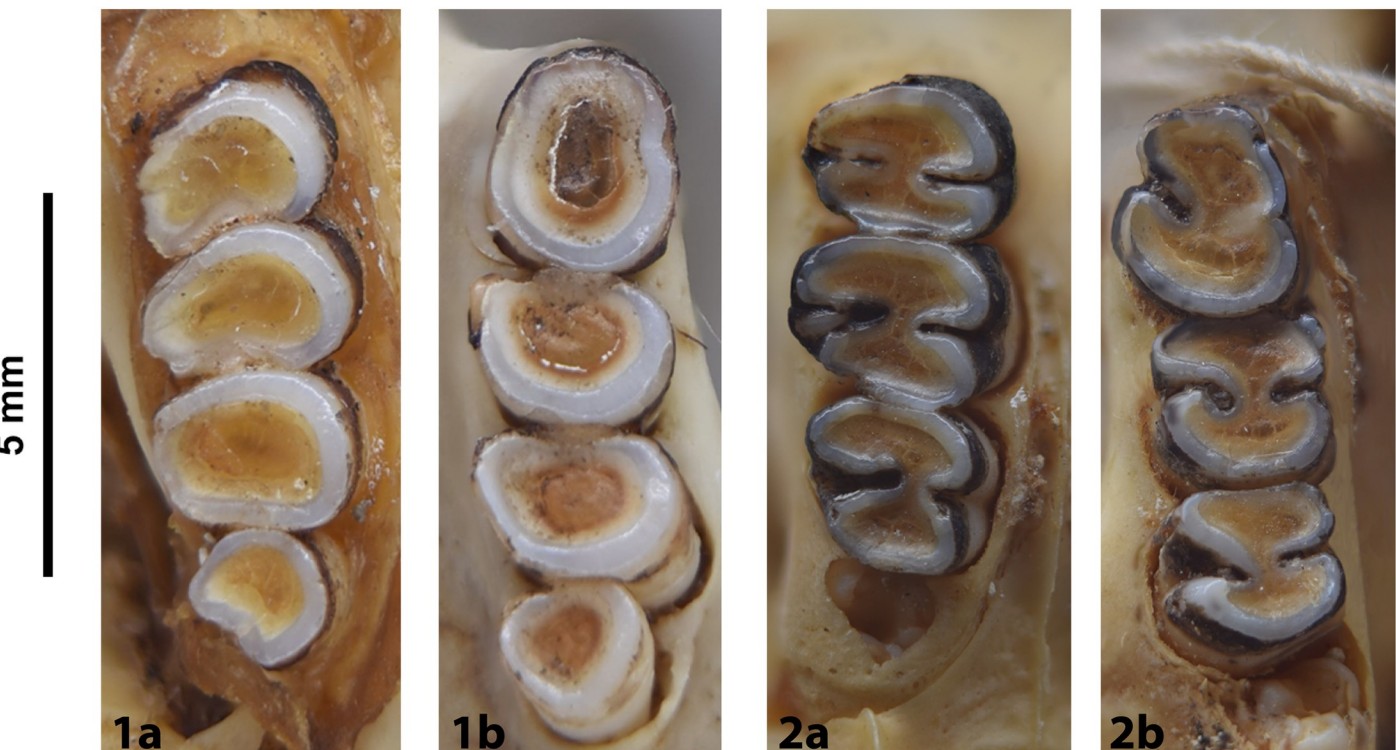

**Fig 93. Cheekteeth of *Bathyergus*. 1)** Upper (a) and lower (b) right toothrow of an adult *B. janetta* (upper DNMNH-543; lower DNMNH-39304); **2)** Upper (a) and lower (b) right toothrow of a juvenile *B. suillus* (DNMNH-2162), with molars $M^3/M_3$ not erupted yet.

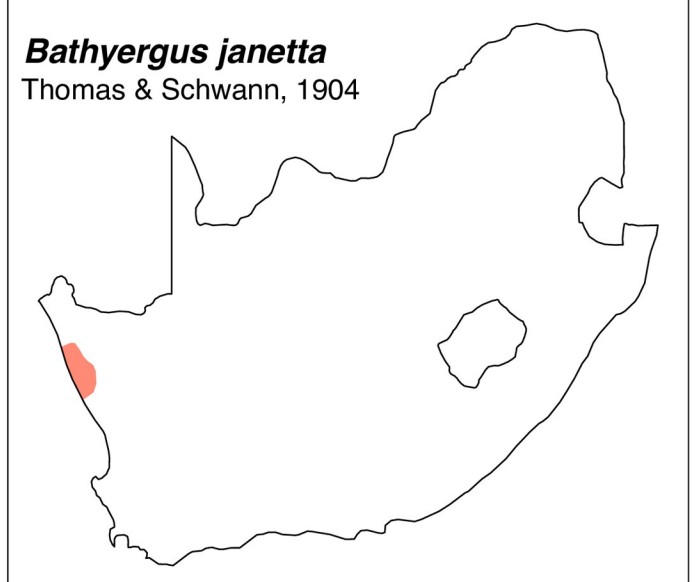

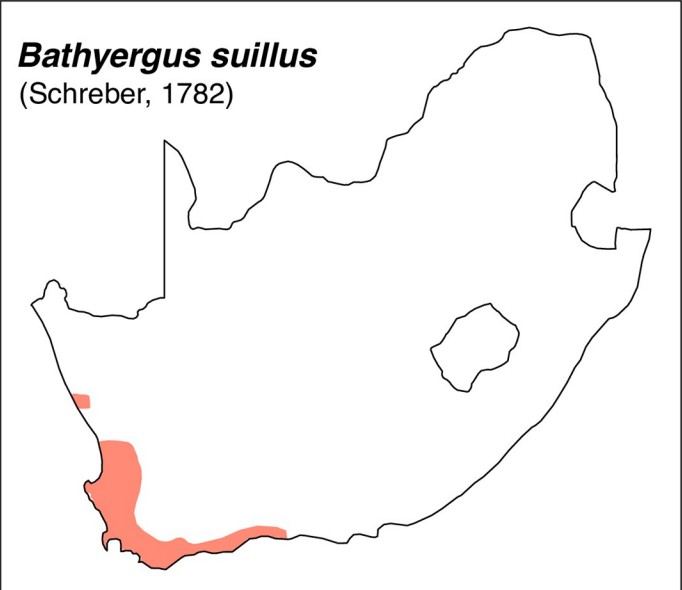

**Fig 94. Distribution maps.**

**Table 34.  Dental measurements (in mm) for *Bathyergus* from South Africa, sexes and species combined.**

|        | Mean | Min | Max | n |
|--------|------|-----|-----|---|
| LLTR   | 11.0 | 8.8 | 13.5 | 19 |
| $WM_1$ | 3.5  | 2.4 | 4.7  | 20 |
| LUTR   | 10.5 | 8.1 | 12.8 | 19 |
| $WM^1$ | 3.7  | 2.7 | 4.6  | 17 |

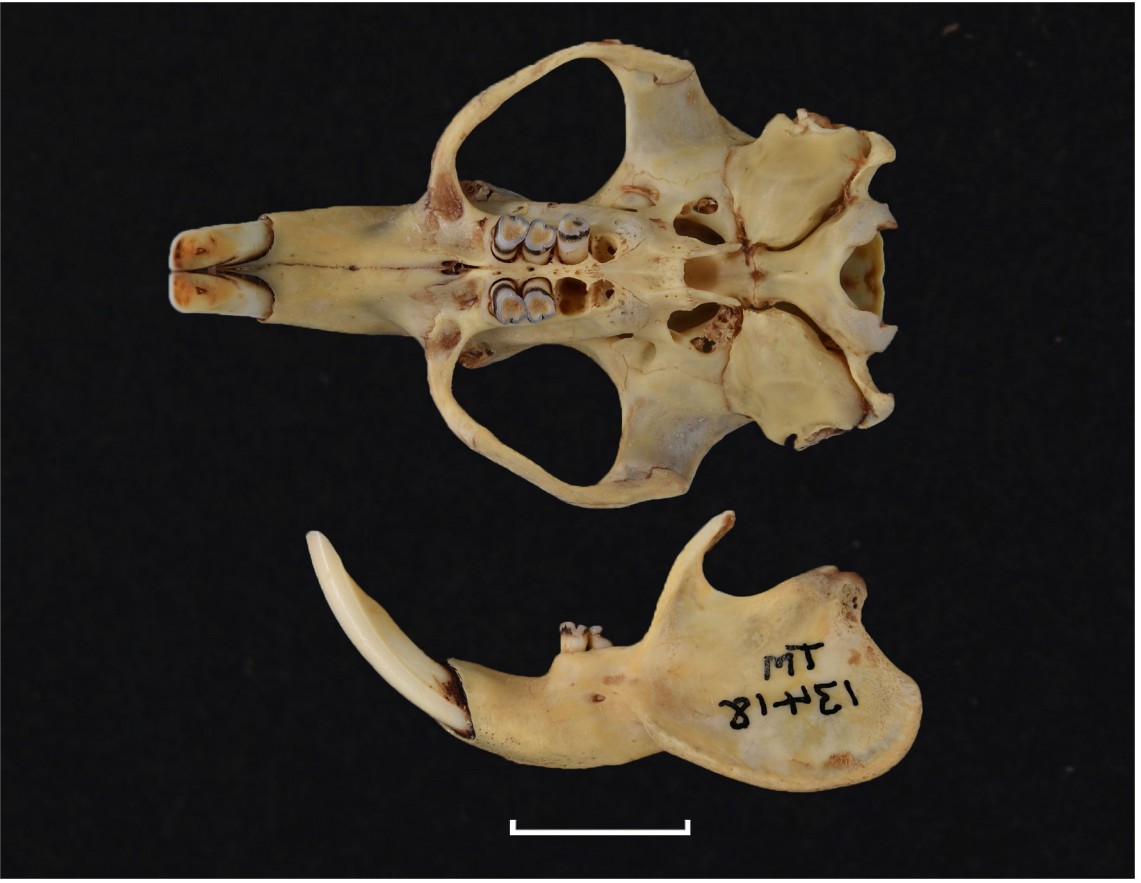

**Fig 95.  Cranium of *Cryptomys hottentotus* (DNMNH-13418), with scale bar of 1 cm.** The molars $M^3$ are missing.

species currently placed within *Fukomys*. A single species is currently recognised in South Africa:

- *Cryptomys hottentotus* (Lesson, 1826)

  Additional fossil species have been described in South Africa:

- †*Cryptomys broomi* Denys, 1998 from Langebaanweg

- †*Cryptomys robertsi* Broom, 1937

  Fossils of *Cryptomys* are known from the Early Pliocene and have been identified in many Quaternary deposits. However, most of this material was identified before the recognition of

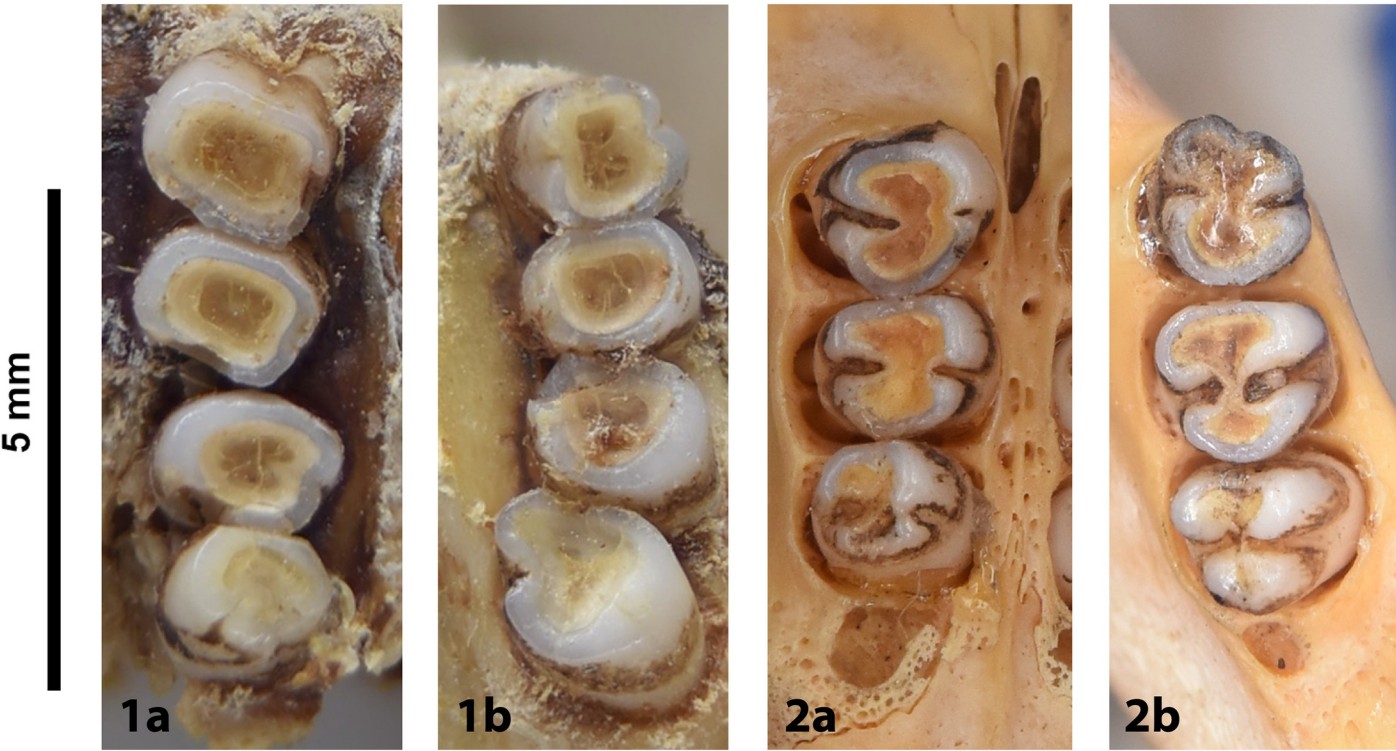

**Fig 96. Cheekteeth of *Cryptomys*. 1)** Upper (a) and lower (b) right toothrow of an adult *C. hottentotus* (DNMNH-13033); **2)** Upper (a) and lower (b) right toothrow of a juvenile *C. hottentotus* (DNMNH-3418), with molars M$^3$/M$_3$ not erupted yet.

*Fukomys* as a distinct genus. It is therefore likely that some of these remains may rather be attributed to this latter genus.

Genus ***Fukomys*** Kock et al., 2006 (Mole-rats)

Figs 98–100; Table 36

Dental formula is 1-0-1-3:1-0-1-3. Alveolar formula varies with age of specimen.

**Upper jaw.** *Cryptomys* and *Fukomys* appear to be indistinguishable on morphological grounds. See *Cryptomys* for anatomical description.

**Lower jaw.** See *Cryptomys*.

**Systematic notes and South African fossil record.** Species of this genus were previously included in *Cryptomys* until molecular analyses revealed the existence of two well separated clades within this genus [75]. A single species is currently recognized in South Africa:

- *Fukomys damarensis* (OGILBY, 1838)

No fossil remains of *Fukomys* have been identified in South Africa, probably due to the fact that *F. damarensis* has only recently been erected as a separate species from *C. hottentotus*.

Genus ***Georychus*** Illiger, 1811 (Cape Mole-rats)

Figs 101–103; Table 37

Dental formula is 1-0-1-3:1-0-1-3. Alveolar formula varies with specimen age.

**Upper jaw.** Incisors are robust, ungrooved and proodont. Anterior palatal foramina are very short, ending before molar M$^1$. Molars have single inner and outer folds of enamel jutting into the dentine. Folds on M$^2$ persist with age, on M$^3$ also to a lesser extent, and folds on P$^4$

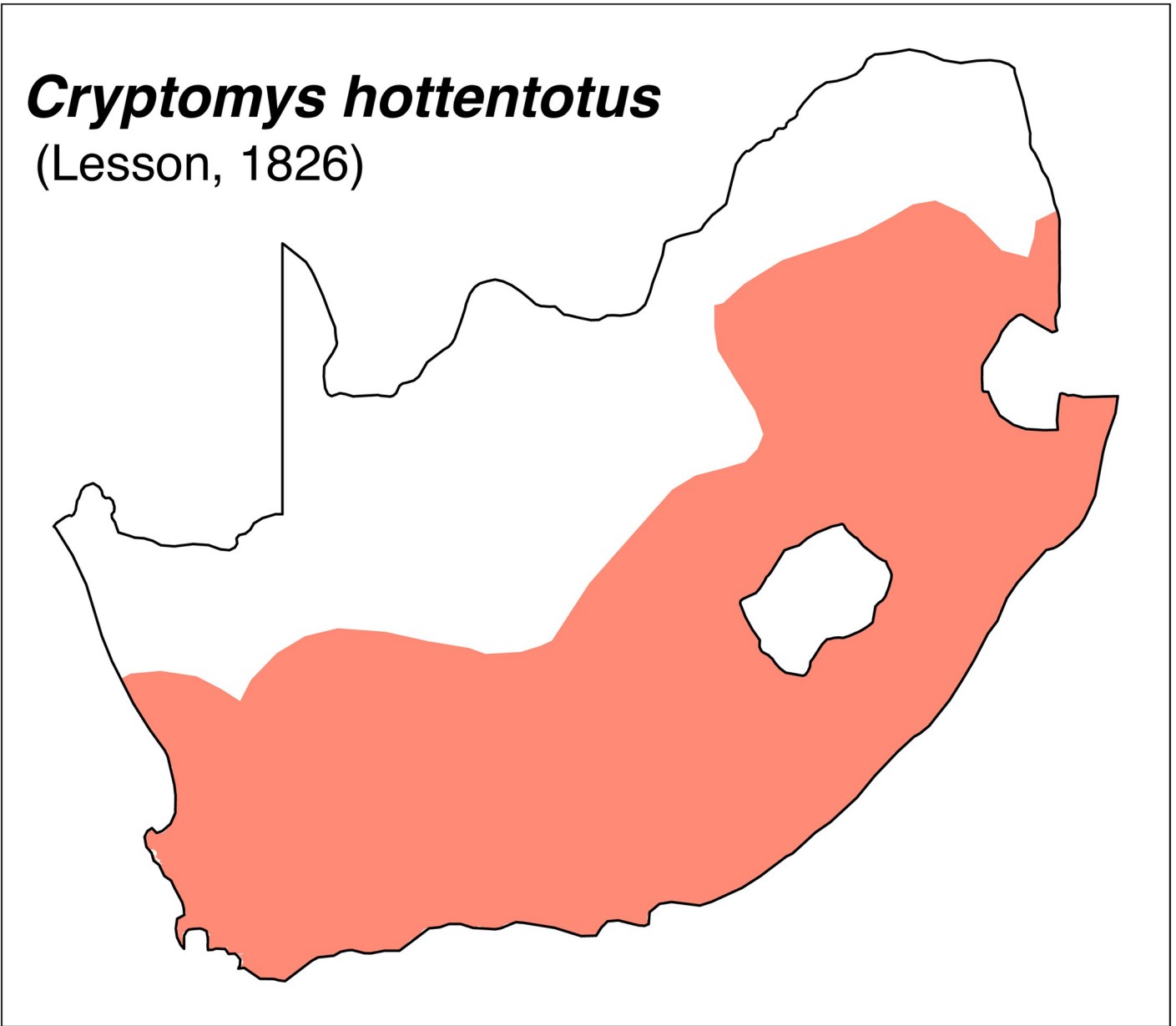

**Fig 97. Distribution map.**

**Table 35. Dental measurements (in mm) for *Cryptomys hottentotus*, sexes combined.**

|  | Mean | Min | Max | n |
|---|---|---|---|---|
| LLTR | 6.1 | 5.1 | 6.8 | 11 |
| $WM_1$ | 1.9 | 1.7 | 2.2 | 11 |
| LUTR | 6.3 | 5.2 | 6.6 | 11 |
| $WM^1$ | 2.1 | 1.9 | 2.3 | 11 |

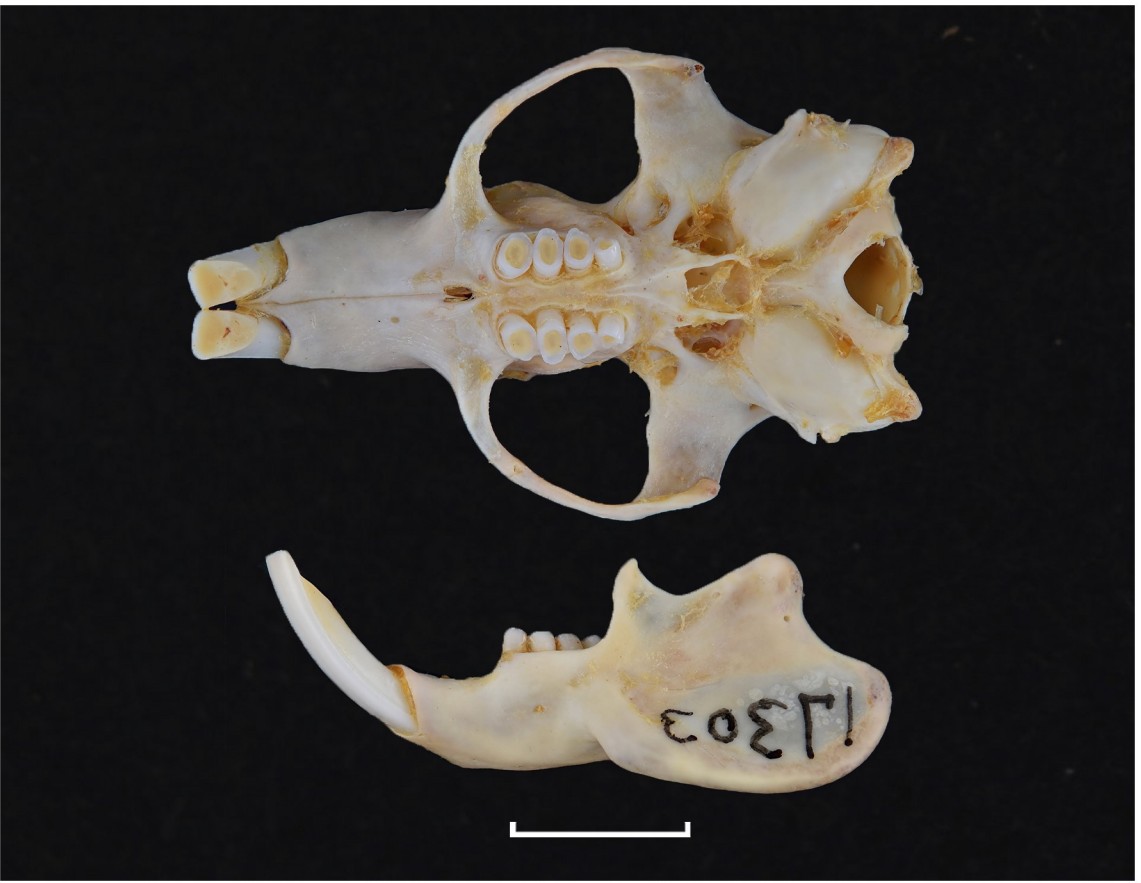

**Fig 98. Cranium of *Fukomys damarensis* (DNMNH-17303), with scale bar of 1 cm.**

and $M^1$ tend to disappear when teeth are worn. The skull is of intermediate size, being larger than *Cryptomys/Fukomys* but smaller than *Bathyergus*; it is robust and has a narrow rostrum.

**Lower jaw.** Lower incisors are smooth. Their width is more than 2.3 mm, even for young specimens. Lower molars have one outer fold that persists with age and one inner fold that tends to disappear with wear. The mental foramen is positioned posteriorly to the diastema, aligned with the posterior border of the $M_1$.

**Systematic notes and South African fossil record.** This genus is monotypic:

- *Georychus capensis* (PALLAS, 1778)

Fossil data suggest that this genus is present since the Late Pliocene in South Africa [50, 58].

## Pedetidae

The family Pedetidae contains only the genus *Pedetes*. It is a very large rodent, surpassed in weight only by *Hystrix* and large specimens of *Thryonomys*. Due to its size and high weight, eagle owls (*Bubo africanus*, *Bubo capensis* and *Bubo lacteus*) are the only nocturnal raptors capable of predating adult individuals [21, 76]. The nomenclature of the teeth of fossil and modern representatives of Pedetidae was detailed in Pickford & Mein [77].

Family **PEDETIDAE** Gray, 1825
Genus ***Pedetes*** Fitzinger, 1867 (Springhares)

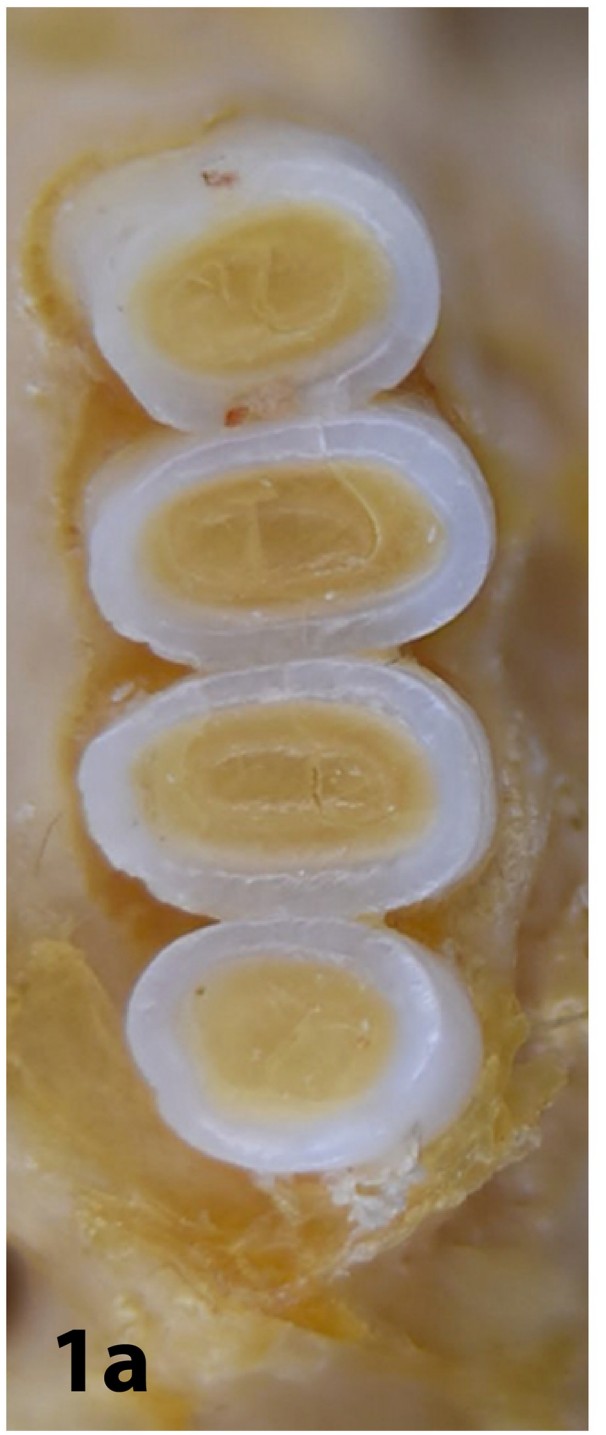 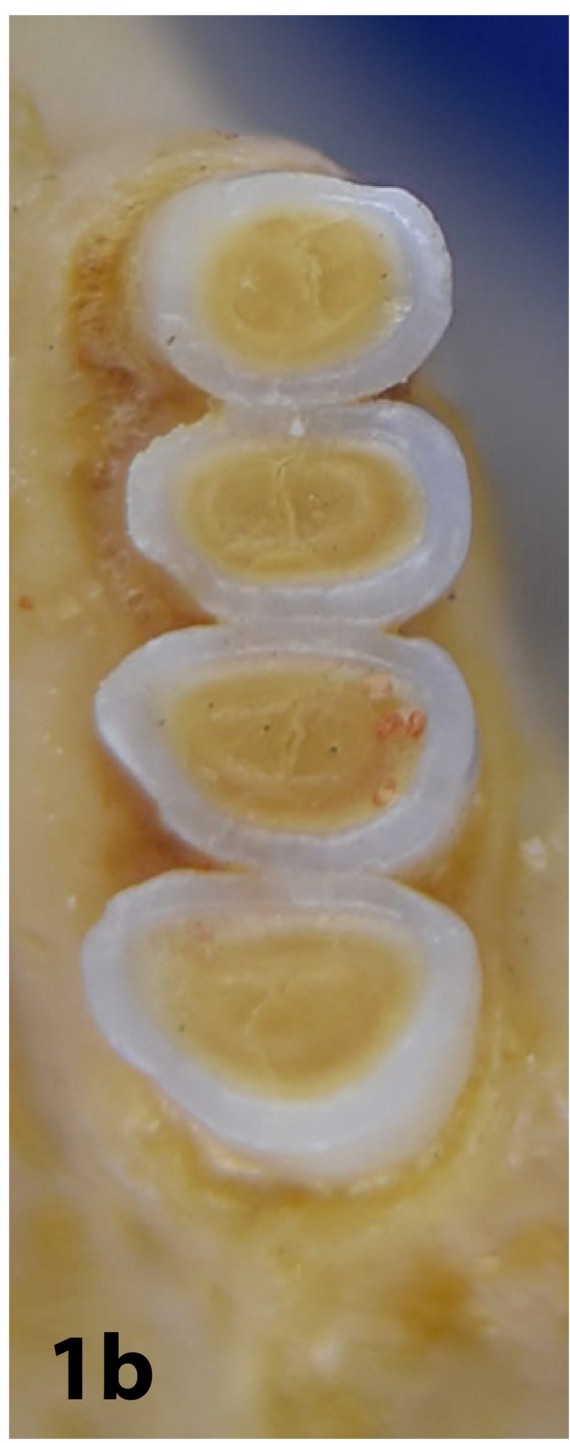

**Fig 99. Upper (a) and lower (b) right toothrow of an adult *F. damarensis* (DNMNH-17304).**

Figs 104–106; Table 38

Dental formula is 1-0-1-3:1-0-1-3.

**Upper jaw.**    The skull is large and robust. Upper incisors are opisthodont, thick and ungrooved. The anterior palatal foramina are short. The cheekteeth are all roughly the same

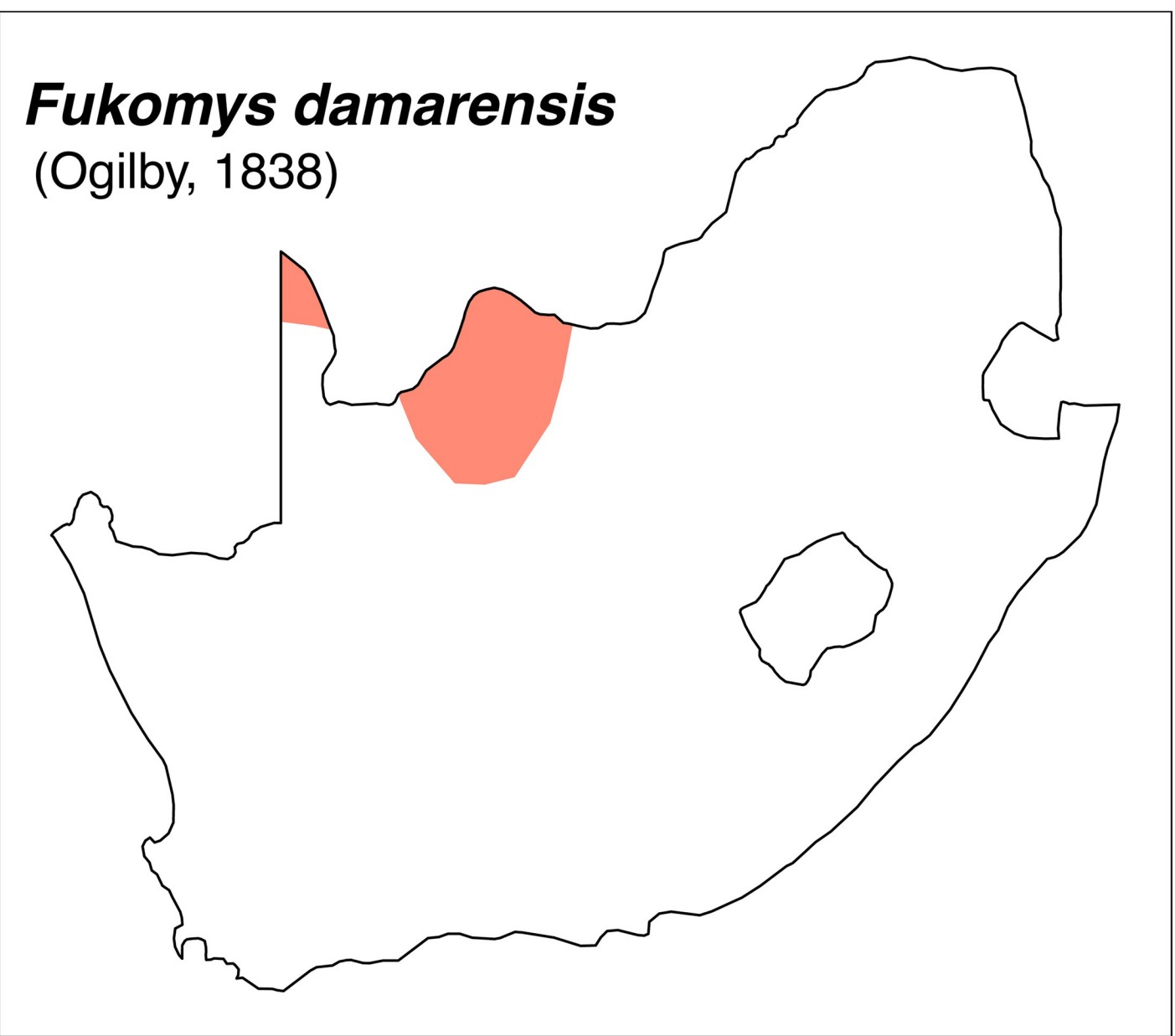

**Fig 100. Distribution map.**

**Table 36. Dental measurements (in mm) for *Fukomys damarensis*, sexes combined.**

|  | Mean | Min | Max | n |
|---|---|---|---|---|
| LLTR | 6.4 | 6.2 | 6.9 | 12 |
| $WM_1$ | 2.1 | 2.0 | 2.3 | 12 |
| LUTR | 6.8 | 6.6 | 7.7 | 11 |
| $WM^1$ | 2.4 | 2.3 | 2.7 | 12 |

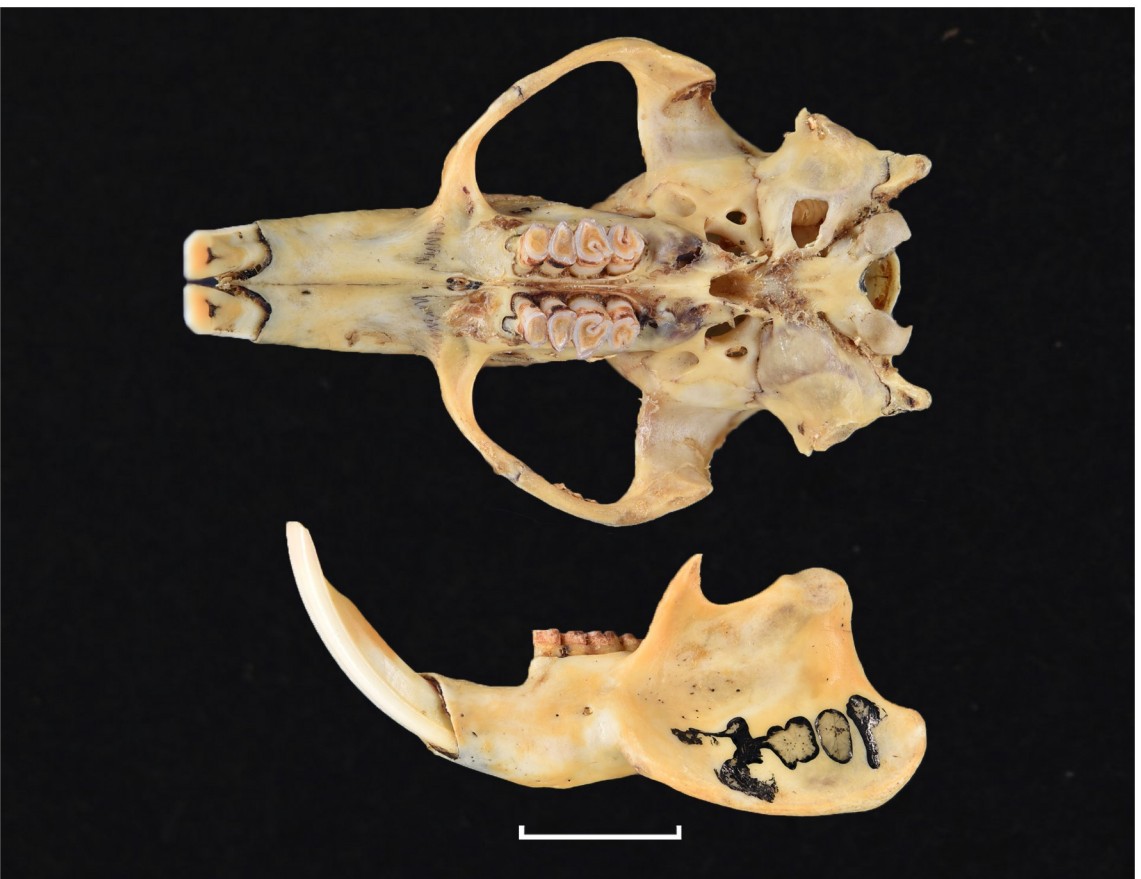

**Fig 101. Cranium of *Georychus capensis* (DNMNH-9145), with scale bar of 1 cm.**

size. They are hypsodont and open-rooted. The cheekteeth are comprised of two lophs, which are separated by a median transverse valley and uniting in the lingual side.

**Lower jaw.** The mandible is sciurognath. The condylar process reaches beyond the level of the angular process, which is poorly defined. The coronoid process consists of a thin ridge. Lower incisors are thick and ungrooved. Lower cheekteeth are bilophed like in upper tooth row but are uniting in the labial side.

**Systematic notes and South African fossil record.** A single species of *Pedetes* occurs in South Africa:

- *Pedetes capensis* (Forster, 1778)

    Two additional fossil species have been listed:

- †*Pedetes gracilis* Broom, 1934 from the Pliocene locality of Taung

- †*Pedetes hagenstadti* Dreyer and Lyle, 1931 from the Middle Pleistocene locality of Florisbad

## Hystricidae

In South Africa, only the genus *Hystrix* occurs. It contains one modern species, *H. africaeaustralis*, and one fossil species, *H. makapanensis*. *H. africaeaustralis* is the largest African rodent

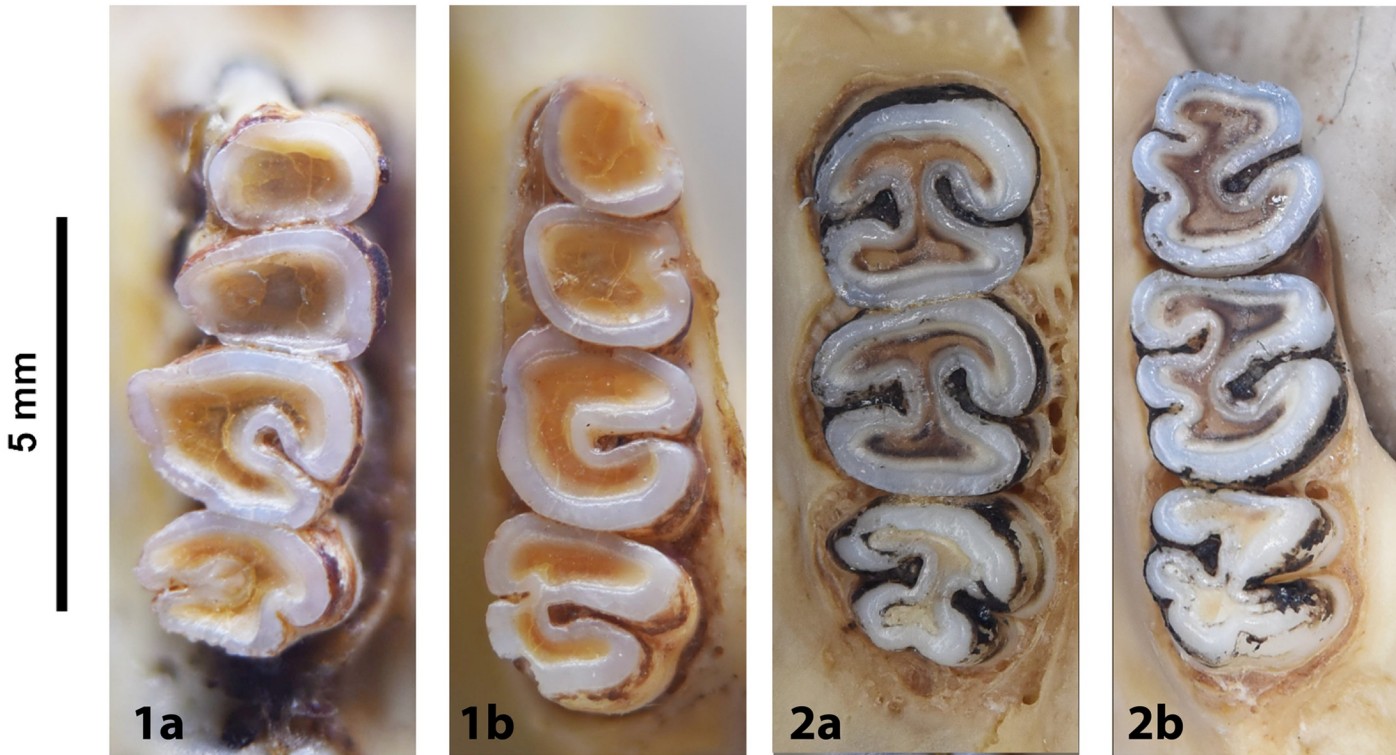

**Fig 102. Cheekteeth of *Georychus*.** 1) Upper (a) and lower (b) right toothrow of an adult *G. capensis* (DNNH-9145); 2) Upper (a) and lower (b) right toothrow of a juvenile *G. capensis* (DNMNH-4159), with molars M$^3$/M$_3$ not erupted yet.

and has long spines (quills) covering its back and flanks. Large carnivores are the most common predator. The nomenclature of the teeth of fossil and modern representatives of African *Hystrix* is detailed in Azzarà et al. [78].

Family **HYSTRICIDAE** G. Fischer, 1817

Genus ***Hystrix*** Linnaeus, 1758 (Crested Porcupines)

Figs 107–109; Table 39

Dental formula is 1-0-1-3:1-0-1-3.

**Upper jaw.**   Upper incisors are large, proodont and smooth. The anterior palatal foramina are short and located far forwards from the cheekteeth. Molars are large and tend towards hypsodonty. They display cusps related by lophs when unworn, but the crowns rapidly reduce in height with age, and wear of the tooth and cusps get individualized into small enamel islands. The teeth all have only one fold (also called *flexus*) on the lingual side and up to three folds on the labial side (this arrangement is revers in the lower teeth).

**Lower jaw.**   Lower incisors are large and ungrooved. Lower teeth display the same characteristic occlusal pattern with crests of enamel, but there is one labial fold and up to three lingual folds. The mandible is hystricognath.

**Systematic notes and South African fossil record.**   The following species occurs in South Africa:

• *Hystrix africaeaustralis* PETERS, 1852

An additional species is listed in the South African fossil record:

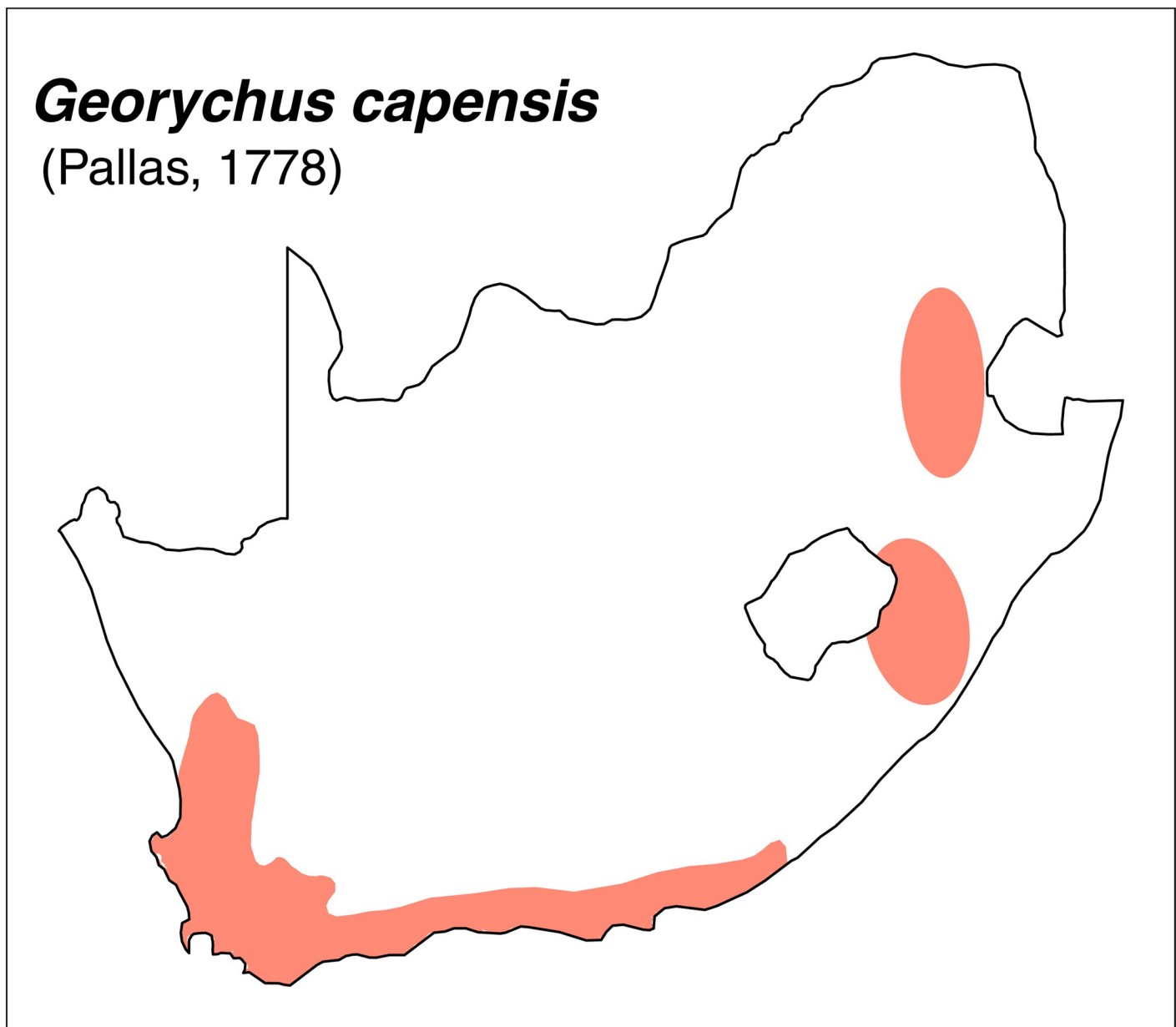

**Fig 103. Distribution map.**

**Table 37. Dental measurements (in mm) for *Georychus capensis*, sexes combined.**

|  | Mean | Min | Max | n |
|---|---|---|---|---|
| LLTR | 7.9 | 7.1 | 8.6 | 10 |
| $WM_1$ | 2.4 | 2.0 | 2.9 | 10 |
| LUTR | 7.7 | 6.7 | 8.3 | 12 |
| $WM^1$ | 2.7 | 2.3 | 3.1 | 12 |

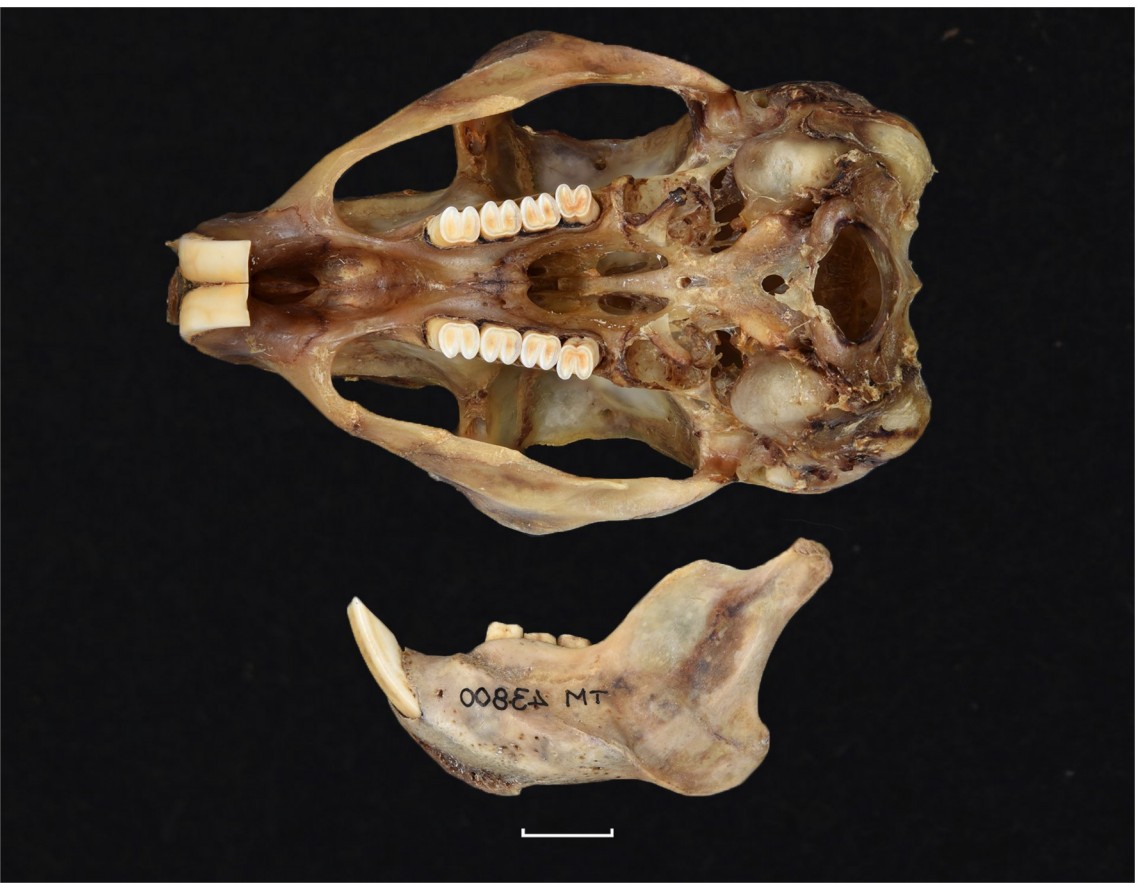

**Fig 104. Cranium of *Pedetes capensis* (skull DNMNH-44480, mandible DNMNH-43800), with scale bar of 1 cm.**

- *†Hystrix makapanensis* GREENWOOD, 1958 found in the Pliocene Limeworks of Makapansgat and several Early Pleistocene deposits from the Sterkfontontein Valley in Gauteng Province [78].

  Remains of *Hystrix* are found in many fossil deposits through the Quaternary.

## Thryonomyidae

The family Thryonomyidae includes the single genus, *Thryonomys*. It is the third largest rodent genus in South Africa, after *Hystrix* and *Pedetes*. Like the latter, eagle owls are the only nocturnal raptors susceptible of predating adults [21] but mesocarnivores can prey upon this genus.

Family **THRYONOMYIDAE** Pocock, 1922

Genus ***Thryonomys*** Fitzinger, 1867 (Cane Rats)

Figs 110–112; Table 40

Dental formula is 1-0-1-3:1-0-1-3.

**Upper jaw.**   Upper incisors are broad, opisthodont, and have three grooves on the inner side. The skull is stickily built with a short rostrum. The anterior palatal foramina reach the $M^1$. The cheekteeth have three to four characteristic enamel infoldings, which are single in the lingual side and double in the labial side. The premolar $P^4$ has four lophs, while the molars

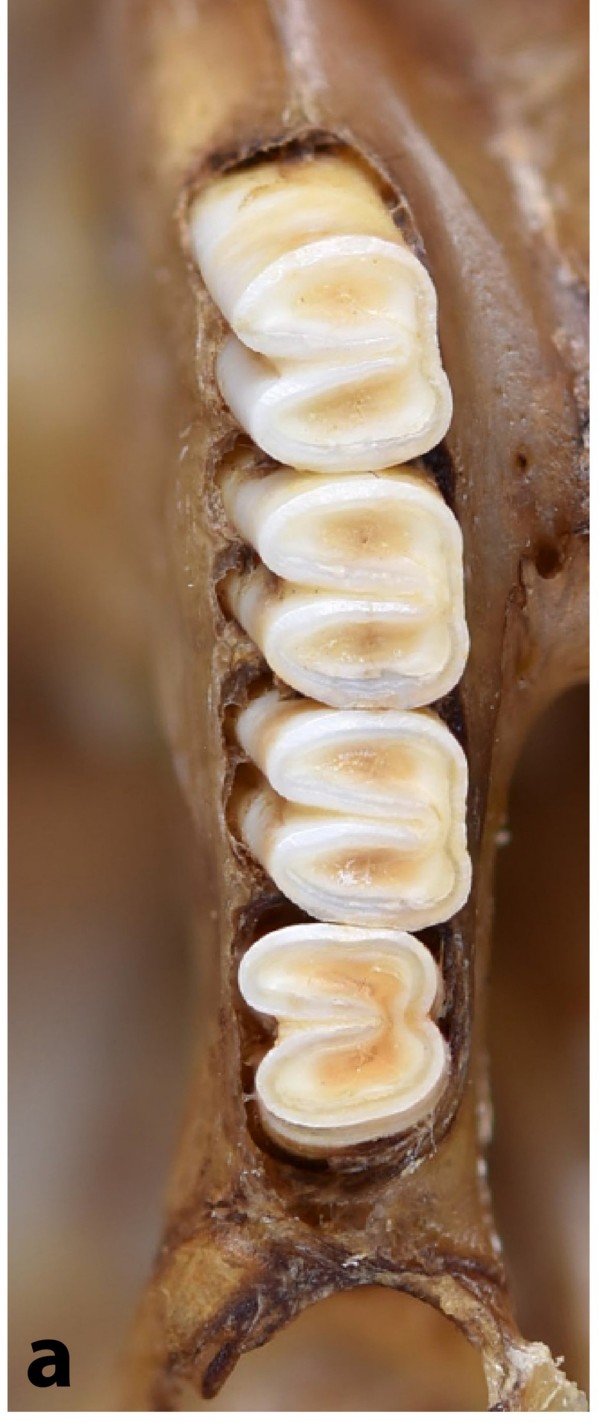
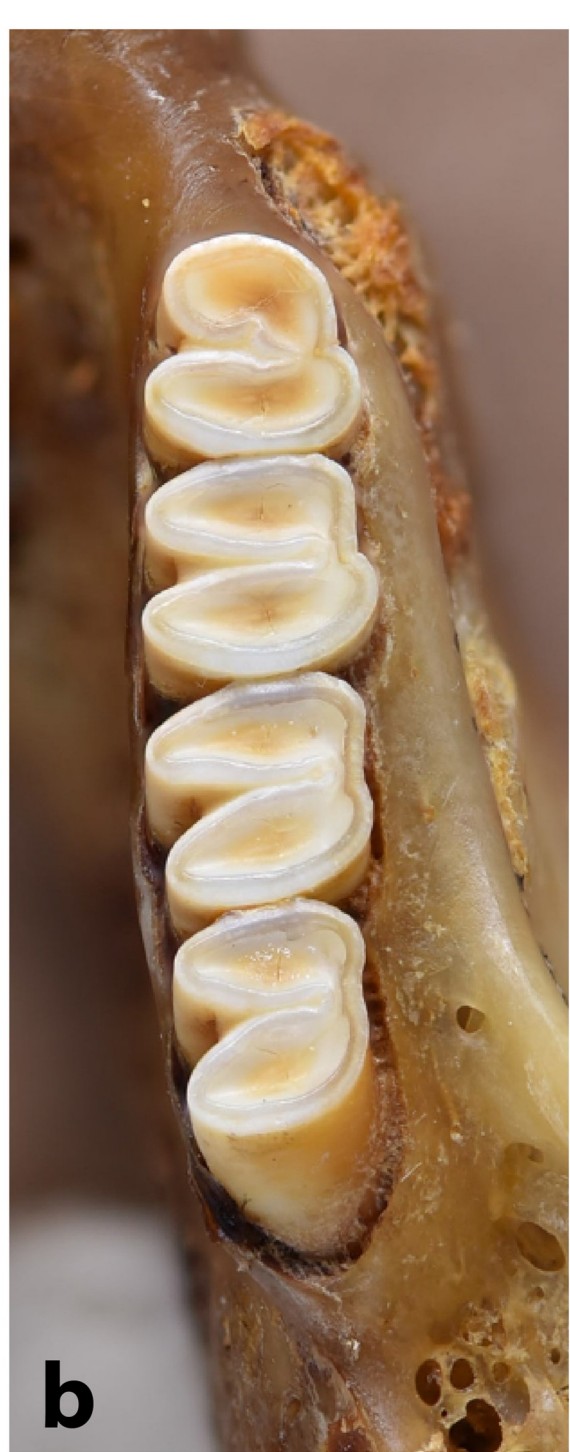

**Fig 105. Upper (a) and lower (b) right toothrow of *P. capensis* (DNMNH-44480).**

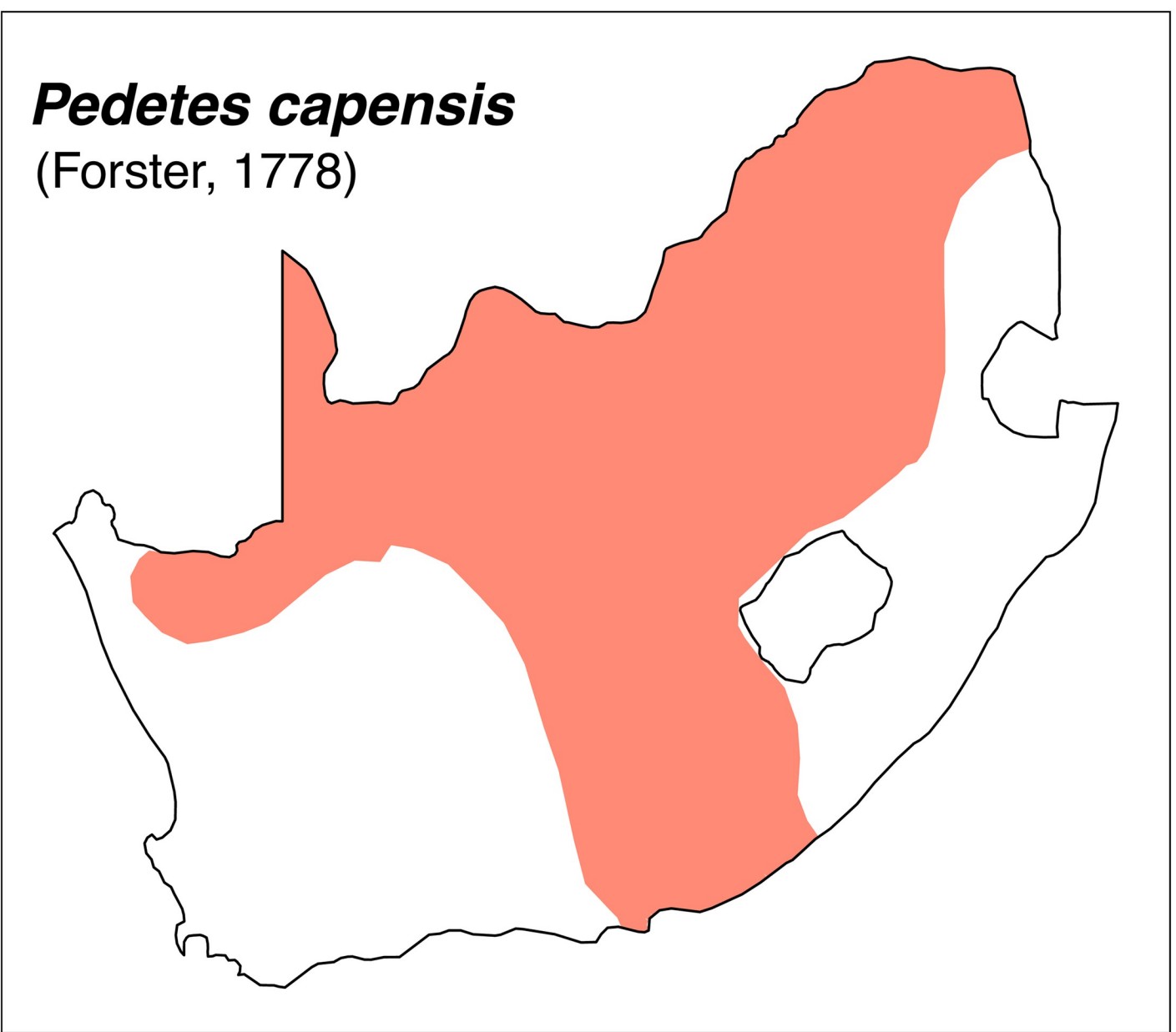

**Fig 106. Distribution map.**

**Table 38. Dental measurements (in mm) for *Pedetes capensis*, sexes combined.**

|  | Mean | Min | Max | n |
|---|---|---|---|---|
| LLTR | 18.3 | 16.2 | 19.9 | 16 |
| $WM_1$ | 4.4 | 3.5 | 5.5 | 16 |
| LUTR | 18.2 | 16.1 | 19.8 | 11 |
| $WM^1$ | 3.9 | 3.4 | 4.2 | 14 |

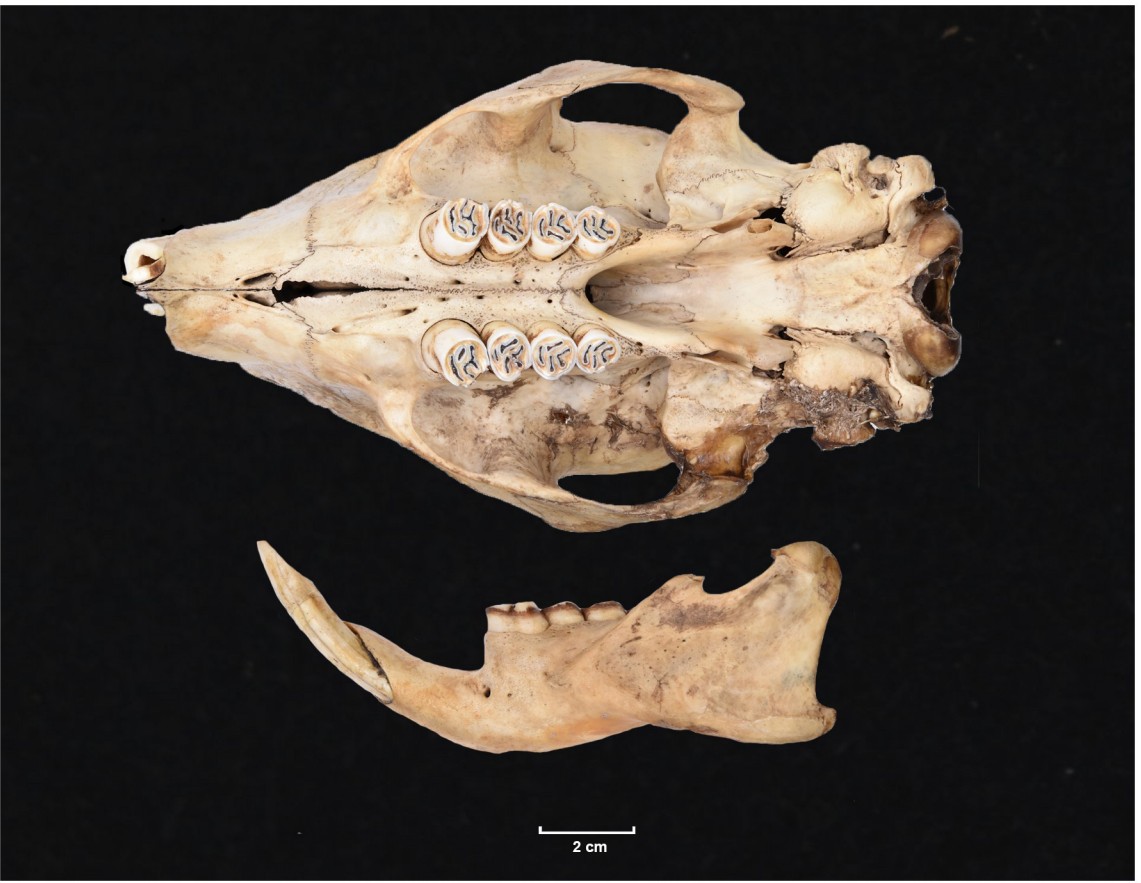

**Fig 107. Cranium of *Hystrix africaeaustralis* (ESI BPI-4-881), with scale bar of 2 cm.**

$M^1$-$M^3$ have only three. The $M^2$ is often the largest tooth in the row, and the $M^3$ erupts late in the life on the individual.

**Lower jaw.** Lower incisors are ungrooved. The cheekteeth have two outer and one inner enamel infoldings (it is reversed in the maxilla). As for the lower teeth, there are four lophs in $P^4$ and three in the molars $M^1$-$M^3$. The mandible is robust, and the well-developed angular process projects far backwards, and is hystricognathous.

**Systematic notes and South African fossil record.** In South Africa, a single species occurs:

- *Thryonomys swinderianus* (Thomas, 1894).

Its remains are scanty in the Quaternary fossil record, and it is found for the first time during the Late Pleistocene from Umhlatuzana [79] and Sibudu [80] caves.

## Petromuridae

The family Petromuridae contains only a single genus, *Petromus*. These medium-sized rodents are squirrel-like in appearance, and their present-day distribution is restricted to the west coast of Southern Africa. They are associated with rocky habitats, occupying rock crevices in boulders, canyons and mountain slopes [21]. Little is known about its predators, but remains were identified within pellet material produced by *Bubo africanus* [81].

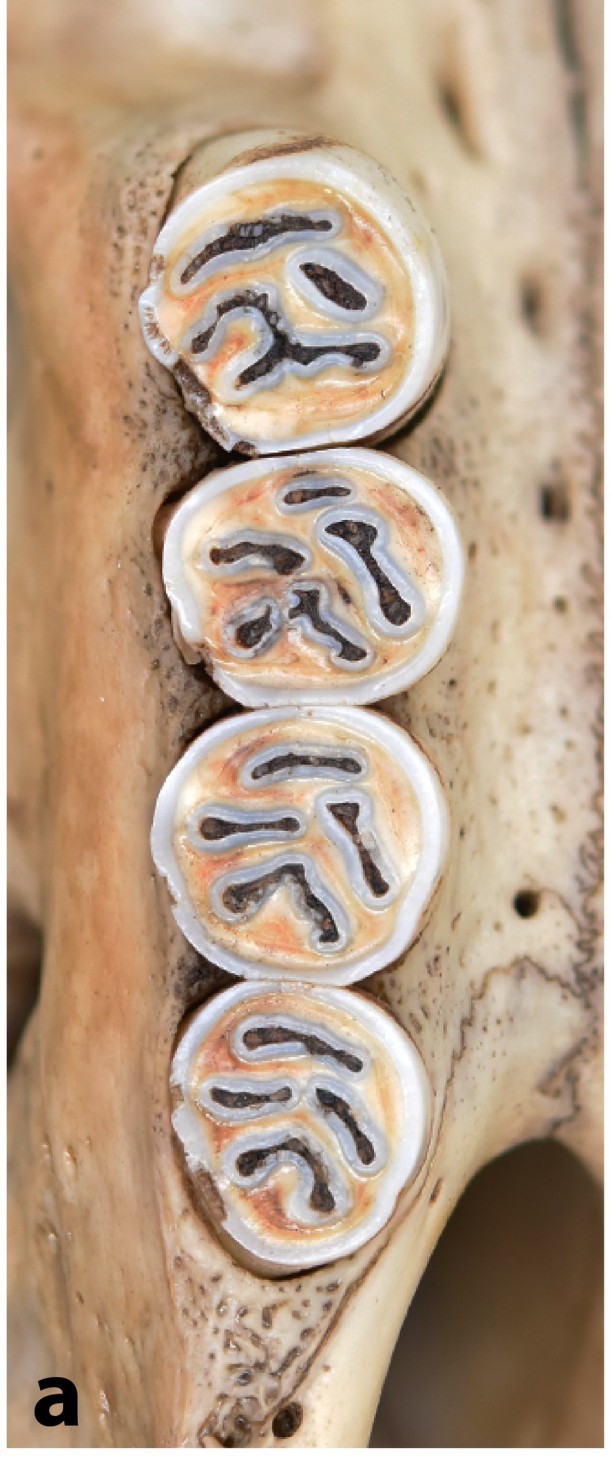
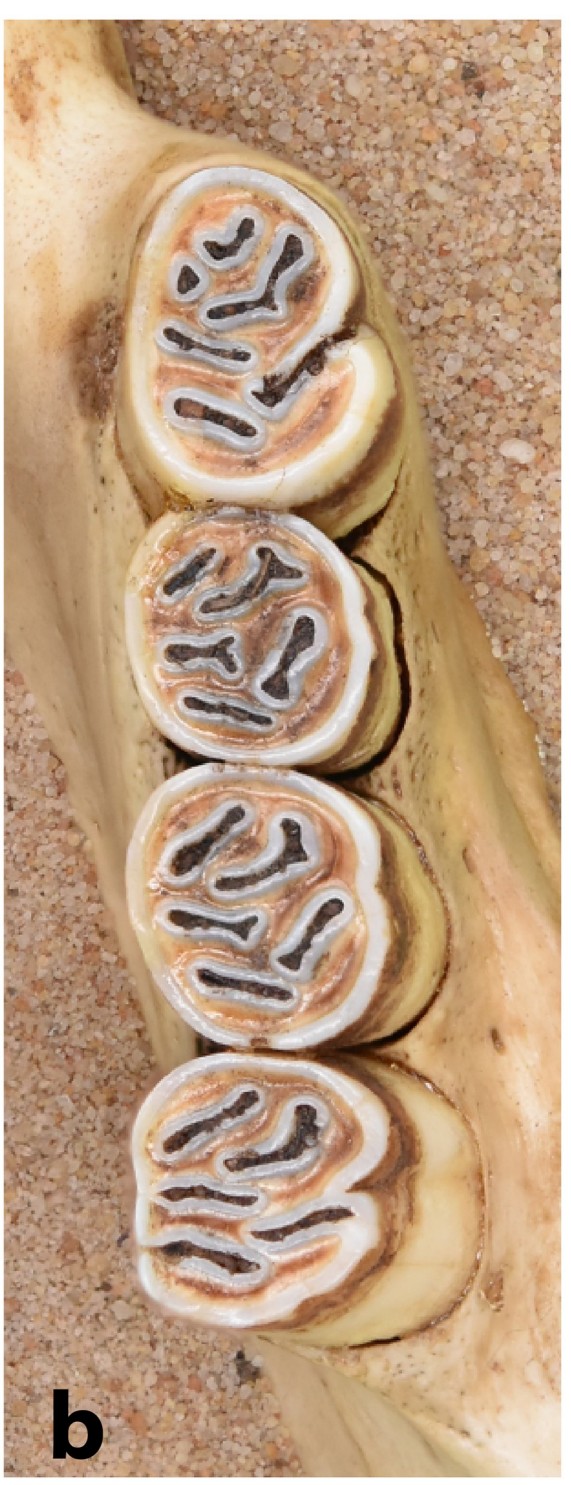

**Fig 108. Upper (a) and lower (b) right toothrow of** *H. africaeaustralis* **(ESI BPI-4-881).**

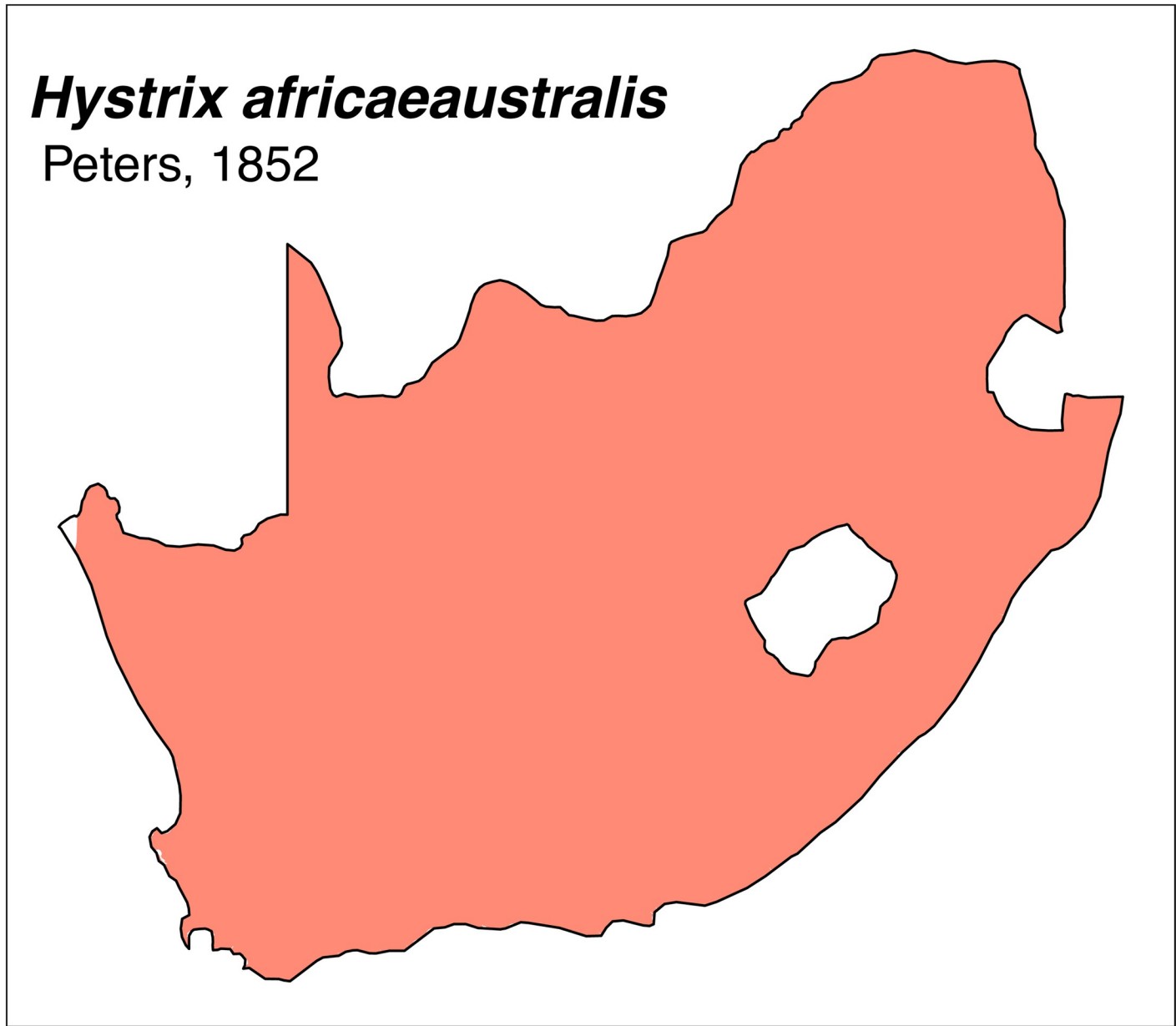

**Fig 109. Distribution map.**

**Table 39. Dental measurements (in mm) for *Hystrix africaeaustralis*, sexes combined.**

|  | Mean | Min | Max | n |
|---|---|---|---|---|
| LLTR | 36.9 | 33.8 | 39.2 | 7 |
| $WM_1$ | 8.2 | 7.7 | 8.7 | 7 |
| LUTR | 36.1 | 31.8 | 39.9 | 8 |
| $WM^1$ | 9.0 | 6.9 | 10.1 | 8 |

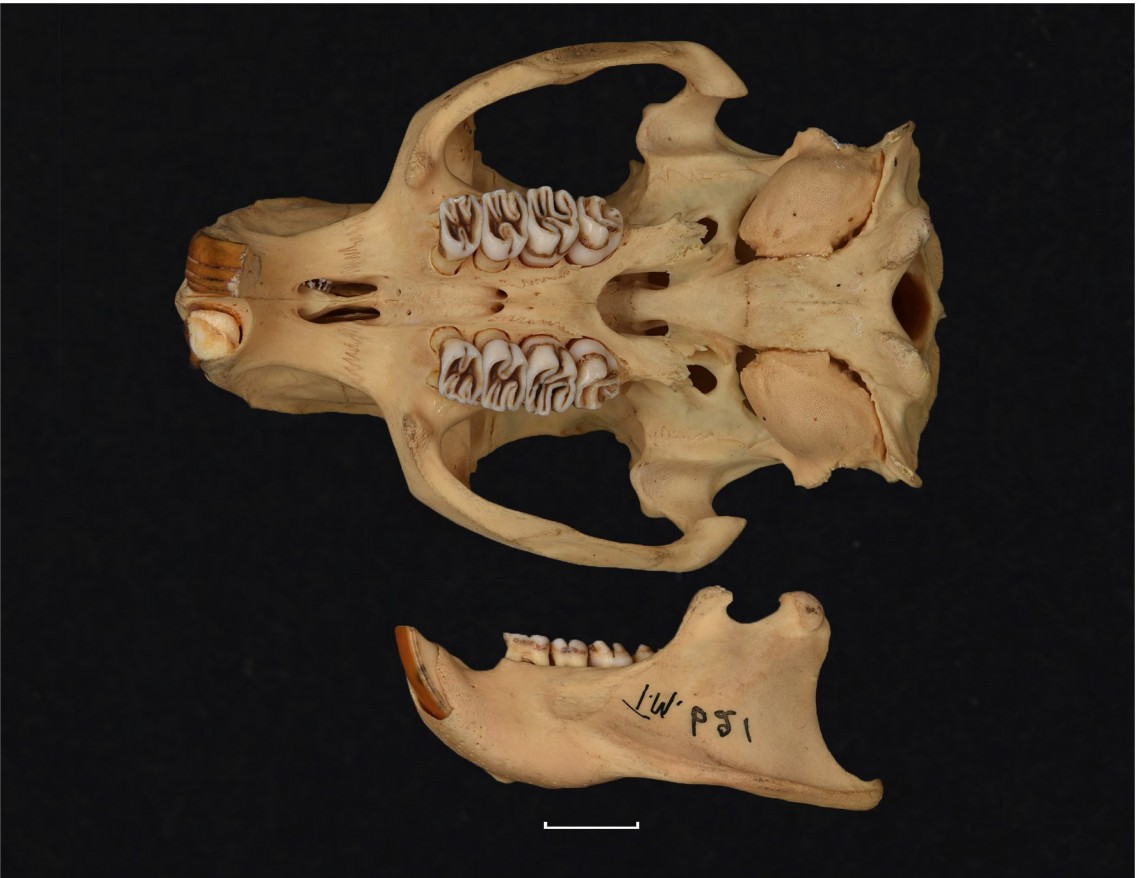

**Fig 110. Cranium of *Thryonomys swinderianus* (DNMNH-621), with scale bar of 1 cm.**

Family **PETROMURIDAE** Tullberg, 1899
Genus ***Petromus*** A. Smith, 1831 (Nokis or Dassie Rats)
Figs 113–115; Table 41
Dental formula is 1-0-1-3:1-0-1-3.

**Upper jaw.** The skull is dorsoventrally flat with inflated bulla. Upper incisors are plain and opisthodont. The palate is narrow, about equal to the width of the $P^4$ in its anterior part. The anterior palatal foramina are long and reach the $P^4$. The four cheekteeth have roughly the same size. They are hypsodont with deep lingual enamel infoldings.

**Lower jaw.** Lower incisors are ungrooved. There are four cheekteeth, which are hypsodont with deep buccal enamel infoldings. The mandible is very long and vertically compressed with a sharp angular process, showing hystricognathy.

**Systematic notes and South African fossil record.** The genus is monotypic:

- *Petromus typicus* A. Smith, 1831

Two additional fossil species have been described in the South African fossil record:

- †*Petromus antiquus* Sénégas, 2004 from the Early Pliocene site of Waypoint 160

- †*Petromus minor* Broom, 1939 from the Late Pliocene site of Taung

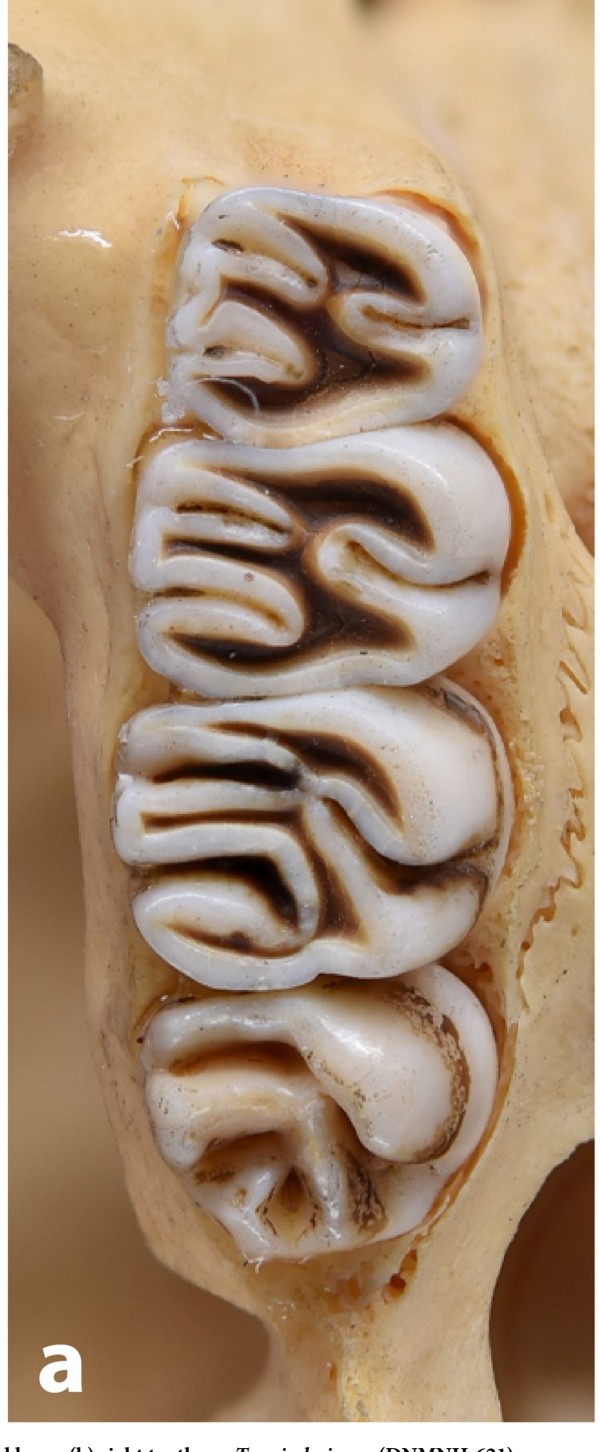
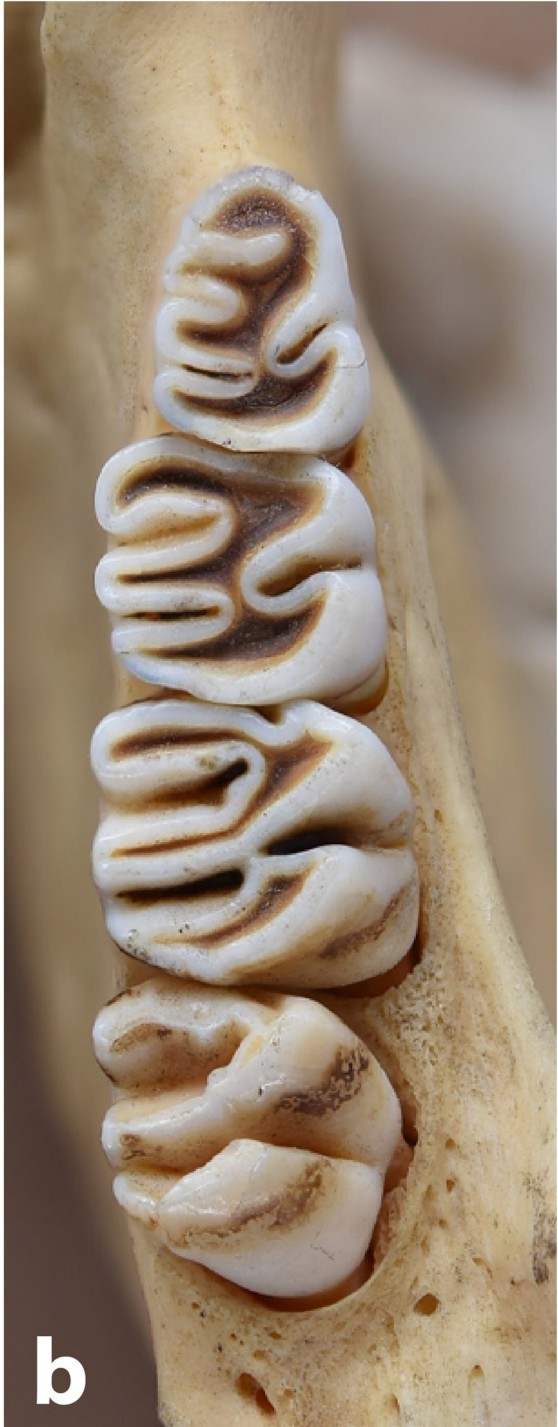

**Fig 111. Upper (a) and lower (b) right toothrow *T. swinderianus* (DNMNH-621).**

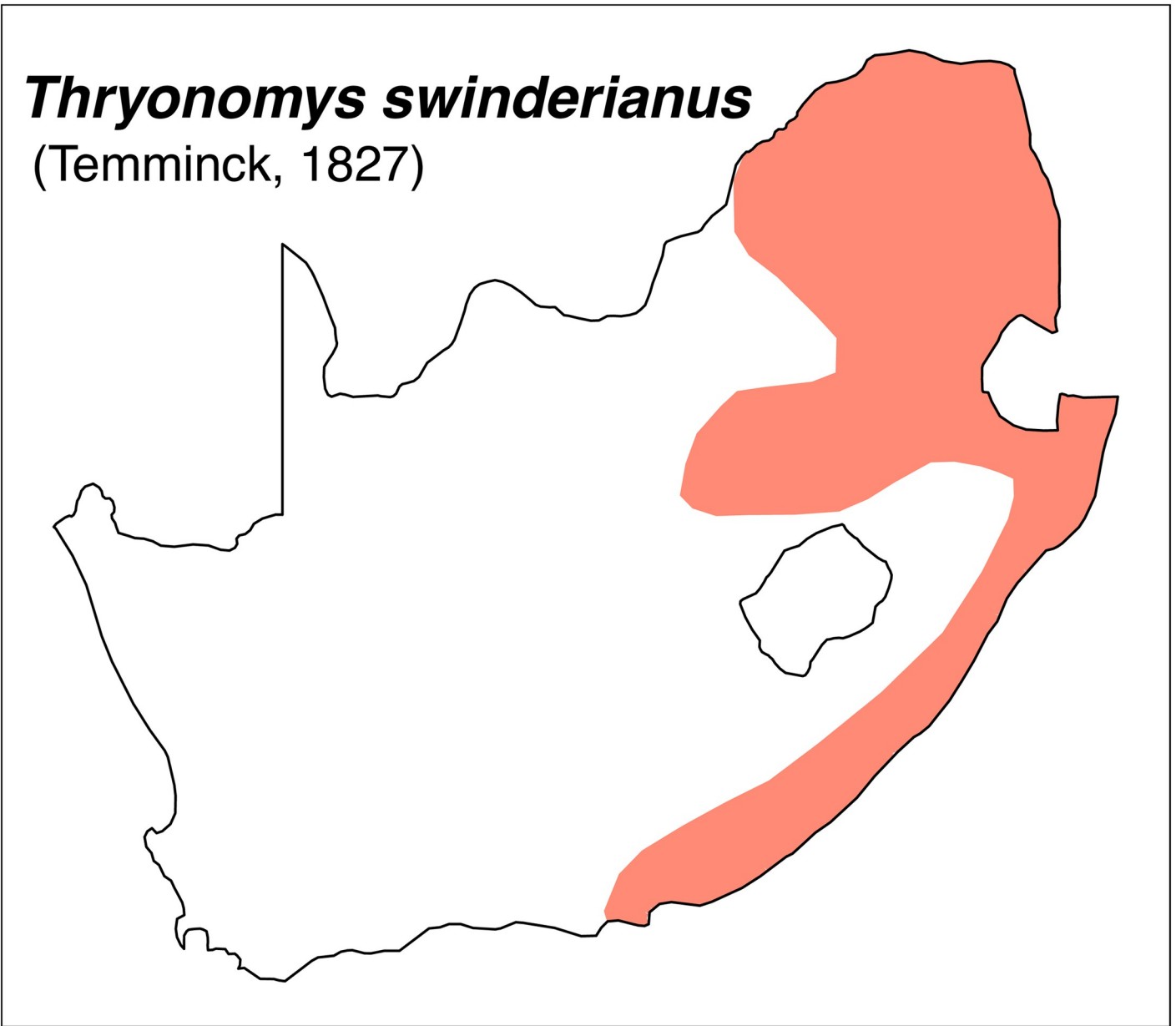

**Fig 112. Distribution map.**

**Table 40. Dental measurements (in mm) for *Thryonomys swinderianus*, sexes combined.**

|  | Mean | Min | Max | n |
|---|---|---|---|---|
| LLTR | 22.0 | 17.8 | 23.7 | 14 |
| $WM_1$ | 5.7 | 4.4 | 8.1 | 26 |
| LUTR | 18.8 | 15.1 | 20.6 | 15 |
| $WM^1$ | 7.0 | 5.3 | 9.3 | 15 |

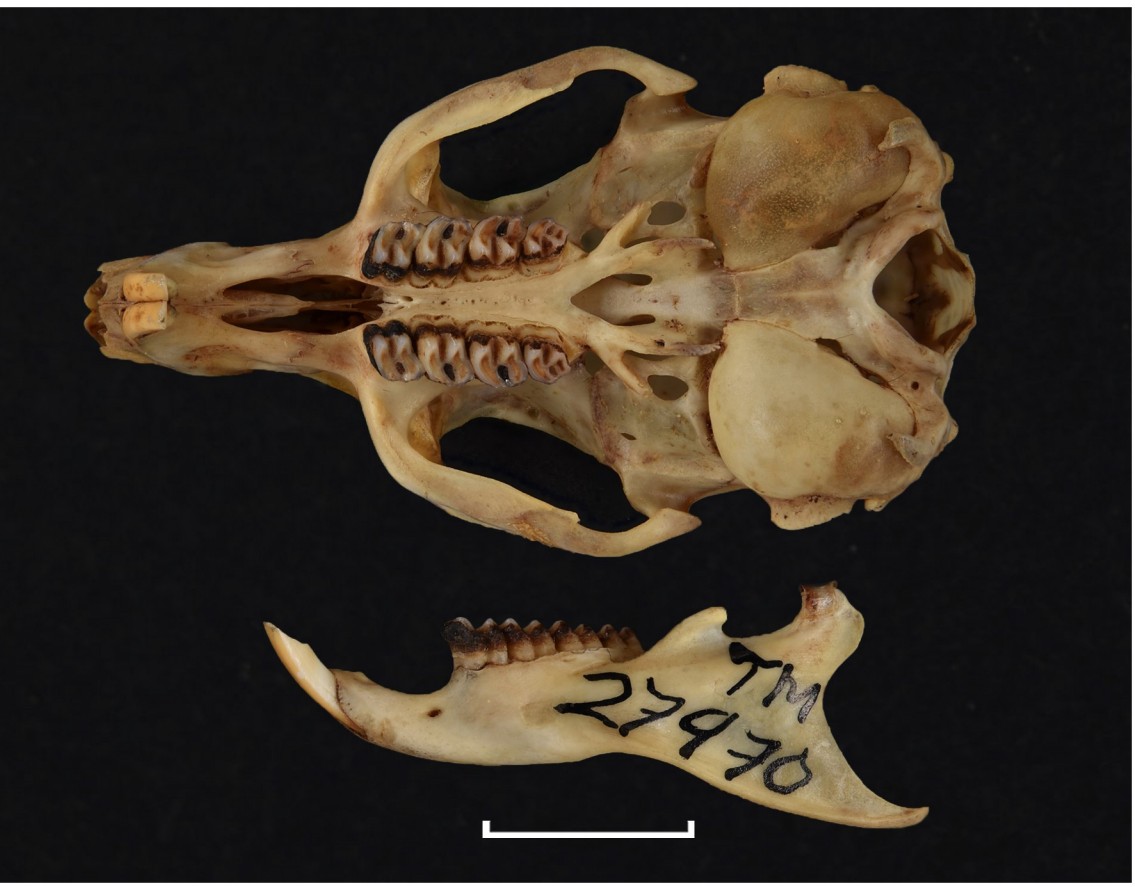

**Fig 113. Cranium of *Petromus typicus* (DNMNH-27970), with scale bar of 1 cm.**

## Gliridae

Only one genus of Gliridae occur in Southern Africa, the African dormouse *Graphiurus*. Species of *Graphiurus* are small-sized, squirrel-like rodents with good climbing abilities. They are predominantly arboreal, or are associated with boulders and rocky outcrops [22]. They are preyed upon by various predators, including owls.

Family **GLIRIDAE** Thomas, 1897
Subfamily **GRAPHIURINAE** Winge, 1887
Genus *Graphiurus* Smuts, 1832 (Dormice)
Figs 116–118; Table 42
Dental formula is 1-0-1-3:1-0-1-3.

**Upper jaw.** Incisors are ungrooved, and orthodont to slightly opisthodont. Palatal foramina are small and reach far before the toothrow. The palate is wide, about equal to the length of the upper toothrow. The premolar $P^4$ is roughly the same size or a bit smaller than $M^3$, except in *G. ocularis* where $P^4$ is much reduced. The occlusal surface of the teeth displays faint transverse ridges.

**Lower jaw.** Incisors are ungrooved. Lower cheekteeth also display faint transverse ridges. The mandible is elongated with a long coronoid process and a tilted angular process. The mental foramen is well marked.

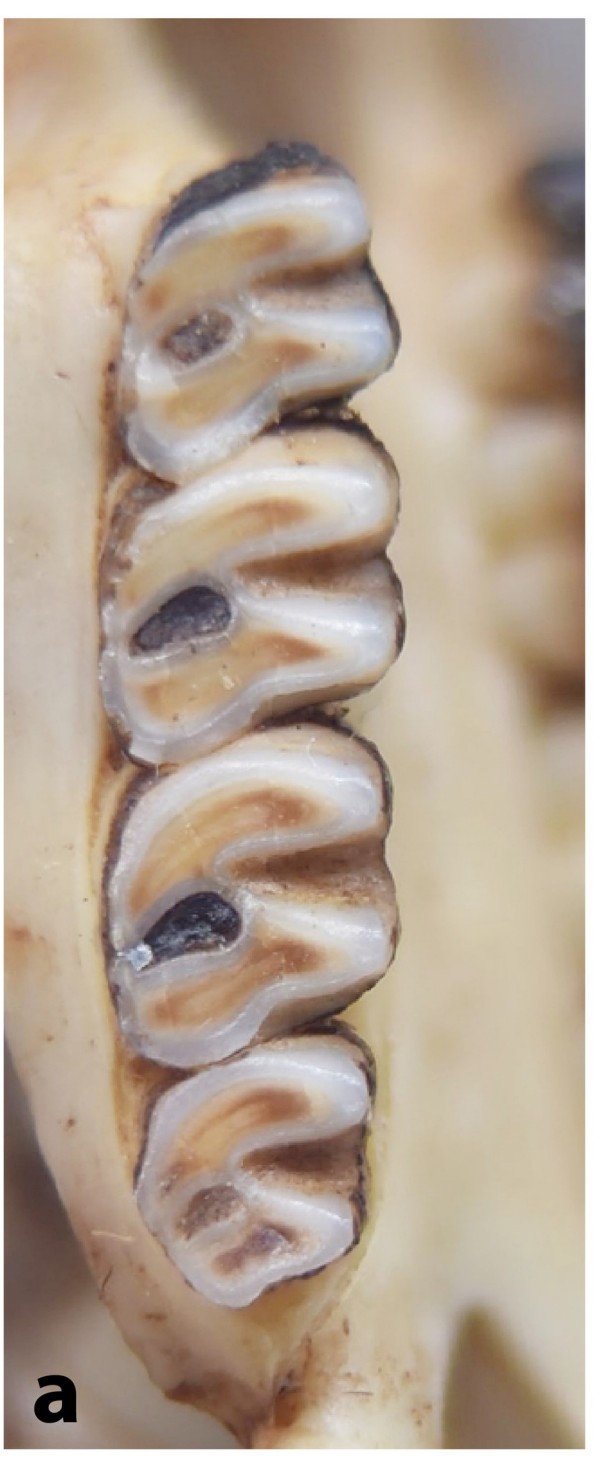 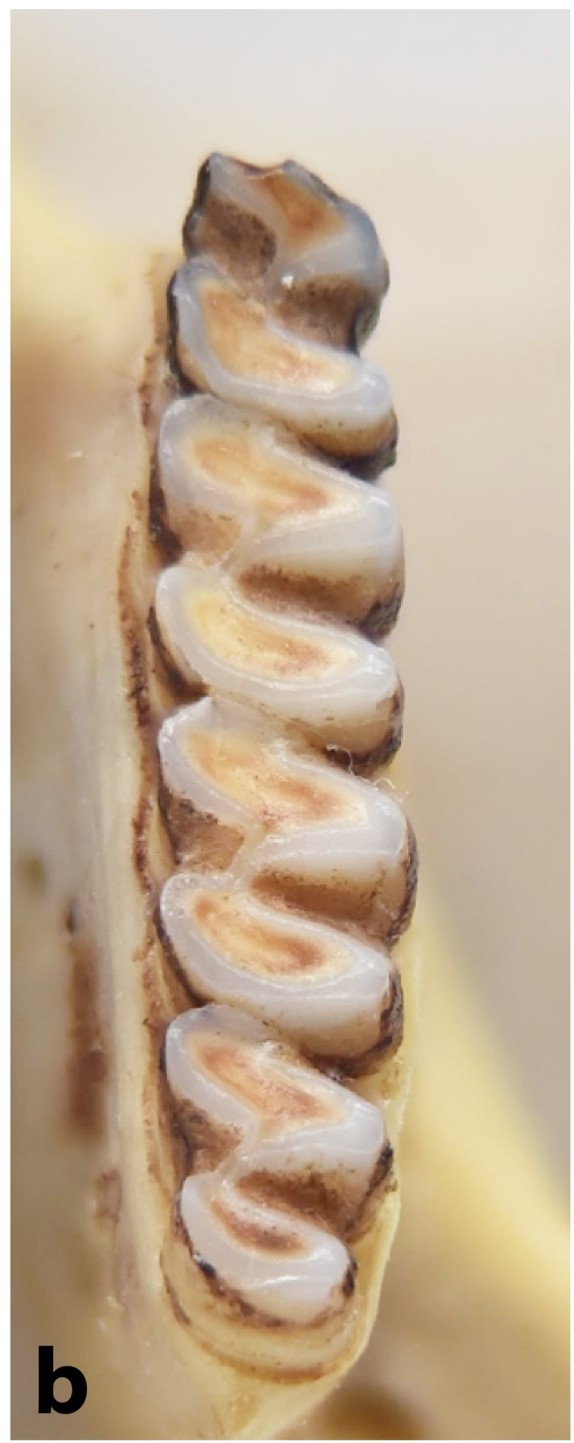

**Fig 114. Upper (a) and lower (b) right toothrow of *P. typicus* (DNMNH-27970).**

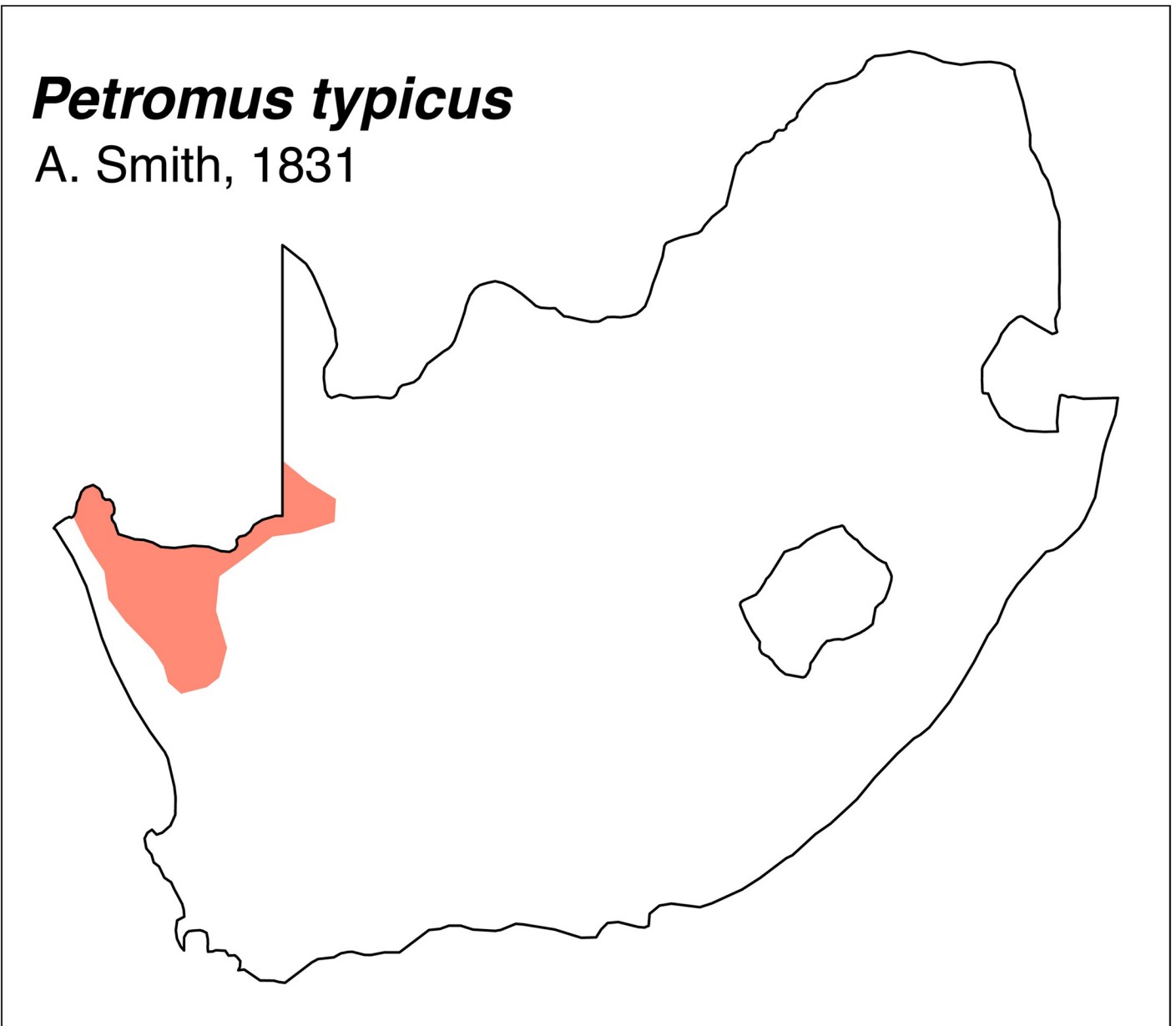

**Fig 115. Distribution map.**

**Table 41. Dental measurements (in mm) for *Petromus typicus*, sexes combined.**

|  | Mean | Min | Max | n |
|---|---|---|---|---|
| LLTR | 10.5 | 9.5 | 11.0 | 7 |
| $WM_1$ | 2.4 | 2.2 | 2.7 | 7 |
| LUTR | 9.9 | 9.5 | 10.8 | 7 |
| $WM^1$ | 2.8 | 2.6 | 3.2 | 7 |

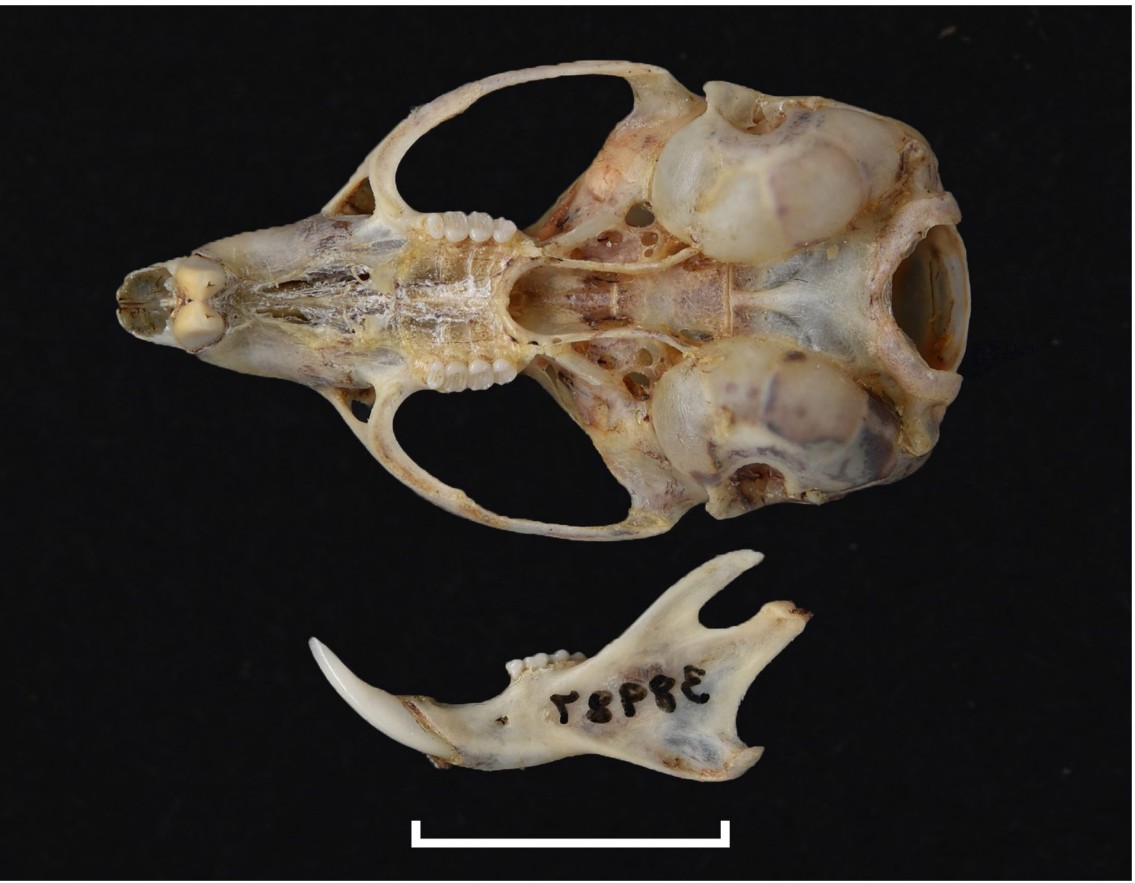

**Fig 116. Cranium of *Graphiurus murinus* (DNMNH-23386), with scale bar of 1 cm.**

**Systematic notes and South African fossil record.**   Five species of *Graphiurus* are currently described in South Africa:

- *Graphiurus microtis* (NOACK, 1887)

- *Graphiurus murinus* (DESMEREST, 1822)

- *Graphiurus ocularis* (SMITH, 1829)

- *Graphiurus platyops* Thomas, 1897

- *Graphiurus rupicola* (THOMAS & HINTON, 1925)

The oldest remains of *Graphiurus* in South Africa are found in several Early Pleistocene deposits from the Sterkfontein Valley, in Gauteng Province [5, 47, 48, 74, 82].

## Sciuridae

Some two genera of Sciuridae (Tables 43 and 44) occur in RSA: *Paraxerus* and *Geosciurus* (which was included until recently in the genus *Xerus*). Species of the genus *Paraxerus* are small to medium-sized tree squirrels, which have predominantly diurnal and arboreal habits. They typically nest in tree holes and occupy forests and woodland areas. They are preyed upon by owls [24]. *Geosciurus* is a larger ground-dwelling squirrel, being terrestrial and resting in

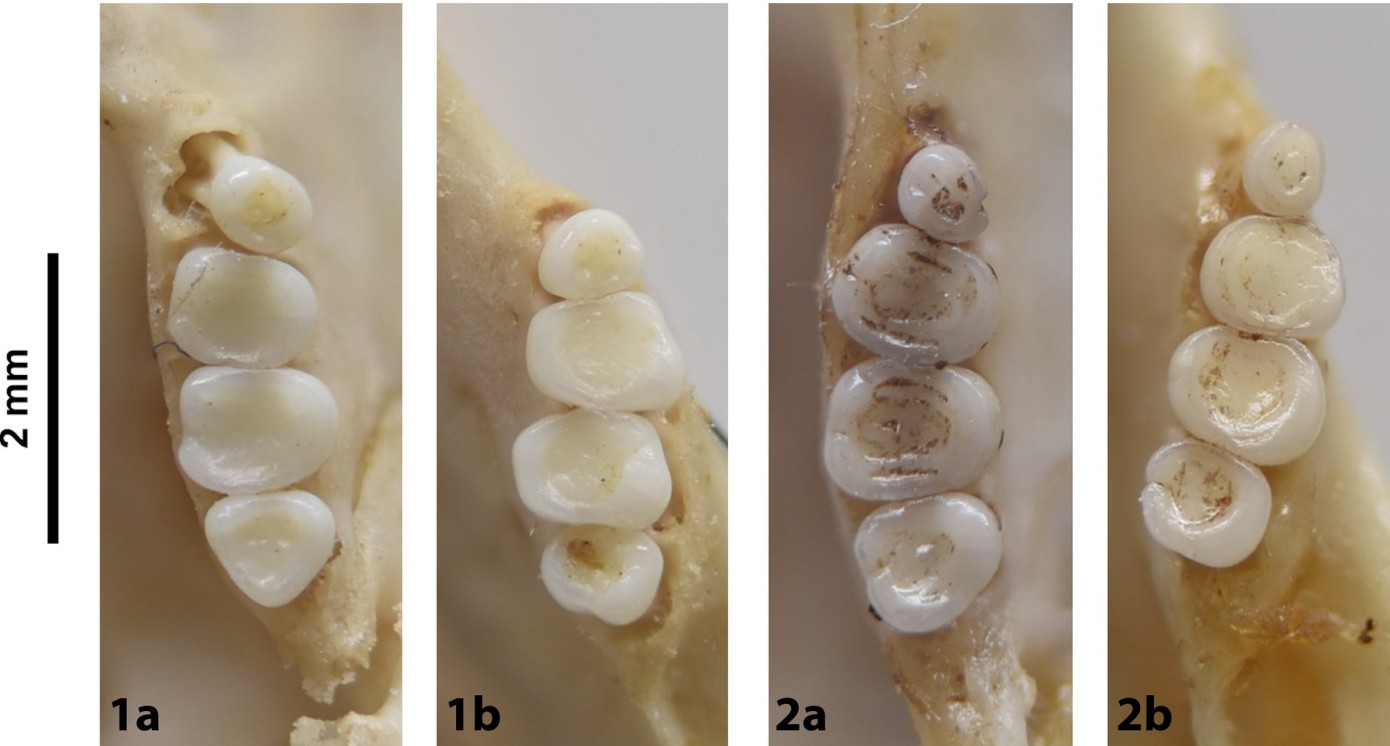

**Fig 117. Cheekteeth of *Graphiurus*.** 1) Upper (a) and lower (b) right toothrow of *G. platyops* (DNMNH-4360); 2) Upper (a) and lower (b) right toothrow of *G. ocularis* (DNMNH-27470).

burrows, piles of rocks and termite mounds [21]. Members of *Geosciurus* are too big to be hunted by most nocturnal raptors, but Spotted eagle owl (*Bubo africanus*) and giant eagle owl (*Bubo lacteus*) have been reported preying upon them [83, 84]. The nomenclature of the teeth of fossil and modern representatives of African sciurids is provided in Denys et al. [85], along with a list of distinctive dental and skulls characters presented below (see Fig 119).

Family **SCIURIDAE** Fischer de Waldheim, 1817

Genus ***Geosciurus*** Smith, 1834 (Ground squirrels)

Figs 120–122; Table 45

Dental formula is 1-0-1-3-:1-0-1-3.

**Upper jaw.**  Upper incisors are ungrooved and opisthodont. The palatine bone extends well behind the molars. There is a well-developed masseter knob. A single premolar ($P^4$) is present in the upper jaw (there are two premolars in *Paraxerus*). Molars have three transverse lophs (anteroloph, metaloph, posteroloph) relating bunodont cusps on the occlusal surface. They display a big protocone on the whole lingual part of the molar, while the hypocone is hardly visible. In $M^1$ and $M^2$, the metaloph is short and is connected to the posteroloph. The presence of a mesostyle varies among specimens.

**Lower jaw.**  Lower incisors are ungrooved. Molars have well individualized bunodont cusps (a prominent hypoconid and protoconid, a metaconid and a crestiform entoconid) connected by transverse ridges (protolophid, entolophid, posterolophid) that develop with wear. The molars have a prominent hypoconid In $M_1$ and $M_2$, the cusps protoconid and metaconid are fused by their distal side (anterior side in *Paraxerus*). The mandible is sciurognath, with its angular part stockily built.

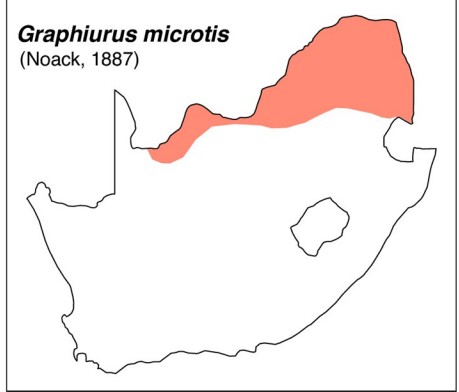

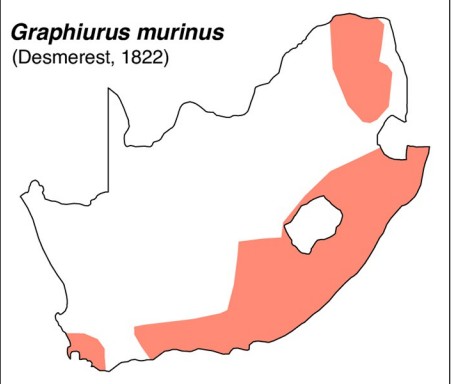

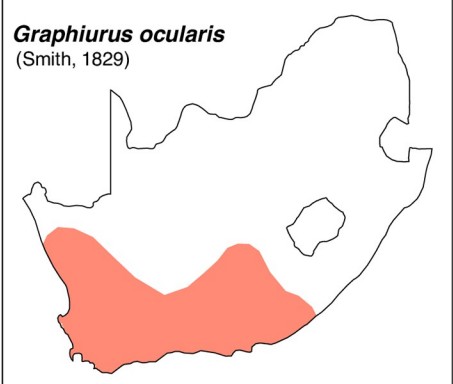

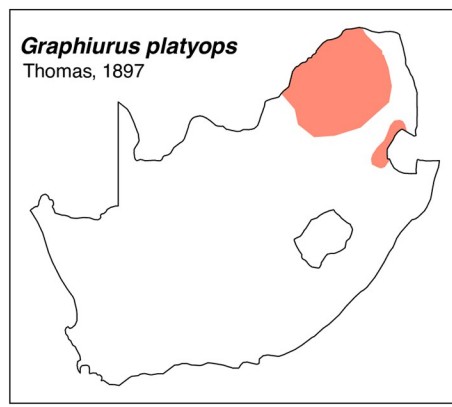

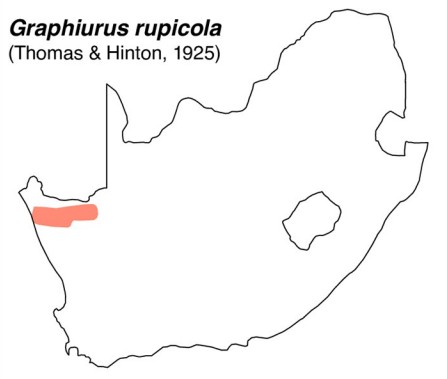

**Fig 118. Distribution maps.**

**Table 42. Dental measurements (in mm) for *Graphiurus* from South Africa, sexes and species combined.**

|            | Mean | Min | Max | n  |
|------------|------|-----|-----|----|
| LLTR       | 3.0  | 2.7 | 3.3 | 18 |
| $WM_1$     | 1.0  | 0.7 | 1.1 | 21 |
| LUTR       | 3.2  | 2.7 | 3.6 | 17 |
| $WM^1$     | 1.1  | 0.9 | 1.2 | 18 |

**Table 43. Key to the sciurid genera: Upper jaw.**

| 1 | four cheekteeth; LUTR > 10 mm; in $M^1$ and $M^2$, metaloph is short and connected to the posteroloph; palatine bone extends well behind the molars | *Geosciurus* |
|---|---|---|
|   | five cheekteeth; LUTR < 10 mm; in $M^1$ and $M^2$, metaloph is long and connected to the protocone; palatine bone extends only the posterior edge of the $M^3$ | *Paraxerus* |

**Table 44. Key to the sciurid genera: Lower jaw.**

| 1 | LLTR > 10 mm; in $M_1$ and $M_2$, protoconid and metaconid fused by posterior edge of cusps | *Geosciurus* |
|---|---|---|
|   | LLTR < 9 mm; in $M_1$ and $M_2$, protoconid and metaconid fused by anterior edge of cusps | *Paraxerus* |

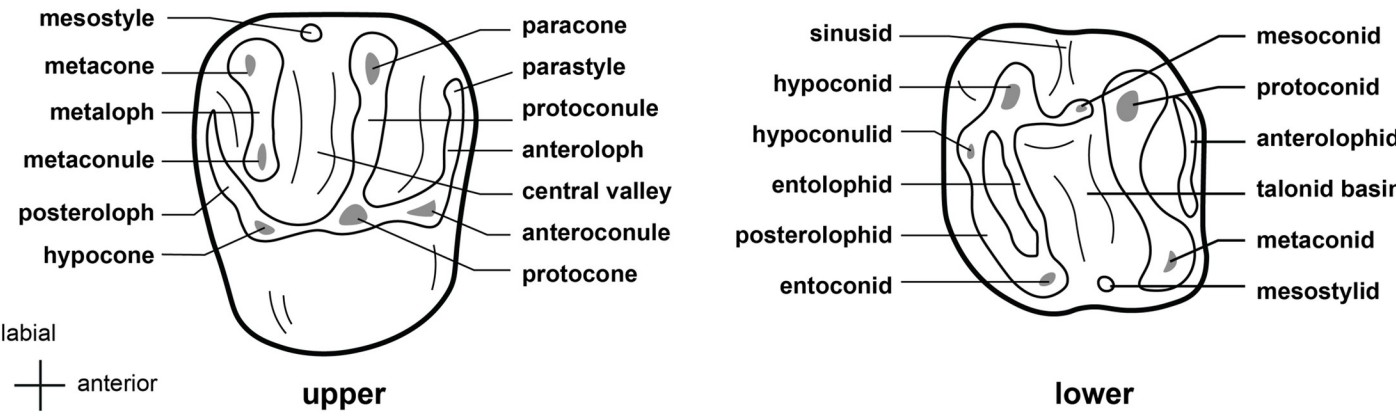

**Fig 119. Sciurid right upper and left lower molars with nomenclature of the cusps, adapted from Cuenca-Bescós (1988) [86] and Viriot et al., (2011) [87].**

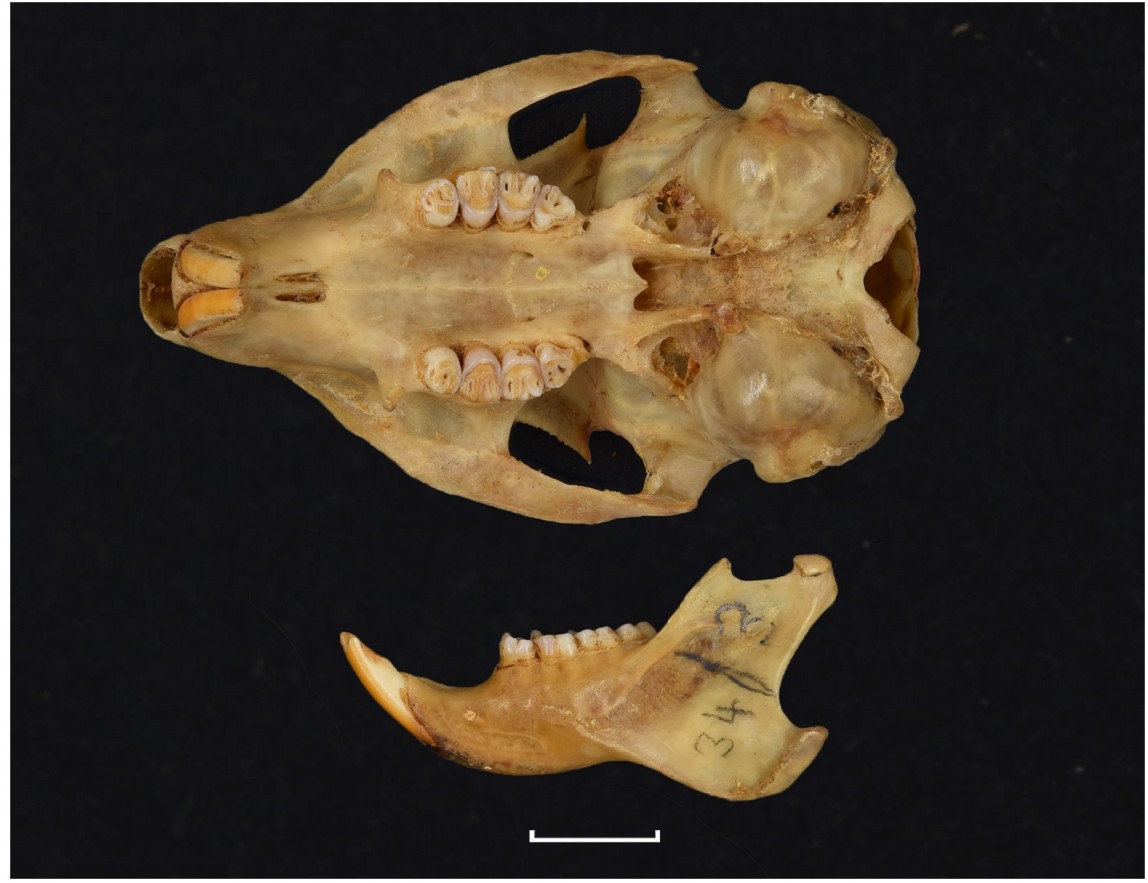

**Fig 120. Cranium of *Geosciurus princeps* (DNMNH-8344), with scale bar of 1 cm.**

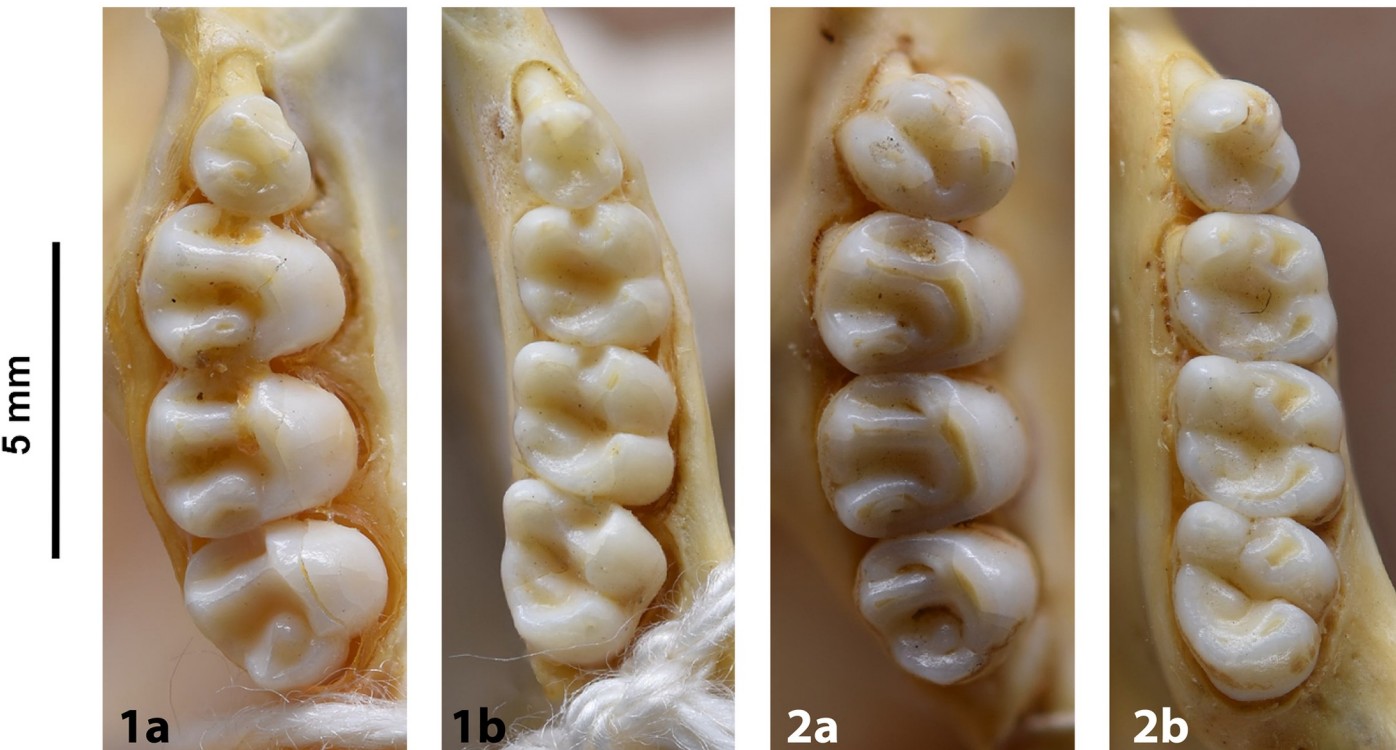

**Fig 121. Craniodental anatomy of *Geosciurus*.** 1) Upper (a) and lower (b) right toothrow of *G. inauris* (DNMNH-15143); 2) Upper (a) and lower (b) right toothrow of *G. princeps* (DNMNH-6327).

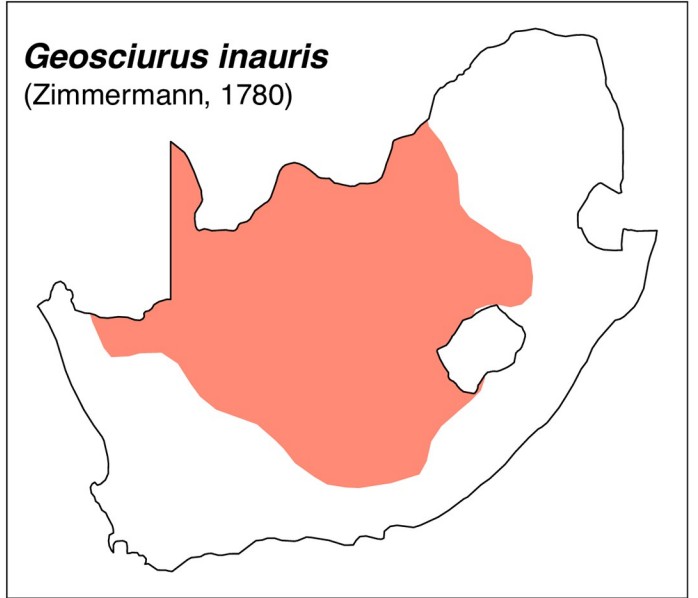
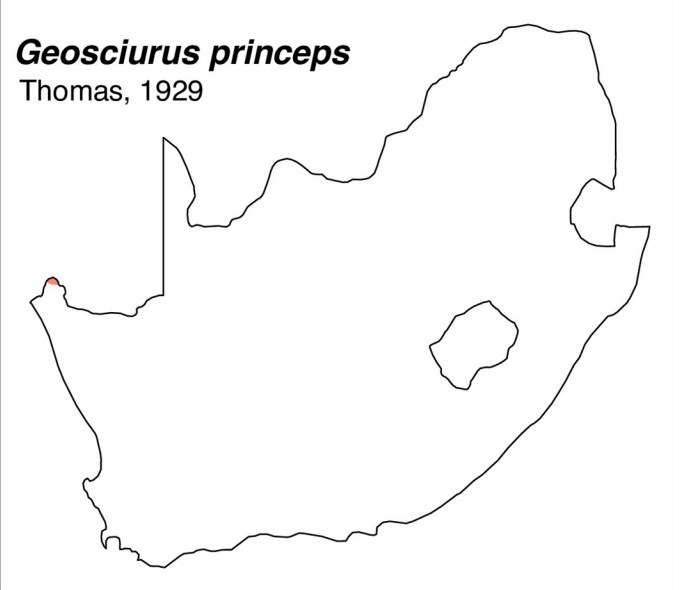

*Geosciurus inauris*
(Zimmermann, 1780)

*Geosciurus princeps*
Thomas, 1929

**Fig 122. Distribution maps.**

**Table 45. Dental measurements (in mm) for *Geosciurus* from South Africa, sexes and species combined.**

|  | Mean | Min | Max | n |
|---|---|---|---|---|
| LLTR | 12.1 | 11.3 | 13.3 | 32 |
| WM$_1$ | 3.3 | 3.0 | 3.9 | 32 |
| LUTR | 11.0 | 10.2 | 11.9 | 32 |
| WM$^1$ | 4.1 | 3.1 | 4.7 | 32 |

**Systematic notes and South African fossil record.** This genus was previously included within *Xerus*, until recent phylogenetic reconstructions led to the recognition of the genus *Geosciurus* for specimens from Southern Africa [88]. Two species are currently recognised:

- *Geosciurus inauris* (ZIMMERMANN, 1780)

- *Geosciurus princeps* THOMAS, 1929

Remains of this genus are scanty in the Quaternary fossil record. The oldest record of *Geosciurus* is from the Middle Pleistocene locality of Florisbad [89, 90].

Genus ***Paraxerus*** Forsyth Major, 1893 (Bush Squirrels)

Figs 123–125; Table 46

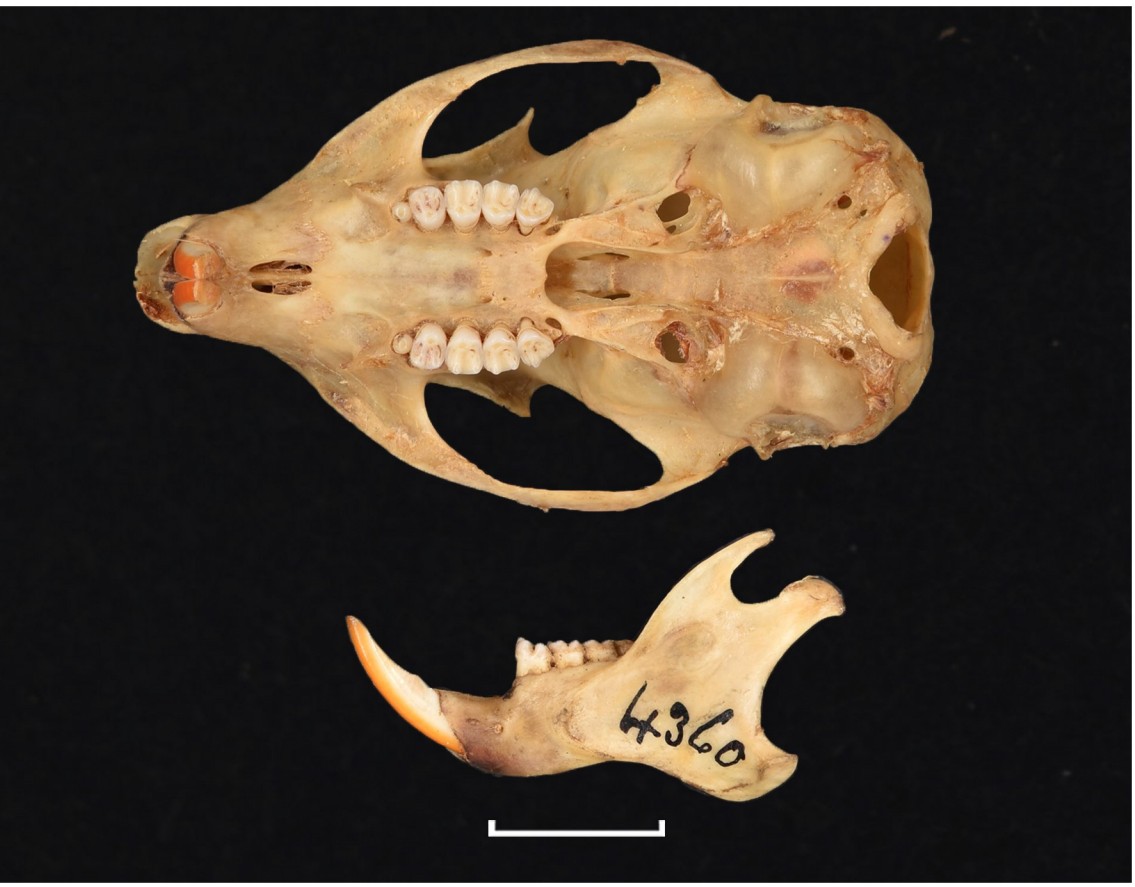

**Fig 123. Cranium of *Paraxerus cepapi* (skull DNMNH-4363, mandible DNMNH-6322), with scale bar of 1 cm.**

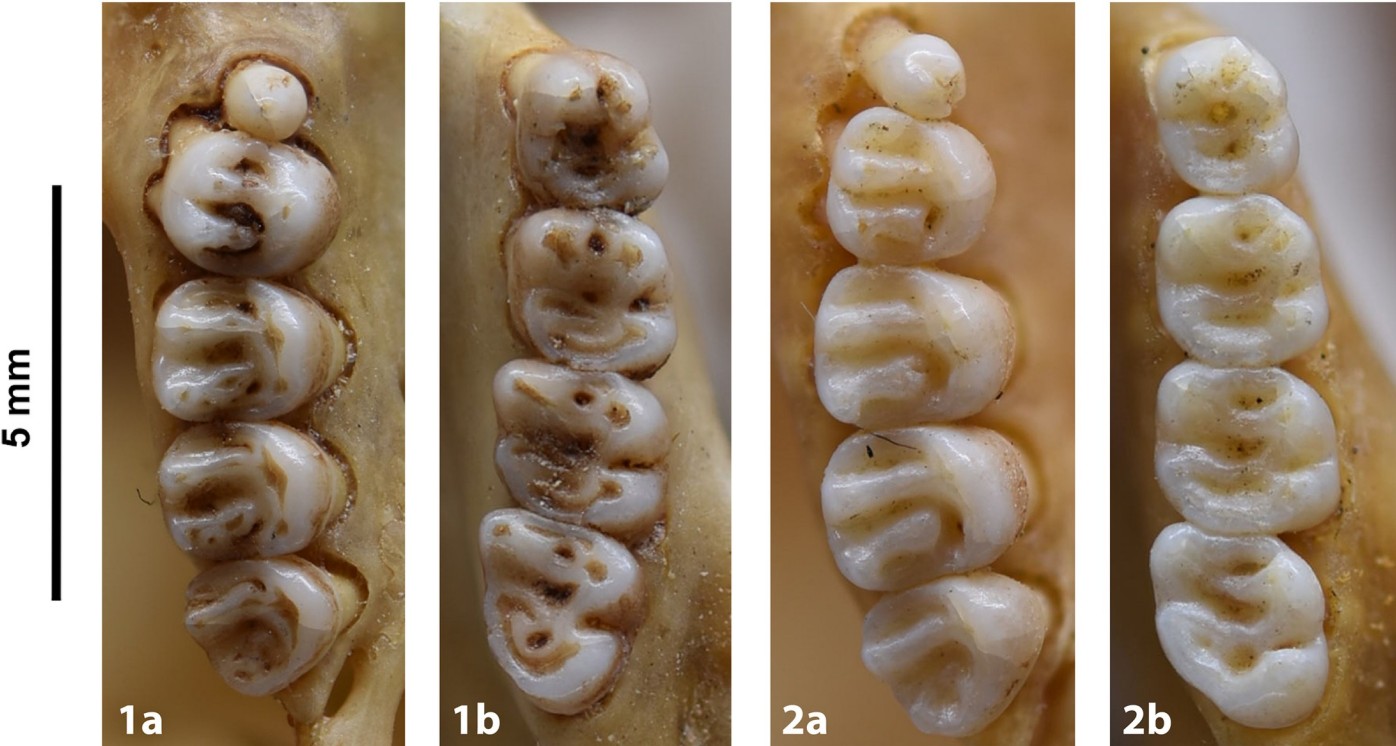

**Fig 124. Cheekteeth of *Paraxerus*. 1)** Upper (a) and lower (b) right toothrow of *P. cepapi* (DNMNH-4360); **2)** Upper (a) and lower (b) right toothrow of *P. palliatus* (DNMNH-6214).

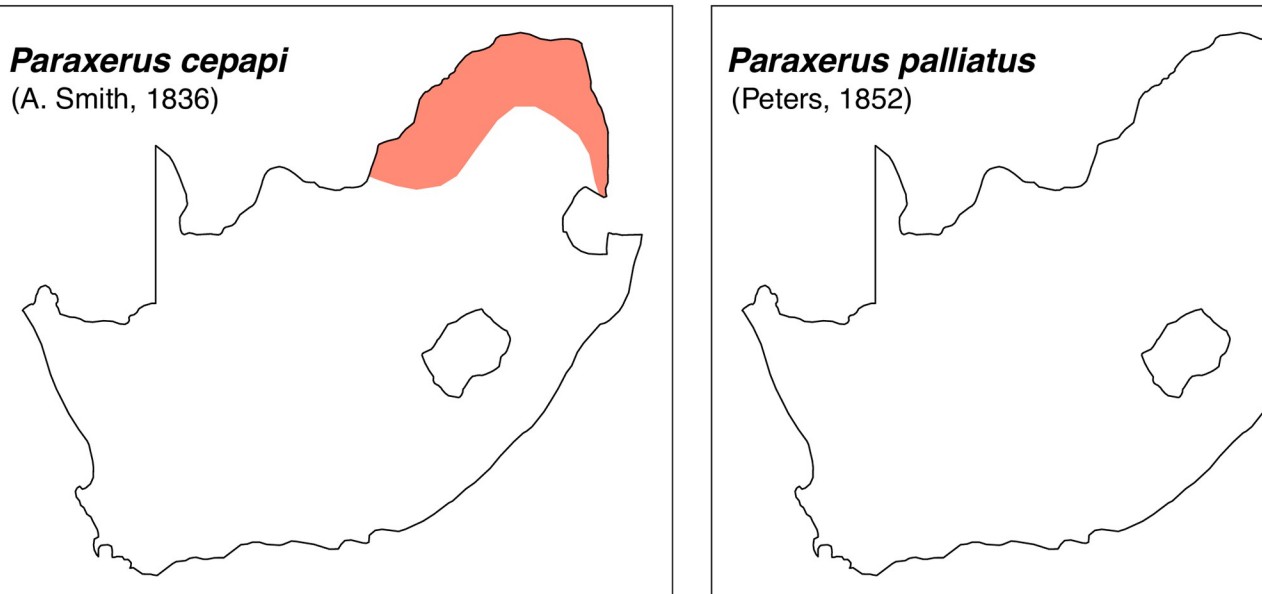

**Fig 125. Distribution maps.**

**Table 46. Dental measurements (in mm) for *Paraxerus* from South Africa, sexes and species combined.**

|                  | Mean | Min | Max | n  |
|------------------|------|-----|-----|----|
| LLTR             | 8.4  | 7.6 | 9.3 | 30 |
| $WM_1$           | 2.2  | 1.9 | 2.4 | 30 |
| LUTR             | 8.4  | 7.2 | 9.7 | 28 |
| $WM^1$           | 2.6  | 2.3 | 3.0 | 29 |

Dental formula is 1-0-2-3:1-0-1-3.

**Upper jaw.** Upper incisors are orthodont and ungrooved. The palatine bone extends only the posterior edge of the $M^3$. The anterior palatal foramina are short and located far forward from the teeth. There are five upper cheekteeth, with the presence of a minute $P^3$. Molars are of approximately the same size and display transverse lophs (anteroloph, protoloph, metaloph, posteroloph) relating bunodont cusps on the occlusal surface. $M^1$ and $M^2$ both have three transverse valleys, $M^3$ has two. In $M^1$-$M^2$, the metaloph is long and connected to the proto-coone, and the mesostyle is absent. In the $M^3$ the hypocone and protocone are connected.

**Lower jaw.** Lower incisors are ungrooved. Molars are ridged more or less transversely and display four main cusps which are not always visible with wear: a big protoconid fused to a metaconid by the anterior side (posterior in *Geosciurus*), a hypoconid and a transverse entoco-nid. The $M_3$ is the largest tooth of the molar row. The mandible is sciurognath; it has a high coronoid process, well segregated from the condylar process.

**Systematic notes and South African fossil record.** Two species are currently recognized in South Africa:

- *Paraxerus cepapi* (A. Smith, 1836)

- *Paraxerus palliates* (Peters, 1852)

Fossils of this genus are rare in the Quaternary fossil record, with a single Pleistocene record from Border Cave [91] and a few Holocene records [50].

## 6. Discussion and conclusion

Recent advances in rodent taxonomy at the subfamily, genus, and species level, together with the need for more detailed description of their dental anatomy, has encouraged us to provide an updated version of identification keys previously published by Coetzee [23] and De Graaff [24], based on standard, and new, morphological characters. This contribution has endeavored to assemble the latest information regarding rodent systematics, taxonomy, and palaeontology. We aim to support and facilitate the identification of rodent specimens from Quaternary and modern coprocenoses by providing a simple identification tool based on a traditional dichoto-mous system and scaled photographs of the skull and dentition.

We have attempted to compile the most reliable characters based on collection specimens identified on cytogenetic or molecular grounds mainly. Some published questionable charac-ters were discarded (for instance, the presence of an accessory anterior median cusp on $M^1$ was previously used to distinguish *Thallomys* from *Grammomys*) and characters likely to dis-play variability have received less emphasis in this work than in previous identification keys (such as the presence of accessory cusplets or posterior cingulum). Instead, we have retained mainly those features that we were able to observe consistently in the collection specimens ourselves.

Regrettably, however, isolated teeth and fragmented specimens from owl pellet or fossil material may remain unidentifiable, even at the genus level. Most of the criteria used in our

key involve having the complete dental row preserved. In a similar way, this key does not include genera found in the savannahs and deserts of Botswana, Zambia, Namibia and Zimbabwe that were possibly found further south or west during the Pleistocene period. This remark concerns, for instance, the genus *Pelomys*, which was identified by some authors among the fossil remains from Makapansgat, Border Cave, and Sterkfontein [47, 70, 92].

Accurate taxonomic identification is essential for a wide range of scientific applications, from ecology, conservation biology and pest management, to archaeozoology and palaeoenvironmental reconstruction. The species level generally constitutes the fundamental unit of investigation. For well-preserved modern specimens, one may have little difficulty in identifying features characteristic of a species using criteria available in literature coupled with biogeographic data. Works of this kind are to be found in general monographs or more specialised journal articles, and we have elaborated on the factors that led us to restrict our key to genus level. Problems arise with broken specimens from modern and palaeo coprocenoses, which lack diagnostic criteria. Moreover, several species cannot be distinguished on the basis of craniodental anatomy unless time-intensive techniques such as geometric morphometrics are employed, and this may not always be feasible. Finally, when it comes to fossils, the use of distribution data is extremely risky, and may explain the disagreement that exists between some researchers on the status of different fossil species, especially those described during the last century (this is the case, for example, of the large number of species described by Broom in the 1930s and 1940s).

Ironically, the huge progress made in systematics and species identification through the use of numerical taxonomy and molecular taxonomy has eroded our confidence in identifying species based on bone and teeth remains. However, the discovery of new fossil rodents in South Africa is continuing at a rapid pace, and taxonomic identification of craniodental material remains the foundation of palaeontological and archaeological research. Identification skills rely mainly on experience and repeated observations of key features. In this respect, this key should be a valuable resource for both professionals and amateurs alike. It is particularly beneficial regarding the numerous genera which have similar morphological features, and which are challenging to differentiate. Nevertheless, we must emphasize that the identification key is intended to complement the use of official natural history collections in order to facilitate accurate identification.

## Supporting information

**S1 Fig. Comparative osteological plates.** Upper jaws.
(PDF)

**S2 Fig. Comparative osteological plates.** Lower jaws.
(PDF)

**S1 Checklist. References of specimens used for osteological plates.**
(DOCX)

**S1 Table. Material examined for measurements.**
(XLSX)

## Acknowledgments

This study could not have been attempted without the help of collection curators and researchers who helped us and permitted us access to their collections. We would like to thank here Josef Bryja and Ondřej Mikula from the Institute of Vertebrate Biology (Czech Republic),

Violaine Nicolas from the Muséum national d'Histoire naturelle (France), Teresa Kearney from the Ditsong National Museum of Natural History (Republic of South Africa) and Emmanuel Gilissen from the Royal Museum for Central Africa (Belgium).

## Author Contributions

**Conceptualization:** Pierre Linchamps, Emmanuelle Stoetzel.

**Data curation:** Christiane Denys.

**Formal analysis:** Pierre Linchamps.

**Investigation:** Pierre Linchamps, D. Margaret Avery, Christiane Denys.

**Methodology:** Pierre Linchamps, Christiane Denys, Emmanuelle Stoetzel.

**Supervision:** D. Margaret Avery, Raphaël Cornette, Christiane Denys, Thalassa Matthews, Emmanuelle Stoetzel.

**Validation:** D. Margaret Avery, Christiane Denys, Thalassa Matthews, Emmanuelle Stoetzel.

**Writing – original draft:** Pierre Linchamps.

**Writing – review & editing:** D. Margaret Avery, Raphaël Cornette, Christiane Denys, Thalassa Matthews, Emmanuelle Stoetzel.

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
