## [Decision Letter · Decision Letter 0]

29 May 2023

PONE-D-23-12553Quaternary Rodents of South Africa: A companion guide for cranio-dental identificationPLOS ONE

Dear Dr. Linchamps,

Thank you for submitting your manuscript to PLOS ONE. After careful consideration, we feel that it has merit but does not fully meet PLOS ONE’s publication criteria as it currently stands. Therefore, we invite you to submit a revised version of the manuscript that addresses the points raised during the review process.

ACADEMIC EDITOR:Kindly read the comment from the reviewer and address it in the revised manuscript.

1. suggest that authors could display some pictures of digestion that could eventually have taxonomic repercussion.

We look forward to receiving your revised manuscript.

Kind regards,

Mohmed Isaqali Karobari, BDS, MScD.Endo, MFDS.RCPS Glasg, Ph.D. scholar

Academic Editor

PLOS ONE

Journal Requirements:

2. In your manuscript, please provide additional information regarding the specimens used in your study. Ensure that you have reported human remain specimen numbers and complete repository information, including museum name and geographic location. 

For more information on PLOS ONE's requirements for paleontology and archeology research, see https://journals.plos.org/plosone/s/submission-guidelines#loc-paleontology-and-archaeology-research.

"Funds were given for accessing specimens from Czech Republic and South Africa by the Partenariat Hubert Curien (PHC) Barrande and the international mobility program Transhumance of the doctoral school 227 “Sciences de la nature et de l’Homme” from the Muséum national d’Histoire naturelle-Sorbonne Université. "

4. We note that you have referenced (-†Dasymys bolti BROOM, UNPUBLISHED, -†Dasymys broomi BROOM, UNPUBLISHED and -†Dasymys lavocati BROOM, UNPUBLISHED) on page 22 which has currently not yet been accepted for publication. Please remove this from your References and amend this to state in the body of your manuscript: (ie “Bewick et al. [Unpublished]”) as detailed online in our guide for authors

http://journals.plos.org/plosone/s/submission-guidelines#loc-reference-style.

Additional Editor Comments:

Dear Authors,

Kindly read the comment from the reviewer and address it in the revised manuscript.

1. suggest that authors could display some pictures of digestion that could eventually have taxonomic repercussion.

Best regards and keep well

Reviewers' comments:

Reviewer's Responses to Questions

**Comments to the Author**

1. Is the manuscript technically sound, and do the data support the conclusions?

Reviewer #1: Yes

Reviewer #2: Yes

2. Has the statistical analysis been performed appropriately and rigorously? 

Reviewer #1: Yes

Reviewer #2: N/A

3. Have the authors made all data underlying the findings in their manuscript fully available?

Reviewer #1: Yes

Reviewer #2: Yes

4. Is the manuscript presented in an intelligible fashion and written in standard English?

Reviewer #1: Yes

Reviewer #2: Yes

5. Review Comments to the Author

Reviewer #1: This manuscript is an excellent work. It will be useful both in current studies that include taxonomic determinations of rodents from southern Africa, as well as in work on faunal assemblages from the Quaternary period of the area.

I congratulate the authors for this contribution.

Reviewer #2: I can only say to the authors: Thanks and congratulations. This is a Master PIece extraordinary useful and no doubt that it will be extensively used and referred. The authors count with the main South African experts in taxonomic small mammal identifications. Among the authors there are also taphonomists and I would maybe suggest that they could display some pictures of digestion that could eventually have taxonomic repercussion. Apart from this suggestion, I again congratulate the initiative and the complete guide of identification that this paper is.

6. PLOS authors have the option to publish the peer review history of their article (what does this mean?). If published, this will include your full peer review and any attached files.

Reviewer #1: No

Reviewer #2: **Yes: **YOLANDA FERNANDEZ JALVO

While revising your submission, please upload your figure files to the Preflight Analysis and Conversion Engine (PACE) digital diagnostic tool, https://pacev2.apexcovantage.com/. PACE helps ensure that figures meet PLOS requirements. To use PACE, you must first register as a user. Registration is free. Then, login and navigate to the UPLOAD tab, where you will find detailed instructions on how to use the tool. If you encounter any issues or have any questions when using PACE, please email PLOS at figures@plos.org. Please note that Supporting Information files do not need this step.<quillbot-extension-portal></quillbot-extension-portal>

---

## [Author Response · Author response to Decision Letter 0]

13 Jul 2023

Dear reviewers, 

Thank you for your feedback and this opportunity to improve the manuscript.

Academic editor and reviewers suggest that we could display some pictures of digestion that could eventually have taxonomic repercussion. We have therefore added a small paragraph with a photo of digestion. We have taken advantage of this new version to make a few additional changes to the structure of the manuscript.

We thank you for your time and attention and look forward to hearing from you.

Sincerely yours,

---

## [Decision Letter · Decision Letter 1]

27 Jul 2023

Quaternary Rodents of South Africa: A companion guide for cranio-dental identification

PONE-D-23-12553R1

Dear Dr. Linchamps,

We’re pleased to inform you that your manuscript has been judged scientifically suitable for publication and will be formally accepted for publication once it meets all outstanding technical requirements.

Kind regards,

Mohmed Isaqali Karobari, BDS, MScD.Endo, MFDS.RCPS Glasg, Ph.D. scholar

Academic Editor

PLOS ONE

Additional Editor Comments (optional):

Dear Authors,

The authors have addressed all the comments and manuscript has improved much. I would like to wish all the very best for the authors for their future endeavors.

Best regards and keep well

Reviewers' comments:

Reviewer's Responses to Questions

**Comments to the Author**

1. If the authors have adequately addressed your comments raised in a previous round of review and you feel that this manuscript is now acceptable for publication, you may indicate that here to bypass the “Comments to the Author” section, enter your conflict of interest statement in the “Confidential to Editor” section, and submit your "Accept" recommendation.

Reviewer #1: All comments have been addressed

2. Is the manuscript technically sound, and do the data support the conclusions?

Reviewer #1: Yes

3. Has the statistical analysis been performed appropriately and rigorously? 

Reviewer #1: (No Response)

4. Have the authors made all data underlying the findings in their manuscript fully available?

Reviewer #1: Yes

5. Is the manuscript presented in an intelligible fashion and written in standard English?

Reviewer #1: Yes

6. Review Comments to the Author

Reviewer #1: I suggest the publication of this manuscript. It is very interesting and it will be a very useful tool for scholars of the fauna of the study area.

7. PLOS authors have the option to publish the peer review history of their article (what does this mean?). If published, this will include your full peer review and any attached files.

Reviewer #1: No

<quillbot-extension-portal></quillbot-extension-portal>

---

## [Editor Report · Acceptance letter]

31 Jul 2023

PONE-D-23-12553R1 

Quaternary Rodents of South Africa: A companion guide for cranio-dental identification 

Dear Dr. Linchamps:

I'm pleased to inform you that your manuscript has been deemed suitable for publication in PLOS ONE. Congratulations! Your manuscript is now with our production department. 

Kind regards, 

on behalf of

Dr. Mohmed Isaqali Karobari 

Academic Editor

PLOS ONE